# Algorithms and SQ Lower Bounds for Robustly Learning Real-valued Multi-index Models

Ilias Diakonikolas
UW Madison
ilias@cs.wisc.edu

Giannis Iakovidis
UW Madison
iakovidis@wisc.edu

Daniel M. Kane
UC San Diego
dakane@ucsd.edu

Lisheng Ren
UW Madison
lren29@wisc.edu

## Abstract

We study the complexity of learning real-valued Multi-Index Models (MIMs) under the Gaussian distribution. A $K$-MIM is a function $f : \mathbb{R}^d \to \mathbb{R}$ that depends only on the projection of its input onto a $K$-dimensional subspace. We give a general algorithm for PAC learning a broad class of MIMs with respect to the square loss, even in the presence of adversarial label noise. Moreover, we establish a nearly matching Statistical Query (SQ) lower bound, providing evidence that the complexity of our algorithm is qualitatively optimal as a function of the dimension. Specifically, we consider the class of bounded variation MIMs with the property that degree at most $m$ distinguishing moments exist with respect to projections onto any subspace. In the presence of adversarial label noise, the complexity of our learning algorithm is $d^{O(m)}2^{\mathrm{poly}(K/\epsilon)}$. For the realizable and independent noise settings, our algorithm incurs complexity $d^{O(m)}2^{\mathrm{poly}(K)}(1/\epsilon)^{O(K)}$. To complement our upper bound, we show that if for some subspace degree-$m$ distinguishing moments do not exist, then any SQ learner for the corresponding class of MIMs requires complexity $d^{\Omega(m)}$. As an application, we give the first efficient learner for the class of positive-homogeneous $L$-Lipschitz $K$-MIMs. The resulting algorithm has complexity $\mathrm{poly}(d)2^{\mathrm{poly}(KL/\epsilon)}$. This gives a new PAC learning algorithm for Lipschitz homogeneous ReLU networks with complexity independent of the network size, removing the exponential dependence incurred in prior work.

## 1 Introduction

A common assumption in supervised learning is that real-world data possess hidden low-dimensional structure, in the sense that the relationship between the features is of a lower-dimensional nature. A natural formalization of this principle leads to the notion of a Multi-index model [FJS81, Hub85, Li91, HL93, XTLZ02, Xia08], defined below.

**Definition 1.1** (Multi-Index Model (MIM)). A function $f : \mathbb{R}^d \to \mathbb{R}$ is a $K$-MIM if there exists a $K$-dimensional subspace $W \subseteq \mathbb{R}^d$ such that $f(\mathbf{x}) = f(\mathbf{x}_W)$ for all $\mathbf{x} \in \mathbb{R}^d$, where $\mathbf{x}_W$ is the projection of $\mathbf{x}$ onto $W$. The special case of $K = 1$ corresponds to Single-Index Models (SIMs).

A few comments are in order. First, the dimension $K$ of the hidden subspace is typically assumed to be significantly smaller than the ambient dimension $d$. Second, certain regularity assumptions

---

Authors are listed in alphabetical order.

39th Conference on Neural Information Processing Systems (NeurIPS 2025).

on the target class are required for learning to be (even information-theoretically) possible. MIMs can be viewed as a lens for studying neural networks and other natural function classes. In recent years, we have witnessed a resurgence of research interest on learning SIMs and special cases of MIMs; see,e.g., [DH18, DKKZ20, DK20, AGJ21, DKTZ22, CKM22, WZDD23, GGKS23, DPLB24, ZWDD24, WZDD24, DIKZ25, ZWDD25] and references therein. See Section 1.3 for a summary of related work. Yet, our understanding of the computational complexity of learning MIMs remains limited, especially in the presence of noisy data.

The main result of this paper is an efficient robust regression algorithm, with respect to the square loss, for a broad class of MIMs. We complement our upper bound with a nearly matching Statistical Query lower bound, providing evidence that the sample complexity of our algorithm is qualitatively optimal—as a function of the dimension—for computationally efficient algorithms.

We start with the definition of learning in our context.

**Definition 1.2** (Agnostic PAC Learning under Gaussian Distribution). Let $\mathcal{C}$ be a class of functions $f : \mathbb{R}^d \to \mathbb{R}$ and $D$ be a distribution of $(\mathbf{x}, y)$ over $\mathbb{R}^d \times \mathbb{R}$ with $D_{\mathbf{x}}$ equal to the standard Gaussian. Given i.i.d. samples from $D$, the goal is to output a hypothesis $h : \mathbb{R}^d \to \mathbb{R}$ such that with high probability the error $\mathrm{err}_D(h) \overset{\text{def}}{=} \mathbf{E}_{(\mathbf{x},y)\sim D}[(y - h(\mathbf{x}))^2]$ is small, compared to $\mathrm{OPT} \overset{\text{def}}{=} \inf_{f \in \mathcal{C}} \mathrm{err}_D(f)$.

Definition 1.2 corresponds to the agnostic model [Hau92, KSS94] that does not make any assumptions on the labels. The special case corresponding to $\mathrm{OPT} = 0$ (when each label is consistent with a function in the class) is known as realizable PAC learning [Val84]. Moreover, the goal is to find a hypothesis with small loss—as opposed to identifying the parameters of the target function.

## 1.1 Our Results

We give a new algorithm for learning MIMs under fairly general assumptions. Essentially, our algorithm is an iterative subspace finding method that learns better and better approximations $V$ to the hidden subspace $W$. Our method succeeds under suitable conditions on the target MIM class. Roughly, we need to know that, for any subspace $V$, either $V$ is a good enough approximation to $W$ (i.e., we can use it to learn $f$); or that by computing moments of $\mathbf{x}$ conditioned on the value of $f$ and the projection onto $V$, we can learn some previously undiscovered direction in $W$. We additionally require the technical conditions that the target function has bounded norm and bounded variation, as the sample complexity of learning inherently scales with these bounds.

For two subspaces $W, V$ of $\mathbb{R}^d$, denote by $W_V \overset{\text{def}}{=} \{\mathbf{w}_V : \mathbf{w} \in W\}$ which is itself a subspace. The necessary condition for our function class is given in the following definition.

**Definition 1.3** (Well-Behaved MIMs). Let $d, K, m \in \mathbb{Z}_+$ and $\zeta, \tau, \sigma > 0$. We define the class $\mathcal{F}(K, m, \zeta, \tau, \sigma)$ as the set of all continuous and continuously differentiable almost everywhere $K$-MIM functions $f : \mathbb{R}^d \to \mathbb{R}$ which have the following properties:

1. $\mathbf{E}_{\mathbf{x}\sim\mathcal{N}_d}[\|\nabla f(\mathbf{x})\|^2], \mathbf{E}_{\mathbf{x}\sim\mathcal{N}_d}[f^2(\mathbf{x})]$ are finite and $f$ is close to a bounded function in $L_2$-norm[1].
2. For any subspace $V \subseteq \mathbb{R}^d$ and any distribution $D$ on $\mathbb{R}^d \times \mathbb{R}$ with $D_{\mathbf{x}} = \mathcal{N}_d$ such that $\mathrm{err}_D(f) \leq \zeta$ either (a) there exists $g : V \to \mathbb{R}$ such that $\mathbf{E}_{\mathbf{x}\sim\mathcal{N}_d}[(f(\mathbf{x}) - g(\mathbf{x}_V))^2] \leq \tau$, or (b) with nontrivial probability over $\mathbf{z} \sim \mathcal{N}_d$ independent of $\mathbf{x}$ there exists a degree at most $m$, zero-mean, unit variance polynomial $p : U \to \mathbb{R}$, where $U = W_{V^\perp}$ and $W$ is the hidden $K$-dimensional subspace corresponding to $f$, such that $\mathbf{E}_{y_0\sim(D_y|\mathbf{x}_V=\mathbf{z}_V)}\left[\mathbf{E}_{\mathbf{x}\sim\mathcal{N}_d}[p(\mathbf{x}_U)|\mathbf{x}_V = \mathbf{z}_V, y = y_0]^2\right] \geq \sigma$.

Our main algorithmic result is the following:

**Theorem 1.4** (Robust Regression for Well-behaved MIMs). *Let $D$ be a distribution on $\mathbb{R}^d \times \mathbb{R}$ with $D_{\mathbf{x}} = \mathcal{N}_d$. There exists an agnostic PAC learner for $\mathcal{F}(K, m, \zeta, \tau, \sigma)$, where $\zeta \geq \mathrm{OPT} + \epsilon$, that draws $N = d^{O(m)}2^{\mathrm{poly}_m(K/(\epsilon\sigma))}$ i.i.d. samples, runs in $\mathrm{poly}(N)$ time, and computes a hypothesis $h$ such that with high probability $\mathrm{err}_D(h) \leq \tau + \mathrm{OPT} + \epsilon$.*

We establish a similar algorithmic result for the realizable and independent label noise settings. In these (easier) settings, the complexity of our algorithm becomes $d^{O(m)}2^{\mathrm{poly}(K)}(1/\epsilon)^{O(K)}\mathrm{poly}(1/\sigma)$, i.e., we incur exponential dependence only on $K$. This is because in these settings the label is independent of the irrelevant subspace $W^\perp$, ensuring that

---

[1] This is a mild assumption which holds, e.g., when the function has bounded 2.1-degree moment.

every direction extracted by our algorithm lies (up to estimation error) within $W$. For the details, we refer the reader to Section D.2.

As we establish in Theorem 1.10, the $d^m$ complexity dependence is qualitatively optimal in the Statistical Query model, even in the realizable (clean label) setting.

As an application of our general algorithmic technique, we obtain the first learner for positive-homogeneous Lipschitz MIMs whose complexity is a fixed-degree polynomial in the dimension.

**Definition 1.5** (Positive-Homogeneous Lipschitz MIMs). For $K \in \mathbb{Z}_+$ and $L > 0$, we define $\mathcal{H}_{K,L}$ to be the class of all $L$-Lipschitz and unit 2-norm $K$-MIMs $f : \mathbb{R}^d \to \mathbb{R}$ such that $f$ is positive-homogeneous, i.e., $f(t\mathbf{x}) = t\, f(\mathbf{x})$ for all $t > 0, \mathbf{x} \in \mathbb{R}^d$.

We note that $\mathcal{H}_{K,L}$ is a broad nonparametric class containing various MIMs of interest. For example, it contains the class of Lipschitz and homogeneous ReLU networks (since the ReLU activation is itself positive-homogeneous). As an application of our general algorithm, we show:

**Theorem 1.6** (PAC Learning $\mathcal{H}_{K,L}$). *Let $D$ be the distribution of $(\mathbf{x}, f(\mathbf{x}))$, where $\mathbf{x} \sim \mathcal{N}_d$ and $f \in \mathcal{H}_{K,L}$. There exists an algorithm that draws $N = d^2\, 2^{O(K^3 L^2 / \epsilon^2)}$ i.i.d. samples from $D$, runs in time $\mathrm{poly}(N)$, and returns a hypothesis $h$ such that with high probability $\mathrm{err}_D(h) \leq \epsilon$.*

As an immediate corollary of Theorem 1.6, we obtain a new algorithm—with qualitatively better complexity—for homogeneous Lipschitz ReLU networks. Let $\mathcal{F}_{S,K,L}$ be the class of $L$-Lipschitz functions of the form $f(\mathbf{x}) = \mathbf{W}_D \phi(\mathbf{W}_{D-1}(\cdots \phi(\mathbf{W}_1 \mathbf{x}) \cdots))$, where $\phi(z) = \max\{z, 0\}$ is the ReLU activation, $\mathbf{W}_i \in \mathbb{R}^{k_{i+1} \times k_i}$, $i \in [D-1]$, with $k_1 = d$ and $k_D = 1$, $\mathrm{rank}(\mathbf{W}_1) \leq K$, and $S = \sum_{i=2}^{D} k_i$. Since $\mathcal{F}_{S,K,L} \subset \mathcal{H}_{K,L}$, we obtain the following.

**Corollary 1.7** (Learning ReLU Networks). *Let $D$ be the distribution of $(\mathbf{x}, f(\mathbf{x}))$, with $\mathbf{x} \sim \mathcal{N}_d$ and $f \in \mathcal{F}_{S,K,L}$. There is an algorithm that draws $N = d^2\, 2^{O(K^3 L^2 / \epsilon^2)}$ samples from $D$, runs in $\mathrm{poly}(N)$ time , and returns a hypothesis $h$ such that with high probability $\mathrm{err}_D(h) \leq \epsilon$.*

Corollary 1.7 improves on the prior work of [CKM22] by eliminating the complexity dependence on the network size $S$ (on which the prior algorithm of [CKM22] had an exponential dependence).

We now proceed to describe our Statistical Query lower bounds. We start with the model definition.

**Definition 1.8** (Statistical Query Model). Let $D$ be a distribution on $\mathbb{R}^d$. A *statistical query* is a bounded function $q : \mathbb{R}^d \to [0, 1]$. We define $\mathrm{STAT}(\tau)$ to be the oracle that given any such query $q$, outputs a value $v$ such that $|v - \mathbf{E}_{\mathbf{x} \sim D}[q(\mathbf{x})]| \leq \tau$, where $\tau > 0$ is the *tolerance* of the query. A *Statistical Query (SQ) algorithm* is an algorithm whose objective is to learn some information about an unknown distribution $D$ by making adaptive calls to the corresponding $\mathrm{STAT}(\tau)$ oracle.

Our SQ lower bound relies on the existence of a distribution on labeled examples that has similar low-degree moments as the standard Gaussian projected onto some subspace. Namely, for a distribution $(\mathbf{x}, y)$ supported on $\mathbb{R}^{d+1}$ and an appropriate subspace $V \subseteq \mathbb{R}^d$, the distribution of $\mathbf{x}_{V^\perp}$ conditioned on any fixed value of $\mathbf{x}_V$ and $y$ matches its first $m$ moments with $\mathcal{N}(0, \Pi_{V^\perp})$ (the standard Gaussian projected onto the subspace $V^\perp$), where $\Pi_{V^\perp}$ denotes the projection matrix of the subspace $V^\perp$.

**Definition 1.9** (Relative Matching of Degree-$m$ Moments). Let $m \in \mathbb{Z}_+$, $A$ be a distribution of $\mathbf{v}$ supported on $\mathbb{R}^n$ and $U \subseteq \mathbb{R}^n$ be a subspace. We say that $A$ matches degree-$m$ moments relative to the subspace $U$ (with the standard Gaussian projected onto $U^\perp$) if for almost all $\hat{\mathbf{v}} \in U$, under the distribution of $\hat{\mathbf{v}} = \mathbf{v}_U$, for all $m' \leq m$ it holds $\mathbf{E}_{\mathbf{v} \sim A | \mathbf{v}_U = \hat{\mathbf{v}}}[(\mathbf{v}_{U^\perp})^{\otimes m'}] = \mathbf{E}_{\mathbf{v} \sim \mathcal{N}(0, \Pi_{U^\perp})}[\mathbf{v}^{\otimes m'}]$, where we denote by $\mathbf{v}^{\otimes m'}$ the $m'$-fold tensor product.

We are now ready to state our SQ lower bound for agnostic PAC learning of $K$-MIMs under the Gaussian distribution.

**Theorem 1.10** (SQ Lower Bound for Learning $K$-MIMs). *Let $\mathcal{C}$ be a class of rotationally invariant $K$-MIMs on $\mathbb{R}^d$. Suppose there exist $m \in \mathbb{Z}_+$, $\tau > 0$, and a joint distribution $D$ of $(\mathbf{x}, y)$ supported on $\mathbb{R}^d \times \mathbb{R}$ with $D_{\mathbf{x}}$ equal to $\mathcal{N}_d$ such that for some subspace $V \subseteq \mathbb{R}^d$, we have:*

*1. The distribution $D$ matches degree-$m$ moments relative to the subspace $V \times \mathbb{R}$, where the extra $\mathbb{R}$ contains the label; and*

2. *Any function* $h : \mathbb{R}^d \to \mathbb{R}$ *has* $\mathbf{E}_{(\mathbf{x},y) \sim D}[(h(\mathbf{x}_V) - y)^2] \geq \tau$.

*Then, under the mild assumption that the extreme values of $y$ have small contribution to the variance, the following holds: for $d$ sufficiently large compared to $K$ and $m$, and $c \in (0,1)$, any SQ algorithm that learns $\mathcal{C}$ within error substantially better than $\tau$ given* $\mathrm{OPT} \leq \inf_{c \in \mathcal{C}} \mathrm{err}_D(c)$ *requires either a query to* $\mathrm{STAT}\left(d^{-(1-c)m/4}\right)$ *or* $2^{d^{\Omega(c)}}$ *many queries.*

Since a query to $\mathrm{STAT}(\tau)$ requires $\Omega(1/\tau^2)$ samples to simulate in general, the intuitive interpretation of our SQ lower bound is the following: any simulation of an SQ algorithm for our learning task using samples, either requires $d^{(1-c)/m/2}$ samples or exponential in $d^c$ time.

Note that Theorem 1.10 is essentially (up to some technical conditions on each side) a converse to Theorem 1.4. In particular, Theorem 1.10 says that if there is a subspace $V$ so that it is neither the case that $y$ is $\tau$-close to a function of $\mathbf{x}_V$ nor is there a non-trivial moment conditioned on $\mathbf{x}_V$ and $y$ of degree at most $m$, then it is SQ-hard to learn (with queries of $d^{-O(m)}$ accuracy) to error much better than $\tau$. On the other hand, Theorem 1.4 says that if for every subspace $V$ we either are approximated by a function of $\mathbf{x}_V$ or have a non-trivial conditional moment, then we can learn to error roughly $\tau$ in time $d^{O(m)}$ times some function of the other parameters.

It is worth pointing out that an SQ lower bound for realizable learning of $K$-MIMs can be obtained here as a corollary of Theorem 1.10 by additionally having that $\mathrm{OPT} = \inf_{c \in \mathcal{C}} \mathrm{err}_D(c) = 0$.

Both Definition 1.9 and the corresponding SQ lower bounds for learning MIMs can be generalized for *approximate* moment-matching and for more general label spaces; see Section C.

## 1.2 Technical Overview

**General Algorithm.**  Intuitively, our plan is to first estimate the hidden subspace, $W$, and then to use a brute-force technique to learn a distribution that depends on $K$ dimensions. A straightforward approach to implement this plan is to use the method of moments. Since (in the noiseless case) $y$ depends only on the components of $\mathbf{x}$ within $W$, any non-vanishing moments must lie entirely within $W$ . Unfortunately, this approach can perform poorly—even for simple function classes, such as linear combinations of ReLUs. Specifically, [DKKZ20] showed that there exist linear combinations of $k$ ReLUs whose first $k$ moments vanish. This implies that any purely "moment-based" strategy would require at least $d^{\Omega(k)}$ sample and time complexity. The work [CKM22] improved on this (for Lipschitz and homogenous ReLU networks) by considering a more powerful test: examining moments of $\mathbf{x}$ conditioned on $y$ falling within a specified range (or, equivalently, analyzing moments of indicator functions applied to $y$). While this broadens the power of the algorithm, simply computing moments in one shot may still be insufficient to obtain near-optimal algorithms. In particular, [DIKZ25] presents a class of Boolean functions for which no constant number of moments suffices to learn the hidden subspace. However, a two-stage procedure—first using moments to identify a lower-dimensional subspace $V$, and then leveraging additional moments conditioned on the projection onto $V$—can successfully learn the full subspace.

This approach underlies our algorithm (see **LearnMIMs**). We employ an iterative approach that constructs progressively larger subspaces $V$. At each stage, we analyze the moments of $\mathbf{x}$ conditioned on $y$ lying within a small range and the projection of $\mathbf{x}$ onto $V$ falling within another localized region. If any of these conditional moments exhibits significant correlation with a particular direction (which we can detect using spectral methods), we augment $V$ by adding that direction. We repeat this process for several iterations, and then learn a function of the projection onto $V$ via brute-force search.

This method does not work for all functions, but is successful for functions that are suitably well-behaved. In particular, we require that at each stage, either at least one of the discovered directions correlates non-trivially with the hidden subspace $W$ (indicating progress), or that the current subspace $V$ already contains sufficient information to learn the target function to suitable error. In particular, we aim to ensure that for every function $f$ in our class (possibly with added noise) and every subspace $V$, either $f$ is well-approximated by some function of the projection onto $V$ (within the allowable error tolerance of our learner), or there exists a neighborhood $N \subseteq V$ and an interval $I \subseteq \mathbb{R}$ such that, conditioned on $\mathbf{x}_V \in N$ and $y \in I$, the distribution of $\mathbf{x}$ exhibits a non-trivial moment in some direction in $W_{V^\perp}$. To achieve this, we prove that a weaker condition actually suffices. This condition essentially states that either the function is close to a function of the projection

onto $V$, or that every noisy version of the function—with a small amount of additional additive noise—exhibits distinguishing moments (see Proposition 2.2). To make the algorithm work, we also need a few other minor technical assumption to ensure that it is sufficient to condition on small neighborhoods. For the full condition, see Definition 1.3.

**SQ Lower Bound.** While the aforementioned condition might not appear especially natural, we show that it is essentially *necessary*—in the sense that we establish a nearly-matching lower bound in the Statistical Query (SQ) model. In particular, if we have a rotationally-invariant function class containing some function $f$ that does not satisfy this condition—namely, for some subspace $V$, $f$ is neither close to a function of $\mathbf{x}_V$ nor is there some conditioning on $y$ and $\mathbf{x}_V$ that leads to non-trivial low degree moments—then we prove a lower bound for learning this function class to suitably small error in the SQ model. In particular, if we rotate this function $f$ and the joint distribution of $(\mathbf{x}, y)$ about $V$, we have a distribution that—once we condition on the value of $y$ and $\mathbf{x}_V$—we end-up with a random rotation of the distribution $A_{y,\mathbf{x}_V}$, where $A_{y,\mathbf{x}_V}$ is the distribution of $\mathbf{x}_{V^\perp}$ conditioned on $y$ and $\mathbf{x}_V$. Furthermore, $A_{y,\mathbf{x}_V}$ matches its first $m$ moments with the standard Gaussian projected onto the subspace $V^\perp$. This is an example of a *Relativized* Non-Gaussian Component Analysis (RNGCA) problem. Given the moment-matching property, one would expect the following: the SQ-complexity of distinguishing between this distribution and the one where $\mathbf{x}_{V^\perp}$ is independent of $\mathbf{x}_V$ and the $y$ is $d^{\Omega(m)}$. Since the latter distribution cannot be learned within any error better than the error of learning $y$ as a function of $\mathbf{x}_V$ (which by our assumption is large), this provides our learning SQ lower bound.

Unfortunately, while this kind of SQ lower bounds for Non-Gaussian Component Analysis (NGCA) are well-established [DKS17], the distributions $A_{y,\mathbf{x}_V}$ will likely not be continuous with respect to the standard Gaussian. In particular, they will not have finite chi-squared norm with respect to the standard Gaussian. This rules out the traditional SQ dimension-based arguments for proving the desired lower bounds. Recent work [DKRS23] showed that these kinds of SQ lower bounds can be proven with just moment-matching and no assumption on the Chi-squared norm. However, that work did not prove these bounds for RNGCA, i.e., could prove lower bounds for learning a single $A_{y,\mathbf{x}_V}$, but not the mixture over many of them (as we vary $y$ and $\mathbf{x}_V$). Fortunately, this can be fixed by generalizing the techniques of [DKRS23] to our more challenging context. Specifically, we show that an arbitrary bounded SQ query function $q$ is overwhelmingly likely to have expectation over the joint distribution of $(\mathbf{x}, y)$ very close to the averaged expectation over random rotations of this distribution described above. By mirroring the analysis of [DKRS23], we prove this by using Fourier analysis. We note that the low-degree Fourier coefficients of $A_{y,\mathbf{x}_V}$ vanish (or nearly vanish) and so contribute little to the expectation of $q$; and that the higher-degree Fourier coefficients are unlikely to correlate well with $q$ after the random rotation is applied.

**Concrete Applications.** Given our general algorithm, our applications hinge on establishing structural results for the relevant function classes. In particular, in order to obtain an algorithm for a function class $\mathcal{F}$, we need to show that it satisfies Definition 1.3 with suitably favorable parameters. Specifically, we need to establish that, unless a function in $\mathcal{F}$ is already close to depending only on the projection onto $V$, it exhibits non-trivial conditional low degree moments.

Our main application is to the class of positive-homogeneous Lipschitz functions— a broad, non-parametric generalization of the ReLU networks studied in [CKM22]. Here we show that second moments are sufficient. The basic idea is that if $f$ is not close to zero, then there exists some $\mathbf{x}$ for which $|f(\mathbf{x})|$ is reasonably large. This implies that $|f(\lambda\mathbf{x})|$ will be quite large for suitably large $\lambda$. On the other hand, by the Lipschitz property, $|f(\mathbf{x})|$ can only be large if $\|\mathbf{x}_W\|$ is large. Therefore, the set $S_t = \{\mathbf{x} : |f(\mathbf{x})| > \tau\}$ will exhibit a non-trivial second moment along $W$ for sufficiently large $\tau$. This argument yields at least one relevant direction. Moreover, given a subspace $V$, we can apply the same reasoning to the residual function $f(\mathbf{x}) - f(\mathbf{x}_V)$. This shows that either $f(\mathbf{x})$ is close to $f(\mathbf{x}_V)$ (i.e., a function of the projection onto $V$), or $f$ exhibits a non-vanishing conditional moment. Consequently, by approximating the Boolean function $\mathbb{1}(|f(\mathbf{x}) - f(\mathbf{x}_V)| \geq \tau)$ by a piecewise constant function over a partition consisting of cubes in $\mathbf{x}_V$ and intervals in $y$, we show that there exists a partition element for which the conditional distribution exhibits a non-trivial moment. This, in turn, implies that the function class is well-behaved, so our algorithm applies.

An additional application is for the class of polynomials that depend only on projections onto a low-dimensional subspace, recovering the upper bounds of [CM20]. See Section D.3.

## 1.3 Related Work

Due to space limitations, here we record the most directly relevant works. For a detailed overview, see Section A. Roughly speaking, our algorithmic understanding of learning SIMs is currently fairly complete, both for parameter recovery [DH18, AGJ21, DPLB24] and agnostic PAC learning [DKTZ22, WZDD24, ZWDD24, ZWDD25]. On the other hand, our understanding of the efficient learnability of MIMs is somewhat more limited. A number of papers have developed efficient learners for interesting special cases, including low-dimensional polynomials [CM20] and homogeneous ReLU networks [CKM22]. [ABAB+21, ABAM22, ABAM23] introduced a complexity notion (leap complexity) for learning structured MIMs, which turns out to essentially characterize the Correlational SQ (CSQ) complexity of learning under certain assumptions. More recently, [JMS24] adapted the notion of leap complexity to characterize the SQ hardness of hidden-junta functions (a natural special case of MIMs). The reader is referred to [BH25] for a very recent survey on the topic.

[DPLB24] defined the notion of the generative exponent, which plays the role of our parameter $m$ in characterizing the complexity of parameter recovery for SIMs. As explained in Appendix C.3, our Definition 1.3 reduces to a modification of the generative exponent when $K = 1$. Such a modification is necessary, to account for the fact that we characterize the complexity of PAC learning, rather than parameter estimation, even in the presence of adversarial label noise. Thus, our techniques can be viewed as a generalization of [DPLB24] to multi-index models.

**Comparison with [DIKZ25]** At the technical level, the most closely related work to ours is [DIKZ25], that established a discrete-analogue of our results in the context of classification for MIMs with finite output space. While our work broadly follows the approach of [DIKZ25], the transition from discrete-valued MIMs to those with infinitely many outputs, as well as the shift from $L_0$-loss to $L_2$-loss, requires significant changes in the mechanics of our results and the analysis.

In terms of our algorithm, perhaps the most significant change is that we can no longer condition on specific values of $y$—since we do not expect to observe repeated $y$ values. Instead, we need to condition on $y$ falling within a small interval. Additionally, since $y$ is now unbounded and we are working with the $L_2$ loss, establishing convergence results for our piecewise constant approximations becomes more challenging. Finally, [DIKZ25] used a technical condition on the Gaussian surface area of the level-sets to allow conditioning on small rectangles, and to guarantee that the learned directions are sufficiently distinct from those already identified. Here we need to design new conditions to deal with these issues. Regarding our SQ lower-bound analysis, conditioning on a given value of $y$ in this setting would likely yield a singular distribution. So establishing the desired bounds requires us to develop new machinery for proving lower bounds for relativized NGCA without having bounds on the chi-squared divergence. Another technical complication arises in our reduction from testing lower bounds to learning. In particular, we need to be able to approximate the $L_2$ loss within the SQ framework. While this is essentially trivial for the $L_0$ loss, here we need to add some technical conditions to make it feasible, as $y$ might be unbounded.

**Comparison with [DLB25]** We compare our contributions to the independent work of [DLB25]. Roughly speaking, both works study the task of learning MIMs and obtain efficient algorithms with qualitatively similar sample complexities with respect to the dimension. This similarity notwithstanding, the two works address fundamentally different objectives:

1. **PAC Learning vs. Parameter Recovery**: Our work focuses on the PAC learning task, i.e., the task of obtaining a hypothesis with low prediction error (without requiring to identify the underlying subspace). In contrast, [DLB25] focuses on parameter recovery, aiming to precisely estimate the underlying subspace. Note that PAC learning may be feasible in settings where parameter recovery is information-theoretically impossible. Specifically, it is possible to design very simple functions that admit accurate predictors even when the subspace is not identifiable. In Section C.3 we elaborate on this distinction in the context of the prior work [DPLB24] (which focused on single-index models).

2. **Agnostic Learning/Adversarial Label Noise**: Our work provides an algorithm with provable guarantees for both the independent label noise and the agnostic settings. In contrast, the work of [DLB25] focuses on the realizable setting.

Interestingly, when restricted to the realizable setting and when parameter-recovery is possible, one can show that our Definition 1.3 and the "generative-leap exponent" of [DLB25] are essentially equivalent. This can be shown by roughly the same analysis as presented in Section C.3 for the generative exponent (the single-index special case). In other words, our complexity measure from Definition 1.3 can be viewed as a generalization of the "generative leap" to the potentially noisy setting.

Finally, we note that the sample complexity dependence of the algorithm in [DLB25] is, roughly, $d^{\max\{1,m/2\}}$ (excluding factors that depend on $\epsilon$ and the underlying activation), is quantitatively optimal with respect to the dimension in the class of SQ algorithms (as follows from our SQ lower bound). Our work focused on obtaining a qualitative bound of the form $d^{O(m)}$, which suffices for the purpose of characterizing polynomial learnability of MIMs. Remark D.15 outlines a simple approach to obtain a sample bound of $d^{\lceil m/2 \rceil}$ in our setting.

## 2 General MIM Algorithm

As mentioned in Section 1.2, to apply the moment method effectively to such a general class of functions, we need to condition on $\mathbf{x}$ and $y$ falling within certain ranges. To achieve this, we partition the space of $\mathbf{x}$ and $y$ into sufficiently small regions—specifically, regular cubic regions for $\mathbf{x}$ and intervals for $y$. We prove that, as long as these partitions are fine enough, they can detect distinguishing moments. Formally:

**Definition 2.1** ($\epsilon$-Approximating Discretization). Let $V$ be a subspace of $\mathbb{R}^d$. We define an $\epsilon$-approximating discretization of $V \times \mathbb{R}$ as a pair $(\mathcal{S}, \mathcal{I})$ satisfying the following. The set $\mathcal{S}$ partitions the subset of $V$, consisting of all vectors whose coordinates in a fixed orthonormal basis of $V$ are less than $\sqrt{\log(1/\epsilon)}$ in absolute value, into cubes of side length $\epsilon$ (with respect to the same orthonormal basis). The set $\mathcal{I}$ partitions the interval $[-1/\epsilon, 1/\epsilon]$ into intervals of length $\epsilon$.

Moreover, for a partition $\mathcal{S}$, we denote by $h_{\mathcal{S}}$ the piecewise constant function that for every $S \in \mathcal{S}$ outputs $h_{\mathcal{S}}(\mathbf{x}) = \mathbf{E}[y \mid \mathbf{x} \in S]$ for all $\mathbf{x} \in S$.

As mentioned in Section 1.2, our algorithm, **LearnMIMs**, performs iterative subspace approximation. At each step $t$, it updates a list of vectors $L_t$ (Line 3c of **LearnMIMs**) so that the span $V_t = \mathrm{span}(L_t)$ becomes a better approximation of the hidden subspace $W$. Specifically, at each iteration, the algorithm computes a sufficiently fine discretization $(\mathcal{S}, \mathcal{I})$ of the space $V_t \times \mathbb{R}$ (Line 2 of **FindDirection**). Using the assumption that the distribution is well-behaved (Definition 1.3), we can show that a non-negligible fraction of the discretization cells exhibit distinguishing moments.

As a result, we extract relevant directions by computing the top eigenvectors of the influence matrix corresponding to a regression polynomial fitted within each cell (Line 3 of **FindDirection**). However, since the number of discretization cells depends exponentially on $\dim(V_t)$, we must apply a filtering step to avoid adding too many vectors. To this end, we construct a matrix $\mathbf{U}$, which is the weighted sum of influence matrices across all discretization cells, with weights given by the probability mass of each cell (Line 4 of **FindDirection**). It is not difficult to show that, since a constant fraction of the cells exhibit distinguishing moments, there exists an eigenvector of $\mathbf{U}$ with a sufficiently large eigenvalue that correlates with a distinguishing moment, thereby revealing a relevant direction. Once no further distinguishing moments can be found, since the target function satisfies Definition 1.3, the current subspace $V_t$ forms a good enough approximation of $W$. Finally, the algorithm returns a piecewise constant function $h_{\mathcal{S}}$, defined over a sufficiently fine partition $\mathcal{S}$ of $V_t$.

The main part of our analysis is to show that, at each iteration, as long as $V_t$ is not sufficient to compute a hypothesis with small error, the algorithm will add a direction that correlates with $W$. By applying this argument iteratively, we can show that improvement will eventually stop and we will have a good predictor.

**Proposition 2.2** (Estimating a Relevant Direction). *Let $D$ be distribution supported on $\mathbb{R}^d \times \mathbb{R}$ whose $\mathbf{x}$-marginal is $\mathcal{N}_d$. Let $f : \mathbb{R}^d \to \mathbb{R}$ be such that $f \in \mathcal{F}(K, m, \mathrm{OPT} + \epsilon, \tau, \sigma)$, and denote by $W$ a $K$-dimensional subspace defining $f$. Let $V$ be a $k$-dimensional subspace of $\mathbb{R}^d$ and let $\mathcal{S}$ be a partition of $V$ into cubes of width $(\epsilon/k)^{O(1)}$. If $\mathbf{E}_{(\mathbf{x},y) \sim D}[(h_{\mathcal{S}}(\mathbf{x}) - y)^2] > \tau + \mathrm{OPT} + \epsilon$, then* **FindDirection**, *when given $N = d^{O(m)}(k/\epsilon)^{O(k)}/\sigma^{O(1)}$ samples, runs in time $\mathrm{poly}(N)$, and with*

---

**LearnMIMs**: Robust Regression for Well-Behaved MIMs

**Input:** Accuracy $\epsilon > 0$, sample access to a distribution $D$ over $\mathbb{R}^d \times \mathbb{R}$ for which there exists a $K$-MIM function $f \in \mathcal{F}(K, m, \text{OPT} + \epsilon, \tau, \sigma)$, parameters $m, \sigma, K$.
**Output:** A hypothesis $h$ such that with high probability $\text{err}_D(h) \leq \tau + \text{OPT} + \epsilon$.

1. Let $T$ be a sufficiently large constant-degree polynomial in $m, K, 1/\sigma, 1/\epsilon$.

2. Initialize $L_1 \leftarrow \emptyset$, $N \leftarrow d^{O(m)} 2^T \log(1/\delta)$.

3. For $t = 1, \ldots, T$

    (a) Draw a set $S_t$ of $N$ i.i.d. samples from $D$.
    (b) $\mathcal{E}_t \leftarrow$ **FindDirection**$(\text{span}(L_t), S_t, \epsilon, \sigma, m, K)$.
    (c) $L_{t+1} \leftarrow L_t \cup \mathcal{E}_t$.

4. Construct an $\epsilon$-approximating discretization $(\mathcal{S}, \mathcal{I})$ of $\text{span}(L_t) \times \mathbb{R}$.

5. Draw $N$ i.i.d. samples from $D$ and empirically approximate the piecewise constant function $h_\mathcal{S}$.

6. Return $h_\mathcal{S}$.

---

**Algorithm 1:** Learning Well-Behaved MIMs

---

**FindDirection**: Estimating a relevant direction

**Input:** A subspace $V$ of $\mathbb{R}^d$, and a set of $N$ samples from a distribution $D$ over $\mathbb{R}^d \times \mathbb{R}$ for which there exists a $K$-MIM function $f \in \mathcal{F}(K, m, \text{OPT} + \epsilon, \tau, \sigma)$, parameters $\epsilon, \sigma, m, K$.
**Output:** A set of unit vectors $\mathcal{E}$.

1. Let $\lambda$ be a sufficiently small polynomial in $\sigma, \epsilon, 1/K$.

2. Construct an $\epsilon$-approximating discretization $(\mathcal{S}, \mathcal{I})$ of $V \times \mathbb{R}$.

3. For each $S \in \mathcal{S}$ and $I \in \mathcal{I}$, perform degree-$m$ polynomial regression on $\mathbb{1}(y \in I)$ over the samples, resulting in a polynomial $p_{S,I}(\mathbf{x}_{V^\perp})$.

4. Let $\mathbf{U} = \sum_{S \in \mathcal{S}, I \in \mathcal{I}} \mathbf{E}_{\mathbf{x} \sim D_\mathbf{x}}[\nabla p_{S,I}(\mathbf{x}_{V^\perp}) \nabla p_{S,I}(\mathbf{x}_{V^\perp})^\top \mid \mathbf{x} \in S] \mathbf{Pr}_{(\mathbf{x},y) \sim D}[S]$.

5. Return the set $\mathcal{E}$ of unit eigenvectors of $\mathbf{U}$ with corresponding eigenvalues at least $\lambda$.

---

**Algorithm 2:** Estimating a relevant direction

*high probability returns a list of unit vectors $\mathcal{E}$ of size $|\mathcal{E}| = (mK/(\epsilon\sigma))^{O(1)}$, such that for some $\mathbf{v} \in \mathcal{E}$, $\|\mathbf{v}_W\| = (\epsilon\sigma/(mK))^{O(1)}$.*

We sketch the analysis of our driving proposition bellow. Full details of the proof are provided in Section D.

*Proof Sketch of Proposition 2.2.* Let $\mathbf{w}^{(1)}, \ldots, \mathbf{w}^{(K)}$ be an orthonormal basis of $W$ and denote by $\mathbf{U}$ the matrix computed at Line 4. Let $\mathcal{I}$ a partition of $[-1/\epsilon^{O(1)}, 1/\epsilon^{O(1)}]$ to intervals of width $\epsilon^{O(1)}$.

Our strategy for proving the proposition essentially involves three steps: (i) show that Condition (2b) of Definition 1.3 is satisfied; (ii) prove that the discretization of $V \times \mathbb{R}$ into cube-interval pairs is sufficient to detect moments; and (iii) argue that, given the observed moments, there exists an eigenvector of $\mathbf{U}$ corresponding to a large eigenvalue that has a non-trivial projection onto $W$. We briefly discuss the proof of each of these steps.

Notice that, to establish the first step, it suffices to show that if $\mathbf{E}[(f(\mathbf{x}) - h_S(\mathbf{x}))^2] \geq \tau + \epsilon$, then $\mathbf{E}[(f(\mathbf{x}) - g(\mathbf{x}^V))^2] > \tau$ for all $g : V \to \mathbb{R}$. This follows from the assumption that $f$ is a function of bounded variation, i.e., $\mathbf{E}_{\mathbf{x} \sim \mathcal{N}_d}[\|\nabla f(\mathbf{x})\|^2]$ is bounded, and that $f$ is approximately bounded: any such function can be approximated arbitrarily well by piecewise-constant functions over a sufficiently fine partition of cubes covering all of $\mathbb{R}^d$ except for a set of small mass under $\mathcal{N}_d$. Hence, Condition (2a) is not satisfied, therefore Condition (2b) is (see Definition 1.3).

Step (ii) holds essentially because, by assumption, the distinguishing moment condition applies to all label random variables $y'$ that are $(\text{OPT}+\epsilon)$-close to $f$ in $L_2$. Specifically, we construct a label $y'$ that remains close to $f$ in two steps: first discretizing and then averaging $y$ over boxes. We discretize $y$ by rounding it to the nearest multiple of $\epsilon$, thereby partitioning the label distribution into intervals of width $\epsilon$. Then, since $f$ has bounded variation, for each cube $S \in \mathcal{S}$ the value $f(\mathbf{x})$ is close to the average label over that cube, so we can do the same for the label. By combining these two steps, we obtain a label random variable $y'$ that is discretized over small intervals, is conditionally independent of a specific point $\mathbf{x}$ given a cube $S \in \mathcal{S}$, and remains close to $f$. Using this independence yields the distinguishing-moment condition on the joint discretization of $\mathbf{x}_V$ and $y$. Moreover, since Condition (2b) ensures that distinguishing moments hold for a non-trivial fraction of $\mathbf{x}_V$, it follows that we observe these moments conditioned on a cube $S$ with probability at least $\alpha$ over $S$, for some $\alpha > 0$.

Step (iii) follows because the regression polynomial $p_{S,I}$ must match low-degree Hermite coefficients with the function $g(\mathbf{x}) \stackrel{\text{def}}{=} \mathbf{E}_{(\mathbf{x},y)\sim D}[\mathbb{1}(y \in I) \mid \mathbf{x} \in S]$, and hence must exhibit sufficient variation along directions where $g$ has a nontrivial low-degree moment, which in turn implies a nonzero directional derivative in these directions.

Recall that with probability $\alpha > 0$ over $S \in \mathcal{S}$ there exists some $i \in [K]$ such that $\mathbf{E}[(\mathbf{w}^{(i)} \cdot \nabla p_{S,I}(\mathbf{x}_{V^\perp}))^2] = \Omega(\sigma/K)$. This implies that, for some $i \in [K]$, with probability $\alpha/K$ over $S$ it holds that $\mathbf{E}[(\mathbf{w}^{(i)} \cdot \nabla p_{S,I}(\mathbf{x}_{V^\perp}))^2] = \Omega(\sigma/K)$. Therefore, the quadratic form of $\mathbf{U}$ for the corresponding $\mathbf{w}^{(i)}$ is large, i.e., $(\mathbf{w}^{(i)})^\top \mathbf{U} \mathbf{w}^{(i)} = \Omega(\sigma/K^2)$. Moreover, from well-known facts about polynomials over the standard Gaussian, we obtain that $\|\mathbf{U}\|_F \leq m/\text{poly}(\epsilon)$.

Finally, by a standard linear algebraic fact, if we consider the unit eigenvectors of $\mathbf{U}$ corresponding to eigenvalues greater than $O(\sigma/K^2)$, we obtain at most $(|\mathbf{U}|_F K/\sigma)^{O(1)}$ such vectors. Among them, at least one achieves correlation at least $(\sigma/(|\mathbf{U}|_F K))^{O(1)}$ with the aforementioned $\mathbf{w}^{(i)}$.

Furthermore, we note that the number of samples specified in the statement is precisely the number required to perform polynomial regression with enough accuracy to observe these low-degree moments with high probability. This completes the proof sketch of Proposition 2.2. $\qquad\square$

## 3 SQ Lower Bounds for MIMs

In order to prove our SQ lower bound for learning MIMs, we develop the framework of Relativized Non-Gaussian Component Analysis (RNGCA), a generalization of the previously developed Non-Gaussian Component Analysis (NGCA) framework [DKS17, DKRS23]—where we allow the hidden distribution to be a labeled distribution so that we can tackle the supervised MIM setting. The main technical contribution of this section (Theorem 3.3) is an SQ lower bound for RNGCA. Our SQ lower bounds for learning MIMs follow as an application of this general result. We believe that our generic SQ lower bound for RNGCA will be of broader applicability.

We start by defining the family of relativized hidden-subspace distributions, which is a core ingredient of the RNGCA framework.

We require some additional notation. We use $\mathbf{O}_{d,k} \subseteq \mathbb{R}^{d \times k}$ with $k \leq d$ to denote the set of all $d \times k$ orthogonal matrices, i.e., the set of all matrices $\mathbf{V}$ such that $\mathbf{V}^\top \mathbf{V} = \mathbf{I}_k$. For two distributions $D_1, D_2$ over $X_1, X_2$, we use $D_1 \otimes D_2$ to denote the product distribution of $D_1$ and $D_2$ over $X_1 \times X_2$.

**Definition 3.1** (Relativized Hidden-Subspace Distribution)**.** For a joint distribution $A$ of $(\mathbf{z}, \mathbf{y})$ supported on $\mathbb{R}^k \times \mathbb{R}^n$ and a matrix $\mathbf{U} \in \mathbf{O}_{d,k}$, we define the distribution $\mathbf{P}_{\mathbf{U}}^A$ as the joint distribution of $(\mathbf{z}', \mathbf{y}')$ supported on $\mathbb{R}^d \times \mathbb{R}^n$ such that

1. the joint distribution of $(\mathbf{U}^\top \mathbf{z}', \mathbf{y}')$ is $A$; and
2. $\mathbf{z}'_{U^\perp}$ is distributed according to $\mathcal{N}(\mathbf{0}, \Pi_{U^\perp})$ independent of the value of $(\mathbf{U}^\top \mathbf{z}', \mathbf{y}')$, where $U$ is the column space of $\mathbf{U}$ and $\mathcal{N}(\mathbf{0}, \Pi_{U^\perp})$ is the standard Gaussian projected onto $U^\perp$.

That is, up to a rotation on $\mathbb{R}^d$, $\mathbf{P}_{\mathbf{U}}^A$ is the distribution on $\mathbb{R}^{d-k} \times (\mathbb{R}^k \times \mathbb{R}^n)$ given by $\mathcal{N}_{d-k} \otimes A$.

We now define the natural hypothesis testing version of the RNGCA problem. This suffices for the purpose of proving hardness, as the learning version typically reduces to the testing problem. Intuitively, the task here is to test whether there is a subspace such that the marginal distribution on the subspace is not a standard Gaussian.

**Definition 3.2** (Hypothesis Testing Version of Relativized Non-Gaussian Component Analysis)**.** Let $d > k \geq 1$ be integers. For a joint distribution $A$ of $(\mathbf{x}, \mathbf{y})$ supported on $\mathbb{R}^k \times \mathbb{R}^n$, one is given access to a distribution $D$ such that either: $H_0$: $D = \mathcal{N}_d \otimes A_{\mathbf{y}}$, or $H_1$: $D$ is given by $\mathbf{P}_{\mathbf{U}}^A$, where $\mathbf{U} \sim U(\mathbf{O}_{d,k})$. The goal is to distinguish between these two hypotheses $H_0$ and $H_1$.

We are now ready to give our main SQ lower bound result for this problem. Intuitively, our lower bound states that if the distribution $A$ of $(\mathbf{z}, \mathbf{y})$ matches degree-$m$ moments relative to the subspace with the standard Gaussian, then any SQ algorithm solving the RNGCA testing problem requires complexity $d^{\Omega(m)}$. The reader is referred to Section C for the generalization of Theorem 3.3 with approximate moment matching and generalized label spaces.

**Theorem 3.3** (SQ Lower Bound for RNGCA)**.** *Let* $\lambda \in (0, 1)$ *and* $d, k, m \in \mathbb{N}$ *with* $m$ *even and* $k, m \leq d^\lambda / \log d$. *Let* $A$ *be a distribution over* $\mathbb{R}^k \times \mathbb{R}^n$ *that matches degree-$m$ moments relative to the subspace* $\mathbb{R}^n$ *(with the standard Gaussian on* $\mathbb{R}^k$*). Let* $0 < c < (1 - \lambda)/4$ *and* $d$ *be sufficiently large. Then any SQ algorithm solving the* $d$*-dimensional RNGCA problem with hidden distribution* $A$ *(as defined in Definition 3.2) with* $2/3$ *success probability requires either a query to* STAT $\left( O_{k,m} \left( d^{-((1-\lambda)/4 - c)m} \right) \right)$ *or* $2^{d^{\Omega(c)}}$ *many queries.*

It is worth noting that the non-relativized special case of Theorem 3.3 (i.e., when $n = 0$) was already proven in prior work [DKRS23]. It is important to note that Theorem 3.3 cannot be derived using [DKRS23] as a black-box. While a weaker version of Theorem 3.3 could be potentially obtained using techniques in previous works (see, e.g., [DKS17, DKS19]), this would necessarily require the additional assumption that $\chi^2(A, \mathcal{N}_k \otimes A_{\mathbf{y}})$ is finite. As a result, one would not be able to apply it to even the simplest settings like realizable MIMs—as having noiseless labels would induce infinite $\chi^2(A, \mathcal{N}_k \otimes A_{\mathbf{y}})$. Our proof here builds on the earlier proof in [DKRS23]. Namely, we apply a similar technique of truncating the $\mathbf{x}$ part of the distribution inside a ball, and then use Fourier analysis on the truncated $A$. However, doing so for the labeled distribution $A$ here would also mess up the marginal distribution $A_{\mathbf{y}}$ and change the notion of the norm in Fourier analysis. To deal with this problem, our analysis employs a new reweighting technique to ensure the equivalence of norm before and after the truncation. The detailed proof is given in Section C.1.

Given Theorem 3.3, we are now ready to prove Theorem 1.10 (see full proof in Section C.2).

*Proof sketch of Theorem 1.10.* The proof follows directly by embedding an RNGCA problem to agnostic PAC learning of the class $\mathcal{C}$. Let $A'$ be the distribution $D$ in Theorem 1.10 and $W$ be the $K$-dimensional relevant subspace of the $K$-MIM $c$ that minimizes the error $\text{err}_{A'}(c)$. Let $V$ be the subspace satisfying the conditions in Theorem 1.10 and $U = W_{V^\perp}$, where $W_{V^\perp} \overset{\text{def}}{=} \{\mathbf{w}_{V^\perp} : \mathbf{w} \in W\}$. Without loss of generality, we assume that $V$ is the subspace spanned by the last $\dim(V)$ coordinates, and $U$ is the subspace spanned by the $\dim(U)$ coordinates immediately preceding those of $V$, which can be arranged by an appropriate rotation.

Let $(\mathbf{x}, y) \sim A'$. We define the distribution $A$ for the RNGCA (Definition 3.2) as the joint distribution of $(\mathbf{x}', (\mathbf{x}'', y))$ over $\mathbb{R}^{\dim(U)} \times \mathbb{R}^{\dim(V)+1}$, where $\mathbf{x}'$ and $\mathbf{x}''$ each contains the coordinates of $\mathbf{x}$ corresponding to $U$ and $V$, i.e., $\mathbf{x}'$ contains the part of the relevant subspace (of the optimal hypothesis) outside $V$ and $(\mathbf{x}'', y)$ contains $\mathbf{V}$ and the label $y$. Let $D$ be the input distribution of this RNGCA problem. Notice that $D$ can be equivalently thought of as a labeled distribution supported on $\mathbb{R}^d \times \mathbb{R}$, where we treat the coordinate corresponding to the $y$ part as the label. If $D$ is the null hypothesis distribution, we would simply observe the production distribution of $\mathcal{N}(\mathbf{0}, \mathbf{I}_{d-\dim(V)}) \otimes A_{\mathbf{y}}$, where $A_{\mathbf{y}}$ is the marginal distribution of $(\mathbf{x}'', y)$. If we treat $D$ as a labeled distribution, then any hypothesis can only predict the label by the value of $\mathbf{x}_V$, therefore, no hypothesis $h : \mathbb{R}^d \to \mathbb{R}$ can have error $\text{err}_D(h) < \tau$ from the assumption. However, if $D$ is the alternative distribution, the distribution we observe is the product distribution of $\mathcal{N}(\mathbf{0}, \mathbf{I}_{d-\dim(U)-\dim(V)}) \otimes A$ (up to applying a rotation). If we treat $D$ as a labeled distribution, since $A$ contains the coordinates of $A'$ that span the relevant subspace $W$ of the optimal hypothesis, when given to the MIM algorithm, it is obliged to return a hypothesis with squared error substantially better than $\tau$.

Given the above discussion, we can simply give the distribution $D$ to the MIM algorithm as a labeled distribution over $\mathbb{R}^d \times \mathbb{R}$ and check the error of the output hypothesis. If the error is better than $\tau$, $D$ must be the alternative hypothesis distribution. Otherwise, $D$ is the null hypothesis distribution. This completes the proof sketch of Theorem 1.10. $\qquad\square$

## 4    Acknowledgments

Ilias Diakonikolas was supported by NSF Medium Award CCF-2107079, ONR award number N00014-25-1-2268, and an H.I. Romnes Faculty Fellowship. Giannis Iakovidis was supported in part by ONR award number N00014-25-1-2268 and NSF Award DMS-2023239 (TRIPODS). Daniel Kane was supported by NSF Award CCF-1553288 (CAREER) and a Sloan Research Fellowship. Lisheng Ren was supported in part by NSF Medium Award CCF-2107079.

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

# Appendix

**Organization** The appendix is structured as follows: In Section A, we discuss additional related work. In Section B, we record the notation and mathematical background required in our technical sections. The technical content of the appendix consists of two sections: Section C presents our SQ lower bounds and Section D presents our algorithmic results.

## A   Related Work

The most closely related works to ours is [DIKZ25] which studies the problem of learning discrete-valued MIMs. In Section 1.2, we highlight the technical and conceptual distinctions between our approach and that of [DIKZ25]. However, for learning real-valued MIMs, there has been no prior work establishing a characterization of the SQ complexity of the problem.

For the special case of SIMs, the problem is much better understood. Specifically, recent work [DPLB24] examined the complexity of parameter estimation for SIMs and identified a complexity measure that, under certain assumptions, characterizes the SQ sample complexity. As we demonstrate in Section C.3, our SQ lower bound strictly generalizes theirs, applying both to learning to small-$L_2$ error learning and parameter estimation whenever the MIM matches moments. Moreover, there has been a lot of algorithmic works for general classes of SIMs/GLMs from classical works like [KKKS11] to more recent works obtaining near optimal complexity and error guarantees [DGK+20, WZDD24, ZWDD24, ZWDD25, WZDD23].

Several works introduce CSQ complexity measures and algortihms for learning MIMs and SIMs—e.g. the information exponent for SIM link functions [AGJ21, DH18], the leap complexity [ABAB+21, ABAM22, ABAM23] for MIMs. However, all of these measures yield only CSQ guarantees, since they neither condition on the label $y$. Notably, [JMS24] further generalized the notion of leap complexity to characterize the SQ hardness of hidden-junta functions (which is a special case of MIMs).

Moreover, recently there is a significant interest in learning several structured subclasses of MIMs. Specifically [OSSW24] studied the problem of learning sums of SIMs under a near-orthonormality and [RL24] under a strict orthonormality assumption, providing both algorithms and lower bounds. Iterative dimensionality reduction techniques have been used in the past for learning certain functions families such as homogeneous ReLU networks [CKM22] and polynomials in a few relevant directions [CM20]. There has also been a lot of work [DDM+25, TDD+24, KZM25] on the problem of weak subspace recovery for MIMs using a linear number of samples within the approximate message passing framework.

Other works offer alternative guarantees, complexity under random bias [CMM25], gradient-flow convergence and time bounds [SBH24], mean-field Langevin dynamics yielding global convergence in infinite-width nets [MHWE25] and agnostic subspace-recovery learning with an oracle [MHJE24].

## B   Preliminaries

**Basic Notation** For $n \in \mathbb{Z}_+$, let $[n] \overset{\text{def}}{=} \{1, \ldots, n\}$. We will use lowercase boldface letters for vectors and capitalized boldface letters for matrices and tensors. For $\mathbf{x} \in \mathbb{R}^d$ and $i \in [d]$, $\mathbf{x}_i$ denotes the $i$-th coordinate of $\mathbf{x}$, and $\|\mathbf{x}\|_k := (\sum_{i=1}^d |\mathbf{x}_i|^k)^{1/k}$ denotes the $\ell_k$-norm of $\mathbf{x}$. Throughout this text, we will often omit the subscript and simply write $\|\mathbf{x}\|$ for the $\ell_2$-norm of $\mathbf{x}$. For a matrix $\mathbf{V} \in \mathbb{R}^{n \times m}$, we denote by $\|\mathbf{V}\|_2, \|\mathbf{V}\|_F$ to be the operator norm and Frobenius norm respectively. We will use $\mathbf{x} \cdot \mathbf{y}$ for the inner product of $\mathbf{x}, \mathbf{y} \in \mathbb{R}^d$.

For a subspace $V$ of $\mathbb{R}^d$, we denote by $V^\perp$ its orthogonal complement and by $\Pi_V$ its projection matrix. For vectors $\mathbf{x}, \mathbf{v} \in \mathbb{R}^d$ and a subspace $V \subseteq \mathbb{R}^d$ denote by $\mathbf{x}_V$ the projection of $\mathbf{x}$ onto $V$ and by $\mathbf{x}_{\mathbf{v}}$ the projection of $\mathbf{x}$ onto the line spanned by $\mathbf{v}$. For two subspaces $V, W \subseteq \mathbb{R}^d$, we denote by $W_V = \{\mathbf{w}_V : \mathbf{w} \in W\}$ and by $V + W = \{\mathbf{w} + \mathbf{v} : \mathbf{w} \in W, \mathbf{v} \in V\}$, note that $W_V$ and $V + W$ are both subspaces. Furthermore, for a set of vectors $L \subseteq \mathbb{R}^d$, we denote by $\text{span}(L)$ the subspace of $\mathbb{R}^d$ defined by their span. We slightly abuse notation and denote by $\mathbf{e}_i$ the $i$-th standard basis vector in $\mathbb{R}^d$. We use $\mathbb{S}^{n-1} = \{\mathbf{x} \in \mathbb{R}^n : \|\mathbf{x}\|_2 = 1\}$ to denote the $n$-dimensional unit sphere.

We use the standard asymptotic notation, where $\widetilde{O}(\cdot)$ is used to omit polylogarithmic factors. Furthermore, we use $a \lesssim b$ to denote that there exists an absolute universal constant $C > 0$ (independent of the variables or parameters on which $a$ and $b$ depend) such that $a \leq Cb$, $\gtrsim$ is defined similarly. We use the notation $g(t) \leq \mathrm{poly}(t)$ for a quantity $t \geq 1$ to indicate that there exists constants $c, C > 0$ such that $g(t) \leq Ct^c$. Similarly we use $g(t) \geq \mathrm{poly}(t)$ for a quantity $t < 1$ to denote that there exists constants $c, C > 0$ such that $g(t) \geq Ct^c$.

**Tensor Notation** For tensors, we will consider a $k$-tensor to be an element in $(\mathbb{R}^d)^{\otimes k} \cong \mathbb{R}^{d^k}$. This can be thought of as a vector with $d^k$ coordinates. We will use $\mathbf{A}_{i_1,\dots,i_k}$ to denote the coordinate of a $k$-tensor $\mathbf{A}$ indexed by the $k$-tuple $(i_1, \dots, i_k)$. By abuse of notation, we will sometimes also use this to denote the entire tensor. The inner product and $\ell^k$-norm of a $k$-tensor are defined by viewing the tensor as a vector with $d^k$ coordinates and then applying the standard definitions of the inner product and $\ell^k$-norm for vectors. The inner product of two tensors will be denoted by $\langle \cdot, \cdot \rangle$. For a vector $\mathbf{v} \in \mathbb{R}^d$, we denote by $\mathbf{v}^{\otimes k}$ to be a vector (linear object) in $\mathbb{R}^{d^k}$. In addition, for a matrix $\mathbf{V} \in \mathbb{R}^{d \times m}$, we denote by $\mathbf{V}^{\otimes k}$ to be a matrix (linear operator) mapping $\mathbb{R}^{m^k}$ to $\mathbb{R}^{d^k}$. Also, we define the set of orthogonal $d \times m$ matrices by $\mathbf{O}_{d,m} = \left\{ \mathbf{V} \in \mathbb{R}^{d \times m} \mid \mathbf{V}^\top \mathbf{V} = \mathbf{I}_m \right\}$.

**Probability Notation** We use $\mathbf{E}_{x \sim D}[x]$ for the expectation of the random variable $x$ according to the distribution $D$ and $\mathbf{Pr}[\mathcal{E}]$ for the probability of event $\mathcal{E}$. For simplicity of notation, we may omit the distribution when it is clear from the context. For a continuous distribution $D$ over $\mathbb{R}^d$, we sometimes use $D$ for both the distribution itself and its probability density function. For two distributions $D_1, D_2$ over a probability space $\Omega$, let $d_{\mathrm{TV}}(D_1, D_2) \stackrel{\text{def}}{=} \sup_{S \subseteq \Omega} |\mathbf{Pr}_{D_1}(S) - \mathbf{Pr}_{D_2}(S)|$ denote the total variation distance between $D_1$ and $D_2$. For two continuous distributions $D_1, D_2$ both over $\mathbb{R}^d$, we use $\chi^2(D_1, D_2) = \int_{\mathbb{R}^d} D_1(\mathbf{x})^2 / D_2(\mathbf{x}) d\mathbf{x} - 1$ to denote the chi-square norm of $D_1$ w.r.t. $D_2$.

For a distribution $D$ on a space $X$ and two measurable functions $f_1, f_2 : X \to \mathbb{R}^d$, we define their inner product w.r.t. $D$ as $\langle f_1, f_2 \rangle_D \stackrel{\text{def}}{=} \mathbb{E}_{\mathbf{x} \sim D}[\langle f_1(\mathbf{x}), f_2(\mathbf{x}) \rangle]$, and define the $L^2$ norm of a function $f$ w.r.t. $D$ as $\|f\|_D \stackrel{\text{def}}{=} \langle f, f \rangle_D^{1/2}$. For two distributions $D_1, D_2$ over $X_1, X_2$, we use $D_1 \otimes D_2$ to denote the product distribution over $X_1 \times X_2$.

For a subset $S \subseteq \mathbb{R}^d$ with finite measure or finite surface measure, we use $U(S)$ to denote the uniform distribution over $S$ (w.r.t. Lebesgue measure for the volume/surface area of $S$).

We use $\mathbb{1}$ to denote the indicator function of a set, specifically $\mathbb{1}(t \in S) = 1$ if $t \in S$ and $0$ otherwise. For a joint distribution $D$ of $(\mathbf{x}, y)$ over $\mathcal{X} \times \mathcal{Y}$, we use $D_\mathbf{x}$ and $D_y$ to denote the marginal distribution of $\mathbf{x}$ and $y$ and use $D_{\mathbf{x}|y=y'}$ to denote the conditional distribution of $\mathbf{x}$ given $y = y'$ (we will use the notation $D_{\mathbf{x}|y}$ as a shorthand when the variable $y$ is used in the context). Let $\mathcal{N}(\boldsymbol{\mu}, \boldsymbol{\Sigma})$ denote the $d$-dimensional Gaussian distribution with mean $\boldsymbol{\mu} \in \mathbb{R}^d$ and covariance $\boldsymbol{\Sigma} \in \mathbb{R}^{d \times d}$. For simplicity of notation, we use $\mathcal{N}_d$ for the $d$-dimensional standard normal $\mathcal{N}(\mathbf{0}, \mathbf{I})$.

**Basics of Hermite Polynomials** We require the following definitions.

**Definition B.1** (Normalized Hermite Polynomial). For $k \in \mathbb{N}$, we define the $k$-th *probabilist's* Hermite polynomials $He_k : \mathbb{R} \to \mathbb{R}$ as $He_k(t) = (-1)^k e^{t^2/2} \cdot \frac{d^k}{dt^k} e^{-t^2/2}$. We define the $k$-th *normalized* Hermite polynomial $h_k : \mathbb{R} \to \mathbb{R}$ as $h_k(t) = He_k(t)/\sqrt{k!}$.

Furthermore, we will use multivariate Hermite polynomials in the form of Hermite tensors (as the entries in the Hermite tensors are rescaled multivariate Hermite polynomials). We define the *Hermite tensor* as follows.

**Definition B.2** (Hermite Tensor). For $k \in \mathbb{N}$ and $\mathbf{x} \in \mathbb{R}^d$, we define the $k$-th Hermite tensor as

$$(\mathbf{H}_k(\mathbf{x}))_{i_1, i_2, \dots, i_k} = \frac{1}{\sqrt{k!}} \sum_{\substack{\text{Partitions } P \text{ of } [k] \\ \text{into sets of size 1 and 2}}} \bigotimes_{\{a,b\} \in P} (-\mathbf{I}_{i_a, i_b}) \bigotimes_{\{c\} \in P} \mathbf{x}_{i_c} .$$

For a function $f : \mathbb{R}^d \to \mathbb{R}$ and $\ell \in \mathbb{N}$, we use $f^{\leq \ell}$ to denote $f^{\leq \ell}(\mathbf{x}) = \sum_{k=0}^\ell \langle \mathbf{A}_k, \mathbf{H}_k(\mathbf{x}) \rangle$, where $\mathbf{A}_k = \mathbf{E}_{\mathbf{x} \sim \mathcal{N}_d}[f(\mathbf{x})\mathbf{H}_k(\mathbf{x})]$, which is the degree-$\ell$ approximation of $f$. We use $f^{>\ell} = f - f^{\leq \ell}$ to denote its residue. We also remark that both our definition of Hermite polynomial and Hermite tensor are "normalized" in the following sense: For Hermite polynomials, it holds $\|h_k\|_2 = 1$. For Hermite tensors, given any symmetric tensor $A$, we have $\|\langle \mathbf{A}, \mathbf{H}_k(\mathbf{x}) \rangle\|_2^2 = \langle \mathbf{A}, \mathbf{A} \rangle$.

# C   Statistical Query Lower Bounds

In this section, we establish our SQ lower bounds for learning Multi-Index models, thereby proving Theorem 1.10.

**Organization.**   The structure of this section is as follows: In Section C.1, we define a relativized version of Non-Gaussian Component Analysis that is appropriate for supervised learning tasks and establish an optimal SQ lower bound for it under appropriate conditions. In Section C.2, we leverage this general result to show our SQ lower bounds for learning MIMs, for both the realizable and the agnostic settings. Finally, in Section C.3, we relate the conditions of our SQ lower bounds for learning MIMs with prior complexity measures in the literature.

## C.1   Statistical Query Lower Bounds for Relativized NGCA

In this section, we prove an SQ lower bound for Relativized Non-Gaussian Component Analysis (RNGCA). The main result of this section is a generalization of Theorem 3.3, handling more general label spaces and approximate moment matching. We leverage this technical result in the following subsection to prove our main SQ lower bounds for Multi-Index Models.

To be compatible with more general label spaces, we start with the following definitions generalizing the relativized hidden-subspace distribution of Definition 3.1, and the hypothesis testing version of RNGCA of Definition 3.2. The main difference here is that we replace the space $\mathbb{R}^n$, appearing in Definition 3.1 and Definition 3.2, with a general space $\mathcal{Y}$.

**Definition C.1** (Relativized Hidden-Subspace Distribution; Generalization of Definition 3.1)**.** For a joint distribution $A$ of $(\mathbf{z}, y)$ supported on $\mathbb{R}^k \times \mathcal{Y}$ and a matrix $\mathbf{U} \in \mathbf{O}_{d,k}$, we define the distribution $\mathbf{P}_{\mathbf{U}}^A$ as the joint distribution of $(\mathbf{z}', y')$ supported on $\mathbb{R}^d \times \mathcal{Y}$ such that

1. the joint distribution of $(\mathbf{U}^\top \mathbf{z}', y')$ is $A$; and
2. $\mathbf{z}'_{U^\perp}$ is distributed according to $\mathcal{N}(\mathbf{0}, \Pi_{U^\perp})$ independent of the value of $(\mathbf{U}^\top \mathbf{z}', y')$, where $U$ is the column space of $\mathbf{U}$ and $\mathcal{N}(\mathbf{0}, \Pi_{U^\perp})$ is the standard Gaussian projected onto $U^\perp$.

We next give the generalization of Definition 3.2.

**Definition C.2** (Hypothesis Testing Version of RNGCA; Generalization of Definition 3.2 )**.** Let $d > k \geq 1$ be integers. For a joint distribution $A$ of $(\mathbf{x}, y)$ supported on $\mathbb{R}^k \times \mathcal{Y}$, one is given access to a distribution $D$ such that either: $H_0$: $D = \mathcal{N}_d \otimes A_y$, or $H_1$: $D$ is given by $\mathbf{P}_{\mathbf{U}}^A$, where $\mathbf{U} \sim U(\mathbf{O}_{d,k})$. The goal is to distinguish between these two hypotheses $H_0$ and $H_1$.

For the hidden distribution $A$ in the definition of RNGCA, the lower bound construction here requires that the conditional distribution of $A_{\mathbf{x}|y}$ is well-defined for every $y$. In order to ensure that this conditional distribution is well-defined, we first introduce the following technical condition.

**Definition C.3** (Regular Distribution)**.** Let $A$ be a joint distribution of $(\mathbf{x}, y)$ supported on $\mathbb{R}^k \times \mathcal{Y}$. We say that $A$ is *regular* if there is a family of distributions $A_{\mathbf{x}|y}$ on $\mathbb{R}^k$ for each $y \in \mathcal{Y}$ such that for any measurable set $S$ of $A$, $\mathbf{Pr}_{(\mathbf{x},y)\sim A}[(\mathbf{x}, y) \in S] = \int_{A_y} \mathbf{Pr}_{\mathbf{x}\sim A_{\mathbf{x}|y}}[(\mathbf{x}, y) \in S]dy$ . We will call such distributions $A_{\mathbf{x}|y}$ the conditional distributions of $\mathbf{x}$ given $y$.

**Remark C.4.** Note that $A$ is always regular if $\mathcal{Y} = \mathbb{R}^n$, which is a Polish space.

Our SQ lower bound construction crucially relies on the assumption that the conditional distributions $A_{\mathbf{x}|y}$ approximately match their low-degree moments with the standard Gaussian, i.e., that $A$ has similar low-degree moments with $\mathcal{N}_k \otimes A_y$. Roughly speaking, for each conditional distribution $A_{\mathbf{x}|y}$, we characterize the mismatch between $A_{\mathbf{x}|y}$ and the standard Gaussian as $\sup_p \left( \mathbf{E}_{\mathbf{x}\sim A_{\mathbf{x}|y}}[p(\mathbf{x})] - \mathbf{E}_{\mathbf{x}\sim \mathcal{N}_k}[p(\mathbf{x})] \right)$, where $p$ is any low-degree polynomial with $\mathbf{E}_{\mathbf{x}\sim \mathcal{N}_k}[p(\mathbf{x})^2] \leq 1$. Then we take the $L^2$ norm of this quantity over the marginal distribution $A_y$ as the overall mismatch between $A$ and $\mathcal{N}_k \otimes A_y$, as described in the following definition (generalizing the exact moment-matching in Definition 1.9).

**Condition C.5** (Relatively $\nu$-Matching Degree-$m$ Moments; Generalization of Definition 1.9)**.** *Let* $0 < \nu < 2$, $m \in \mathbb{N}$, *and* $A$ *be a regular distribution of* $(\mathbf{x}, y)$ *supported on* $\mathbb{R}^k \times \mathcal{Y}$. *We say that* $A$ *$\nu$-matches degree-$m$ moments with the standard Gaussian relative to* $\mathcal{Y}$ *if for any* $f : \mathbb{R}^k \times \mathcal{Y} \to \mathbb{R}$ *such that*

1. *the function $f(\cdot, y)$ is a polynomial of degree at most $m$ for any $y \in \mathcal{Y}$; and*

2. *$\|f\|_{\mathcal{N}_k \otimes A_y} \leq 1$, where $A_y$ is the $y$-marginal of $A$ and $\mathcal{N}_k \otimes A_y$ is the product distribution of $\mathcal{N}_k$ and $A_y$,*

*it holds that $\left| \mathbf{E}_{(\mathbf{x},y) \sim A}[f(\mathbf{x},y)] - \mathbf{E}_{(\mathbf{x},y) \sim \mathcal{N}_k \otimes A_y}[f(\mathbf{x},y)] \right| \leq \nu$ .*

With this context, we are ready to state our main SQ lower bound theorem for RNGCA. Roughly speaking, we show that for any regular distribution $A$ that satisfies Condition C.5, there is an SQ lower bound for RNGCA using $A$ as the hidden distribution.

**Theorem C.6** (SQ Lower Bound for RNGCA; Generalization of Theorem 3.3)**.** *Let $\lambda \in (0,1)$ and $d, k, m \in \mathbb{N}$ with $m$ even and $k, m \leq d^\lambda$. Let $0 < \nu < 2$ and $A$ be a regular distribution over $\mathbb{R}^k \times \mathcal{Y}$ such that $A$ $\nu$-matches degree-$m$ moments with the standard Gaussian relative to $\mathcal{Y}$. Let $0 < c < (1-\lambda)/4$ and $d$ be at least a sufficiently large constant depending on $c$. Then any SQ algorithm solving the $d$-dimensional RNGCA problem with hidden distribution $A$, as defined in Definition C.2, with $2/3$ success probability requires either a query to $\mathrm{STAT}(\tau)$, where $\tau < O_{k,m}\left(d^{-((1-\lambda)/4-c)m}\right) + (1 + o(1))\nu$, or $2^{d^{\Omega(c)}}$ many queries.*

To prove the desired lower bound, we need to show that for any query function $f$ the algorithm selects, over the choice of the hidden subspace $\mathbf{U} \sim U(\mathbf{O}_{d,k})$, the expectation $\mathbf{E}_{(\mathbf{x},y) \sim \mathbf{P}_{\mathbf{U}}^A}[f(\mathbf{x},y)]$ is concentrated around $\mathbf{E}_{(\mathbf{x},y) \sim \mathcal{N}_d \otimes A_y}[f(\mathbf{x},y)]$. Therefore, the algorithm cannot tell if the distribution is the alternative hypothesis distribution $\mathbf{P}_{\mathbf{U}}^A$ or the null hypothesis distribution $\mathcal{N}_d \otimes A_y$.

Such a concentration result is given in the following proposition.

**Proposition C.7.** *Let $\lambda \in (0,1)$ and $d, k, m \in \mathbb{N}$ with $m$ even and $k, m \leq d^\lambda$. Let $0 < \nu < 2$ and $A$ be a regular distribution over $\mathbb{R}^k \times \mathcal{Y}$ such that $A$ $\nu$-matches degree-$m$ moments with the standard Gaussian relative to $\mathcal{Y}$. Let $0 < c < (1-\lambda)/4$, $d$ be at least a sufficiently large constant depending on $c$, and $f : \mathbb{R}^d \times \mathcal{Y} \to [0,1]$. Then it holds that*

$$\mathbf{Pr}_{\mathbf{U} \sim U(\mathbf{O}_{d,k})}\left[ \left| \mathbf{E}_{(\mathbf{x},y) \sim \mathbf{P}_{\mathbf{U}}^A}[f(\mathbf{x},y)] - \mathbf{E}_{(\mathbf{x},y) \sim \mathcal{N}_d \otimes A_y}[f(\mathbf{x},y)] \right| \geq \tau \right] \leq 2^{-d^{\Omega(c)}} ,$$

*where*

$$\tau = \left( \frac{\Gamma(m/2 + k/2)}{\Gamma(k/2)} \right) d^{-((1-\lambda)/4-c)m} + (1 + o(1))\nu .$$

Given Proposition C.7, the proof of Theorem C.6 is straightforward. We just need to show that for all the queries the algorithm makes, with high probability, the expected values $\mathbf{E}_{(\mathbf{x},y) \sim \mathbf{P}_{\mathbf{U}}^A}[f(\mathbf{x},y)]$ and $\mathbf{E}_{(\mathbf{x},y) \sim \mathcal{N}_d \otimes A_y}[f(\mathbf{x},y)]$ are always close to each other for any query function $f$. Therefore, the SQ oracle can always answer the queries with $\mathbf{E}_{(\mathbf{x},y) \sim \mathcal{N}_d \otimes A_y}[f(\mathbf{x},y)]$ and the algorithm cannot differentiate between the alternative and null hypotheses.

*Proof of Theorem C.6.* Suppose there is an SQ algorithm $\mathcal{A}$ using $q < 2^{d^{\Omega(c)}}$ many queries of accuracy $\tau \geq \frac{\Gamma(m/2+k/2)}{\Gamma(k/2)} d^{-((1-\lambda)/4-c)m} + (1 + o(1))\nu$ and succeeds with at least $2/3$ probability.

We prove by contradiction that such an $\mathcal{A}$ cannot exist. Suppose that the input distribution is $\mathcal{N}_d \otimes A_y$, and the SQ oracle always answers $\mathbf{E}_{(\mathbf{x},y) \sim \mathcal{N}_d \otimes A_y}[f(\mathbf{x},y)]$ for any query $f$. Then the assumption on $\mathcal{A}$ implies that it answers "null hypothesis" with probability $\alpha > 2/3$. Now consider the case that the input distribution is $\mathbf{P}_{\mathbf{U}}^A$ and $\mathbf{U} \sim U(\mathbf{O}_{d,k})$. Suppose the SQ oracle still always answers $\mathbf{E}_{(\mathbf{x},y) \sim \mathcal{N}_d \otimes A_y}[f(\mathbf{x},y)]$ whenever possible. Let $f_1, \ldots, f_q$ be the queries the algorithm makes, where $q = 2^{d^{O(c)}}$ for a sufficiently small implied constant in the big-$O$. By Proposition C.7 and a union bound, we have

$$\mathbf{Pr}_{\mathbf{U} \sim U(\mathbf{O}_{d,k})}[\exists i \in [q], |\mathbf{E}_{(\mathbf{x},y) \sim \mathbf{P}_{\mathbf{U}}^A}[f_i(\mathbf{x},y)] - \mathbf{E}_{(\mathbf{x},y) \sim \mathcal{N}_d \otimes A_y}[f_i(\mathbf{x},y)]| \geq \tau] = o(1) .$$

Therefore, with probability $1 - o(1)$, the oracle will be able to always answer $\mathbf{E}[f_i(\mathcal{N}_d \otimes \mathcal{A}_y)]$. From our assumption on $\mathcal{A}$, the algorithm needs to answer the "alternative hypothesis" with probability at least $\frac{2}{3}(1 - o(1))$.

But since the oracle always answers $\mathbf{E}_{(\mathbf{x},y) \sim \mathcal{N}_d \otimes A_y}[f_i(\mathbf{x},y)]$ (which is the same as in the above discussed null hypothesis case), we know that the algorithm will return "null hypothesis" with probability $\alpha > 2/3$. This gives a contradiction and completes the proof of Theorem C.6. $\square$

The rest of the section is devoted to proving Proposition C.7. In the next subsection, we will first show that in order to prove Proposition C.7, it suffices for us to apply Fourier analysis on the distribution $A'$, a modification of $A$ (for convenience of the analysis) that has bounded total variation distance with $A$. The approach here shares similarities with [DKRS23]). Then, in Section C.1.2, we put everything together and establish Proposition C.7.

### C.1.1 Fourier Analysis using Hermite Polynomials

We will first try to use Fourier Analysis to analyze the value of $\mathbf{E}_{(\mathbf{x},y)\sim\mathbf{P}_{\mathbf{U}}^A}[f(\mathbf{x},y)]$. We can calculate $\mathbf{E}_{\mathbf{x}\sim\mathbf{P}_{\mathbf{U}}^A}[f(\mathbf{x},y)]$ by its Hermite decomposition as stated in the following lemma.

**Lemma C.8** (Fourier Decomposition Lemma). *Let $A$ be a regular joint distribution $(\mathbf{x},y)$ supported on $\mathbb{R}^k \times \mathcal{Y}$, $\mathbf{U} \in \mathbb{R}^{d\times k}$ and $\mathbf{U}^\top\mathbf{U} = \mathbf{I}_k$. Then for any $f : \mathbb{R}^d \times \mathcal{Y} \to \mathbb{R}$ and $\ell \in \mathbb{N}$,*

$$\mathbf{E}_{(\mathbf{x},y)\sim\mathbf{P}_{\mathbf{U}}^A}[f(\mathbf{x},y)] = \sum_{i=0}^{\ell}\langle\mathbf{U}^{\otimes i}\mathbf{A}_i(y),\mathbf{T}_i(y)\rangle_{A_y} + \mathbf{E}_{(\mathbf{x},y)\sim\mathbf{P}_{\mathbf{U}}^A}\left[f^{>\ell}(\mathbf{x},y)\right] \ ,$$

*where $\mathbf{A}_i(y) = \mathbf{E}_{\mathbf{x}\sim A_{\mathbf{x}|y}}[\mathbf{H}_i(\mathbf{x})]$ and $\mathbf{T}_i(y) = \mathbf{E}_{\mathbf{x}\sim\mathcal{N}_d}[f(\mathbf{x},y)\mathbf{H}_i(\mathbf{x})]$ and $f^{>\ell}(\mathbf{x},y) = (f(\cdot,y))^{>\ell}(\mathbf{x})$.*

*Proof of Lemma C.8.* The proof of the theorem directly follows by applying the law of total expectation on Lemma 3.3 from [DKRS23]. We first state Lemma 3.3 from [DKRS23] below.

**Fact C.9** (Lemma 3.3 of [DKRS23]). *Let $A$ be any distribution supported on $\mathbb{R}^k$, $\mathbf{U} \in \mathbb{R}^{d\times k}$ and $\mathbf{U}^\top\mathbf{U} = \mathbf{I}_k$. Then for any $\ell \in \mathbb{N}$,*

$$\mathbf{E}_{\mathbf{x}\sim\mathbf{P}_{\mathbf{U}}^A}[f(\mathbf{x})] = \sum_{i=0}^{\ell}\langle\mathbf{U}^{\otimes i}\mathbf{E}_{\mathbf{x}\sim A}[\mathbf{H}_i(\mathbf{x})],\mathbf{E}_{\mathbf{x}\sim\mathcal{N}_d}[f(\mathbf{x})\mathbf{H}_i(\mathbf{x})]\rangle + \mathbf{E}_{(\mathbf{x},y)\sim\mathbf{P}_{\mathbf{U}}^A}\left[f^{>\ell}(\mathbf{x})\right] \ .$$

Applying the law of total expectation, we get

$$\mathbf{E}_{(\mathbf{x},y)\sim\mathbf{P}_{\mathbf{U}}^A}[f(\mathbf{x},y)]$$

$$=\mathbf{E}_{y\sim A_y}\left[\mathbf{E}_{\mathbf{x}\sim\left(\mathbf{P}_{\mathbf{U}}^A\right)_{\mathbf{x}|y}}[f(\mathbf{x},y)]\right]$$

$$=\mathbf{E}_{y\sim A_y}\left[\mathbf{E}_{\mathbf{x}\sim\mathbf{P}_{\mathbf{U}}^{\left(A_{\mathbf{x}|y}\right)}}[f(\mathbf{x},y)]\right]$$

$$=\mathbf{E}_{y\sim A_y}\left[\sum_{i=0}^{\ell}\langle\mathbf{U}^{\otimes i}\mathbf{E}_{\mathbf{x}\sim A_{\mathbf{x}|y}}[\mathbf{H}_i(\mathbf{x})],\mathbf{E}_{\mathbf{x}\sim\mathcal{N}_d}[f(\mathbf{x},y)\mathbf{H}_i(\mathbf{x})]\rangle + \mathbf{E}_{(\mathbf{x},y)\sim\mathbf{P}_{\mathbf{U}}^{\left(A_{\mathbf{x}|y}\right)}}\left[(f(\cdot,y))^{>\ell}(\mathbf{x})\right]\right]$$

$$=\mathbf{E}_{y\sim A_y}\left[\sum_{i=0}^{\ell}\langle\mathbf{U}^{\otimes i}\mathbf{A}_i(y),\mathbf{T}_i(y)\rangle + \mathbf{E}_{(\mathbf{x},y)\sim\mathbf{P}_{\mathbf{U}}^{\left(A_{\mathbf{x}|y}\right)}}\left[(f(\cdot,y))^{>\ell}(\mathbf{x})\right]\right]$$

$$=\sum_{i=0}^{\ell}\langle\mathbf{U}^{\otimes i}\mathbf{A}_i,\mathbf{T}_i\rangle_{A_y} + \mathbf{E}_{(\mathbf{x},y)\sim\mathbf{P}_{\mathbf{U}}^A}\left[f^{>\ell}(\mathbf{x},y)\right] \ .$$

This completes the proof of Lemma C.8. $\qquad\square$

We note that, ideally, we would like to have

$$\mathbf{E}_{(\mathbf{x},y)\sim\mathbf{P}_{\mathbf{U}}^A}[f(\mathbf{x},y)] = \sum_{i=0}^{\infty}\langle\mathbf{U}^{\otimes i}\mathbf{A}_i,\mathbf{T}_i\rangle_{A_y} \ . \tag{1}$$

However, Equation (1) is not true in general for technical reasons. Namely, it is possible that $\chi^2(A,\mathcal{N}_k \otimes A_y)$ is infinite (this is true even assuming $A_{\mathbf{x}} = \mathcal{N}_k$); therefore, the convergence in Equation (1) may not hold (see Remark 3.4 of [DKRS23] for a more detailed discussion). Instead,

we will show that for a sufficiently large $l$, we can have $\|f^{\geq l}\|_{\mathcal{N}_k \otimes A_y}$ be arbitrarily close to 0. Combining this with some other technical facts will suffice to obtain that $\mathbf{E}_{(\mathbf{x},y) \sim \mathbf{P}_{\mathbf{U}}^A}[f^{>\ell}(\mathbf{x},y)]$ is also arbitrarily close to 0. Now notice that

$$\langle \mathbf{U}^{\otimes 0} \mathbf{A}_0, \mathbf{T}_0 \rangle_{A_y} = \mathbf{E}_{y \sim A_y}[\mathbf{E}_{\mathbf{x} \sim \mathcal{N}_k}[f(\mathbf{x},y)]] = \mathbf{E}_{(\mathbf{x},y) \sim \mathcal{N}_d \otimes A_y}[f(\mathbf{x},y)] .$$

Therefore, the quantity we want to bound is just

$$\mathbf{E}_{(\mathbf{x},y) \sim \mathbf{P}_{\mathbf{U}}^A}[f(\mathbf{x},y)] - \mathbf{E}_{(\mathbf{x},y) \sim \mathcal{N}_d \otimes A_y}[f(\mathbf{x},y)] = \sum_{i=1}^{\ell} \langle \mathbf{U}^{\otimes i} \mathbf{A}_i, \mathbf{T}_i \rangle_{A_y} + \mathbf{E}_{(\mathbf{x},y) \sim \mathbf{P}_{\mathbf{U}}^A}\left[f^{>\ell}(\mathbf{x},y)\right] .$$

As we have mentioned above, for sufficiently large $l$, the second term $\mathbf{E}_{(\mathbf{x},y) \sim \mathbf{P}_{\mathbf{U}}^A}\left[f^{>\ell}(\mathbf{x},y)\right]$ will be arbitrarily close to 0. Therefore, we just need to bound the first term $\sum_{i=1}^{\ell} \langle \mathbf{U}^{\otimes i} \mathbf{A}_i, \mathbf{T}_i \rangle_{A_y}$.

To bound $\sum_{i=1}^{\ell} \langle \mathbf{U}^{\otimes i} \mathbf{A}_i, \mathbf{T}_i \rangle_{A_y}$, notice that

$$\sum_{i=1}^{\ell} \langle \mathbf{U}^{\otimes i} \mathbf{A}_i, \mathbf{T}_i \rangle_{A_y} \leq \sum_{i=1}^{\ell} |\langle \mathbf{A}_i, (\mathbf{U}^{\top})^{\otimes i} \mathbf{T}_i \rangle_{A_y}| \leq \sum_{i=1}^{\ell} \|\mathbf{A}_i\|_{A_y} \|(\mathbf{U}^{\top})^{\otimes i} \mathbf{T}_i\|_{A_y} .$$

To proceed, we just need to bound the terms $\|\mathbf{A}_i\|_{A_y}$ and $\|(\mathbf{U}^{\top})^{\otimes i} \mathbf{T}_i\|_{A_y}$ for all $i$. We first establish the following fact, which can be derived from Lemma 3.7, Lemma 3.8 and Corollary 3.9 of [DKRS23]. This fact bounds from above the $a$th moment $\|(\mathbf{U}^{\top})^{\otimes i} \mathbf{T}_i\|_{A_y}^a$ and implies that $\|(\mathbf{U}^{\top})^{\otimes i} \mathbf{T}_i\|_{A_y}$ is $o(1)$ with high probability.

**Fact C.10.** *Let $i, k, d \in \mathbb{Z}_+$ with $k < d$, $a \in \mathbb{Z}_+$ be even and $i' = ai/2$. Let $D$ be a distribution over $\mathcal{Y}$ and $\mathbf{T} : \mathcal{Y} \to \mathbb{R}^{d^{\otimes i}}$. Then*

$$\mathbf{E}_{\mathbf{U} \sim U(\mathbf{O}_{d,k})}\left[\|(\mathbf{U}^{\top})^{\otimes i}\mathbf{T}(y)\|_{y \sim D}^a\right] = O\left(\frac{\Gamma\left(\frac{i'+k}{2}\right)\Gamma\left(\frac{d}{2}\right)}{\Gamma\left(\frac{i'+d}{2}\right)\Gamma\left(\frac{k}{2}\right)}\right) \|\mathbf{T}(y)\|_{y \sim D}^a .$$

*Furthermore,*

$$\mathbf{E}_{\mathbf{U} \sim U(\mathbf{O}_{d,k})}\left[\|(\mathbf{U}^{\top})^{\otimes i}\mathbf{T}(y)\|_{y \sim D}^a\right] = O(2^{i'/2}(d/\max(k,i'))^{-i'/2})\|\mathbf{T}(y)\|_{y \sim D}^a .$$

*In addition, if there exists some constant $c \in (0,1)$ such that $k \leq d^c < i'$, then*

$$\mathbf{E}_{\mathbf{U} \sim U(\mathbf{O}_{d,k})}\left[\|(\mathbf{U}^{\top})^{\otimes i}\mathbf{T}(y)\|_{y \sim D}^a\right] = \exp(-\Omega(d^c \log d)) O\left(\left(\frac{d^c+d}{i'+d}\right)^{(d-k)/2}\right) \|\mathbf{T}(y)\|_{y \sim D}^a .$$

*Proof of Fact C.10.* Notice that

$$\mathbf{E}_{\mathbf{U}\sim U(\mathbf{O}_{d,k})}[\|(\mathbf{U}^\top)^{\otimes i}\mathbf{T}(y)\|_{y\sim D}^a]$$

$$=\mathbf{E}_{\mathbf{U}\sim U(\mathbf{O}_{d,k})}\left[\mathbf{E}_{y\sim D}\left[\|(\mathbf{U}^\top)^{\otimes i}\mathbf{T}(y)\|_2^2\right]^{a/2}\right]$$

$$=\mathbf{E}_{\mathbf{U}\sim U(\mathbf{O}_{d,k})}\left[\mathbf{E}_{y_1,\cdots,y_{a/2}\sim D^{\otimes a/2}}\left[\prod_{j=1}^{a/2}\|(\mathbf{U}^\top)^{\otimes i}\mathbf{T}(y_j)\|_2^2\right]\right]$$

$$=\mathbf{E}_{\mathbf{U}\sim U(\mathbf{O}_{d,k})}\left[\mathbf{E}_{y_1,\cdots,y_{a/2}\sim D^{\otimes a/2}}\left[\left\|(\mathbf{U}^\top)^{\otimes ai/2}\bigotimes_{j=1}^{a/2}\mathbf{T}(y_j)\right\|_2^2\right]\right]$$

$$=\mathbf{E}_{\mathbf{U}\sim U(\mathbf{O}_{d,k})}\left[\mathbf{E}_{y_1,\cdots,y_{a/2}\sim D^{\otimes a/2}}\left[\left\langle\mathbf{U}^{\otimes ai/2}(\mathbf{U}^\top)^{\otimes ai/2},\left(\bigotimes_{j=1}^{a/2}\mathbf{T}(y_j)\right)^{\otimes 2}\right\rangle\right]\right]$$

$$\leq\left\|\mathbf{E}_{\mathbf{U}\sim U(\mathbf{O}_{d,k})}\left[\mathbf{U}^{\otimes ai/2}(\mathbf{U}^\top)^{\otimes ai/2}\right]\right\|_{\text{spectral}}\left\|\mathbf{E}_{y_1,\cdots,y_{a/2}\sim D^{\otimes a/2}}\left[\left(\bigotimes_{j=1}^{a/2}\mathbf{T}(y_j)\right)^{\otimes 2}\right]\right\|_2$$

$$=\left\|\mathbf{E}_{\mathbf{U}\sim U(\mathbf{O}_{d,k})}\left[\mathbf{U}^{\otimes ai/2}(\mathbf{U}^\top)^{\otimes ai/2}\right]\right\|_{\text{spectral}}\left\|\mathbf{E}_{y\sim D}\left[\mathbf{T}(y)^{\otimes 2}\right]^{\otimes a/2}\right\|_2$$

$$=\left\|\mathbf{E}_{\mathbf{U}\sim U(\mathbf{O}_{d,k})}\left[\mathbf{U}^{\otimes ai/2}(\mathbf{U}^\top)^{\otimes ai/2}\right]\right\|_{\text{spectral}}\left\|\mathbf{E}_{y\sim D}\left[\mathbf{T}(y)^{\otimes 2}\right]\right\|_2^{a/2}$$

$$\leq\left\|\mathbf{E}_{\mathbf{U}\sim U(\mathbf{O}_{d,k})}\left[\mathbf{U}^{\otimes ai/2}(\mathbf{U}^\top)^{\otimes ai/2}\right]\right\|_{\text{spectral}}\mathbf{E}_{y\sim D}\left[\|\mathbf{T}(y)\|_2^2\right]^{a/2}$$

$$=\left\|\mathbf{E}_{\mathbf{U}\sim U(\mathbf{O}_{d,k})}\left[\mathbf{U}^{\otimes ai/2}(\mathbf{U}^\top)^{\otimes ai/2}\right]\right\|_{\text{spectral}}\|\mathbf{T}(y)\|_{y\sim D}^a\ ,$$

where we used the notation $\left\|\mathbf{E}_{\mathbf{U}\sim U(\mathbf{O}_{d,k})}\left[\mathbf{U}^{\otimes ai/2}(\mathbf{U}^\top)^{\otimes ai/2}\right]\right\|_{\text{spectral}}$ for the spectral norm of $\mathbf{E}_{\mathbf{U}\sim U(\mathbf{O}_{d,k})}\left[\mathbf{U}^{\otimes ai/2}(\mathbf{U}^\top)^{\otimes ai/2}\right]$, which we consider as a $(\mathbb{R}^K)^{\otimes ai/2}\times(\mathbb{R}^K)^{\otimes ai/2}$ symmetric matrix.

Therefore, we just need to bound $\left\|\mathbf{E}_{\mathbf{U}\sim U(\mathbf{O}_{d,k})}\left[\mathbf{U}^{\otimes ai/2}(\mathbf{U}^\top)^{\otimes ai/2}\right]\right\|_{\text{spectral}}$. The calculation here follows Lemma 3.7 of [DKRS23]. Namely, let $\mathbf{A}=\mathbf{E}_{\mathbf{U}\sim U(\mathbf{O}_{d,k})}[\mathbf{U}^{\otimes ai/2}(\mathbf{U}^\intercal)^{\otimes ai/2}]$, $\mathbf{T}_0$ be the eigenvector associated with the largest absolute eigenvalue, and let $\mathbf{u}=\operatorname{argmax}_{\mathbf{u}\in\mathbb{S}^{d-1}}|\langle\mathbf{T}_0,\mathbf{u}^{\otimes ai/2}\rangle|$. Then, we have

$$\|\mathbf{A}\|_2=|\langle\mathbf{A}\mathbf{T}_0,\mathbf{u}^{\otimes ai/2}\rangle|/|\langle\mathbf{T}_0,\mathbf{u}^{\otimes ai/2}\rangle|=|\langle\mathbf{T}_0,\mathbf{A}\mathbf{u}^{\otimes ai/2}\rangle|/|\langle\mathbf{T}_0,\mathbf{u}^{\otimes ai/2}\rangle|$$

$$=|\langle\mathbf{T}_0,\mathbf{E}_{\mathbf{U}\sim U(\mathbf{O}_{n,m})}[(\mathbf{U}\mathbf{U}^\intercal\mathbf{u})^{\otimes ai/2}]\rangle|/|\langle\mathbf{T}_0,\mathbf{u}^{\otimes ai/2}\rangle|$$

$$=|\mathbf{E}_{\mathbf{U}\sim U(\mathbf{O}_{n,m})}[\langle\mathbf{T}_0,(\mathbf{U}\mathbf{U}^\intercal\mathbf{u})^{\otimes ai/2}\rangle]|/|\langle\mathbf{T}_0,\mathbf{u}^{\otimes ai/2}\rangle|$$

$$\leq\mathbf{E}_{\mathbf{U}\sim U(\mathbf{O}_{n,m})}[|\langle\mathbf{T}_0,(\mathbf{U}\mathbf{U}^\intercal\mathbf{u})^{\otimes ai/2}\rangle|]/|\langle\mathbf{T}_0,\mathbf{u}^{\otimes ai/2}\rangle|$$

$$\leq\mathbf{E}_{\mathbf{U}\sim U(\mathbf{O}_{n,m})}[\|(\mathbf{U}\mathbf{U}^\intercal\mathbf{u})^{\otimes ai/2}\|_2|\langle\mathbf{T}_0,\mathbf{u}^{\otimes ai/2}\rangle|]/|\langle\mathbf{T}_0,\mathbf{u}^{\otimes ai/2}\rangle|$$

$$=\mathbf{E}_{\mathbf{U}\sim U(\mathbf{O}_{n,m})}[\|(\mathbf{U}\mathbf{U}^\intercal\mathbf{u})^{\otimes ai/2}\|_2]$$

$$=\mathbf{E}_{\mathbf{U}\sim U(\mathbf{O}_{n,m})}\left[\|\mathbf{U}^\intercal\mathbf{u}\|_2^{ai/2}\right]\ ,$$

where we used $\mathbf{u}=\operatorname{argmax}_{\mathbf{u}\in\mathbb{S}^{d-1}}|\langle\mathbf{T}_0,\mathbf{u}^{\otimes ai/2}\rangle|$ in the second inequality. Plugging everything back into the representation for $\mathbf{E}_{\mathbf{U}\sim U(\mathbf{O}_{d,k})}[\|(\mathbf{U}^\intercal)^{\otimes i}\mathbf{T}\|_{y\sim D}^a]$, we get

$$\mathbf{E}_{\mathbf{U}\sim U(\mathbf{O}_{d,k})}[\|(\mathbf{U}^\intercal)^{\otimes i}\mathbf{T}\|_{y\sim D}^a]\leq\mathbf{E}_{\mathbf{U}\sim U(\mathbf{O}_{d,k})}\left[\|\mathbf{U}^\intercal\mathbf{u}\|_2^{ai/2}\right]\|\mathbf{T}\|_{y\sim D}^a\ .$$

Therefore, it only remains to bound the term $\mathbf{E}_{\mathbf{U}\sim U(\mathbf{O}_{d,k})}\left[\|\mathbf{U}^\intercal\mathbf{u}\|_2^{ai/2}\right]$, which can be bounded by Lemma 3.8 of [DKRS23] as stated below.

**Fact C.11** (Lemma 3.8 of [DKRS23])**.** *For any even $i \in \mathbb{N}$, and $\mathbf{u} \in \mathbb{S}^{d-1}$, we have that*

$$\mathbf{E}_{\mathbf{U}^\top \sim U(\mathbf{O}_{d,k})}[\|\mathbf{U}\mathbf{u}\|_2^i] = \Theta\left(\frac{\Gamma\left(\frac{i+k}{2}\right)\Gamma\left(\frac{d}{2}\right)}{\Gamma\left(\frac{i+d}{2}\right)\Gamma\left(\frac{k}{2}\right)}\right) \ .$$

Plugging it back into the equation right before this fact gives

$$\mathbf{E}_{\mathbf{U} \sim U(\mathbf{O}_{d,k})}[\|(\mathbf{U}^\top)^{\otimes i}\mathbf{T}(y)\|_{y\sim D}^a] = O\left(\frac{\Gamma\left(\frac{i'+k}{2}\right)\Gamma\left(\frac{d}{2}\right)}{\Gamma\left(\frac{i'+d}{2}\right)\Gamma\left(\frac{k}{2}\right)}\right) \|\mathbf{T}(y)\|_{y\sim D}^a \ .$$

The remaining statements follow directly by simplifying the term $\frac{\Gamma\left(\frac{i'+k}{2}\right)\Gamma\left(\frac{d}{2}\right)}{\Gamma\left(\frac{i'+d}{2}\right)\Gamma\left(\frac{k}{2}\right)}$, which follows via the exact same calculation as in Corollary 3.9 of [DKRS23]. $\qquad\square$

Given that $\|(\mathbf{U}^\top)^{\otimes i}\mathbf{T}_i\|_{A_y}$ is $o(1)$ with high probability as discussed above, we just need to show that $\|\mathbf{A}_i\|_{A_y} = \|\mathbf{E}_{\mathbf{x}\sim A_{\mathbf{x}|y}}[\mathbf{H}_i(\mathbf{x})]\|_{A_y}$ does not grow too fast with respect to $i$ compared to $\|(\mathbf{U}^\top)^{\otimes i}\mathbf{T}_i\|_{A_y}$, so that the summation converges. However, for a general hidden distribution $A$, the quantity $\|\mathbf{A}_i\|_{A_y}$ is not bounded. To overcome this obstacle, we leverage an idea from [DKRS23]. Specifically, we can truncate the $\mathbf{x}$ part of $A$ inside a ball to obtain a distribution $A'$. This incurs negligible total variation distance error between $A$ and $A'$ and forces $\|\mathbf{E}_{\mathbf{x}\sim A'_{\mathbf{x}|y}}[\mathbf{H}_i(\mathbf{x})]\|_{A_y}$ to not grow too fast with respect to $i$. We can then proceed with the analysis with respect to $A'$ instead of $A$.

However, this naive approach of directly truncating $\mathbf{x}$ will not work in our context for the following reason: this truncation also changes the marginal distribution of $y$ and the norm we need to bound (we now need to bound $\|\cdot\|_{A'_y}$, instead of $\|\cdot\|_{A_y}$). To overcome this issue, we will do a proper importance sampling on $y$ after the truncation, so that the new distribution $A'$ is close in total variation distance to $A$ and has $\|\mathbf{E}_{\mathbf{x}\sim A'_{\mathbf{x}|y}}[\mathbf{H}_i(\mathbf{x})]\|_{A'_y}$ bounded. Since the total variation distance between $A$ and $A'$ is small, if we can show an SQ lower bound for the RNGCA problem with the hidden distribution $A'$, this implies an SQ lower bound for the RNGCA problem with hidden distribution $A$.

For $B \in \mathbb{R}_+$, we use $\mathbb{B}^k(B) \subseteq \mathbb{R}^k$ to denote the ball defined as $\mathbb{B}^k(B) \overset{\text{def}}{=} \{\mathbf{x} \in \mathbb{R}^k \mid \|\mathbf{x}\|_2 \leq B\}$. We first give the following definition and lemma about $A'$.

**Definition C.12** (Truncated and Reweighted Distribution inside a Ball)**.** Let $A$ be a regular joint distribution of $(\mathbf{x}, y)$ over $\mathbb{R}^k \times \mathcal{Y}$ and $B \in \mathbb{R}_+$. We define the truncated and reweighted distribution $A'$ as the joint distribution of $(\mathbf{x}', y')$ supported on $\mathbb{B}^k(B) \times \mathcal{Y}$ obtained by the following process. We first sample $y' \sim A_y$, then we reject the sample with probability $1 - \mathbf{Pr}_{\mathbf{x}\sim A_{\mathbf{x}|y=y'}}\left[\mathbf{x} \in \mathbb{B}^k(B)\right]^2$. If the sample is not rejected, then we sample $\mathbf{x}' \sim A_{\mathbf{x}|y=y' \wedge \mathbf{x}\in\mathbb{B}^k(B)}$.

We note that $A'$ is by definition a regular distribution since it is defined by $A'_{\mathbf{x}|y}$ for each $y$ and $A'_y$ is also well defined. We now give the following lemma, which shows that $A$ and $A'$ are close in total variation distance and $\|\mathbf{E}_{\mathbf{x}\sim A'_{\mathbf{x}|y}}[\mathbf{H}_i(\mathbf{x})]\|_{A'_y}$ is bounded.

**Lemma C.13.** *Let $k, m \in \mathbb{N}$ with $m$ be even. Let $0 < \nu < 2$ and $A$ be a regular distribution on $\mathbb{R}^k \times \mathcal{Y}$ that $\nu$-matches degree-$m$ moments with the standard Gaussian relative to $\mathcal{Y}$. Let $B \in \mathbb{R}_+$ such that $B^m \geq c_1 \left(2^{m/2}\sqrt{\frac{\Gamma(m+k/2)}{\Gamma(k/2)}}\right)$, where $c_1$ is at least a sufficiently large universal constant. Let $A'$ be the truncated and reweighted distribution over $\mathbb{B}^k(B) \times \mathcal{Y}$, as defined in Definition C.12. Then we have that*

*1. $d_{\mathrm{TV}}(A, A') = O\left(2^{m/2}\sqrt{\frac{\Gamma(m+k/2)}{\Gamma(k/2)}}\right) B^{-m}$; and*

*2. For any $i \in \mathbb{Z}_+$,*

$$\left\| \mathbf{E}_{\mathbf{x} \sim A'_{\mathbf{x}|y}}[\mathbf{H}(\mathbf{x})] \right\|_{A'_y} = \begin{cases} 2^{O(i)} \left( 2^{m/2} \sqrt{\frac{\Gamma(m+k/2)}{\Gamma(k/2)}} \right) B^{i-m} \\ + \left( 1 + O\left( 2^{m/2} \sqrt{\frac{\Gamma(m+k/2)}{\Gamma(k/2)}} \right) B^{-m} \right) \nu, & i < m \, ; \\ 2^{O(i)} \left( 2^{m/2} \sqrt{\frac{\Gamma(m+k/2)}{\Gamma(k/2)}} \right) B^{i-m}, & i \geq m \, . \end{cases}$$

*Proof of Lemma C.13.* We first bound $d_{\mathrm{TV}}(A, A')$. For convenience of the analysis, we define the distribution $\bar{A}$ as the distribution of $(\mathbf{x}, y) \sim A$ conditioned on $\mathbf{x} \in \mathbb{B}^K(B)$. Since $d_{\mathrm{TV}}(A, A') \leq d_{\mathrm{TV}}(A, \bar{A}) + d_{\mathrm{TV}}(\bar{A}, A')$, it suffices for us to bound each term separately.

We first bound $d_{\mathrm{TV}}(A, \bar{A})$ using the fact that $A$ $\nu$-matches degree-$m$ moments with the standard Gaussian relative to $\mathcal{Y}$. Namely, we have that

$$\begin{aligned} \mathbf{E}_{(\mathbf{x},y) \sim A}[\|\mathbf{x}\|_2^m] &\leq \mathbf{E}_{\mathbf{x} \sim \mathcal{N}_k}[\|\mathbf{x}\|_2^m] + \nu \mathbf{E}_{\mathbf{x} \sim \mathcal{N}_k}[\|\mathbf{x}\|_2^{2m}]^{1/2} \\ &= \mathbf{E}_{t \sim \chi^2(k)}[t^{m/2}] + \nu \mathbf{E}_{t \sim \chi^2(k)}[t^m]^{1/2} \\ &= 2^{m/2} \frac{\Gamma((m+k)/2)}{\Gamma(k/2)} + 2^{m/2} \sqrt{\frac{\Gamma((2m+k)/2)}{\Gamma(k/2)}} \nu \\ &\leq c_2 \left( 2^{m/2} \sqrt{\frac{\Gamma(m+k/2)}{\Gamma(k/2)}} \right) , \end{aligned}$$

where $c_2$ is a universal constant. Using Markov's inequality and the union bound, we have

$$\mathbf{Pr}_{(\mathbf{x},y) \sim A}[\mathbf{x} \notin \mathbb{B}^k(B)] \leq c_2 \left( 2^{m/2} \sqrt{\frac{\Gamma(m+k/2)}{\Gamma(k/2)}} \right) B^{-m} \, .$$

By the definition of $\bar{A}$, we have that $d_{\mathrm{TV}}(A, \bar{A}) \leq \mathbf{Pr}_{(\mathbf{x},y) \sim A}[\mathbf{x} \notin \mathbb{B}^m(B)] \leq c_2 \left( 2^{m/2} \sqrt{\frac{\Gamma(m+k/2)}{\Gamma(k/2)}} \right) B^{-m}$.

Then, for $d_{\mathrm{TV}}(\bar{A}, A')$, notice that $\bar{A}_{\mathbf{x}|y}$ and $A'_{\mathbf{x}|y}$ are the same for any $y$. Therefore,

$$\begin{aligned} d_{\mathrm{TV}}(\bar{A}, A') &\leq d_{\mathrm{TV}}(\bar{A}_y, A'_y) \leq d_{\mathrm{TV}}(\bar{A}_y, A_y) + d_{\mathrm{TV}}(A'_y, A_y) \\ &= c_2 \left( 2^{m/2} \sqrt{\frac{\Gamma(m+k/2)}{\Gamma(k/2)}} \right) B^{-m} + d_{\mathrm{TV}}(A'_y, A_y) \, . \end{aligned}$$

So we just need to bound $d_{\mathrm{TV}}(A'_y, A_y)$. From the definition of $A'$, notice that

$$\begin{aligned} d_{\mathrm{TV}}(A'_y, A_y) &\leq \mathbf{E}_{y \sim A_y}[1 - \mathbf{Pr}_{\mathbf{x} \sim A_{\mathbf{x}|y}}[\mathbf{x} \in \mathbb{B}^k(B)]^2] \\ &\leq \mathbf{E}_{y \sim A_y}[2\mathbf{Pr}_{\mathbf{x} \sim A_{\mathbf{x}|y}}[\mathbf{x} \notin \mathbb{B}^k(B)]] \\ &\leq 2\mathbf{Pr}_{(\mathbf{x},y) \sim A}[\mathbf{x} \notin \mathbb{B}^k(B)] \leq 2c_2 \left( 2^{m/2} \sqrt{\frac{\Gamma(m+k/2)}{\Gamma(k/2)}} \right) B^{-m} \, . \end{aligned}$$

Combining the above, we get $d_{\mathrm{TV}}(A, A') \leq 4c_2 \left( 2^{m/2} \sqrt{\frac{\Gamma(m+k/2)}{\Gamma(k/2)}} \right) B^{-m}$.

It remains to verify the bound on $\left\| \mathbf{E}_{\mathbf{x} \sim A'_{\mathbf{x}|y}}[\mathbf{H}_i(\mathbf{x})] \right\|_{A'_y}$. We will analyze the cases $1 \leq i < m$ and $i \geq m$ respectively. We first prove the following bound that will be convenient for the analysis that

follows. Notice that, by the definition of $A'$, we have that for any function $f : \mathbb{R}^k \to \mathbb{R}^n$,

$$\left\| \mathbf{E}_{\mathbf{x} \sim A'_{\mathbf{x}|y}} [f(\mathbf{x})] \right\|_{A'_y}$$

$$= \mathbf{E}_{y \sim A'_y} \left[ \left\| \mathbf{E}_{\mathbf{x} \sim A'_{\mathbf{x}|y}} [f(\mathbf{x})] \right\|_2^2 \right]^{1/2}$$

$$= \mathbf{E}_{y \sim A'_y} \left[ \left\| \mathbf{E}_{\mathbf{x} \sim A_{\mathbf{x}|y}} \left[ f(\mathbf{x}) \mathbb{1} \left( \mathbf{x} \in \mathbb{B}^k(B) \right) \right] / \mathbf{Pr}_{\mathbf{x} \sim A_{\mathbf{x}|y}} \left[ \mathbf{x} \in \mathbb{B}^k(B) \right] \right\|_2^2 \right]^{1/2} \tag{2}$$

$$= \mathbf{E}_{y \sim A_y} \left[ \left\| \mathbf{E}_{\mathbf{x} \sim A_{\mathbf{x}|y}} \left[ f(\mathbf{x}) \mathbb{1} \left( \mathbf{x} \in \mathbb{B}^k(B) \right) \right] \right\|_2^2 \right]^{1/2} \mathbf{E}_{y \sim A_y} \left[ \mathbf{Pr}_{\mathbf{x} \sim A_{\mathbf{x}|y}} \left[ \mathbf{x} \in \mathbb{B}^k(B) \right]^2 \right]^{-1}$$

$$\leq \left\| f(\mathbf{x}) \mathbb{1} \left( \mathbf{x} \in \mathbb{B}^k(B) \right) \right\|_{A_y} \left( 1 - 2 \mathbf{Pr}_{(\mathbf{x},y) \sim A_{\mathbf{x}|y}} \left[ \mathbf{x} \notin \mathbb{B}^k(B) \right] \right)^{-1}$$

$$= \left( 1 + O \left( 2^{m/2} \sqrt{\frac{\Gamma(m+k/2)}{\Gamma(k/2)}} \right) B^{-m} \right) \left\| f(\mathbf{x}) \mathbb{1} \left( \mathbf{x} \in \mathbb{B}^k(B) \right) \right\|_{A_y} ,$$

where the last equality follows from the earlier bound that

$$\mathbf{Pr}_{(\mathbf{x},y) \sim A} \left[ \mathbf{x} \notin \mathbb{B}^k(B) \right] \leq c_2 \left( 2^{m/2} \sqrt{\frac{\Gamma(m+k/2)}{\Gamma(k/2)}} \right) B^{-m} ,$$

and the assumption that $B^m \geq c_1 \left( 2^{m/2} \sqrt{\frac{\Gamma(m+k/2)}{\Gamma(k/2)}} \right)$. Given Equation (2), we have

$$\left\| \mathbf{E}_{\mathbf{x} \sim A'_{\mathbf{x}|y}} [\mathbf{H}(\mathbf{x})] \right\|_{A'_y}$$

$$= \left( 1 + O \left( 2^{m/2} \sqrt{\frac{\Gamma(m+k/2)}{\Gamma(k/2)}} \right) B^{-m} \right) \left\| \mathbf{E}_{\mathbf{x} \sim A_{\mathbf{x}|y}} \left[ \mathbf{H}(\mathbf{x}) \mathbb{1} \left( \mathbf{x} \in \mathbb{B}^k(B) \right) \right] \right\|_{A_y} .$$

Therefore, we just need to bound $\left\| \mathbf{E}_{\mathbf{x} \sim A_{\mathbf{x}|y}} \left[ \mathbf{H}(\mathbf{x}) \mathbb{1} \left( \mathbf{x} \in \mathbb{B}^k(B) \right) \right] \right\|_{A_y}$.

For the case $1 \leq k < m$, notice that

$$\left\| \mathbf{E}_{\mathbf{x} \sim A_{\mathbf{x}|y}} \left[ \mathbf{H}(\mathbf{x}) \mathbb{1} \left( \mathbf{x} \in \mathbb{B}^k(B) \right) \right] \right\|_{A_y}$$

$$\leq \left\| \mathbf{E}_{\mathbf{x} \sim A_{\mathbf{x}|y}} [\mathbf{H}(\mathbf{x})] \right\|_{A_y} + \left\| \mathbf{E}_{\mathbf{x} \sim A_{\mathbf{x}|y}} \left[ \mathbf{H}(\mathbf{x}) \mathbb{1} \left( \mathbf{x} \notin \mathbb{B}^k(B) \right) \right] \right\|_{A_y}$$

$$\leq \nu + \left\| \mathbf{E}_{\mathbf{x} \sim A_{\mathbf{x}|y}} \left[ \|\mathbf{H}(\mathbf{x})\|_2 \mathbb{1} \left( \mathbf{x} \in \mathbb{B}^k(B) \right) \right] \right\|_{A_y} .$$

To bound the second term, we will use the following fact from [DKRS23].

**Fact C.14** (Fact B.1 of [DKRS23]). *Let $\mathbf{H}_i$ be the $i$-th Hermite tensor in $k$ dimensions. Suppose that $\|\mathbf{x}\|_2 \geq k^{1/4}$. Then $\|\mathbf{H}_i(\mathbf{x})\|_2 = 2^{O(i)} \|\mathbf{x}\|_2^i$.*

Given that $B^m \geq c_1 \left( 2^{m/2} \sqrt{\frac{\Gamma(m+k/2)}{\Gamma(k/2)}} \right)$, we have $B^2 > k$. Therefore, using Fact C.14, we get

$$\left\| \mathbf{E}_{\mathbf{x} \sim A_{\mathbf{x}|y}} \left[ \|\mathbf{H}(\mathbf{x})\|_2 \mathbb{1} \left( \mathbf{x} \notin \mathbb{B}^k(B) \right) \right] \right\|_{A_y}$$

$$\leq \left\| \mathbf{E}_{\mathbf{x} \sim A_{\mathbf{x}|y}} \left[ 2^{O(i)} \|\mathbf{x}\|_2^i \mathbb{1} \left( \mathbf{x} \notin \mathbb{B}^k(B) \right) \right] \right\|_{A_y}$$

$$\leq 2^{O(i)} \left\| \mathbf{E}_{\mathbf{x} \sim A_{\mathbf{x}|y}} \left[ \|\mathbf{x}\|_2^i \mathbb{1} \left( \mathbf{x} \notin \mathbb{B}^k(B) \right) \right] \right\|_{A_y}$$

$$\leq 2^{O(i)} \left\| \int_0^\infty \mathbf{Pr}_{\mathbf{x} \sim A_{\mathbf{x}|y}} \left[ \|\mathbf{x}\|_2 \geq u \wedge \mathbf{x} \notin \mathbb{B}^k(B) \right] du^i \right\|_{A_y}$$

$$\leq 2^{O(i)} \int_0^\infty \left\| \mathbf{Pr}_{\mathbf{x} \sim A_{\mathbf{x}|y}} \left[ \|\mathbf{x}\|_2 \geq u \wedge \mathbf{x} \notin \mathbb{B}^k(B) \right] \right\|_{A_y} du^i ,$$

where

$$\left\| \mathbf{Pr}_{\mathbf{x} \sim A_{\mathbf{x}|y}} \left[ \|\mathbf{x}\|_2 \geq u \right] \right\|_{A_y} \leq \left\| \mathbf{E}_{\mathbf{x} \sim A_{\mathbf{x}|y}} \left[ \|\mathbf{x}\|_2^m \right] / u^m \right\|_{A_y} = \left\| \mathbf{E}_{\mathbf{x} \sim A_{\mathbf{x}|y}} \left[ \|\mathbf{x}\|_2^m \right] \right\|_{A_y} / u^m .$$

Therefore, using the earlier bound on $\left\|\mathbf{E}_{\mathbf{x}\sim A_{\mathbf{x}|y}}\left[\|\mathbf{x}\|_2^m\right]\right\|_{A_y}$, we get

$$\left\|\mathbf{E}_{\mathbf{x}\sim A_{\mathbf{x}|y}}\left[\|\mathbf{H}(\mathbf{x})\|_2\,\mathbb{1}\left(\mathbf{x}\notin\mathbb{B}^k(B)\right)\right]\right\|_{A_y}$$

$$\leq 2^{O(i)}\int_0^\infty\left(2^{m/2}\sqrt{\frac{\Gamma(m+k/2)}{\Gamma(k/2)}}\right)\min(B^{-m},u^{-m})du^i$$

$$\leq 2^{O(i)}\left(2^{m/2}\sqrt{\frac{\Gamma(m+k/2)}{\Gamma(k/2)}}\right)\left(\int_0^B B^{-m}du^i+\int_B^\infty u^{-m}du^i\right)$$

$$\leq 2^{O(i)}\left(2^{m/2}\sqrt{\frac{\Gamma(m+k/2)}{\Gamma(k/2)}}\right)B^{i-m}\ .$$

Plugging this back in the ealier equation for $\left\|\mathbf{E}_{\mathbf{x}\sim A'_{\mathbf{x}|y}}[\mathbf{H}(\mathbf{x})]\right\|_{A'_y}$, we get that for $1\leq i<m$,

$$\left\|\mathbf{E}_{\mathbf{x}\sim A'_{\mathbf{x}|y}}[\mathbf{H}(\mathbf{x})]\right\|_{A'_y}=2^{O(i)}\left(2^{m/2}\sqrt{\frac{\Gamma(m+k/2)}{\Gamma(k/2)}}\right)B^{i-m}$$

$$+\left(1+O\left(2^{m/2}\sqrt{\frac{\Gamma(m+k/2)}{\Gamma(k/2)}}\right)B^{-m}\right)\nu\ .$$

Now we bound $\left\|\mathbf{E}_{\mathbf{x}\sim A'_{\mathbf{x}|y}}[\mathbf{H}_i(\mathbf{x})]\right\|_{A'_y}$ for $i\geq m$. For an order-$i$ tensor $\mathbf{A}$, we use $\mathbf{A}^\pi$ to denote the matrix $\mathbf{A}^\pi_{j_1,\cdots,j_k}=\mathbf{A}_{\pi(j_1,\cdots,j_k)}$ and $\|\mathbf{A}\|_2=\|\mathbf{A}^\pi\|_2$. From the definition of the Hermite tensor, we have

$$\mathbf{H}(\mathbf{x})=\frac{1}{\sqrt{i!}}\sum_{t=0}^{\lfloor i/2\rfloor}\sum_{\text{Permutation }\pi\text{ of }[i]}\frac{1}{2^t t!(i-2t)!}\left((-\mathbf{I})^{\otimes t}\mathbf{x}^{\otimes(i-2t)}\right)^\pi\ .$$

This implies that

$$\left\|\mathbf{E}_{\mathbf{x}\sim A'_{\mathbf{x}|y}}[\mathbf{H}_i(\mathbf{x})]\right\|_{A'_y}$$

$$=\left\|\frac{1}{\sqrt{i!}}\sum_{t=0}^{\lfloor i/2\rfloor}\sum_{\text{Permutation }\pi\text{ of }[i]}\frac{1}{2^t t!(i-2t)!}\left((-\mathbf{I})^{\otimes t}\mathbf{E}_{\mathbf{x}\sim A'_{\mathbf{x}|y}}\left[\mathbf{x}^{\otimes(i-2t)}\right]\right)^\pi\right\|_{A'_y}$$

$$\leq\sum_{t=1}^{\lfloor i/2\rfloor}\frac{\sqrt{i!}}{2^t t!(i-2t)!}\|\mathbf{I}^{\otimes t}\|_2\left\|\mathbf{E}_{\mathbf{x}\sim A'_{\mathbf{x}|y}}\left[\mathbf{x}^{\otimes(i-2t)}\right]\right\|_{A'_y}$$

$$\leq\sum_{t=1}^{\lfloor i/2\rfloor}\frac{\sqrt{i!}}{2^t t!(i-2t)!}\|\mathbf{I}^{\otimes t}\|_2\left\|\mathbf{E}_{\mathbf{x}\sim A'_{\mathbf{x}|y}}\left[\left\|\mathbf{x}^{\otimes(i-2t)}\right\|_2\right]\right\|_{A'_y}$$

$$=\sum_{t=1}^{\lfloor i/2\rfloor}\frac{\sqrt{i!}}{2^t t!(i-2t)!}\|\mathbf{I}^{\otimes t}\|_2\left\|\mathbf{E}_{\mathbf{x}\sim A'_{\mathbf{x}|y}}\left[\|\mathbf{x}\|_2^{i-2t}\right]\right\|_{A'_y}$$

$$\leq\sum_{t=1}^{\lfloor i/2\rfloor}\frac{\sqrt{i!}}{2^t t!(i-2t)!}k^{t/2}B^{\max(i-m-2t,0)}\left\|\mathbf{E}_{\mathbf{x}\sim A'_{\mathbf{x}|y}}\left[\|\mathbf{x}\|_2^{\min(m,i-2t)}\right]\right\|_{A'_y}$$

$$\leq\sum_{t=1}^{\lfloor i/2\rfloor}\frac{\sqrt{i!}}{2^t t!(i-2t)!}k^{\min(2t,i-m)/4}B^{\max(i-m-2t,0)}k^{\max(0,2t-i+m)/4}\left\|\mathbf{E}_{\mathbf{x}\sim A'_{\mathbf{x}|y}}\left[\|\mathbf{x}\|_2^{\min(m,i-2t)}\right]\right\|_{A'_y}$$

$$\leq B^{i-m}\sum_{t=1}^{\lfloor i/2\rfloor}\frac{\sqrt{i!}}{2^t t!(i-2t)!}k^{\max(0,2t-i+m)/4}\left\|\mathbf{E}_{\mathbf{x}\sim A'_{\mathbf{x}|y}}\left[\|\mathbf{x}\|_2^{\min(m,i-2t)}\right]\right\|_{A'_y}\ ,$$

where the last inequality follows from the fact that $B^2\geq k$ (which, as already noted, is implied by the fact that $B^m\geq c_1\left(2^{m/2}\sqrt{\frac{\Gamma(m+k/2)}{\Gamma(k/2)}}\right)$).

We now bound the term $\left\|\mathbf{E}_{\mathbf{x}\sim A'_{\mathbf{x}|y}}\left[\|\mathbf{x}\|_2^{\min(m,i-2t)}\right]\right\|_{A'_y}$. For convenience of the analysis, let $m' = \min(m, i-2t)$ and $f(y) = \mathbf{E}_{\mathbf{x}\sim A_{\mathbf{x}|y}}\left[\|\mathbf{x}\|_2^{m'}\right]$. Notice that the quantity we want to bound is $\|f\|_{A'_y}$. Furthermore, using the fact that $A'$ $\nu$-matches degree-$m$ moments relative to $\mathcal{Y}$ with the standard Gaussian, we have that

$$
\begin{aligned}
\|f\|_{A'_y}^2 &= \mathbf{E}_{(\mathbf{x},y)\sim A'}\left[f(y)\|\mathbf{x}\|_2^{m'}\right]\\
&\leq \mathbf{E}_{(\mathbf{x},y)\sim\mathcal{N}_k\otimes A'_y}\left[f(y)\|\mathbf{x}\|_2^{m'}\right] + \nu\|f(y)\|\mathbf{x}\|_2^{m'}\|_{\mathcal{N}_k\otimes A'_y}\\
&= \mathbf{E}_{y\sim A'_y}[f(y)]\mathbf{E}_{\mathbf{x}\sim\mathcal{N}_k}[\|\mathbf{x}\|_2^{m'}] + \nu\|f\|_{A'_y}\mathbf{E}_{\mathbf{x}\sim\mathcal{N}_k}\left[\|\mathbf{x}\|_2^{2m'}\right]^{1/2}\\
&= \mathbf{E}_{\mathbf{x}\sim A'_x}[\|\mathbf{x}\|_2^{m'}]\mathbf{E}_{\mathbf{x}\sim\mathcal{N}_k}[\|\mathbf{x}\|_2^{m'}] + \nu\|f\|_{A'_y}\mathbf{E}_{\mathbf{x}\sim\mathcal{N}_k}\left[\|\mathbf{x}\|_2^{2m'}\right]^{1/2}\\
&= \left(\mathbf{E}_{\mathbf{x}\sim\mathcal{N}_k}[\|\mathbf{x}\|_2^{m'}] + \nu\mathbf{E}_{\mathbf{x}\sim\mathcal{N}_k}\left[\|\mathbf{x}\|_2^{2m'}\right]^{1/2}\right)\mathbf{E}_{\mathbf{x}\sim\mathcal{N}_k}[\|\mathbf{x}\|_2^{m'}] + \nu\|f\|_{A'_y}\mathbf{E}_{\mathbf{x}\sim\mathcal{N}_k}\left[\|\mathbf{x}\|_2^{2m'}\right]^{1/2}\\
&= \mathbf{E}_{\mathbf{x}\sim\mathcal{N}_k}[\|\mathbf{x}\|_2^{m'}]^2 + \nu\mathbf{E}_{\mathbf{x}\sim\mathcal{N}_k}\left[\|\mathbf{x}\|_2^{2m'}\right]^{1/2}\mathbf{E}_{\mathbf{x}\sim\mathcal{N}_k}[\|\mathbf{x}\|_2^{m'}] + \nu\|f\|_{A'_y}\mathbf{E}_{\mathbf{x}\sim\mathcal{N}_k}\left[\|\mathbf{x}\|_2^{2m'}\right]^{1/2}.
\end{aligned}
$$

Since $\nu = O(1)$, we must have

$$
\|f\|_{A'_y} = O\left(\max\left(\mathbf{E}_{\mathbf{x}\sim\mathcal{N}_k}[\|\mathbf{x}\|_2^{m'}], \mathbf{E}_{\mathbf{x}\sim\mathcal{N}_k}\left[\|\mathbf{x}\|_2^{2m'}\right]^{1/2}\right)\right).
$$

Notice that both quantities can be calculated using the $\chi^2$ distribution. From previous calculations, we have that

$$
\mathbf{E}_{\mathbf{x}\sim\mathcal{N}_k}[\|\mathbf{x}\|_2^{m'}] \leq \mathbf{E}_{\mathbf{x}\sim\mathcal{N}_k}\left[\|\mathbf{x}\|_2^{2m'}\right]^{1/2} = 2^{\min(m,i-2t)/2}\sqrt{\frac{\Gamma(\min(m,i-2t)+k/2)}{\Gamma(k/2)}}.
$$

Therefore, we get $\|f\|_{A'_y} = O\left(2^{\min(m,i-2t)/2}\sqrt{\frac{\Gamma(\min(m,i-2t)+k/2)}{\Gamma(k/2)}}\right)$. Plugging it back in the ealier representation for $\left\|\mathbf{E}_{\mathbf{x}\sim A'_{\mathbf{x}|y}}[\mathbf{H}_i(\mathbf{x})]\right\|_{A'_y}$ gives

$$
\begin{aligned}
&\left\|\mathbf{E}_{\mathbf{x}\sim A'_{\mathbf{x}|y}}[\mathbf{H}_i(\mathbf{x})]\right\|_{A'_y}\\
\leq& B^{i-m}\sum_{t=1}^{\lfloor i/2\rfloor}\frac{\sqrt{i!}}{2^t t!(i-2t)!}k^{\max(0,2t-i+m)/4}O\left(2^{\min(m,i-2t)/2}\sqrt{\frac{\Gamma(\min(m,i-2t)+k/2)}{\Gamma(k/2)}}\right)\\
\leq& B^{i-m}O\left(2^{m/2}\sqrt{\frac{\Gamma(m+k/2)}{\Gamma(k/2)}}\right)\sum_{t=1}^{\lfloor i/2\rfloor}\frac{\sqrt{i!}}{2^t t!(i-2t)!},
\end{aligned}
$$

where the second inequality follows from the elementary fact $\max(0, 2t-i+m)+\min(m, i-2t) = m$. One can see that the denominator is minimized when $t = i/2 - O(\sqrt{i})$. Then it follows that the sum is at most $2^{O(i)}B^{i-m}\left(2^{m/2}\sqrt{\frac{\Gamma(m+k/2)}{\Gamma(k/2)}}\right)$.

This completes the proof of Lemma C.13. $\qquad\square$

Recall that the quantity we want to bound is $|\mathbf{E}_{(\mathbf{x},y)\sim\mathbf{P}_{\mathbf{U}}^A}[f(\mathbf{x},y)] - \mathbf{E}_{(\mathbf{x},y)\sim\mathcal{N}_d\otimes A_y}[f(\mathbf{x},y)]|$. Given that $A'$ and $A$ are close in total variation distance, we have that for any $\mathbf{U} \in \mathbf{O}_{d,k}$, the distributions $\mathbf{P}_{\mathbf{U}}^A$ and $\mathbf{P}_{\mathbf{U}}^{A'}$ are close in total variation distance. Therefore, for any query function $f : \mathbb{R}^d \times \mathcal{Y} \to [-1,1]$, $\mathbf{E}_{(\mathbf{x},y)\sim\mathbf{P}_{\mathbf{U}}^A}[f(\mathbf{x},y)] - \mathbf{E}_{(\mathbf{x},y)\sim\mathbf{P}_{\mathbf{U}}^{A'}}[f(\mathbf{x},y)]$ is small and $\mathbf{E}_{(\mathbf{x},y)\sim\mathcal{N}_d\otimes A_y}[f(\mathbf{x},y)] - \mathbf{E}_{(\mathbf{x},y)\sim\mathcal{N}_d\otimes A'_y}[f(\mathbf{x},y)]$ is small. Thus, we can bound $|\mathbf{E}_{(\mathbf{x},y)\sim\mathbf{P}_{\mathbf{U}}^{A'}}[f(\mathbf{x},y)] - \mathbf{E}_{(\mathbf{x},y)\sim\mathcal{N}_d\otimes A'_y}[f(\mathbf{x},y)]|$ instead.

For that it suffices for us to apply the Hermite decomposition (Lemma C.8) to $A'$ instead of $A$ and analyze $\sum_{i=1}^{\ell}\langle\mathbf{U}^{\otimes i}\mathbf{A}_i, \mathbf{T}_i\rangle_{A'_y}$, where $\mathbf{A}_i(y) = \mathbf{E}_{\mathbf{x}\sim A'_{\mathbf{x}|y}}[\mathbf{H}_i(\mathbf{x})]$. We give the following upper bound on $\sum_{i=1}^{\ell}\langle\mathbf{U}^{\otimes i}\mathbf{A}_i, \mathbf{T}_i\rangle A'_y$.

**Lemma C.15.** *Under the conditions of Proposition C.7, and further assuming $m, k \leq d^\lambda / \log d$, $\nu < 2$ and $\left( \frac{\Gamma(m/2+k/2)}{\Gamma(k/2)} \right) d^{-((1-\lambda)/4-c)m} < 2$, the following holds: For any $d$ that is at least a sufficiently large constant depending on $c$, there is a $B < d$ such that the truncated and reweighted distribution $A'$ over $\mathbb{B}^k(B) \times \mathcal{Y}$, defined Definition C.12, satisfies*

$$d_{\mathrm{TV}}(A, A') \leq \left( \frac{\Gamma(m/2 + k/2)}{\Gamma(k/2)} \right) d^{-((1-\lambda)/4-c)m} .$$

*Furthermore for any $\ell \in \mathbb{Z}_+$, except with probability at most $2^{-d^{\Omega(c)}}$ with respect to $\mathbf{U} \sim U(\mathbf{O}_{d,k})$, it holds*

$$\left| \sum_{i=1}^{\ell} \langle \mathbf{A}_i, (\mathbf{U}^\top)^{\otimes i} \mathbf{T}_i \rangle_{A'_y} \right| \leq \left( \frac{\Gamma(m/2 + k/2)}{\Gamma(k/2)} \right) d^{-((1-\lambda)/4-c)m} + (1 + o(1))\nu ,$$

*where $\mathbf{A}_i(y) = \mathbf{E}_{\mathbf{x} \sim A'_{\mathbf{x}|y}}[\mathbf{H}_i(\mathbf{x})]$ and $\mathbf{T}_i(y) = \mathbf{E}_{\mathbf{x} \sim \mathcal{N}_d}[f(\mathbf{x}, y)\mathbf{H}_i(\mathbf{x})]$.*

*Proof.* For convenience in the relevant calculations, we will break the summation into four ranges. We can write

$$\left| \sum_{i=1}^{\ell} \left\langle \mathbf{A}_i, \left(\mathbf{V}^\top\right)^{\otimes i} \mathbf{T}_i \right\rangle_{A'_y} \right| \leq \sum_{i=1}^{\ell} \left| \left\langle \mathbf{A}_i, \left(\mathbf{V}^\top\right)^{\otimes i} \mathbf{T}_i \right\rangle_{A'_y} \right|$$

$$= \sum_{i=1}^{m-1} \left| \left\langle \mathbf{A}_i, \left(\mathbf{V}^\top\right)^{\otimes i} \mathbf{T}_i \right\rangle_{A'_y} \right| + \sum_{i=m}^{d^\lambda} \left| \left\langle \mathbf{A}_i, \left(\mathbf{V}^\top\right)^{\otimes i} \mathbf{T}_i \right\rangle_{A'_y} \right|$$

$$+ \sum_{i=d^\lambda+1}^{T} \left| \left\langle \mathbf{A}_i, \left(\mathbf{V}^\top\right)^{\otimes i} \mathbf{T}_i \right\rangle_{A'_y} \right| + \sum_{i=T+1}^{\ell} \left| \left\langle \mathbf{A}_i, \left(\mathbf{V}^\top\right)^{\otimes i} \mathbf{T}_i \right\rangle_{A'_y} \right| ,$$

where $T$ is a value we will later specify. To analyze each $\left| \left\langle \mathbf{A}_i, \left(\mathbf{V}^\top\right)^{\otimes i} \mathbf{T}_i \right\rangle_{A'_y} \right|$, recall that

$\left| \left\langle \mathbf{A}_i, \left(\mathbf{V}^\top\right)^{\otimes i} \mathbf{T}_i \right\rangle_{A'_y} \right| \leq \|\mathbf{A}_i\|_{A'_y} \left\| \left(\mathbf{V}^\top\right)^{\otimes i} \mathbf{T}_i \right\|_{A'_y}$, where $\mathbf{A}_i = \mathbf{E}_{\mathbf{x} \sim A'_{\mathbf{x}|y}}[\mathbf{H}_i(\mathbf{x})]$ is a constant (not depending on the randomness of $\mathbf{V}$). For $\left\| \left(\mathbf{V}^\top\right)^{\otimes i} \mathbf{T}_i \right\|_{A'_y}$, we can show it is small by bounding its $a$-th moment for even $a$ using Fact C.10. We will apply this strategy on the four different ranges of $i$.

Without loss of generality, we will assume that $\lambda \geq 4c$. Suppose that $\lambda < 4c$. Then we can simply consider a new pair $\lambda', c'$, where $\lambda' = \lambda + 2c$ and $c' = c/2$. Notice that $(1 - \lambda)/4 - c = (1 - \lambda')/4 - c'$; therefore, the SQ lower bound in the statement remains unchanged.

We start by picking the following parameters (the "sufficiently close" here only depends on $c$):

- We require $m, k \leq d^\lambda / \log d$;

- $B = d^\alpha$, where $\alpha < (1 - \lambda_3)/4$ and $(1 - \lambda_3)/4 - \alpha$ is a sufficiently small constant fraction of $c$;

- $T = d^{\max(2\alpha, \lambda)}$;

- We let $\lambda_3 > \lambda_2 > \lambda_1 > \lambda$ to be sufficiently close (the difference between these quantities will be a sufficiently small constant fraction of $c$).

We now bound the summation $\sum_{i=1}^{\ell} \left| \left\langle \mathbf{A}_i, \left(\mathbf{V}^\top\right)^{\otimes i} \mathbf{T}_i \right\rangle_{A'_y} \right|$ as follows:

$\sum_{i=1}^{m-1} \left| \left\langle \mathbf{A}_i, \left(\mathbf{V}^\top\right)^{\otimes i} \mathbf{T}_i \right\rangle_{A'_y} \right|$ **is small with high probability:**

Since $\left(\frac{\Gamma(m+k/2)}{\Gamma(k/2)}\right)d^{-((1-\lambda)/4-c)m} < 2$ and $B = d^\alpha$, where $\alpha$ is sufficiently close to $(1-\lambda)/4$, the parameters satisfy the condition $B^m = \omega\left(2^{m/2}\sqrt{\frac{\Gamma(m+k/2)}{\Gamma(k/2)}}\right)$ in Lemma C.13. Since $i < m$, by Lemma C.13, we have

$$\|\mathbf{A}_i\|_{A_y'} = 2^{O(i)}\left(2^{m/2}\sqrt{\frac{\Gamma(m+k/2)}{\Gamma(k/2)}}\right)B^{i-m} + \left(1 + O\left(2^{m/2}\sqrt{\frac{\Gamma(m+k/2)}{\Gamma(k/2)}}\right)B^{-m}\right)\nu\ .$$

Let $a$ be the largest even number such that $ai/2 \le d^\lambda$, where $m = o\left(d^\lambda\right)$ implies $a \ge 2$. Then using Fact C.10, we have

$$\mathbf{E}_{\mathbf{V}\sim U(\mathbf{O}_{d,k})}\left[\left\|(\mathbf{V}^\top)^{\otimes i}\mathbf{T}_i\right\|_{A_y'}^a\right] = O\left(2^{ai/4}d^{-(1-\lambda)ai/4}\right) = O\left(d^{-(1-\lambda_1)ai/4}\right)\ .$$

Using Markov's Inequality, this implies the tail bound

$$\mathbf{Pr}\left[\left\|(\mathbf{V}^\top)^{\otimes i}\mathbf{T}_i\right\|_{A_y'} \ge d^{-(1-\lambda_2)i/4}\right] \le 2^{-\Omega\left(cd^\lambda\right)} = 2^{-d^{\Omega(c)}}\ .$$

Therefore, we have

$$\sum_{i=1}^{m-1}\left|\left\langle\mathbf{A}_i, (\mathbf{V}^\top)^{\otimes i}\mathbf{T}_i\right\rangle_{A_y'}\right| \le \sum_{i=1}^{m-1}\|\mathbf{A}_i\|_{A_y'}\left\|(\mathbf{V}^\top)^{\otimes i}\mathbf{T}_i\right\|_{A_y'}$$

$$\le \sum_{i=1}^{m-1}d^{-(1-\lambda_2)i/4}2^{O(i)}\left(2^{m/2}\sqrt{\frac{\Gamma(m+k/2)}{\Gamma(k/2)}}\right)B^{i-m}$$

$$+ \sum_{i=1}^{m-1}d^{-(1-\lambda_2)i/4}\left(1 + O\left(2^{m/2}\sqrt{\frac{\Gamma(m+k/2)}{\Gamma(k/2)}}\right)B^{-m}\right)\nu$$

$$\le(1+o(1))\left(2^{m/2}\sqrt{\frac{\Gamma(m+k/2)}{\Gamma(k/2)}}B^{-m} + \nu\right)$$

$$=(1+o(1))\left(2^{m/2}\sqrt{\frac{\Gamma(m+k/2)}{\Gamma(k/2)}}d^{-\alpha m} + \nu\right)\ ,$$

except with probability $2^{-d^{\Omega(c)}}$.

$\sum_{i=m}^{d^\lambda}\left|\left\langle\mathbf{A}_i, (\mathbf{V}^\top)^{\otimes i}\mathbf{T}_i\right\rangle_{A_y'}\right|$ **is small with high probability:**

In the previous case, we have argued that the parameters satisfy the condition $B^m = \omega\left(2^{m/2}\sqrt{\frac{\Gamma(m+k/2)}{\Gamma(k/2)}}\right)$ in Lemma C.13. Since $k \ge m$, by Lemma C.13 we have $\|\mathbf{A}_k\|_{A_y'} = 2^{O(i)}\left(2^{m/2}\sqrt{\frac{\Gamma(m+k/2)}{\Gamma(k/2)}}\right)B^{i-m}$. Let $a$ be the largest even number that $ai/2 \le d^\lambda$, where $m = o\left(d^\lambda\right)$ implies $a \ge 2$. Applying Fact C.10 yields

$$\mathbf{E}_{\mathbf{V}\sim U(\mathbf{O}_{d,k})}\left[\left\|(\mathbf{V}^\top)^{\otimes i}\mathbf{T}_i\right\|_{A_y'}^a\right] = O\left(2^{ai/4}d^{-(1-\lambda)ai/4}\right) = O\left(d^{-(1-\lambda_1)ai/4}\right)\ .$$

Therefore, we have

$$\sum_{i=m}^{d^\lambda}\left|\left\langle\mathbf{A}_i, (\mathbf{V}^\top)^{\otimes i}\mathbf{T}_i\right\rangle_{A_y'}\right| \le \sum_{i=m}^{d^\lambda}\|\mathbf{A}_i\|_{A_y'}\left\|(\mathbf{V}^\top)^{\otimes i}\mathbf{T}_i\right\|_{A_y'}$$

$$\le \sum_{i=m}^{d^\lambda}d^{-(1-\lambda_2)i/4}2^{O(i)}\left(2^{m/2}\sqrt{\frac{\Gamma(m+k/2)}{\Gamma(k/2)}}\right)B^{i-m}$$

$$= 2^{O(m)}d^{-((1-\lambda_2)/4)m} = d^{-((1-\lambda_3)/4)m}\ ,$$

except with probability $2^{-d^{\Omega(c)}}$ (the first equality above follows from $B = d^\alpha = o\left(d^{(1-\lambda)/4}\right) = o\left(d^{(1-\lambda_2)/4}\right)$).

$\sum_{i=d^\lambda+1}^{T}\left|\left\langle \mathbf{A}_i, \left(\mathbf{V}^\top\right)^{\otimes i}\mathbf{T}_i\right\rangle_{A'_y}\right|$ **is small with high probability:** We assume without loss of generality that $d^\lambda < T$, since otherwise, this term is just 0. Notice that this implies that $\lambda < 2\alpha$. We will then use Fact 3.5 of [DKRS23] to bound $\|\mathbf{A}_i\|_{A'_y}$.

**Fact C.16** (Fact 3.5 of [DKRS23])**.** *Let $\mathbf{H}_i$ be the $i$-th Hermite tensor in $k$ dimensions. Suppose that $\|\mathbf{x}\|_2 \le B$. Then $\|\mathbf{H}_i(\mathbf{x})\|_2 \le 2^i k^{i/4} B^i i^{-i/2}\exp\left(\binom{i}{2}/B^2\right)$.*

Using Fact C.16, we have

$$\|\mathbf{A}_i\|_{A'_y} = \left\|\mathbf{E}_{\mathbf{x}\sim A'_{\mathbf{x}|y}}\left[\mathbf{H}_i(\mathbf{x})\right] - \mathbf{E}_{\mathbf{x}\sim\mathcal{N}_k}\left[\mathbf{H}_i(\mathbf{x})\right]\right\|_{A'_y} = \left\|\mathbf{E}_{\mathbf{x}\sim A'_{\mathbf{x}|y}}\left[\mathbf{H}_i(\mathbf{x})\right]\right\|_{A'_y}$$

$$\le 2^i k^{i/4} B^i i^{-i/2}\exp\left(\binom{i}{2}/B^2\right) \le 2^{O(i)} k^{i/4} B^i i^{-i/2} \ ,$$

where the last inequality follows from $\binom{i}{2}/B^2 \le iT/B^2 \le id^{2\alpha}/B^2 = i$. Then, let $a$ be the largest even number such that $ai/2 \le T$, where $i \le T$ implies $a \ge 2$. Applying Fact C.10 yields

$$\mathbf{E}_{\mathbf{V}\sim U(\mathbf{O}_{d,k})}\left[\left\|\left(\mathbf{V}^\top\right)^{\otimes i}\mathbf{T}_i\right\|_{A'_y}^a\right] = O\left(2^{ai/4}(ai/2d)^{ai/4}\right) = O\left(\frac{d}{ai}\right)^{-ai/4} \ ,$$

which implies the tail bound

$$\mathbf{Pr}\left[\left\|\left(\mathbf{V}^\top\right)^{\otimes i}\mathbf{T}_i\right\|_{A'_y} \ge d^{-((1-\lambda)/4)i}(i^{i/4})\right] \le d^{-\lambda ai/4} \le 2^{-\Omega(T)} = 2^{-\Omega\left(d^{2\alpha}\right)} = 2^{-d^{\Omega(c)}} \ .$$

Therefore, we have

$$\sum_{i=d^\lambda+1}^{T}\left|\left\langle\mathbf{A}_i, \left(\mathbf{V}^\top\right)^{\otimes i}\mathbf{T}_i\right\rangle_{A'_y}\right| \le \sum_{i=d^\lambda+1}^{T}\|\mathbf{A}_i\|_{A'_y}\left\|\left(\mathbf{V}^\top\right)^{\otimes i}\mathbf{T}_i\right\|_{A'_y}$$

$$\le \sum_{i=d^\lambda+1}^{T} 2^{O(k)} k^{i/4} B^i i^{-i/4} d^{-((1-\lambda)/4)i}$$

$$\le \sum_{i=d^\lambda+1}^{T} 2^{O(k)} B^i d^{-((1-\lambda)/4)i}$$

$$= O\left(B^{d^\lambda+1} d^{-((1-\lambda)/4)\left(d^\lambda+1\right)}\right) = d^{-\Omega\left(cd^\lambda\right)} = d^{-\Omega(cm\log d)} \le d^{-m} \ ,$$

except with probability $2^{-d^{\Omega(c)}}$, where the third line follows from $i > d^\lambda > k$.

$\sum_{i=T+1}^{\ell}\left|\left\langle\mathbf{A}_i, \left(\mathbf{V}^\top\right)^{\otimes i}\mathbf{T}_i\right\rangle_{A'_y}\right|$ **is small with high probability:**

We will first need Fact 3.6 of [DKRS23] to bound $\|\mathbf{A}_i\|_{A'_y}$.

**Fact C.17** (Fact 3.6 of [DKRS23])**.** *Let $\mathbf{H}_i$ be the $i$-th Hermite tensor in $k$ dimensions. Then*

$$\|\mathbf{H}_i(\mathbf{x})\|_2 \le 2^{O(k)}\binom{i+k-1}{k-1}^{1/2}\exp(\|\mathbf{x}\|_2^2/4) \ .$$

Combining Fact C.17 with the fact that $A'$ is bounded inside $\mathbb{B}^k(B)$, we have that

$$\|\mathbf{A}_i\|_{A'_y} = \left\|\mathbf{E}_{\mathbf{x}\sim A'_{\mathbf{x}|y}}\left[\mathbf{H}_i(\mathbf{x})\right] - \mathbf{E}_{\mathbf{x}\sim\mathcal{N}^k}\left[\mathbf{H}_i(\mathbf{x})\right]\right\|_{A'_y} = \left\|\mathbf{E}_{\mathbf{x}\sim A'_{\mathbf{x}|y}}\left[\mathbf{H}_i(\mathbf{x})\right]\right\|_{A'_y}$$

$$\le 2^{O(k)}\binom{i+k-1}{k-1}^{1/2}\exp\left(B^2/4\right) \ .$$

We pick $a = 2$. Note that $ai/2 \geq T = d^{\max(2\alpha, \lambda)}$. Applying Fact C.10 yields

$$\mathbf{E}_{\mathbf{V} \sim U(\mathbf{O}_{d,k})} \left[ \left\| (\mathbf{V}^\top)^{\otimes i} \mathbf{T}_i \right\|_{A'_y}^a \right] = \exp\left(-\Omega\left(T \log d\right)\right) O\left( \left( \frac{T+d}{i+d} \right)^{(d-k)/2} \right).$$

Applying Markov's inequality yields the tail bound

$$\mathbf{Pr} \left[ \left\| (\mathbf{V}^\top)^{\otimes i} \mathbf{T}_i \right\|_{A'_y} \geq 2^{-\Omega(T \log d)} O\left( \left( \frac{T+d}{i+d} \right)^{(d-k)/5} \right) \right] \leq \left( \frac{T+d}{i+d} \right) 2^{-d^{\Omega(c)}}.$$

Therefore, we have

$$\sum_{i=T+1}^{\ell} \left| \left\langle \mathbf{A}_i, (\mathbf{V}^\top)^{\otimes i} \mathbf{T}_i \right\rangle_{A'_y} \right|$$

$$\leq \sum_{i=T+1}^{\infty} \left| \left\langle \mathbf{A}_i, (\mathbf{V}^\top)^{\otimes i} \mathbf{T}_i \right\rangle_{A'_y} \right| \leq \sum_{i=T+1}^{\infty} \| \mathbf{A}_i \|_{A'_y} \left\| (\mathbf{V}^\top)^{\otimes i} \mathbf{T}_i \right\|_{A'_y}$$

$$\leq \sum_{i=T+1}^{\infty} 2^{O(k)} \binom{i+k-1}{k-1}^{1/2} \exp\left(B^2/4\right) 2^{-\Omega(T \log d)} O\left( \left( \frac{T+d}{i+d} \right)^{(d-i)/5} \right)$$

$$\leq \sum_{i=T}^{\infty} 2^{-\Omega(T \log d)} \binom{T+k}{k}^{1/2} \left( \frac{i+k}{T+k} \right)^{k/2} \left( \frac{T+d}{i+d} \right)^{(d-i)/5}$$

$$\leq \sum_{i=T}^{\infty} 2^{-\Omega(T \log n)} \left( \frac{i+k}{T+k} \right)^{k/2} \left( \frac{T+d}{i+d} \right)^{d/8},$$

where the last inequality follows from our choice of parameters. Therefore, we have that

$$\sum_{i=T+1}^{\ell} \left| \left\langle \mathbf{A}_i, (\mathbf{V}^\top)^{\otimes i} \mathbf{T}_i \right\rangle_{A'_y} \right| \leq \sum_{i=T}^{\infty} 2^{-\Omega(T \log d)} \left( 1 + \frac{i-T}{T+k} \right)^{k/2} \left( 1 + \frac{i-T}{T+d} \right)^{-d/8}$$

$$\leq \sum_{i=T}^{\infty} 2^{-\Omega(T \log d)} \left( 1 + \frac{i-T}{T+d} \right)^{(k/2)(2d/T)} \left( 1 + \frac{i-T}{T+d} \right)^{-d/8}$$

$$\leq \sum_{i=T}^{\infty} 2^{-\Omega(T \log d)} \left( 1 + \frac{i-T}{T+d} \right)^{-d/8+dk/T}$$

$$\leq \sum_{i=T}^{\infty} 2^{-\Omega(T \log d)} \left( \frac{d+i}{T+d} \right)^{-d/16}$$

$$\leq 2^{-\Omega(T \log d)} \int_{i=T-1}^{\infty} \left( \frac{d+i}{T+d} \right)^{-d/16} di$$

$$= 2^{-\Omega(T \log d)} \frac{(T+d)^{-d/16}}{(d/16-1)(T+d-1)^{d/16-1}}$$

$$= 2^{-\Omega(d^{2\alpha})},$$

except with probability $\sum_{i=T}^{\infty} \left(\frac{T+d}{i+d}\right) 2^{-d^{\Omega(c)}} = 2^{-d^{\Omega(c)}}$. Adding the four cases above together, we get for any $m, k \leq d^{\lambda}/\log d$ and $d$ at least a sufficiently large constant depending on $c$,

$$\sum_{i=1}^{\ell} \left| \left\langle \mathbf{A}_i, \left(\mathbf{V}^{\top}\right)^{\otimes i} \mathbf{T}_i \right\rangle_{A'_y} \right|$$

$$\leq (1+o(1)) \left( 2^{m/2} \sqrt{\frac{\Gamma(m+k/2)}{\Gamma(k/2)}} d^{-\alpha m} + \nu \right) + d^{-((1-\lambda_3)/4)m} + d^{-m} + 2^{-\Omega\left(d^{2\alpha}\right)}$$

$$\leq \left( \frac{\Gamma(m/2+k/2)}{\Gamma(k/2)} \right) d^{-((1-\lambda)/4-c/2)m} + (1+o(1))\nu$$

$$= \left( \frac{\Gamma(m/2+k/2)}{\Gamma(k/2)} \right) d^{-((1-\lambda)/4-c)m} + (1+o(1))\nu \,,$$

except with probability $2^{-d^{\Omega(c)}}$, where the second line above follows from $\frac{\Gamma(m/2+k/2)}{\Gamma(k/2)} \geq 2^{m/2}\sqrt{\frac{\Gamma(m+k/2)}{\Gamma(k/2)}}$. This completes the proof of Lemma C.15. $\qquad\square$

### C.1.2 Proof of Proposition C.7

We are now ready to prove Proposition C.7 which is the main technical ingredient of our lower bound. Proposition C.7 states that $\left| \mathbf{E}_{(\mathbf{x},y)\sim\mathbf{P}_{\mathbf{U}}^A}[f(\mathbf{x},y)] - \mathbf{E}_{(\mathbf{x},y)\sim\mathcal{N}_d\otimes A_y}[f(\mathbf{x},y)] \right|$ is small with high probability. The main idea of the proof is to use Fourier analysis on $\mathbf{E}_{(\mathbf{x},y)\sim\mathbf{P}_{\mathbf{U}}^{A'}}[f(\mathbf{x},y)]$ as we discussed in the last section, where $A'$ is the distribution obtained by truncating and reweighting $A$ (see Definition C.12) and is close to $A$ in total variation distance.

*Proof of Proposition C.7.* For convenience, we let $\zeta = (1-\lambda)/4 - c$. We will first truncate and reweigh $A$, as defined in Definition C.12 and then apply Lemma C.15. Notice that Lemma C.15 additionally assumes $m, k \leq d^{\lambda}/\log d$, $\nu < 2$ and $\left( \frac{\Gamma(m/2+k/2)}{\Gamma(k/2)} \right) d^{-\zeta m} < 2$. We show that all these three conditions can be assumed true without loss of generality. If either the second or the third condition is not true, then our lower bound here is trivialized and is always true since $f$ is bounded between $[-1, +1]$. For $m, k \leq d^{\lambda}/\log d$, consider a $\lambda' > \lambda$ such that $(1-\lambda')/4 - \zeta = \frac{(1-\lambda)/4-\zeta}{2}$. Then it is easy to see that for any sufficiently large $d$ depending on $(1-\lambda)/4 - \zeta$, we have $m, k \leq d^{\lambda'}/\log d$ and $\zeta \leq (1-\lambda)/4 - \zeta$. Therefore, we can apply Lemma C.15 for $\lambda'$.

Now let $B = d^{\alpha}$, where $\alpha < (1-\lambda)/4$ is the constant in Lemma C.15. Then we consider the truncated and reweighted distribution $A'$, as defined in Definition C.12. By Lemma C.15, we have $d_{\mathrm{TV}}(A, A') \leq \left( \frac{\Gamma(m/2+k/2)}{\Gamma(k/2)} \right) d^{-\zeta m}$. Given that $f$ is bounded between $[-1, 1]$, this implies

$$\left| \mathbf{E}_{(\mathbf{x},y)\sim\mathbf{P}_{\mathbf{U}}^A}[f(\mathbf{x},y)] - \mathbf{E}_{(\mathbf{x},y)\sim\mathbf{P}_{\mathbf{U}}^{A'}}[f(\mathbf{x},y)] \right|$$

$$\leq d_{\mathrm{TV}}(\mathbf{P}_{\mathbf{U}}^A, \mathbf{P}_{\mathbf{U}}^{A'}) = d_{\mathrm{TV}}(A, A') \leq \left( \frac{\Gamma(m/2+k/2)}{\Gamma(k/2)} \right) d^{-\zeta m} \,.$$

Similarly, we have

$$\left| \mathbf{E}_{(\mathbf{x},y)\sim\mathcal{N}_d\otimes A_y}[f(\mathbf{x},y)] - \mathbf{E}_{(\mathbf{x},y)\sim\mathcal{N}_d\otimes A'_y}[f(\mathbf{x},y)] \right|$$

$$\leq d_{\mathrm{TV}}(\mathcal{N}_d\otimes A_y, \mathcal{N}_d\otimes A'_y) = d_{\mathrm{TV}}(A_y, A'_y) \leq d_{\mathrm{TV}}(A, A') \leq \left( \frac{\Gamma(m/2+k/2)}{\Gamma(k/2)} \right) d^{-\zeta m} \,.$$

Therefore, by the triangle inequality, it suffices for us to analyze the difference $\left| \mathbf{E}_{(\mathbf{x},y)\sim\mathbf{P}_{\mathbf{U}}^{A'}}[f(\mathbf{x},y)] - \mathbf{E}_{(\mathbf{x},y)\sim\mathcal{N}_d\otimes A'_y}[f(\mathbf{x},y)] \right|$ instead of the difference $\left| \mathbf{E}_{(\mathbf{x},y)\sim\mathbf{P}_{\mathbf{U}}^A}[f(\mathbf{x},y)] - \mathbf{E}_{(\mathbf{x},y)\sim\mathcal{N}_d\otimes A_y}[f(\mathbf{x},y)] \right|$.

Let $\ell = \ell_f(d) \in \mathbb{N}$ be a function depending only on the query function $f$ and the dimension $d$ ($\ell$ to be specified later). By Lemma C.8, we have that

$$\mathbf{E}_{(\mathbf{x},y)\sim\mathbf{P}_{\mathbf{U}}^{A'}}[f(\mathbf{x},y)] = \sum_{i=0}^{\ell} |\langle \mathbf{A}_i, (\mathbf{U}^\top)^{\otimes i}\mathbf{T}_i\rangle_{A_y'}| + \mathbf{E}_{(\mathbf{x},y)\sim\mathbf{P}_{\mathbf{U}}^{A'}}[f^{>\ell}(\mathbf{x},y)] \,,$$

where $\mathbf{A}_i(y) = \mathbf{E}_{\mathbf{x}\sim A_{\mathbf{x}|y}}[\mathbf{H}_i(\mathbf{x})]$ and $\mathbf{T}_i(y) = \mathbf{E}_{\mathbf{x}\sim\mathcal{N}_d}[f(\mathbf{x},y)\mathbf{H}_i(\mathbf{x})]$ and $f^{>\ell}(\mathbf{x},y) = (f(\cdot,y))^{>\ell}(\mathbf{x})$. Recall that we want to bound

$$\left|\mathbf{E}_{(\mathbf{x},y)\sim\mathbf{P}_{\mathbf{U}}^{A'}}[f(\mathbf{x},y)] - \mathbf{E}_{(\mathbf{x},y)\sim\mathcal{N}_d\otimes A_y'}[f(\mathbf{x},y)]\right|$$

with high probability, where we note that $\mathbf{E}_{(\mathbf{x},y)\sim\mathcal{N}_d\otimes A_y'}[f(\mathbf{x},y)] = \langle \mathbf{A}_0, \mathbf{T}_0\rangle_{A_y'}$. Therefore, we can write $\left|\mathbf{E}_{(\mathbf{x},y)\sim\mathbf{P}_{\mathbf{U}}^{A'}}[f(\mathbf{x},y)] - \mathbf{E}_{(\mathbf{x},y)\sim\mathcal{N}_d\otimes A_y'}[f(\mathbf{x},y)]\right| \leq \left|\sum_{i=1}^{\ell}\langle \mathbf{A}_i, (\mathbf{U}^\top)^{\otimes i}\mathbf{T}_i\rangle_{A_y'}\right| + \left|\mathbf{E}_{(\mathbf{x},y)\sim\mathbf{P}_{\mathbf{U}}^{A'}}[f^{>\ell}(\mathbf{x},y)]\right|$. For the first term, by Lemma C.15, we have that

$$\left|\sum_{i=1}^{\ell}\langle \mathbf{A}_i, (\mathbf{U}^\top)^{\otimes i}\mathbf{T}_i\rangle_{A_y'}\right| = \left(\frac{\Gamma(m/2+k/2)}{\Gamma(k/2)}\right)d^{-\zeta m} + (1+o(1))\nu \,,$$

except with probability $2^{-d^{\Omega(c)}}$.

It now remains for us to show that $\left|\mathbf{E}_{(\mathbf{x},y)\sim\mathbf{P}_{\mathbf{U}}^{A'}}[f^{>\ell}(\mathbf{x},y)]\right|$ is also small with high probability.

Consider the distribution $D = \mathbf{E}_{\mathbf{v}\sim U(\mathbf{O}_{d,k})}[\mathbf{P}_{\mathbf{U}}^{A'}]$. We then use Lemma 3.11 of [DKRS23] to show that $D_{\mathbf{x}|y}$ is continuous for any $y$ and $\chi^2(D, \mathcal{N}_d\otimes A_y')$ is at most a constant only depending on $d$ (independent of the choice of the distribution $A$).

**Fact C.18** (Lemma 3.11 of [DKRS23]). *Let $A$ be any distribution supported on $\mathbb{B}^k(d)$ for $d\in\mathbb{Z}_+$ which is at least a sufficiently large universal constant. Let $D = \mathbf{E}_{\mathbf{U}\sim U(\mathbf{O}_{d,k})}[\mathbf{P}_{\mathbf{U}}^A]$. Then, $D$ is a continuous distribution and $\chi^2(D, \mathcal{N}_d) = O_d(1)$.*

Now for our regular distribution $A'$ supported on $\mathbb{B}^k(d)\times\mathcal{Y}$, by applying Fact C.18 for each $A_{\mathbf{x}|y}$, we get that

$$\chi^2\left(\mathbf{E}_{\mathbf{U}\sim U(\mathbf{O}_{d,k})}\left[\mathbf{P}_{\mathbf{U}}^{A'}\right], \mathcal{N}_d\otimes A_y'\right) = \left\|\chi^2\left(\mathbf{E}_{\mathbf{U}\sim U(\mathbf{O}_{d,k})}\left[\mathbf{P}_{\mathbf{U}}^{A'_{\mathbf{x}|y}}\right], \mathcal{N}_d\right)\right\|_{A_y'}^2 = O(1) \,.$$

Therefore, we have that

$$\begin{aligned}
\mathbf{E}_{\mathbf{U}\sim U(\mathbf{O}_{d,k})}\left[\left|\mathbf{E}_{(\mathbf{x},y)\sim\mathbf{P}_{\mathbf{U}}^{A'}}[f^{>\ell}(\mathbf{x},y)]\right|\right] &\leq \mathbf{E}_{\mathbf{U}\sim U(\mathbf{O}_{d,k})}\left[\mathbf{E}_{(\mathbf{x},y)\sim\mathbf{P}_{\mathbf{U}}^{A'}}[|f^{>\ell}(\mathbf{x},y)|]\right] \\
&\leq \mathbf{E}_{(\mathbf{x},y)\sim D}[|f(\mathbf{x},y)^{>\ell}|] \\
&\leq \chi^2(D, \mathcal{N}_d\otimes A_y')^{1/2}\|f^{>\ell}\|_{\mathcal{N}_d\otimes A_y'} \\
&\leq \delta(d)\|f^{>\ell}\|_{\mathcal{N}_d\otimes A_y'} \,.
\end{aligned}$$

We can take $\ell = \ell_f(d)$ ($\ell$ only depends on the query function $f$ and dimension $d$) to be a sufficiently large function such that $\|f^{>\ell}\|_{\mathcal{N}_d\otimes A_y'} \leq \left(\frac{2^{-d}}{\delta(d)}\right)\left(\frac{\Gamma(m/2+k/2)}{\Gamma(k/2)}\right)d^{-\zeta m}$. Then we get

$$\mathbf{E}_{\mathbf{U}\sim U(\mathbf{O}_{d,k})}\left[\left|\mathbf{E}_{(\mathbf{x},y)\sim\mathbf{P}_{\mathbf{U}}^{A'}}[f^{>\ell}(\mathbf{x},y)]\right|\right] \leq \delta(d)\|f^{>\ell}\|_{\mathcal{N}_d\otimes A_y'} \leq 2^{-d}\left(\frac{\Gamma(m/2+k/2)}{\Gamma(k/2)}\right)d^{-\zeta m} \,.$$

This gives the tail bound $\mathbf{Pr}_{\mathbf{U}\sim U(\mathbf{O}_{d,k})}\left[\left|\mathbf{E}_{(\mathbf{x},y)\sim\mathbf{P}_{\mathbf{U}}^{A'}}[f^{>\ell}(\mathbf{x},y)]\right| \geq \left(\frac{\Gamma(m/2+k/2)}{\Gamma(k/2)}\right)d^{-\zeta m}\right] \leq 2^{-d}$.

Using the above upper bounds, we have

$$\begin{aligned}
\left|\mathbf{E}_{(\mathbf{x},y)\sim\mathbf{P}_{\mathbf{U}}^{A'}}[f(\mathbf{x})] - \mathbf{E}_{(\mathbf{x},y)\sim\mathcal{N}_d\otimes A_y'}[f(\mathbf{x})]\right| &\leq \left|\sum_{i=1}^{\ell}\langle \mathbf{A}_i, (\mathbf{U}^\top)^{\otimes i}\mathbf{T}_i\rangle\right| + |\mathbf{E}_{\mathbf{x}\sim\mathbf{P}_{\mathbf{U}}^{A'}}[f^{>\ell}(\mathbf{x})]| \\
&= 2\left(\frac{\Gamma(m/2+k/2)}{\Gamma(k/2)}\right)d^{-\zeta m} + (1+o(1))\nu \,,
\end{aligned}$$

except with probability $2^{-d^{\Omega(c)}}$ using the fact that $c = O(1)$. Therefore,

$$|\mathbf{E}_{(\mathbf{x},y)\sim\mathbf{P}_{\mathbf{U}}^A}[f(\mathbf{x})] - \mathbf{E}_{(\mathbf{x},y)\sim\mathcal{N}_d\otimes A_y}[f(\mathbf{x})]| \leq 6\left(\frac{\Gamma(m/2+k/2)}{\Gamma(k/2)}\right)d^{-\zeta m} + (1+o(1))\nu\ ,$$

except with probability $2^{-d^{\Omega(c)}}$.

In summary, notice that the above argument remains true if we take $\zeta' > \zeta$ such that $(1-\lambda)/4 - \zeta' = \frac{(1-\lambda)/4-\zeta}{2}$. Using the above argument for $\zeta'$, and given $d$ is a sufficiently large constant depending on $(1-\lambda)/4 - \zeta = 2((1-\lambda)/4 - \zeta')$, we get

$$|\mathbf{E}_{(\mathbf{x},y)\sim\mathbf{P}_{\mathbf{U}}^A}[f(\mathbf{x})] - \mathbf{E}_{(\mathbf{x},y)\sim\mathcal{N}_d\otimes A_y}[f(\mathbf{x})]| \leq \left(\frac{\Gamma(m/2+k/2)}{\Gamma(k/2)}\right)d^{-\zeta m} + (1+o(1))\nu\ ,$$

except with probability $2^{-d^{\Omega((1-\lambda)/4-\zeta')}} = 2^{-d^{\Omega(c)}}$. Replacing $\zeta$ with $(1-\lambda)/4 - c$ completes the proof of Proposition C.7]. $\qquad\square$

## C.2 SQ Lower Bounds for Learning Multi-index Models

In this section, we prove our SQ lower bound for learning Multi-index Models, as an application of Theorem C.6. We first give the formal statement of Theorem 1.10 below.

**Theorem C.19** (SQ Lower Bound for Learning $K$-MIMs; Formal Version of Theorem 1.10). *Let $\mathcal{C}$ be a class of rotationally invariant $K$-MIMs on $\mathbb{R}^d$. Suppose there exist $m \in \mathbb{Z}_+$, $\tau > 0$, and a joint distribution $D$ of $(\mathbf{x}, y)$ supported on $\mathbb{R}^d \times \mathbb{R}$ with $D_{\mathbf{x}}$ equal to $\mathcal{N}_d$ such that for some subspace $V \subseteq \mathbb{R}^d$, we have:*

1. *The distribution $D$ $\nu$-matches degree-$m$ moments relative to the subspace $V \times \mathbb{R}$, where the extra $\mathbb{R}$ contains the label;*
2. *Any function $h : \mathbb{R}^d \to \mathbb{R}$ has $\mathbf{E}_{(\mathbf{x},y)\sim D}[(h(\mathbf{x}_V) - y)^2] \geq \tau$; and*
3. *There exist $B, \delta \in \mathbb{R}_+$ such that $\mathbf{E}_y[y^2 \mathbb{1}(|y| > B)] \leq \delta$.*

*Then, for $m, K \leq d^\lambda$ for some $\lambda \in (0,1)$, $\dim(V) \leq d/2$, $c \in (0, (1-\lambda)/4)$ and $d$ at least a sufficiently large constant depending on $c$, the following holds: any SQ algorithm that learns $\mathcal{C}$ within error $\tau - 7\delta - 3\zeta B^2$ given $\mathrm{OPT} \leq \inf_{c\in\mathcal{C}} \mathrm{err}_D(c)$ requires either a query to $\mathrm{STAT}(\zeta)$ or $2^{d^{\Omega(c)}}$ many queries, where $\zeta = O_{K,m}\left(d^{-((1-\lambda)/4-c)m}\right) + (1+o(1))\nu$.*

Some comments regarding the difference between Theorem 1.10 and Theorem C.19 are in order here. We first note that Condition (1) in Theorem C.19 generalizes Condition (1) in Theorem 1.10 with approximate moment matching, as defined in Condition C.5. We then note that Condition (3) in Theorem C.19 is required for technical reasons, namely assuming that the extreme values of $y$ (i.e., $|y| > B$) have contribution at most $\delta$ to the variance. Without such a condition, it is possible that almost all the variance of the label comes from an arbitrarily small mass of the input distribution. For most applications, we will have $B = O(LK^{1/2}\omega(d))$ and $\delta = 2^{-\omega(d)}$, where $L$ is the Lipschitzness of the functions in the concept class. Under such circumstances, Theorem C.19 rules out any algorithm that outperforms the best function in subspace $V$ by some additive factor of $o(1)$ (with respect to $d$).

*Proof of Theorem C.19.* The proof follows directly by embedding an RNGCA problem to agnostic PAC learning of the class $\mathcal{C}$. Let $A'$ be the distribution $D$ supported on $\mathbb{R}^d \times \mathbb{R}$ in Theorem C.19 and $W$ be the $K$-dimensional relevant subspace of a $K$-MIM $c \in \mathcal{C}$ that minimizes the error $\mathrm{err}_{A'}(c)$. Let $V$ be the subspace satisfying the conditions in Theorem C.19 and $U = W_{V^\perp}$, where $W_{V^\perp} \overset{\text{def}}{=} \{\mathbf{w}_{V^\perp} : \mathbf{w} \in W\}$. Let $\mathbf{U} \in \mathbb{R}^{d\times\dim(U)}$ and $\mathbf{V} \in \mathbb{R}^{d\times\dim(V)}$ be matrices whose column vectors are arbitrary orthonormal basis vectors that span the subspaces $U$ and $V$ respectively.

Let $(\mathbf{x}, y) \sim A'$. We define the distribution $A$ for RNGCA (Definition 3.2) as the joint distribution of $(\mathbf{x}', (\mathbf{x}'', y))$ over $\mathbb{R}^{\dim(U)} \times \mathbb{R}^{\dim(V)+1}$ (with $\mathcal{Y} = \mathbb{R}^{\dim(V)+1}$), where $\mathbf{x}' = \mathbf{U}^\top\mathbf{x}$ and $\mathbf{x}'' = \mathbf{V}^\top\mathbf{x}$, i.e., $\mathbf{x}'$ contains the part of the relevant subspace (of the optimal hypothesis) outside $V$ and $(\mathbf{x}'', y)$ contains $V$ and the label $y$. Then we consider the RNGCA problem of Definition C.2 with hidden distribution $A$ and input distribution supported on $\mathbb{R}^{d-\dim(V)} \times \mathbb{R}^{\dim(V)+1}$. By Condition 1 in Theorem C.19 and Theorem C.6, by choosing the parameters $\lambda, c$ in Theorem C.6 to be the same

as the parameters $\lambda, c$ in Theorem C.19, we have that any SQ algorithm that solves this RNGCA problem must use either a query to $\text{STAT}(\zeta)$ or $2^{(d-\dim(V))^{\Omega(c)}} = 2^{(d)^{\Omega(c)}}$ many queries, where

$$\zeta = O_{\dim(U),m}\left((d-\dim(V))^{-((1-\lambda)/4-c)m}\right) + (1+o(1))\nu$$

$$= O_{K,m}\left(d^{-((1-\lambda)/4-c)m}\right) + (1+o(1))\nu .$$

Therefore, we just need to show that the $K$-MIM learning algorithm described in Theorem C.19 can solve this RNGCA problem.

Let $\mathcal{A}$ be such an algorithm for learning Multi-index models and $D'$ be the input distribution of $(\mathbf{x}', \mathbf{y}')$ supported on $\mathbb{R}^{d-\dim(V)} \times \mathbb{R}^{\dim(V)+1}$ for the RNGCA problem. First notice that $D'$ can be equivalently thought of as a labeled distribution supported on $\mathbb{R}^d \times \mathbb{R}$, where we treat the coordinate corresponding to the $y$ part as the label. Namely, we define the new input distribution $D$ as the joint distribution of $(\mathbf{x}, y)$ supported on $\mathbb{R}^d \times \mathbb{R}$, where $\mathbf{x}$ contains $\mathbf{x}'$ and all except the last coordinate of $\mathbf{y}'$ and $y$ is the last coordinate of $\mathbf{y}'$. We then give $D$ as the input distribution to the algorithm $\mathcal{A}$ (notice that any SQ query on $D$ can be answered with an SQ query on $D'$). Let $h : \mathbb{R}^d \to \mathbb{R}$ be the output hypothesis of the algorithm. Then we will check the value of $\mathbf{E}_{(\mathbf{x},y)\sim D}[(h(\mathbf{x}) - y)^2 \mathbb{1}(y \in [-B, B])]$ to error at most $\zeta B^2$. Notice that this can be done by using the query function $q(\mathbf{x}, y) = (h(\mathbf{x}) - y)^2 \mathbb{1}(y \in [-B, B])/B^2$ with query tolerance $\zeta$.

Now suppose that the original $D'$ is from the null hypothesis distribution. Then, by the definition of RNGCA, we have that for $(\mathbf{x}', \mathbf{y}') \sim D'$, $\mathbf{x}'$ and $\mathbf{y}'$ are independent and $\mathbf{y}'$ has the same distribution as the marginal distribution of $A_{\mathbf{y}'}$, which is the marginal distribution of $(\mathbf{V}^\top \mathbf{x}, y)$ for $(\mathbf{x}, y) \sim A'$. Notice that for any $h$ the algorithm satisfies that

$$\mathbf{E}_{(\mathbf{x},y)\sim D}[(h(\mathbf{x}) - y)^2 \mathbb{1}(y \in [-B, B])] \geq \min_{g:\mathbb{R}^k \to \mathbb{R}} \mathbf{E}_{(\mathbf{x},y)\sim A'}[(g(\mathbf{V}^\top \mathbf{x}) - y)^2 \mathbb{1}(y \in [-B, B])] .$$

To bound this quantity, we will use the following fact, which states that if the squared error of a function $f$ is large and the labels outside of $[-B, B]$ only have bounded variance, then there is a lower bound on the squared error of $f$ on the labels inside $[-B, B]$.

**Fact C.20.** *Let $A$ be a joint distribution of $(\mathbf{x}, y)$ over $X \times \mathbb{R}$ such that for any function $f : X \to \mathbb{R}$, $\mathbf{E}_{(\mathbf{x},y)\sim A}[(y - f(\mathbf{x}))^2] \geq \tau$ and $\mathbf{E}_{y\sim A_y}[y^2 \mathbb{1}(|y| > B)] \leq \delta$. Then for any $g : X \to \mathbb{R}$ we have $\mathbf{E}_{(\mathbf{x},y)\sim A}[(y - g(\mathbf{x}))^2 \mathbb{1}(y \in [-B, B])] \geq \tau - 7\delta$ .*

*Proof of Fact C.20.* Notice that

$$\min_{g:X \to \mathbb{R}} \mathbf{E}_{(\mathbf{x},y)\sim A}[(g(\mathbf{x}) - y)^2 \mathbb{1}(y \in [-B, B])]$$

$$= \mathbf{E}_{\mathbf{x}\sim A_\mathbf{x}}\left[\mathbf{Pr}_{y\sim A_{y|\mathbf{x}}}[y \in [-B, B]] \mathbf{Var}(A_{y|\mathbf{x}\wedge y\in[-B,B]})\right] .$$

Therefore, we just need to consider the distribution of $A_{y|\mathbf{x}}$. For convenience of analysis, we give the following intermediate fact.

**Fact C.21.** *Let $D$ be a distribution of $y$ over $\mathbb{R}$ such that $\mathbf{Var}(D) \geq \tau$ and $\mathbf{E}_{y\sim D}[\mathbb{1}(y \notin [-B, B])y^2] \leq \delta$. Then $\mathbf{Pr}_{y\sim D}[y \in [-B, B]]\mathbf{Var}(D \mid y \in [-B, B]) \geq \tau - 7\delta$.*

*Proof of Fact C.21.* Applying the law of total variance and the fact that $\mathbf{Pr}_{y\sim D}[y \notin [-B, B]] \leq \delta/B^2$, we have that

$$\mathbf{Pr}_{y\sim D}[y \in [-B, B]]\mathbf{Var}(D \mid y \in [-B, B])$$

$$= \mathbf{Var}(D) - \mathbf{Pr}_{y\sim D}[y \notin [-B, B]]\mathbf{Var}(D \mid y \notin [-B, B])$$

$$\quad - \mathbf{Pr}_{y\sim D}[y \in [-B, B]]\mathbf{Pr}_{y\sim D}[y \notin [-B, B]] \left(\mathbf{E}_{y\sim D|y\in[-B,B]}[y] - \mathbf{E}_{y\sim D|y\notin[-B,B]}[y]\right)^2$$

$$\geq \tau - \delta - \mathbf{Pr}_{y\sim D}[y \notin [-B, B]]\mathbf{E}_{y\sim D|y\in[-B,B]}[y]^2$$

$$\quad - 2\mathbf{Pr}_{y\sim D}[y \notin [-B, B]]\mathbf{E}_{y\sim D|y\in[-B,B]}[y]\mathbf{E}_{y\sim D|y\notin[-B,B]}[y]$$

$$\quad - \mathbf{Pr}_{y\sim D}[y \notin [-B, B]]\mathbf{E}_{y\sim D|y\notin[-B,B]}[y]^2$$

$$\geq \tau - \delta - \mathbf{Pr}_{y\sim D}[y \notin [-B, B]]B^2 - 2\left|\mathbf{E}_{y\sim D}[\mathbb{1}(y \notin [-B, B])y]B\right| - \mathbf{E}_{y\sim D}[\mathbb{1}(y \notin [-B, B])y^2]$$

$$\geq \tau - 3\delta - 2\left|\mathbf{E}_{y\sim D}[\mathbb{1}(y \notin [-B, B])y]B\right| .$$

So it only remains to bound $|\mathbf{E}_{y\sim D}[\mathbb{1}(y \notin [-B, B])y]|$. Notice that by Markov's inequality, we have

$$|\mathbf{E}_{y\sim D}[\mathbb{1}(y \notin [-B, B])y]| \leq \mathbf{E}_{y\sim A}[|y|\mathbb{1}(y \notin [-B, B])] \leq \int_0^B \delta/B^2 dt + \int_0^\infty \delta/t^2 dt \leq 2\delta/B \ .$$

Plugging it back gives $\mathbf{Pr}_{y\sim D}[y \in [-B, B]]\mathbf{Var}(D \mid y \in [-B, B]) \geq \tau - 7\delta$. This completes the proof of Fact C.21. $\qquad \square$

Now, using Fact C.21, we get

$$\min_{g:X\to\mathbb{R}} \mathbf{E}_{(\mathbf{x},y)\sim A}[(g(\mathbf{x}) - y)^2\mathbb{1}(y \in [-B, B])]$$
$$=\mathbf{E}_{\mathbf{x}\sim A_\mathbf{x}}\left[\mathbf{Pr}_{y\sim A_{y|\mathbf{x}}}[y \in [-B, B]]\mathbf{Var}(A_{y|\mathbf{x}\wedge y\in[-B,B]})\right]$$
$$\geq\mathbf{E}_{\mathbf{x}\sim A_\mathbf{x}}\left[\mathbf{Var}(A_{y|\mathbf{x}}) - 7\mathbf{E}_{y\sim A_{y|\mathbf{x}}}[\mathbb{1}(y \notin [-B, B])y^2]\right]$$
$$\geq\tau - 7\delta \ .$$

This completes the proof of Fact C.20. $\qquad \square$

Applying Fact C.20 gives that

$$\mathbf{E}_{(\mathbf{x},y)\sim D}[(h(\mathbf{x}) - y)^2\mathbb{1}(y \in [-B, B])] \geq \tau - 7\delta \ ,$$

if the original $D'$ is from the null hypothesis case.

Now suppose that the original $D'$ is from the alternative hypothesis. Then it is immediate that $D$ is the same as the product distribution $\mathcal{N}^{d-\dim(V)-\dim(U)} \otimes A$ up to a rotation in the first $d - \dim(V)$ coordinates. From the definition of $A$ ($A$ contains the part of $\mathbf{x}$ in the optimal relevant subspace $W$) and $\mathcal{C}$ being rotation-invariant, we must have $\inf_{c\in\mathcal{C}} \mathrm{err}_{A'}(c) = \inf_{c\in\mathcal{C}} \mathrm{err}_D(c)$. Therefore, from the definition of $\mathcal{A}$, we must have

$$\mathbf{E}_{(\mathbf{x},y)\sim D}[(h(\mathbf{x}) - y)^2\mathbb{1}(y \in [-B, B])] \leq \mathbf{E}_{(\mathbf{x},y)\sim D}[(h(\mathbf{x}) - y)^2] \leq \tau - 7\delta - 3\zeta B^2 \ .$$

Given the analysis above, we can simply check if our estimate of $\mathbf{E}_{(\mathbf{x},y)\sim D}[(h(\mathbf{x}) - y)^2\mathbb{1}(y \in [-B, B])]$ (which has error at most $\tau B^2$) is greater than $\tau - 7\delta - 3\zeta B^2/2$. If so, then it must be from the null hypothesis. Otherwise, it must be from the alternative hypothesis. This completes the proof of Theorem C.19. $\qquad \square$

## C.3 Relation between Our Result and Other Complexity Measures

In this section, we discuss the relationship between our conditions on efficient learnability of MIMs and other complexity measures.

As noted in the related work section, prior work [ABAM23] defined the notion of leap complexity and showed that it characterizes the CSQ complexity of learning hidden junta functions over the uniform distributions on the Boolean hypercube (these are discrete Multi-index models). We remind the reader that CSQ lower bounds are in general strictly weaker compared to SQ lower bounds.

For the special case of SIMs under the Gaussian distribution, [DPLB24] defined the notion of generative exponent and showed that it essentially characterizes the complexity of parameter estimation. It is important to remark that our work focuses on the related but distinct notion of (agnostic) PAC learning, i.e., learning the label distribution to small error. Indeed, PAC learning can be feasible even when parameter estimation is not. For instance, if we consider distributions where the conditional distribution $y \mid \mathbf{x}$ is constant for all $\mathbf{x}$, then parameter estimation is impossible (and the generative exponent becomes infinity). In this case, while it is information-theoretically impossible to recover the hidden direction, it is trivial to output a hypothesis achieving small squared error.

The structure of this section is as follows: In Section C.3.1, we show that our techniques imply as a corollary the main SQ lower bound of [DPLB24] for Single-Index models without a finite chi-squared condition implicitly used in their work. In Section C.3.2, we show that for realizable PAC learning of Singe-Index models, our condition is essentially equivalent to an appropriate adaptation of the generative exponent.

### C.3.1 SQ Lower Bound for Parameter Estimation of SIMs

We begin by providing the definition of the generative exponent for parameter recovery of SIMs. For convenience of the discussion, all the statements presented in this section are simplified for exact moment matching, which—while qualitative the same as the full statements—will be weaker quantitatively in some parameters.

**Definition C.22** (Generative Exponent). For realizable learning of Single-Index models under the Gaussian distribution, we define the Generative Exponent of the link function $f : \mathbb{R} \to \mathbb{R}$ as the smallest $m^* \in \mathbb{Z}_+$ such that $\|\mathbf{E}_{t \sim A_{t|y}}[h_{m^*}(t)]\|_{A_y} > 0$ , where $A$ is the joint distribution of $(t, y)$ on $\mathbb{R} \times \mathbb{R}$ with $t \sim \mathcal{N}_1$ and $y = f(t)$, and $h_i$ is the $i$-th normalized Hermite polynomial. For a SIM $g : \mathbb{R}^d \to \mathbb{R}$ defined as $g(\mathbf{x}) = f(\mathbf{w} \cdot \mathbf{x})$ for some vector $\mathbf{w} \in \mathbb{R}^d$ and link function $f : \mathbb{R} \to \mathbb{R}$, we define the generative exponent of $g$ to be that of the link function $f$.

[DPLB24] gives the following lower bound on the problem.

**Fact C.23** (Theorem 3.2 of [DPLB24]). *Let $\mathbf{w} \in \mathbb{S}^{d-1}$ be a 1-dimensional subspace unknown to the algorithm and $D$ be a joint distribution of $(\mathbf{x}, y)$ over $\mathbb{R}^d \times \mathbb{R}$ such that $\mathbf{x} \sim \mathcal{N}_d$ and $y$ only depends on $\mathbf{x_w}$ (i.e., $D_{y|\mathbf{x}} = D_{y|\mathbf{x}'}$ for any $\mathbf{x_w} = \mathbf{x}'_\mathbf{w}$). Let $m^*$ be the generative exponent for the joint distribution of $(\mathbf{x_w}, y)$, where $(\mathbf{x}, y) \sim D$, and assume that $\chi^2(D, \mathcal{N}_d \otimes D_y)$ is finite [2]. Then any SQ algorithm that returns a $\hat{\mathbf{w}}$ such that $|\mathbf{w} \cdot \hat{\mathbf{w}}| \geq \tilde{\omega}(d^{-1/2})$ with probability at least $2/3$ requires either a query to $\mathrm{STAT}(\Omega_{m^*}\left(d^{-0.24(m^*-1)}\right))$, or $2^{d^{\Omega(1)}}$ many queries.*

We note that since Fact C.23 requires the condition $\chi^2(D, \mathcal{N}_d \otimes D_y)$ being finite, it cannot be applied to the setting of realizable Single-index models, where $y = f(\mathbf{x})$ without noise, as this induces an infinite $\chi^2(D, \mathcal{N}_d \otimes D_y)$.

As a corollary of our techniques, this condition can be removed. In particular, for our Condition C.5, the generative exponent $m^*$ for a distribution $D$ is simply the smallest integer $m^*$ such that $D$ does not relatively match degree-$m^*$ moments with the standard Gaussian. An application of Theorem C.6 would give an SQ lower bound to the same decision problem that is used to reduce to the subspace recovery problem in [DPLB24]. This in turn gives a similar lower bound on the subspace recovery problem as Fact C.23, but without the assumption that $\chi^2(D, \mathcal{N}_d \otimes D_y)$ is finite. Specifically, we obtain:

**Corollary C.24** (SQ Lower Bound for Parameter Recovery in Single-index Model). *Let $\mathbf{w} \in \mathbb{S}^{d-1}$ be a 1-dimensional subspace unknown to the algorithm and $D$ be a joint distribution of $(\mathbf{x}, y)$ over $\mathbb{R}^d \times \mathbb{R}$ such that $\mathbf{x} \sim \mathcal{N}_d$ and $y$ only depends on $\mathbf{x_w}$ (i.e., $D_{y|\mathbf{x}} = D_{y|\mathbf{x}'}$ for any $\mathbf{x_w} = \mathbf{x}'_\mathbf{w}$). Let $m^*$ be the generative exponent for the joint distribution of $(\mathbf{x_w}, y)$, where $(\mathbf{x}, y) \sim D$, and assume that $m^* \leq d^c$ for a sufficiently small constant c. Then any SQ algorithm that returns a $\hat{\mathbf{w}} \in \mathbb{S}^{d-1}$ such that $|\mathbf{w} \cdot \hat{\mathbf{w}}| \geq \tilde{\omega}(d^{-1/2})$ with probability at least $2/3$ requires either a query to $\mathrm{STAT}(\Omega_{m^*}\left(d^{-0.24(m^*-1)}\right))$, or $2^{d^{\Omega(1)}}$ many queries.*

*Proof of Corollary C.24.* Let $A$ be the joint distribution of $(\mathbf{x_w}, y)$ where $(\mathbf{x}, y) \sim D$ where $(\mathbf{x}, y) \sim D$. From the definition of generative exponent, we have that $A$ must be $(0, m^* - 1)$-relatively matching moments with the standard Gaussian. Therefore, according to Theorem C.6, any SQ algorithm that solve the RNGCA for input distribution over $\mathbb{R}^d \times \mathbb{R}$ with hidden distribution $A$ with probability $2/3$ requires either a query to $\mathrm{STAT}\left(\Omega_{m^*}\left(d^{-0.24(m^*-1)}\right)\right)$, or $2^{d^{\Omega(1)}}$ many queries. Therefore, it suffices for us to reduce the RNGCAdecision problem above to the parameter recovery problem for Single-index model.

Let $A$ be such an algorithm for parameter recovery in Single-index model and $D$ be the input distribution of $(\mathbf{x}, \mathbf{y})$ over $\mathbb{R}^d \times \mathbb{R}$ for the RNGCAproblem. We will sample a random rotation matrix $\mathbf{A} \sim U(\mathbf{O}_{d,d})$, which applies an random rotation over $\mathbb{R}^d$. Then we give the joint distribution of $(\mathbf{Ax}, y)$ (where $(\mathbf{x}, y) \sim D$) to the algorithm $A$ as the input distribution and let $\hat{\mathbf{w}}$ be the output vector. We will repeat the above process for $t = d^4$ times and let $\mathbf{B}$ be the empirical estimation of $(\mathbf{A}^{-1}\hat{\mathbf{w}})(\mathbf{A}^{-1}\hat{\mathbf{w}})^\top$, and let $\lambda$ be the max eigenvalue of $\mathbf{B}$.

---

[2] The assumption $\chi^2(D, \mathcal{N}_d \otimes D_y)$ being finite is required here in order for the Fourier expansion to converge in $L^2$ norm, which is not explicitly stated in [DPLB24].

Notice that if the original $D$ is the null hypothesis distribution, then we must have $\mathbf{A}^{-1}\hat{\mathbf{w}} \sim U(S^{d-1})$. Let $\mathbf{w}'_1, \cdots, \mathbf{w}'_t$ be the value of $\mathbf{A}^{-1}\hat{\mathbf{w}}$ for each round and let $\mathbf{M} = \mathbf{E}_{\mathbf{w}' \sim U(S^{d-1})}[\mathbf{w}'\mathbf{w}'^\top]$, then we have that

$$\mathbf{E}_{\mathbf{w}'_1, \cdots, \mathbf{w}'_t \sim U(S^{d-1})^{\otimes t}} \left[ \left\| \sum_{i=1}^{t} \mathbf{w}'_i\mathbf{w}'^\top_i / t - \mathbf{M} \right\|_F^2 \right]$$

$$= \mathbf{E}_{\mathbf{w}'_1, \cdots, \mathbf{w}'_t \sim U(S^{d-1})^{\otimes t}} \left[ \left\langle \frac{1}{t} \sum_{i=1}^{t} \left( \mathbf{w}'_i\mathbf{w}'^\top_i - \mathbf{M} \right), \frac{1}{t} \sum_{i=1}^{t} \left( \mathbf{w}'_i\mathbf{w}'^\top_i - \mathbf{M} \right) \right\rangle \right]$$

$$= \frac{1}{t} \mathbf{E}_{\mathbf{w}' \sim U(S^{d-1})} \left[ \|\mathbf{w}'\mathbf{w}'^\top - \mathbf{M}\|_F^2 \right] = O(d^{-4}).$$

Notice that for any $\mathbf{w} \in S^{d-1}$, we must have $\langle \mathbf{w}\mathbf{w}^\top, M \rangle \leq d^{-1}$ from the symmetry argument. Given $\mathbf{E}_{\mathbf{w}'_1, \cdots, \mathbf{w}'_t \sim U(S^{d-1})^{\otimes t}} \left[ \left\| \sum_{i=1}^{t} \mathbf{w}'_i\mathbf{w}'^\top_i / t - \mathbf{M} \right\|_F^2 \right] = O(d^{-4})$, by Markov's inequality, with probability $1 - o(1)$, we have that $\left\| \sum_{i=1}^{t} \mathbf{w}'_i\mathbf{w}'^\top_i / t - \mathbf{M} \right\|_F \leq d^{-1}$. Therefore, we must have $\lambda = O(d^{-1/2})$ with probability at least $1 - o(1)$.

On the other hand, if $D$ is from the alternative hypothesis, then since the algorithm succeeds with probability at least $2/3$ and outputs a $\hat{\mathbf{w}}$ such that $|\mathbf{w}, \hat{\mathbf{w}}| = \omega(d^{-1/2})$, we must have that $\mathbf{w}^\top \mathbf{B} \mathbf{w} = \omega(d^{-1})$. Therefore, we must have the max eigenvalue $\lambda = \omega(d^{-1/2})$.

Given the above analysis, we can simply check if $\lambda \geq cd^{1/2}$ for a sufficiently large constant $c$. If so, the input distribution $D$ must be from the alternative hypothesis. Otherwise, input distribution $D$ must be from the null hypothesis. This completes the proof of Corollary C.24. $\qquad\square$

### C.3.2 Near-Equivalence with Generative Exponent

We now show that, for realizable SIMs, the generative exponent and the conditions in our lower bound result (Theorem 1.10) are essentially equivalent up to some minor technicality, as stated by the proposition below. Notice that the first condition below is essentially the same condition in our lower bound result (Theorem 1.10), but without the technical assumption that the extreme values of labels have small contribution to the variance.

**Proposition C.25.** *Let $\mathcal{C}$ be a class of rotational invariant SIMs on $\mathbb{R}^d$. Let $\tau \in \mathbb{R}_+$ and $m \in \mathbb{Z}_+$, then the following two conditions are equivalent:*

1. *There exists an $f \in \mathcal{C}$ and a subspace $V \subseteq \mathbb{R}^d$ such that (a) the joint distribution of $(\mathbf{x}, f(\mathbf{x}))$ with $\mathbf{x} \sim \mathcal{N}_d$ matches degree-$m$ moments relative to the subspace $V$ (with the standard Gaussian projected onto $V^\perp$); and (b) for any function $h : V \to \mathbb{R}$, $\mathbf{E}_{\mathbf{x} \sim \mathcal{N}_d}[(f(\mathbf{x}) - h(\mathbf{x}_V))^2] \geq \tau$.*

2. *There exists an $f \in \mathcal{C}$ with Generative exponent strictly greater than $m$ such that the variance of $f(\mathbf{x})$ with $\mathbf{x} \sim \mathcal{N}_d$ is at least $\tau$.*

*Proof.* Notice that it suffices for us to fix a $f \in C$ and prove the equivalence. For convenience of analysis, let $\mathbf{w} \in \mathbb{S}^{d-1}$ be the relevent direction of $f$ and $W$ be the 1-dimensional subspace spanned by $\mathbf{w}$. Let $D$ be the joint distribution of $(\mathbf{x}, y)$ supported on $\mathbb{R}^d \times \mathbb{R}$ with $\mathbf{x} \sim \mathcal{N}_d$ and $y = f(\mathbf{x})$ and $A$ be the joint distribution of $(t, y)$ supported on $\mathbb{R} \times \mathbb{R}$ with $t \sim \mathcal{N}_1$ and $y = f(t\mathbf{w})$. Then it suffices for us to prove that the following two are equivalent.

1. There exists a subspace $V \subseteq \mathbb{R}^d$ such that (a) $D$ matches degree-$m$ moments relative to the subspace $V$ (with the standard Gaussian projected onto $V^\perp$); and (b) for any function $h : V \to \mathbb{R}$, $\mathbf{E}_{(\mathbf{x},y) \sim D}[(y - h(\mathbf{x}_V))^2] \geq \tau$.

2. $f$ has Generative exponent strictly greater than $m$ and the variance of $f(\mathbf{x})$ with $\mathbf{x} \sim \mathcal{N}_d$ is at least $\tau$.

The direction that Condition 2 implies Condition 1 is immediate. We simply take $V = \{0\}$. Then Condition 1(b) follows directly from the fact that $\inf_{c \in \mathbb{R}} \mathbf{E}_{(t,y) \sim A}[(y - c)^2] \geq \tau$. Since

the Generative exponent of $f$ is greater than $m$, we get that $\|\mathbf{E}_{t\sim A_{t|y}}[h_k(t)]\|_{A_y} = 0 = \|\mathbf{E}_{t\sim\mathcal{N}_1}[h_k(t)]\|_{A_y}$ for any $1 \leq k \leq m$, which is $\mathbf{E}_{t\sim A_{t|y}}[h_k(t)] = \mathbf{E}_{t\sim\mathcal{N}_1}[h_k(t)]$ for almost all $y \sim A_y$. Notice that the matching degree-$m$ moments condition (Definition 1.9) is the same as the 0-matching degree-$m$ moments condition (Condition C.5). Therefore, we just need show that for any function $f : \mathbb{R}^{d+1} \to \mathbb{R}$ such that $f(\cdot, y)$ is a polynomial for any fixed $y$, then $\left|\mathbf{E}_{(\mathbf{x},y)\sim D}[f(\mathbf{x}, y)] - \mathbf{E}_{(\mathbf{x},y)\sim\mathcal{N}_d\otimes D_y}[f(\mathbf{x}, y)]\right| = 0$. To do so, notice that

$$\mathbf{E}_{(\mathbf{x},y)\sim D}[f(\mathbf{x}, y)] = \mathbf{E}_{y\sim D_y}[\mathbf{E}_{\mathbf{x}\sim D_{\mathbf{x}|y}}[f(\mathbf{x}, y)]] = \mathbf{E}_{y\sim D_y}[\mathbf{E}_{t\sim D_{\mathbf{x}|y}}[\mathbf{E}_{\mathbf{x}'\sim\mathcal{N}(\mathbf{0},\Pi_{W^\perp})}[f(t\mathbf{w}+\mathbf{x}', y)]]] .$$

Let $f' : \mathbb{R}^2 \to \mathbb{R}$ be the function of $f'(t, y) = \mathbf{E}_{\mathbf{x}'\sim\mathcal{N}(\mathbf{0},\Pi_{W^\perp})}[f(t\mathbf{w}+, y)]$. Notice that $f'(\cdot, y)$ is a polynomial for any fixed $y$. Then using the fact that Hermite polynomials form an orthornomal basis, we have

$$\mathbf{E}_{(\mathbf{x},y)\sim D}[f(\mathbf{x}, y)] = \mathbf{E}_{(t,y)\sim A}[f'(t, y)] = \mathbf{E}_{(t,y)\sim\mathcal{N}_1\otimes A_y}[f'(t, y)]$$
$$= \mathbf{E}_{y\sim D_y}[\mathbf{E}_{t\sim\mathcal{N}_1}[\mathbf{E}_{\mathbf{x}'\sim\mathcal{N}(\mathbf{0},\Pi_{W^\perp})}[f(t\mathbf{w}+\mathbf{x}', y)]]] = \mathbf{E}_{(\mathbf{x},y)\sim\mathcal{N}_d\otimes D_y}[f(\mathbf{x}, y)] ,$$

where we use the Generative exponent condition in the second equality. This proves Condition 1 (b) and completes the proof that Condition 2 implies Condition 1.

For the direction that Condition 1 implies Condition 2, we prove its contrapositive. Assume that Condition 2 does not hold, then we must either have $\inf_{c\in\mathbb{R}} \mathbf{E}_{(t,y)\sim A}[(y-c)^2] < \tau$ or $f$ has Generative exponent at most $m$. If $\inf_{c\in\mathbb{R}} \mathbf{E}_{(t,y)\sim A}[(y-c)^2] < \tau$, then taking the $h$ in Condition 1 (b) to be the function $h(\mathbf{x}) = c$ would imply that Condition 1 (b) does not hold for any subspace $V$. If $f$ has Generative exponent at most $m$, then we get that there must be a $1 \leq k \leq m$ such that $\|\mathbf{E}_{(t,y)\sim A_{t|y}}[h_k(t)]\|_{A_y} > 0$. Now, suppose $V \subseteq \mathbb{R}^d$ is any subspace and we will analyze the following cases.

1. If $W \subseteq V$, then taking $h(\mathbf{x}_V) = f_{\mathbf{x}_W}$ is well-defined and we have $\mathbf{E}_{\mathbf{x}\sim\mathcal{N}_d}[(f(\mathbf{x})-h(\mathbf{x}_V))^2] = 0$, which implies that Condition 1 (b) does not hold.

2. If $W \not\subseteq V$, we assume that Condition 1 (b) holds for the purpose of contradiction. Let $\mathbf{u} = \mathbf{w}_{V^\perp}/\|\mathbf{w}_{V^\perp}\|_2$ and $\mathbf{w}' = \mathbf{u}_W/\|\mathbf{u}_W\|_2$. We now consider the polynomial $p : V^\perp \to \mathbb{R}$ defined as $p(\mathbf{x}_{V^\perp}) = h_k(\langle\mathbf{u}, \mathbf{x}\rangle)$. Notice that

$$\mathbf{E}_{\mathbf{x}\sim D_{\mathbf{x}|y=y_0}}[p(\mathbf{x}_{V^\perp})]$$
$$=\mathbf{E}_{\mathbf{x}\sim D_{\mathbf{x}|y=y_0}}[h_k(\langle\mathbf{u}, \mathbf{x}\rangle)]$$
$$=\mathbf{E}_{\mathbf{x}_0\sim\mathcal{N}(\mathbf{0},\Pi_V)}[\mathbf{E}_{\mathbf{x}\sim D}[h_k(\langle\mathbf{u}, \mathbf{x}\rangle) \mid \mathbf{x}_V = \mathbf{x}_0 \wedge y = y_0]]$$
$$=\mathbf{E}_{\mathbf{x}_0\sim\mathcal{N}(\mathbf{0},\Pi_V)}[\mathbf{E}_{\mathbf{x}\sim D}[h_k(\langle\mathbf{w}, \mathbf{u}\rangle\langle\mathbf{w}, \mathbf{x}\rangle + \langle\mathbf{w}', \mathbf{u}\rangle\langle\mathbf{w}', \mathbf{x}\rangle) \mid \mathbf{x}_V = \mathbf{x}_0 \wedge y = y_0]]$$
$$=\mathbf{E}_{z\sim\mathcal{N}_1,(\mathbf{x},y)\sim D}[h_k(\langle\mathbf{w}, \mathbf{u}\rangle\langle\mathbf{w}, \mathbf{x}\rangle + \langle\mathbf{w}', \mathbf{u}\rangle z) \mid y = y_0]$$
$$=\mathbf{E}_{z\sim\mathcal{N}_1,(t,y)\sim A}[h_k(\langle\mathbf{w}, \mathbf{u}\rangle t + \langle\mathbf{w}', \mathbf{u}\rangle z) \mid y = y_0]$$
$$=\mathbf{E}_{(t,y)\sim A}[(U_{\langle\mathbf{w},\mathbf{u}\rangle}h_k)(t) \mid y = y_0]$$
$$=\langle\mathbf{w}, \mathbf{u}\rangle^k\mathbf{E}_{(t,y)\sim A}[h_k(t) \mid y = y_0] ,$$

where $U_a$ is the Ornstein-Uhlenbeck operator. Therefore, we get

$$\|\mathbf{E}_{\mathbf{x}\sim D_{\mathbf{x}|y}}[p(\mathbf{x}_{V^\perp})]\|_{A_y} = \langle\mathbf{w}, \mathbf{u}\rangle^k\|\mathbf{E}_{t\sim D_{t|y}}[h_k(t)]\|_{A_y} > 0 .$$

Now notice that $p(\mathbf{x}_{V^\perp}) = \sum_{i\in[m]} \mathbf{A}_i (\mathbf{x}_{V^\perp})^{\otimes i}$ for some linear maps $\mathbf{A}_k : (V^\perp)^{\otimes k} \to \mathbb{R}$. Given Condition 1 (b) holds, we have

$$\mathbf{E}_{\mathbf{x}\sim D_{\mathbf{x}|y}}[p(\mathbf{x}_{V^\perp})] = \mathbf{E}_{\mathbf{x}\sim D_{\mathbf{x}|y}}\left[\sum_{i\in[m]} \mathbf{A}_i (\mathbf{x}_{V^\perp})^{\otimes i}\right]$$
$$= \sum_{i\in[m]} \mathbf{A}_i\mathbf{E}_{\mathbf{x}\sim D_{\mathbf{x}|y}}\left[(\mathbf{x}_{V^\perp})^{\otimes i}\right]$$
$$\underset{y\sim D_y}{=} \sum_{i\in[m]} \mathbf{A}_i\mathbf{E}_{\mathbf{x}\sim\mathcal{N}_d}\left[(\mathbf{x}_{V^\perp})^{\otimes i}\right]$$
$$= \mathbf{E}_{\mathbf{x}\sim\mathcal{N}_d}[p(\mathbf{x}_{V^\perp})] = \mathbf{E}_{\mathbf{x}\sim\mathcal{N}_d}[h_k(\langle\mathbf{u}, \mathbf{x}\rangle)] = 0 ,$$

where $\underset{y \sim D_y}{=}$ denotes equivalence for almost all $y \sim D_y$. Therefore, $\|p(\mathbf{x})\|_{D_y} = 0$, which contradicts the fact that $\|p(\mathbf{x})\|_{D_y} > 0$. Thus, Condition 1 (b) does not hold.

This proves the direction that Condition 1 implies Condition 2 and completes the proof of Proposition C.25. $\qquad\square$

# D  Algorithms for Learning Real-valued MIMs

In this section, we establish Theorem 1.4, Theorem 1.6, and Corollary 1.7.

**Organization.**  In Section D.1, we present our agnostic learning algorithm (Algorithm 3), along with the formal conditions that a MIM must satisfy for the algorithm to succeed (see Definition D.1). In Section D.2, we demonstrate that our algorithm exhibits improved efficiency when the labels depend only on a low-dimensional subspace—a regime that encompasses the realizable setting as well as cases with added random noise. Finally, in Section D.3, we describe our applications to positive-homogeneous Lipschitz functions (including homogeneous ReLU networks) and polynomials on a few relevant directions.

## D.1  Agnostically Learning Real-Valued MIMs

### D.1.1  Agnostic Learning Algorithm and Results

In this section, we present an algorithm that agnostically learns MIMs that satisfy a well-defined set of assumptions. The set of conditions we require is given in the following definition, which is a formal version of Definition 1.3.

**Definition D.1** (Well-Behaved MIMs).  We denote by $\mathcal{F}(m, \zeta, \alpha, K, M, L, B, \rho, \tau, \sigma)$ the class of all functions $f : \mathbb{R}^d \to \mathbb{R}$ satisfying the following conditions:

1. There exists a $K$-dimensional subspace $W$ of $\mathbb{R}^d$ such that $f(\mathbf{x}) = f(\mathbf{x}_W)$ for all $\mathbf{x} \in \mathbb{R}^d$.

2. $f$ is continuous everywhere and continuously differentiable almost everywhere, with $\mathbf{E}_{\mathbf{x} \sim \mathcal{N}_d}[\|\nabla f(\mathbf{x})\|^2] \leq L$.

3. $f$ has bounded norm $\mathbf{E}_{\mathbf{x} \sim \mathcal{N}_d}[f^2(\mathbf{x})] \leq M$ and is $\rho$-close to a $B$-bounded function, i.e., there exists $f_B : \mathbb{R}^d \to [-B, B]$ such that $\mathbf{E}_{\mathbf{x} \sim \mathcal{N}_d}[(f(\mathbf{x}) - f_B(\mathbf{x}))^2] \leq \rho$. [3]

4. For any subspace $V$ of $\mathbb{R}^d$ and any distribution $D$ on $\mathbb{R}^d \times \mathbb{R}$ with $D_{\mathbf{x}} = \mathcal{N}_d$ such that $\mathbf{E}_{(\mathbf{x},y) \sim D}[(f(\mathbf{x}) - y)^2] \leq \zeta$ the following hold:

   (a) either $\mathbf{E}_{\mathbf{x} \sim \mathcal{N}_d}[(f(\mathbf{x}) - g(\mathbf{x}_V))^2] \leq \tau$ for some $g : V \to \mathbb{R}$.
   (b) or with probability $\alpha$ over $\mathbf{z} \sim \mathcal{N}_d$ independent of $\mathbf{x}$ there exists a degree at most $m$, zero-mean, unit variance polynomial $p : U \to \mathbb{R}$, where $U = W_{V^\perp}$ such that $\mathbf{E}_{y_0 \sim (D_y|\mathbf{x}_V = \mathbf{z}_V)} \left[ \mathbf{E}_{\mathbf{x} \sim \mathcal{N}_d}[p(\mathbf{x}_U)|\mathbf{x}_V = \mathbf{z}_V, y = y_0]^2 \right] \geq \sigma$.

A *Well-Behaved MIM* is a bounded variation MIM function that exhibits distinguishing moments despite the presence of arbitrarily small $L_2^2$ adversarial noise. Using this robustness property, one can show that for a sufficiently fine partition of a subspace $V$ into cubes and of the real line $\mathbb{R}$ into intervals, there exists a constant fraction of the partition elements for which distinguishing moments are observable. In other words, conditioning on $\mathbf{x}$ belonging to a particular cube and $y$ lying within a particular interval, distinguishing moments persist.

Before presenting our algorithm and results, we first introduce several key concepts used by the algorithm.

One of the crucial components of our algorithm is the discretization of the space of examples $(\mathbf{x}, y)$ into thin regions. To achieve this, we partition $\mathbf{x}$ and $y$ separately into small, equal-width cubes and intervals, respectively.

---

[3]This is a mild assumption which is satisfied for example when the function has bounded 2.1-degree moment, that is $\mathbf{E}[f^{2.1}(\mathbf{x})]$ is appropriately bounded.

In particular, to efficiently approximate distributions over a subspace $V$, we partition $V$ into equal-width cubes, excluding the region where any coordinate exceeds $\sqrt{\log(k/\varepsilon)}$. This approach ensures that we retain nearly all of the mass of the distribution while maintaining regions that are both sufficiently fine and can be sampled efficiently.

**Definition D.2** ($\epsilon$-Approximating Partition)**.** Let $V$ be a $k$-dimensional subspace of $\mathbb{R}^d$ with an orthonormal basis $\mathbf{v}^{(1)}, \ldots, \mathbf{v}^{(k)}$, and let $\epsilon \in (0,1)$. An $\epsilon$-approximating partition with respect to $V$ is a collection of sets $\mathcal{S}$ defined as follows: For each multi-index $\mathbf{j} = (\mathbf{j}_1, \ldots, \mathbf{j}_k) \in [M_\epsilon]^k$, define $S_{\mathbf{j}} = \{\mathbf{x} \in \mathbb{R}^d : z_{\mathbf{j}_i - 1} \leq \mathbf{v}^{(i)} \cdot \mathbf{x} \leq z_{\mathbf{j}_i}, i \in [k]\}$ where $z_i$'s are defined as $z_i = -\sqrt{2\log(k/\epsilon)} + i\epsilon + t$, for $i \in \{0, \ldots, M_\epsilon\}, t \in (0, \epsilon/2)$ and $M_\epsilon = \left\lceil (2\sqrt{2\log(k/\epsilon)})/\epsilon \right\rceil$.

Moreover, we also discretize the label space $\mathbb{R}$ into thin, equal-sized intervals, and refer to the pair of these partitions as an approximating discretization of $V \times \mathbb{R}$.

**Definition D.3** (Approximating Discretization)**.** Let $V$ be a subspace of $\mathbb{R}^d$. We define an $(\epsilon_1, \epsilon_2, B)$-approximating discretization of $V \times \mathbb{R}$ as a pair $(\mathcal{S}, \mathcal{I})$ where

(i) $\mathcal{S}$ is an $\epsilon_1$-approximating partition of $V$;

(ii) $\mathcal{I}$ is the set of intervals $\{[i\epsilon_2 - \epsilon_2/2, i\epsilon_2 + \epsilon_2/2] \ : \ i \in \mathbb{Z}, |i| \leq B/\epsilon_2 - 1\} \cup \{[-\infty, B], [B, +\infty]\}$.

We use the term $(\epsilon_1, \epsilon_2)$-approximating discretization to refer to the special case where $B = 1/\epsilon_2^2$. Moreover, when $\epsilon_1 = \epsilon_2 = \epsilon$, we simply refer to $(\mathcal{S}, \mathcal{I})$ as an $\epsilon$-approximating discretization.

Note that [DIKZ25] does not use a discretization over the label domain and instead obtains a complexity that scales with the number of its elements. As a result, their approach becomes vacuous in the real-valued setting. In contrast, we bin the values of the label domain using a thin but reasonably efficient partition, thereby circumventing this issue.

In order to construct an approximation after identifying the appropriate subspace $W$, we approximate $y$ using a piecewise constant function defined on the projection $\mathbf{x}_W$. For this, we start with a partial partition $\mathcal{S}$ of $W$, and define a function that is constant on each element (region) in $\mathcal{S}$ and minimizes the $L_2$ loss. In particular, given the partition $\mathcal{S}$, the function assigns to each region $S \in \mathcal{S}$ the value $\mathbf{E}[y \mid \mathbf{x} \in S]$. We formalize this definition as follows:

**Definition D.4** (Piecewise Constant Approximation)**.** Let $D$ be a distribution over $\mathbb{R}^d \times \mathbb{R}$, let $V$ be a subspace of $\mathbb{R}^d$ and let $\epsilon \in (0,1)$. Let $\mathcal{S}$ be a partial partition of $V$. A piecewise constant approximation of the distribution $D$, with respect to $\mathcal{S}$, is the function $h_{\mathcal{S}} : \mathbb{R}^d \to \mathbb{R}$ such that for each $S \in \mathcal{S}$ and $\mathbf{x} \in S$, $h_{\mathcal{S}}$ is defined as $h_{\mathcal{S}}(\mathbf{x}) = \mathbf{E}_{(\mathbf{x},y)\sim D}[y \mid \mathbf{x} \in S]$. Furthermore, for any point outside the partition $\mathbf{x} \notin \bigcup_{S \in \mathcal{S}} S$, we define $h_{\mathcal{S}}(\mathbf{x}) = 0$.

We set $h_{\mathcal{S}}(\mathbf{x}) = 0$ for all points outside the partition to simplify variance control, since these regions will carry negligible probability mass do not need to be approximated.

We now present the main result of this section, which establishes that the aforementioned class of well-behaved distributions can be learned efficiently using Algorithm 3.

**Theorem D.5** (Agnostically Learning MIMs)**.** *Let $f : \mathbb{R}^d \to \mathbb{R}$ be a function from the class $\mathcal{F}(m, \zeta, \alpha, K, M, L, B, \epsilon^2/M, \tau, \sigma)$. Let $D$ be a distribution over $\mathbb{R}^d \times \mathbb{R}$ whose $\mathbf{x}$-marginal is $\mathcal{N}_d$ and let $\mathrm{OPT} \overset{\text{def}}{=} \mathbf{E}_{(\mathbf{x},y)\sim D}[(f(\mathbf{x}) - y)^2]$ with $\zeta \geq \mathrm{OPT} + O(\epsilon)$. Then, Algorithm 3 draws at most $N = d^{O(m)} 2^{\mathrm{poly}(2^m BKLM/(\alpha\epsilon\sigma))} \log(1/\delta)$ i.i.d. samples from $D$, runs in time $\mathrm{poly}(N)$, and returns a hypothesis $h$ such that, with probability at least $1 - \delta$, it holds*

$$\mathbf{E}_{(\mathbf{x},y)\sim D}[(h(\mathbf{x}) - y)^2] \leq (\sqrt{\tau + \epsilon} + \sqrt{\mathrm{OPT}})^2 + \epsilon .$$

### D.1.2 Finding a Relevant Direction

First, we show that if there exists a sufficiently accurate approximation of our function within $V$, then a piecewise constant approximation over a sufficiently fine partition of $V$ yields comparable error.

---

**Learn-MIMs**: Robustly Learning Well-Behaved MIMs

**Input:** Accuracy $\epsilon > 0$, failure probability $\delta \in (0, 1)$, sample access to a distribution $D$ over $\mathbb{R}^d \times \mathbb{R}$ with $D_\mathbf{x} = \mathcal{N}_d$ for which there exists $f \in \mathcal{F}(\theta)$ for some $\theta = (m, \mathrm{OPT} + \epsilon, \alpha, K, M, L, B, \rho, \tau, \sigma)$ known to the algorithm.

**Output:** A hypothesis $h$ such that, w.p. $1 - \delta$ $\mathbf{E}_{(\mathbf{x}, y) \sim D}[(h(\mathbf{x}) - y)^2] \leq (\sqrt{\tau + \epsilon} + \sqrt{\mathrm{OPT}})^2 + \epsilon$.

1. Let $T$ be a sufficiently large constant-degree polynomial in $1/\alpha, 1/\epsilon, 1/\sigma, K, L, M, B, 2^m$, and let $C$ be a sufficiently large universal constant..

2. Let $L_1 \leftarrow \emptyset$, $N \leftarrow d^{Cm} 2^{T^C} \log(1/\delta)$, $\epsilon_1 \leftarrow 1/T, \eta \leftarrow 1/T, \epsilon_2 \leftarrow \epsilon^2/(CM)$, $\lambda \leftarrow (a\sigma\epsilon/(MBK2^m))^C$.

3. For $t = 1, \ldots, T$

   (a) Draw a set $S_t$ of $N$ i.i.d. samples from $D$.

   (b) $\mathcal{E}_t \leftarrow$ Algorithm 4$(\eta, \epsilon_1, \epsilon_2, B, \lambda, \mathrm{span}(L_t), S_t, \theta)$.

   (c) $L_{t+1} \leftarrow L_t \cup \mathcal{E}_t$.

4. Construct $\mathcal{S}$, an $\epsilon_1$-approximating partition with respect to $\mathrm{span}(L_t)$ (see Definition D.2).

5. Draw $N$ i.i.d. samples from $D$ and construct the piecewise constant function $h_\mathcal{S}$ as follows: For each $S \in \mathcal{S}$, assign the median of $O(\log(1/\delta))$ means of the labels from the samples falling in $S$.

6. Return $h_\mathcal{S}$.

---

**Algorithm 3:** Robustly Learning Well-Behaved MIMs

---

**FindDirection**: Estimating a relevant direction

**Input:** $\eta, \epsilon_1, \epsilon_2, B, \lambda > 0$, a subspace $V \subseteq \mathbb{R}^d$, samples $\{(\mathbf{x}^{(i)}, y_i)\}_{i=1}^N$ from a distr. $D$ over $\mathbb{R}^d \times \mathbb{R}$.

**Output:** A set of unit vectors $\mathcal{E}$.

1. Construct an $(\epsilon_1, \epsilon_2, B)$-approximating discretization of $V \times \mathbb{R}$, $(\mathcal{S}, \mathcal{I})$ (see Definition D.3)

2. For each $S \in \mathcal{S}$ and each $I \in \mathcal{I}$, find a polynomial $p_{S,I}(\mathbf{x})$ such that

$$\mathbf{E}_{(\mathbf{x}, y) \sim D}[(\mathbb{1}(y \in I) - p_{S,I}(\mathbf{x}_{V^\perp}))^2 \mid \mathbf{x} \in S]$$
$$\leq \min_{p' \in \mathcal{P}_m} \mathbf{E}_{(\mathbf{x}, y) \sim D}[(\mathbb{1}(y \in I) - p'(\mathbf{x}_{V^\perp}))^2 \mid \mathbf{x} \in S] + \eta^2 .$$

3. Let $\widehat{\mathbf{U}} = \sum_{S \in \mathcal{S}, I \in \mathcal{I}} \mathbf{E}_{\mathbf{x} \sim D_\mathbf{x}}[\nabla p_{S,I}(\mathbf{x}_{V^\perp}) \nabla p_{S,I}(\mathbf{x}_{V^\perp})^\top \mid \mathbf{x} \in S] \mathbf{Pr}_{(\mathbf{x}, y) \sim \widehat{D}}[S]$.

4. Return the set $\mathcal{E}$ of unit eigenvectors of $\widehat{\mathbf{U}}$ with corresponding eigenvalues at least $\lambda$.

---

**Algorithm 4:** Estimating a relevant direction

**Lemma D.6** (Piecewise constant approximation suffices). *Let $\epsilon, L, M, \tau \in \mathbb{R}_+$, $k, d \in \mathbb{Z}_+$ with $\tau \leq M$ and $c > 0$ be a sufficiently small absolute constant. Let $f : \mathbb{R}^d \to \mathbb{R}$ be an almost everywhere continuously differentiable function such that $\mathbf{E}_{\mathbf{x} \sim \mathcal{N}_d}[\|\nabla f(\mathbf{x})\|^2] \leq L$ and $\mathbf{E}_{\mathbf{x} \sim \mathcal{N}_d}[f^2(\mathbf{x})] \leq M$. Moreover, assume that there exists a $B > 0$ and $f_B : \mathbb{R}^d \to [-B, B]$ such that $\mathbf{E}[(f(\mathbf{x}) - f_B(\mathbf{x}))^2] \leq c\epsilon^2/M$. Let $V$ be a $k$-dimensional subspace of $\mathbb{R}^d$ and let $\mathcal{S}$ be an $\eta$-approximating partition of $V$ with $\eta \leq c\epsilon^2/(MBLk)$. If there exists a function $g : V \to \mathbb{R}$ such that $\mathbf{E}_{\mathbf{x} \sim \mathcal{N}_d}[(f(\mathbf{x}) - g(\mathbf{x}_V))^2] < \tau$, then we have that $\mathbf{E}_{\mathbf{x} \sim \mathcal{N}_d}[(f(\mathbf{x}) - h_\mathcal{S}(\mathbf{x}))^2] < \tau + \epsilon$, where $h_\mathcal{S}$ denotes any piecewise constant approximation of $f$.*

*Proof.* Note that for any function $g : V \to \mathbb{R}$ the following holds

$$\mathbf{E}_{\mathbf{x} \sim \mathcal{N}_d}[(f(\mathbf{x}) - \mathbf{E}_{\mathbf{z} \sim \mathcal{N}_d}[f(\mathbf{z}) \mid \mathbf{z}_V = \mathbf{x}_V])^2] \leq \mathbf{E}_{\mathbf{x} \sim \mathcal{N}_d}[(f(\mathbf{x}) - g(\mathbf{x}_V))^2] \,.$$

Therefore, we have that the function $s(\mathbf{x}_V) \stackrel{\text{def}}{=} \mathbf{E}_{\mathbf{z} \sim \mathcal{N}_d}[f(\mathbf{z}) \mid \mathbf{z}_V = \mathbf{x}_V]$ achieves squared error at most $\tau$. Note that since $\mathbf{x}_V$ and $\mathbf{x}_{V^\perp}$ are independent for $\mathbf{x} \sim \mathcal{N}_d$, we have that

$$s(\mathbf{x}_V) = \mathbf{E}_{\mathbf{z} \sim \mathcal{N}_d}[f(\mathbf{z}) \mid \mathbf{z}_V = \mathbf{x}_V] = \mathbf{E}_{\mathbf{x}_{V^\perp}}[f(\mathbf{x}_V + \mathbf{x}_{V^\perp})] \,.$$

Hence from the linearity of the derivative operator and Jensen's inequality we have that $\|\nabla s(\mathbf{x}_V)\|^2 \leq \mathbf{E}_{\mathbf{x}_{V^\perp}}\left[\|\nabla_{\mathbf{x}_V} f(\mathbf{x}_V + \mathbf{x}_{V^\perp})\|^2\right]$. Consequently

$$\mathbf{E}_{\mathbf{x}_V}\left[\|\nabla s(\mathbf{x}_V)\|^2\right] \leq \mathbf{E}_{\mathbf{x} \sim \mathcal{N}_d}\left[\|\nabla_{\mathbf{x}_V} f(\mathbf{x}_V + \mathbf{x}_{V^\perp})\|^2\right] \leq \mathbf{E}_{\mathbf{x} \sim \mathcal{N}_d}\left[\|\nabla f(\mathbf{x})\|^2\right] \leq L.$$

Moreover by Jensen's inequality we have that

$$\mathbf{E}_{\mathbf{x} \sim \mathcal{N}_d}[(s(\mathbf{x}_V) - \mathbf{E}_{\mathbf{x}_{V^\perp}}[f_B(\mathbf{x}_V + \mathbf{x}_{V^\perp})])^2] = \mathbf{E}_{\mathbf{x} \sim \mathcal{N}_d}[(\mathbf{E}_{\mathbf{x}_{V^\perp}}[f(\mathbf{x}_V + \mathbf{x}_{V^\perp}) - f_B(\mathbf{x}_V + \mathbf{x}_{V^\perp})])^2]$$
$$\leq \mathbf{E}_{\mathbf{x} \sim \mathcal{N}_d}[(f(\mathbf{x}) - f_B(\mathbf{x}))^2] \leq \rho \,.$$

Therefore, $s$ is close to a bounded function. Let $h$ be an approximation to $s$ that achieves squared error $\alpha$. We have that the function $h(\mathbf{x})\mathbb{1}(\mathbf{x} \in A^c)$ achieves squared error $2B^2\epsilon + 2\rho + \alpha$, where $A$ is any region of $\mathbb{R}^d$ with mass less than $\epsilon$.

As a result we can apply Fact E.11 for the function $s$ have that the piecewise constant function $h_\mathcal{S}(\mathbf{x}_V) \stackrel{\text{def}}{=} \mathbf{E}[s(\mathbf{x}_V) | \mathbf{x} \in S] = \mathbf{E}[f(\mathbf{x}) | \mathbf{x} \in S]$ for all $\mathbf{x} \in S$ and $S \in \mathcal{S}$ and $h_\mathcal{S}(\mathbf{x}_V) = 0$ otherwise achieves error $\mathbf{E}_{\mathbf{x} \sim \mathcal{N}_d}[(s(\mathbf{x}_V) - h_\mathcal{S}(\mathbf{x}_V))^2] \leq 4\rho + \eta + 2\eta B^2$.

Finally, we have that $\mathbf{E}_{\mathbf{x} \sim \mathcal{N}_d}[(f(\mathbf{x}) - h_\mathcal{S}(\mathbf{x}))^2] = \tau + \epsilon$, which concludes the proof of Lemma D.6. $\qquad \square$

Moreover, under the conditions defined in Definition D.1, we show that there exists a non-negligible fraction of finite-width cubes over $\mathbf{x}_V$ and intervals over $y$ where distinguishing moments can be observed. Furthermore, we demonstrate that if a direction of the target model is learned to accuracy $\epsilon$, then there exists an observable moment that depends only on the remaining directions.

Intuitively, the proof proceeds as follows. Since $f$ is a well-behaved MIM, we can round $y$ to some small accuracy while maintaining distinguishing moments. This discretizes the label space into intervals, and so with non-trivial probability over $\mathbf{x}_V$ there is a degree-$m$ moment that correlates with the rounded label. Next, because $f$ has bounded variation, $f(\mathbf{x})$ and $f(\mathbf{x}')$—where $\mathbf{x}'$ is obtained by averaging $\mathbf{x}_V$ within $\mathbf{x}$'s cube in $\mathcal{S}$—remain close in mean-squared error. This insensitivity lets us discretize over $V$ as well. Finally, for any direction with small projection onto $W$, bounded variation and the well-behaved MIM condition imply that averaging along that direction preserves the distinguishing moment, yielding a moment independent of these directions.

**Lemma D.7** (Cube-interval Discretization Suffices). *There exists a sufficiently large constant $C > 0$ such that the following holds. Let $d, k, m \in \mathbb{Z}_+$, and $L, M, \zeta, \alpha, \tau, \sigma, \epsilon > 0$ with $\zeta \leq M$. Let $f : \mathbb{R}^d \to \mathbb{R}$ be in $\mathcal{F}(m, \zeta + C\epsilon, \alpha, K, M, L, B, \epsilon^2/(CM), \tau, \sigma)$ and let $W$ be a $K$-dimensional subspace such that $f(\mathbf{x}) = f(\mathbf{x}_W)$. Let $D$ be a distribution over $\mathbb{R}^d \times \mathbb{R}$ such that $D_\mathbf{x} = \mathcal{N}_d$ and $\mathbf{E}_{(\mathbf{x},y) \sim D}[(f(\mathbf{x}) - y)^2] \leq \zeta$. Let $V$ be a $k$-dimensional subspace, $k \geq 1$ and denote by $U = W_{V^\perp}$. Let $(\mathcal{S}, \mathcal{I})$ be an $(\epsilon_1, \epsilon_2, B)$-approximating discretization of $V \times \mathbb{R}$, with $\epsilon_1 \leq \epsilon^2/(2LM^2\sqrt{k})$ and $\epsilon_2 \leq \epsilon^2/(CM)$. Moreover, let $E \subseteq V^\perp$ be a subspace such that $\|\mathbf{v}_W\| \leq \epsilon/(CKLM)$ for every unit vector $\mathbf{v} \in E$.*

*If* $\mathbf{E}[(f(\mathbf{x}) - g(\mathbf{x}_V))^2] > \tau$ *for all* $g : V \to \mathbb{R}$*, then there exists* $\mathcal{T} \subseteq \mathcal{S}$ *with* $\sum_{S \in \mathcal{T}} \mathbf{Pr}[S] \geq \alpha$ *such that for all* $S \in \mathcal{T}$ *there exist* $I \in \mathcal{I}$ *and zero mean variance one polynomial* $p : U \to \mathbb{R}$ *of degree at most* $m$ *such that the following hold:*

*i)* $\mathbf{E}[p(\mathbf{x}_U)\mathbb{1}(y \in I) \mid \mathbf{x}_V \in S] \geq \mathrm{poly}(\sigma\epsilon/(MB3^m))$.

*ii)* $\nabla p(\mathbf{x}_U) \cdot \mathbf{v} = 0$ *for all* $\mathbf{v} \in E$ *and* $\mathbf{x} \in \mathbb{R}^d$.

*Proof of Lemma D.7.* We prove the two items in order. Specifically, we first show that there exists a sufficiently fine discretization $(\mathcal{S}, \mathcal{I})$ of $V \times \mathbb{R}$ such that, for a fraction of cubes $S \in \mathcal{S}$, there exists a degree-$m$ polynomial that correlates nontrivially with the boolean function $\mathbb{1}(y \in I)$ conditioned on $\mathbf{x} \in S$. Then, we show that averaging each such polynomial over the subspace $E$ preserves its correlation on $S$ while forcing all directional derivatives along $E$ to vanish.

**Existence of correlating polynomials** Without loss of generality, we can let $y$ be the random variable obtained by truncating the original random variable $y$ to $[-B, B]$ (meaning that we assign the value $B\mathrm{sign}(y)$ if $|y| \geq B$) and rounding to the nearest multiple of $\epsilon_2$. Indeed, let $y_B \overset{\text{def}}{=} \mathrm{sign}(y)\min(|y|, B)$ be the truncation of the original random variable $y$, by expanding the square and applying Cauchy-Schwarz $\mathbf{E}[(f_B(\mathbf{x}) - y)^2] \leq \zeta + O(\epsilon)$. Hence, $\mathbf{E}[(f_B(\mathbf{x}) - y_B)^2] \leq \mathbf{E}[(f_B(\mathbf{x}) - y)^2] \leq \zeta + O(\epsilon)$, which by a similar argument implies that $\mathbf{E}[(f(\mathbf{x}) - y_B)^2] \leq \zeta + O(\epsilon)$. Similarly, rounding $y_B$ to multiples of $\epsilon_2 \leq \epsilon^2/(CM)$ also keeps the random variable $(\zeta + O(\epsilon))$-close to $f$ in squared error. Therefore, for the new random variable $y$ it holds that $\mathbf{E}[(f(\mathbf{x}) - y)^2] = \zeta + O(\epsilon)$.

For any $\mathbf{x} \in \mathbb{R}^d$, denote by $\mathcal{S}_{\mathbf{x}}$ the set $S \in \mathcal{S}$ such $\mathbf{x} \in S$. We define a random vector $\mathbf{x}'$ such that $\mathbf{x}'_{V^\perp} = \mathbf{x}_{V^\perp}$ and $\mathbf{x}'_V$ is the sampled from the standard Gaussian over $V$ conditioned on $\mathcal{S}_{\mathbf{x}}$, i.e., $\mathcal{N}_d \mid \mathcal{S}_{\mathbf{x}}$. Note that $\mathbf{x}'$ also follows the standard normal distribution over $\mathbb{R}^d$ because its $V$-component is resampled from $\mathcal{N}_d$. Define $y'$ such that for all $\mathbf{z} \in \mathbb{R}^d$, the distribution of $y'$ given $\mathbf{x}' = \mathbf{z}$ is the same as the distribution of $y$ given $\mathbf{x} = \mathbf{z}$. Notice that $y'$ and $y$ have the same support, i.e., $y'$ is also a multiple of $\epsilon$. Denote by $D'$ the joint distribution of $(\mathbf{x}, y')$.

By applying Fact E.12 we have that $\mathbf{E}[(f(\mathbf{x}) - f(\mathbf{x}'))^2] \leq \epsilon^2/M$. Moreover, it holds that $\mathbf{E}[(f(\mathbf{x}) - y')^2] = \mathbf{E}[(f(\mathbf{x}') - y)^2]$. Therefore, by expanding the square we obtain

$$\mathbf{E}[(f(\mathbf{x}) - y')^2] = \mathbf{E}[(f^2(\mathbf{x}) - y)^2] - 2\mathbf{E}[(f(\mathbf{x}) - y)(f(\mathbf{x}) - f(\mathbf{x}'))] + \mathbf{E}[(f(\mathbf{x}) - f(\mathbf{x}'))^2]$$
$$= \zeta + O(\epsilon) ,$$

where in the last inequality we used Cauchy-Schwarz and the assumption that $\zeta \leq M$. Consequently, by Condition (4) of Definition D.1 we have that with probability $\alpha$ over $\mathbf{z} \sim \mathcal{N}_d$ there exist a zero mean, variance one polynomial $p : U \to \mathbb{R}$ such that $\mathbf{E}_{y_0 \sim (D'_{y'}|\mathbf{x}_V = \mathbf{z}_V)}\left[\mathbf{E}_{(\mathbf{x}, y') \sim D'}[p(\mathbf{x}_U)|\mathbf{x}_V = \mathbf{z}_V, y' = y_0]^2\right] \geq \sigma$.

Fix a $\mathbf{z}$ such that the aforementioned statement is satisfied. Note that $\mathbf{E}[y'^2] = O(M)$. Recall that by the Gaussian hypercontractivity inequality (Fact E.4), we have that $\mathbf{E}[q^4(\mathbf{x})] \leq 3^{2m}$, for any zero-mean, unit-variance polynomial $q : \mathbb{R}^d \to \mathbb{R}$ of degree at most $m$. Let $\eta \in (0, 1)$ be a parameter to be quantified later and denote by $\mathcal{Y}_\eta$ the set of all $y_0$ in the support of $y'$ such that $\mathbf{Pr}_{y' \sim (D'_{y'}|\mathbf{x}_V = \mathbf{z}_V)}[y' = y_0] \leq \eta$. For a label $a \in \mathcal{Y}_\eta$ we have that

$$\mathbf{E}_{y_0 \sim (D'_{y'}|\mathbf{x}_V = \mathbf{z}_V)}[\mathbb{1}(y_0 = a)\mathbf{E}_{(\mathbf{x}, y') \sim D'}[p(\mathbf{x}_U)|\mathbf{x}_V = \mathbf{z}_V, y' = y_0]^2]$$

$$\leq \mathbf{E}_{y_0 \sim (D'_{y'}|\mathbf{x}_V = \mathbf{z}_V)}[\mathbb{1}(y_0 = a)\mathbf{E}_{(\mathbf{x}, y') \sim D'}[p^2(\mathbf{x}_U)|\mathbf{x}_V = \mathbf{z}_V, y' = y_0]]$$

$$\leq \sqrt{\eta}\sqrt{\mathbf{E}_{\mathbf{x} \sim \mathcal{N}_d}[p^2(\mathbf{x}_U)|\mathbf{x}_V = \mathbf{z}_V]]}$$

$$\leq \sqrt{\eta 3^{2m}} ,$$

where in the first inequality we used Jensen and in the second inequality Cauchy-Schwarz and in the last inequality the hypercontractivity bound along with the fact that $U \subseteq V^\perp$.

Note that the number of different values in the support of $y'$ are at most $O(B/\epsilon_2)$. As a result, we have that

$$\mathbf{E}_{y_0 \sim (D'_{y'}|\mathbf{x}_V = \mathbf{z}_V)}\left[\mathbf{E}_{\mathbf{x} \sim \mathcal{N}_d}[p(\mathbf{x}_U)|\mathbf{x}_V = \mathbf{z}_V, y' = y_0]^2\right]$$

$$\leq \mathbf{E}_{y_0 \sim (D'_{y'}|\mathbf{x}_V = \mathbf{z}_V)}\left[\mathbb{1}(y_0 \notin \mathcal{Y}_\eta)\mathbf{E}_{(\mathbf{x}, y') \sim D'}[p(\mathbf{x}_U)|\mathbf{x}_V = \mathbf{z}_V, y' = y_0]^2\right] + O(B/\epsilon_2)\sqrt{\eta 3^{2m}} .$$

Setting the parameter $\eta = (\epsilon^2 \sigma/(2BM3^m))^2$ results to $O(B/\epsilon_2)\sqrt{\eta 3^{2m}} \leq \sigma/2$, and thus $\mathbf{E}_{y_0 \sim (D'_{y'}|\mathbf{x}_V = \mathbf{z}_V)}\left[\mathbb{1}(y_0 \notin \mathcal{Y}_\eta)\mathbf{E}_{(\mathbf{x},y') \sim D'}[p(\mathbf{x}_U)|\mathbf{x}_V = \mathbf{z}_V, y' = y_0]^2]\right] \geq \sigma/2$. Therefore, there exists a $y_0$ with $\mathbf{Pr}_{y' \sim (D'_{y'}|\mathbf{x}_V = \mathbf{z}_V)}[y' = y_0] > \eta$ such that $\mathbf{E}_{(\mathbf{x},y') \sim D'}[p(\mathbf{x}_U)|\mathbf{x}_V = \mathbf{z}_V, y' = y_0] \geq \sqrt{\sigma/2}$. Hence, for the aforementioned $y_0$ it holds that $\mathbf{E}_{(\mathbf{x},y') \sim D'}[p(\mathbf{x}_U)\mathbb{1}(y' = y_0)|\mathbf{x}_V = \mathbf{z}_V] \geq \mathrm{poly}(\sigma\epsilon/(BM3^m))$.

Finally, since $y'$ is independent of $\mathbf{z}_V$ when conditioned on its cube $S_{\mathbf{z}_V}$ we have that $\mathbf{E}_{(\mathbf{x},y') \sim D'}[p(\mathbf{x}_U)\mathbb{1}(y' = y_0) \mid \mathbf{x}_V \in S] \geq \mathrm{poly}(\sigma\epsilon/(BM3^m))$. Noticing that the moments with respect to $U$ on each $S \in \mathcal{S}$ are the same for $y$ and $y'$ we have that also $\mathbf{E}_{(\mathbf{x},y) \sim D}[p(\mathbf{x}_U)\mathbb{1}(y = y_0) \mid \mathbf{x}_V \in S] \geq \mathrm{poly}(\sigma\epsilon/(BM3^m))$. Therefore we have that the first part of the statement follows as conditioning on the truncated and rounded label to equal $i\epsilon_2$ is the equivalent as conditioning on the intervals $[i\epsilon_2 - \epsilon/2, i\epsilon_2 + \epsilon/2]$ for $|i| \leq B/\epsilon - 1$ and $[-\infty, -B], [B, \infty]$.

Using a similar strategy, we show that there exists a polynomial satisfying the second part of the statement also.

**Averaging over $E$**   Define the parameter $\delta \overset{\text{def}}{=} \epsilon/(CLKM)$. Recall that for all unit vectors $\mathbf{v} \in E$, it holds that $|\mathbf{v}_W| \leq \delta$. Let $\mathbf{x}' = \mathbf{z}_E + \mathbf{x}_{E\perp}$, where $\mathbf{z} \sim \mathcal{N}_d$ independent of $\mathbf{x}$.

In the following claim we show that $f(\mathbf{x})$ is very close to $f(\mathbf{x}')$, to do this we simply integrate the change of $f$ across a path from $\mathbf{x}$ to $\mathbf{x}'$. Since $f$ is a function of bounded variation and $\|\Pi_W(\mathbf{x} - \mathbf{x}')\|$ this change is small.

**Claim D.8.** *It holds that* $\mathbf{E}[(f(\mathbf{x}) - f(\mathbf{x}'))^2] \lesssim L(K\delta)^2$.

*Proof Claim D.8.* Note that since we want to utilize the fact that $\mathbf{E}[\|\nabla f(\mathbf{x})\|^2] \leq L$ we want integrate at along a rotation from $\mathbf{x}$ to $\mathbf{x}'$ to preserve the all intermediate points in the path to be standard Gaussian.

For that purpose let $\mathbf{u}(\theta) = (\mathbf{x}_E \cos(\theta) + \mathbf{z}_E \sin(\theta)) + \mathbf{x}_{E\perp}$. Note $\mathbf{u}(0) = \mathbf{x}$ and $\mathbf{u}(\pi/2) = \mathbf{x}'$. Now by the Fundamental Theorem of Calculus

$$f(\mathbf{x}') - f(\mathbf{x}) = \int_0^{\pi/2} \nabla f(\mathbf{u}(\theta)) \cdot \frac{d}{d\theta}\Pi_W \mathbf{u}(\theta) d\theta$$

$$= \int_0^{\pi/2} \nabla f(\mathbf{u}(\theta)) \cdot \Pi_W(\mathbf{z}_E \cos(\theta) - \mathbf{x}_E \sin(\theta)) d\theta$$

$$\leq \int_0^{\pi/2} \|\nabla f(\mathbf{u}(\theta))\|\|\Pi_W(\mathbf{z}_E \cos(\theta) - \mathbf{x}_E \sin(\theta))\| d\theta ,$$

where in the last inequality we used Cauchy-Schwarz. Hence,

$$(f(\mathbf{x}') - f(\mathbf{x}))^2 \leq \left(\int_0^{\pi/2} \|\nabla f(\mathbf{u}(\theta))\|\|\Pi_W(\mathbf{z}_E \cos(\theta) - \mathbf{x}_E \sin(\theta))\| d\theta\right)^2$$

$$= (\pi/2)^2 \left(\int_0^{\pi/2} (2/\pi)\|\nabla f(\mathbf{u}(\theta))\|\|\Pi_W(\mathbf{z}_E \cos(\theta) - \mathbf{x}_E \sin(\theta))\| d\theta\right)^2$$

$$\leq (\pi/2) \int_0^{\pi/2} \|\nabla f(\mathbf{u}(\theta))\|^2 \|\Pi_W(\mathbf{z}_E \cos(\theta) - \mathbf{x}_E \sin(\theta))\|^2 d\theta ,$$

where in the last inequality we used Jensen. Taking the expectation and noticing that $\mathbf{u}(\theta)$ and $\mathbf{z}_E \cos(\theta) - \mathbf{x}_E \sin(\theta)$ are independent (since they are jointly normally distributed and uncorrelated), we get

$$\mathbf{E}[(f(\mathbf{x}') - f(\mathbf{x}))^2] \lesssim \int_0^{\pi/2} \mathbf{E}[\|\nabla f(\mathbf{u}(\theta))\|^2 \|\Pi_W(\mathbf{z}_E \cos(\theta) - \mathbf{x}_E \sin(\theta))\|^2]$$

$$\leq \int_0^{\pi/2} \mathbf{E}[\|\nabla f(\mathbf{u}(\theta))\|^2]\mathbf{E}[\|\Pi_W(\mathbf{z}_E \cos(\theta) - \mathbf{x}_E \sin(\theta))\|^2]$$

$$\lesssim L\mathbf{E}_{\mathbf{x} \sim \mathcal{N}_d}[\|\Pi_W \Pi_E \mathbf{x}\|^2] ,$$

Noticing that $\mathbf{E}_{\mathbf{x}\sim\mathcal{N}_d}[\|\Pi_W\Pi_E\mathbf{x}\|^2] = \|\Pi_W\Pi_E\|_F^2 \leq (\mathrm{rank}(\Pi_W\Pi_E)\|\Pi_W\Pi_E\|_2)^2 \leq (K\delta)^2$ completes the proof of Claim D.8. $\qquad\square$

Hence, by Claim D.8, we have that $\mathbf{E}[(f(\mathbf{x})-f(\mathbf{x}'))^2] \lesssim (\epsilon/(CM))^2$. Consider the random variable $y'$ supported on $\mathbb{R}$ that is distributed like $y$ for $\mathbf{x}'$, i.e., $D(y' = p \mid \mathbf{x}' = \mathbf{z}) = D(y = p \mid \mathbf{x} = \mathbf{z})$, for all $\mathbf{z} \in \mathbb{R}^d, p \in \mathbb{R}$.

Note that $\mathbf{E}[(f(\mathbf{x}) - y')^2] = \mathbf{E}[(f(\mathbf{x}') - y)^2]$, which, similarly to before, can be shown—by expanding the square and applying Cauchy–Schwarz—to be less than $\zeta + \epsilon$ for a sufficiently large constant $C$. As a result, we have that by applying part (i) of the statement to $y'$ there exists $\mathcal{T} \subseteq \mathcal{S}$ with $\sum_{S\in\mathcal{T}} \mathbf{Pr}[S] \geq \alpha$ such that for each $S \in \mathcal{T}$ there exists a zero-mean, variance-one polynomial $p : U \to \mathbb{R}$ of degree at most $m$ along with an interval $I \in \mathcal{I}$ such that $\mathbf{E}_{\mathbf{x},y'}[p(\mathbf{x}_U)\mathbb{1}(y' \in I) \mid \mathbf{x} \in S] > \sigma$, where $U = (V + W) \cap V^\perp$. However, we have that for all $S \in \mathcal{T}$

$$\mathbf{E}_{\mathbf{x},y'}[p(\mathbf{x}_U)\mathbb{1}(y' \in I) \mid \mathbf{x} \in S] = \mathbf{E}[\mathbf{E}_{\mathbf{x}_E}[p(\mathbf{x}_U) \mid \mathbf{x}_{E^\perp}]\mathbf{E}_{\mathbf{x}_E}[\mathbb{1}(y \in I) \mid \mathbf{x}_{E^\perp}] \mid \mathbf{x} \in S] .$$

Notice that $p'(\mathbf{x}_U) \overset{\mathrm{def}}{=} \mathbf{E}_{\mathbf{x}_E}[p(\mathbf{x}_U) \mid \mathbf{x}_{E^\perp}]$ is a mean-zero polynomial of degree at most $m$ with variance at most one by Jensen's inequality. Furthermore, as $p'$ is independent of $\mathbf{x}_E$ we have that $\nabla p'(\mathbf{x}_U) \cdot \mathbf{v} = 0$ for any $\mathbf{v} \in E$ and $\mathbf{x} \in \mathbb{R}^d$. $\qquad\square$

Before proceeding with the proof of Theorem D.5, we establish the following proposition, which states that in each iteration, as long as the current subspace $V$ yields an insufficient approximation of the labels, it is possible to extract a direction that is correlated with the remaining subspace.

**Proposition D.9** (Finding a Relevant Direction)**.** *Let $d, k, m, K \in \mathbb{Z}_+$, $\delta, \alpha \in (0,1)$, $\eta, \epsilon, M, L > 0$ and let $C > 0$ be a sufficiently large universal constant. Let $f : \mathbb{R}^d \to \mathbb{R}$ be a function from the class $\mathcal{F}(m, \zeta, \alpha, K, M, L, B, \epsilon^2/(CM), \tau, \sigma)$ and denote by $W \subseteq \mathbb{R}^d$ with $\dim(W) = K$ the hidden subspace of $f$. Let $D$ be a distribution over $\mathbb{R}^d \times \mathbb{R}$ whose $\mathbf{x}$-marginal is $\mathcal{N}_d$ and let $\mathrm{OPT} \overset{\mathrm{def}}{=} \mathbf{E}_{(\mathbf{x},y)\sim D}[(f(\mathbf{x}) - y)^2]$ with $\zeta \geq \mathrm{OPT} + O(\epsilon)$. Let $V \subseteq \mathbb{R}^d$ be a $k$-dimensional subspace and let $(\mathcal{S}, \mathcal{I})$ be an $(\epsilon_1, \epsilon_2, B)$-approximating discretization of $V \times \mathbb{R}$, with $\epsilon_1 \leq \epsilon^2/(CLM^2\sqrt{k})$ and $\epsilon_2 \leq \epsilon^2/(CM)$. Additionally, let $h_{\mathcal{S}}$ be a piecewise constant approximation of $D$, with respect to $\mathcal{S}$. There exists $N = (dm)^{O(m)}(k/\epsilon_1)^{O(k)} \log(B/(\delta\epsilon_2))/\eta^{O(1)}$ such that, if $\mathbf{E}_{(\mathbf{x},y)\sim D}[(h_{\mathcal{S}}(\mathbf{x}) - y)^2] > (\sqrt{\tau + \epsilon} + \sqrt{\mathrm{OPT}})^2$, then Algorithm 4 given $N$ i.i.d. samples from $D$ and parameters $\eta \leq (\sigma\epsilon/(MB2^m))^C, \epsilon_1, \epsilon_2, \lambda = (\alpha\sigma\epsilon/(MBK2^m))^C$, runs in $\mathrm{poly}(N)$ time, and with probability at least $1 - \delta$, returns a list of unit vectors $\mathcal{E}$ of size $|\mathcal{E}| = \mathrm{poly}(2^m KMB/(\alpha\sigma\epsilon_2))$, such that: For some $\mathbf{v} \in \mathcal{E}$ and unit vector $\mathbf{w} \in W$ it holds that $\mathbf{w}_{V^\perp} \cdot \mathbf{v} \geq \mathrm{poly}(\epsilon\sigma\epsilon_2\alpha/(KLMB2^m))$ .*

*Proof of Proposition D.9.* Let $\mathbf{w}^{(1)}, \ldots, \mathbf{w}^{(K)}$ be an orthonormal basis for the subspace $W$. For each pair $S \in \mathcal{S}$ and $I \in \mathcal{I}$, let $p_{S,I}$ denote the regression polynomial of degree at most $m$ computed at Line 2 of Algorithm 4. Further, let $\widehat{\mathbf{U}}$ be the matrix computed at Line 3 of Algorithm 4, and let $\eta^2$ denote the $L_2^2$ error chosen in the polynomial regression step (see Line 2 of Algorithm 4).

We begin by proving the following general lemma, which states that if a fraction of the discretization cubes exhibit non-trivial moments, then—with a sufficiently large number of samples—the output set $\mathcal{E}$ will be non-empty and will contain only a small number of vectors.

**Lemma D.10** (Existence of Correlating Vectors)**.** *Let $(\mathcal{S}, \mathcal{I})$ be an $(\epsilon_1, \epsilon_2, B)$-approximating discretization of $V \times \mathbb{R}$. Assume that there exists a subset $\mathcal{T} \subseteq \mathcal{S}$ with $\sum_{S\in\mathcal{T}} \mathbf{Pr}[S] \geq \alpha$, such that for each $S \in \mathcal{T}$ there exists an interval $I \in \mathcal{I}$ and a zero-mean, variance-one polynomial $q_S : U \to \mathbb{R}$ of degree at most $m$ such that $\mathbf{E}_{(\mathbf{x},y)\sim D}[q_S(\mathbf{x}_U)\mathbb{1}(y \in I) \mid \mathbf{x} \in S] > \sigma \geq 2\eta$ for some subspace $U$ of $V^\perp$. Furthermore, assume that the eigenvalue threshold $\lambda = (\alpha\sigma/K)^C$, for some sufficiently large universal constant $C > 0$. Then, there exists $N = (dm)^{O(m)}(k/\epsilon_1)^{O(k)} \log(|\mathcal{I}|/\delta)/\eta^{O(1)}$ such that with probability at least $1 - \delta$ the output set $\mathcal{E}$ has cardinality at most $|\mathcal{E}| = \mathrm{poly}(mK/(\alpha\sigma|\mathcal{I}|))$ and contains at least one vector $\mathbf{v} \in \mathcal{E}$ such that $\mathbf{u} \cdot \mathbf{v} \geq \mathrm{poly}(\alpha\sigma|\mathcal{I}|/(mK))$ for some $\mathbf{u} \in U$.*

*Proof of Lemma D.10.* Let $\mathbf{u}^{(1)}, \ldots, \mathbf{u}^{(k')}$ denote an orthonormal basis of the subspace $U$. We prove the lemma in two stages. First, we analyze each cube $S$ that exhibits non-trivial moments by evaluating the quadratic forms of its influence matrix $\mathbf{M}_{S,I} \overset{\mathrm{def}}{=}$

$\mathbf{E}_{\mathbf{x} \sim D_{\mathbf{x}}} \left[ \nabla p_{S,I}(\mathbf{x}_{V^\perp}) \nabla p_{S,I}(\mathbf{x}_{V^\perp})^\top \mid \mathbf{x} \in S \right]$ on the vectors $\mathbf{u}^{(i)}$. Then, we extend the analysis by averaging over all such cubes and examining the eigenvectors of the aggregated influence matrix, $\widehat{\widetilde{\mathbf{U}}}$.

In the following claim, we leverage the existence of non-trivial moments over $U$ for the regions $S \in \mathcal{T}$, to show that if the sample size is sufficiently large then for each region $S \in \mathcal{T}$, there exists an interval $I \in \mathcal{I}$, such that the associated influence matrix, $\mathbf{M}_{S,I}$, has large quadratic form for at least one of the $\mathbf{u}^{(i)}$'s.

**Claim D.11** (Quadratic form of the Influence Matrix). *Let $C > 0$ be a sufficiently large universal constant. Fix, $S \in \mathcal{T}$. If the number of samples that fall in $S$ is $N_S \geq (dm)^{Cm} \log(1/(\delta\epsilon_2))/\eta^C$, then with probability at least $1 - \delta$ there exists $i \in [k']$ and $I \in \mathcal{I}$ such that $(\mathbf{u}^{(i)})^\top \mathbf{M}_{S,I} \mathbf{u}^{(i)} \geq \sigma^2/(2K)$.*

*Proof of Claim D.11.* Notice that membership of a point $\mathbf{x}$ in a cube $S$ depends solely on its projection onto $V$, i.e., on $\mathbf{x}_V$. Therefore, the error guarantee obtained in the polynomial regression step $\mathbf{E}_{(\mathbf{x},y) \sim D}[(\mathbb{1}(y \in I) - p_{S,I}(\mathbf{x}_{V^\perp}))^2 | \mathbf{x} \in S] \leq \min_{p' \in \mathcal{P}_m} \mathbf{E}_{(\mathbf{x},y) \sim D}[(\mathbb{1}(y \in I) - p'(\mathbf{x}_{V^\perp}))^2 | \mathbf{x} \in S] + \eta^2$ is equivalent to $\mathbf{E}_{(\mathbf{x},y) \sim D_{V^\perp}^S}[(\mathbb{1}(y \in I) - p_{S,I}(\mathbf{x}))^2] \leq \min_{p' \in \mathcal{P}_m} \mathbf{E}_{(\mathbf{x},y) \sim D_{V^\perp}^S}[(\mathbb{1}(y \in I) - p'(\mathbf{x}))^2] + \eta^2$, where $D_{V^\perp}^S$ is the marginal obtained by averaging $D$ over $V$ conditioned on $S$, i.e., $D_{V^\perp}^S(\mathbf{x}_{V^\perp}, y) = \mathbf{E}_{\mathbf{x}_V}[D(\mathbf{x}, y) \mid \mathbf{x} \in S]$.

Moreover, by the properties of the Gaussian distribution, we have that the $\mathbf{x}$-marginal of $D_{V^\perp}^S$ is a standard Gaussian. Therefore, since $|\mathcal{I}| = \text{poly}(1/\epsilon_2)$ if $N_S \geq (dm)^{Cm} \log(1/(\delta\epsilon_2))/\eta^C$ for a sufficiently large universal constant $C > 0$, then by applying the union bound and Fact E.9, we have that with probability at least $1 - \delta$:

$$\mathbf{E}_{(\mathbf{x},y) \sim D_{V^\perp}^S}[(\mathbb{1}(y \in I) - p_{S,I}(\mathbf{x}))^2] \leq \min_{p' \in \mathcal{P}_m} \mathbf{E}_{(\mathbf{x},y) \sim D_{V^\perp}^S}[(\mathbb{1}(y \in I) - p'(\mathbf{x}))^2] + \eta^2 ,$$

for all $I \in \mathcal{I}$. Furthermore, by the orthogonality of Hermite polynomials, we have that

$$\begin{aligned}
&\mathbf{E}_{(\mathbf{x},y) \sim D_{V^\perp}^S}[(\mathbb{1}(y \in I) - p_{S,I}(\mathbf{x}))^2] \\
&= \sum_{\beta \subseteq \mathbb{N}^d} (\mathbf{E}_{(\mathbf{x},y) \sim D_{V^\perp}^S}[H_\beta(\mathbf{x})\mathbb{1}(y \in I)] - \mathbf{E}_{(\mathbf{x},y) \sim D_{V^\perp}^S}[H_\beta(\mathbf{x})p_{S,I}(\mathbf{x})])^2 .
\end{aligned}$$

In particular, if we decompose the error into its Hermite polynomial components of degree $t$, we have that $\sum_{t=1}^m \mathbf{E}_{(\mathbf{x},y) \sim D_{V^\perp}^S}[(\mathbb{1}(y \in I)^{[t]} - p_{S,I}^{[t]}(\mathbf{x}))^2] \leq \eta^2$.

From the rotational invariance of the Gaussian without loss of generality, we can denote by $\mathbf{e}_1, \ldots, \mathbf{e}_{k'}$ the orthonormal vectors $\mathbf{u}^{(1)}, \ldots, \mathbf{u}^{(k')}$. Furthermore, since $U$ is a subset of $V^\perp$ we similarly have that $\mathbf{E}_{(\mathbf{x},y) \sim D_{V^\perp}^S}[q_S(\mathbf{x}_U)\mathbb{1}(y \in I)] > \sigma$ for some $I \in \mathcal{I}$. Decomposing $q_S(\mathbf{x}_U)$ to the basis of Hermite polynomials we have that $q_S(\mathbf{x}_U) = \sum_{\beta \in \mathbb{N}^{d'}, 1 \leq \|\beta\|_1 \leq m} \hat{q}_S(\beta)H_\beta(\mathbf{x}_U)$ where $d' = \dim(U)$. Moreover, note that since $q_S$ has no component outside $\mathbf{e}_1, \ldots, \mathbf{e}_{k'}$ we have that $q_S(\mathbf{x}_U) = \sum_{\beta \in J} \hat{q}_S(\beta)H_\beta(\mathbf{x}_U)$, where $J$ denotes the set of $\beta \in \mathbb{N}^{d'}$ such that $\beta_i \geq 1$ for some $i \in [k']$. Considering the correlation of $q_S$ with the label interval $I$, we have that

$$\begin{aligned}
\mathbf{E}_{(\mathbf{x},y) \sim D_{V^\perp}^S}[q_S(\mathbf{x}_U)\mathbb{1}(y \in I)] &= \sum_{\beta \in J} \hat{q}_S(\beta)\mathbf{E}_{(\mathbf{x},y) \sim D_{V^\perp}^S}[H_\beta(\mathbf{x}_U)\mathbb{1}(y \in I)] \\
&\leq \left( \sum_{\beta \in J} \mathbf{E}_{(\mathbf{x},y) \sim D_{V^\perp}^S}[H_\beta(\mathbf{x}_U)\mathbb{1}(y \in I)]^2 \right)^{1/2} ,
\end{aligned}$$

where we used the fact that $\|q_S\|_2 = 1$ and the Cauchy-Schwarz inequality. Hence, we have that the sum of the squares of the Hermite coefficients for of the degree $m$ restriction the random variable $\mathbb{1}(y \in I)$ is at least $\sigma^2$.

Now evaluating the quadratic form of $\mathbf{M}_{S,I}$ using Fact E.6 gives us

$$
\sum_{i=1}^{k'} \mathbf{e}_i^\top \mathbf{M}_{S,I} \mathbf{e}_i = \sum_{i=1}^{k'} \mathbf{E}_{(\mathbf{x},y)\sim D_{V^\perp}^S}[(\nabla p_{S,z}(\mathbf{x}) \cdot \mathbf{e}_i)^2] \geq \sum_{\beta \in \mathbb{N}^d}(\sum_{i=1}^{k'} \beta_i)(\hat{p}_{S,z}(\beta))^2
$$

$$
\geq \sum_{\beta \in J} \mathbf{E}_{(\mathbf{x},y)\sim D_{V^\perp}^S}[H_\beta(\mathbf{x}_U)\mathbb{1}(y \in I)]^2 - 2\eta
$$

$$
\geq \sigma^2 - 2\eta \geq \sigma^2/2
$$

for $\eta \leq \sigma/2$. Thus, since $\dim(U) \leq K$ we have that $\mathbf{e}_j^\top \mathbf{M}_{S,I} \mathbf{e}_j \geq \sigma^2/(2K)$ for some $j \in [k']$. This concludes the proof of Claim D.11. $\qquad\square$

Now note that by the definition of an approximating partition (see Definition D.2) and Fact E.2, for all $S \in \mathcal{S}$ we have that $\mathbf{Pr}_D[S] = (\epsilon_1/k)^{\Omega(k)}$, hence $|\mathcal{S}| = (k/\epsilon_1)^{O(k)}$. Therefore, by union bound and Hoeffding's inequality, it holds that if $N \geq (k/\epsilon_1)^{Ck} \log(|\mathcal{S}|/\delta) = (k/\epsilon_1)^{O(k)} \log(1/\delta)$ for a sufficiently large constant $C > 0$, then with probability at least $1 - \delta$ we have that $|\mathbf{Pr}_{\widehat{D}}[S] - \mathbf{Pr}_D[S]| \leq \mathbf{Pr}_D[S]/2$ for all $S \in \mathcal{S}$. Hence, it is true that $\mathbf{Pr}_{\widehat{D}}[S] \geq \mathbf{Pr}_D[S]/2$, i.e., the number of samples that fall in each set $S \in \mathcal{S}$ is at least $N\mathbf{Pr}_D[S]/2 = N(\epsilon_1/k)^{\Omega(k)}$.

Therefore, for $N \geq (dm)^{Cm}(k/\epsilon_1)^{Ck} \log(|\mathcal{I}|/\delta)/\eta^C$ for a sufficiently large universal constant $C > 0$, by applying Claim D.11, we have that for all $S \in \mathcal{T}$ it holds that $(\mathbf{u}^{(i)})^\top \mathbf{M}_{S,I} \mathbf{u}^{(i)} \geq \sigma^2/(2K)$ for some $i \in [k'], I \in \mathcal{I}$. Hence, since $k' \leq K$, we have that there exists a subset $\mathcal{T}' \subseteq \mathcal{T}$ and $i \in [k']$ with $\sum_{S \in \mathcal{T}'} \mathbf{Pr}_{\widehat{D}}[S] \geq \sum_{S \in \mathcal{T}'} \mathbf{Pr}_D[S]/2 = \Omega(\alpha/K)$ such that for all $S \in \mathcal{T}'$ we have that $(\mathbf{u}^{(i)})^\top \mathbf{M}_{S,I} \mathbf{u}^{(i)} \geq \sigma^2/(2K)$ for some $I \in \mathcal{I}$.

Thus, for some $i \in [K]$ we have that

$$
(\mathbf{u}^{(i)})^\top \widehat{\mathbf{U}} \mathbf{u}^{(i)} \gtrsim \frac{\alpha}{K}\sigma^2/(2K) \geq \mathrm{poly}(\alpha\sigma/K) \,,
$$

where we used the fact that $\widehat{\mathbf{U}}$ is PSD.

Moreover, note that from Fact E.10 we have that $\mathrm{tr}(\mathbf{M}_{S,I}) = O(m)$. Therefore, we have that $\|\mathbf{M}_{S,I}\|_F \leq \mathbf{E}[\|\nabla p_{S,i}\|^2] = \mathrm{tr}(\mathbf{M}_{S,I})$. As a result, by the triangle inequality we can see that

$$
\left\|\widehat{\mathbf{U}}\right\|_F \leq \sum_{S \in \mathcal{S}, I \in \mathcal{I}} \|\mathbf{M}_{S,I}\|_F \mathbf{Pr}_{\widehat{D}}[S] \leq O(m) \sum_{S \in \mathcal{S}, I \in \mathcal{I}} \mathbf{Pr}_{\widehat{D}}[S] \leq m|\mathcal{I}| \,.
$$

Hence, by Fact E.7, we have that there exists a unit eigenvector $\mathbf{v}$ of $\widehat{\mathbf{U}}$ with eigenvalue at least $\mathrm{poly}(\alpha\sigma/K)$ for a sufficiently small polynomial, such that $\mathbf{u}^{(i)} \cdot \mathbf{v} \geq \mathrm{poly}(\alpha\sigma/(mK|\mathcal{I}|))$ for some $i \in [K]$. Moreover, the number of such eigenvectors, $|\mathcal{E}|$, is at most $\mathrm{poly}(mK|\mathcal{I}|/(\alpha\sigma))$. Which concludes the proof of Lemma D.10. $\qquad\square$

We now leverage the assumption that $h_{\mathcal{S}}$ exhibits large error, along with the previously established lemmata, to complete the proof.

Define the subspace $E \stackrel{\mathrm{def}}{=} \mathrm{span}(\{\mathbf{v} \in \mathcal{E} : \|\mathbf{v}_W\| \leq \rho/\sqrt{|\mathcal{E}|}\})$, where $\rho \stackrel{\mathrm{def}}{=} \epsilon^2/(CKLM)$. Notice that $\|\mathbf{v}_W\| \leq \rho$ holds for every unit vector $\mathbf{v} \in E$. Also denote by $U \stackrel{\mathrm{def}}{=} W_{V^\perp}$.

Since $\mathbf{E}_{(\mathbf{x},y)\sim D}[(h_{\mathcal{S}}(\mathbf{x}) - y)^2] > (\sqrt{\tau + \epsilon} + \sqrt{\mathrm{OPT}})^2$ and $f$ is assumed to be in the class $\mathcal{F}(m, \mathrm{OPT} + \epsilon, \alpha, K, M, L, B, \epsilon^2/(CM), \tau, \sigma)$ we have that Lemmas D.6 and D.7 together imply that there exists a subset $\mathcal{T} \subseteq \mathcal{S}$ with $\sum_{S \in \mathcal{T}} \mathbf{Pr}[S] \geq \alpha$, such that for each $S \in \mathcal{T}$ there exists an interval $I \in \mathcal{I}$ and a zero-mean, variance-one polynomial $q_S : U \to \mathbb{R}$ of degree at most $m$ such that $\mathbf{E}_{(\mathbf{x},y)\sim D}[q_S(\mathbf{x}_U)\mathbb{1}(y \in I) \mid \mathbf{x} \in S] > t$ for some $t = \mathrm{poly}(\sigma\epsilon/(MB2^m))$ and $\nabla_E q_S(\mathbf{x}_U) = 0$.

Denote by $\mathbf{u}^{(1)}, \ldots, \mathbf{u}^{(k')}$ an orthonormal basis of the space $U$ projected onto $E^\perp$. We have that $k' \geq 1$, otherwise we would not have the existence of $q_S$ that is defined over this space.

Finally, note that for $C$ a sufficiently large constant $\eta \leq t/2$ and also $\lambda = (t\alpha/K)^C$. Therefore, since $N = (dm)^{O(m)}(k/\epsilon_1)^{O(k)} \log(B/(\delta\epsilon_2))/\eta^{O(1)}$ applying Lemma D.10 for the space

$U$ projected onto $E^\perp$, we have that $|\mathcal{E}| = \text{poly}(2^m KMB/(\alpha\sigma\epsilon\epsilon_2))$ and there exists at least one vector $\mathbf{v} \in \mathcal{E}$ such that $\mathbf{u}^{(i)} \cdot \mathbf{v} \neq 0$ for some $i \in [K]$. However, note that since $\mathbf{u}^{(i)} \cdot \mathbf{v} \neq 0$, we have that $\mathbf{v}$ can not belong in $E$. Thus there exists a unit vector $\mathbf{w} \in W$ such that $|\mathbf{v} \cdot \mathbf{w}| \geq \rho/\sqrt{\mathcal{E}} \geq \text{poly}(\epsilon\sigma\epsilon_2\alpha/(KLMB2^m))$ which completes the proof of Proposition D.9. $\quad\square$

### D.1.3 Proof of Theorem D.5

In this section, given Proposition D.9, we proceed to the proof of Theorem D.5. Recall that, in Proposition D.9, we have shown that if our current approximation to the hidden subspace is not accurate enough to produce a function that has sufficiently small error, then Algorithm 4 efficiently finds a direction that has non-trivial correlation to the hidden subspace. In the proof that follows, we iteratively apply this argument to show that, after a moderate number of iterations, Algorithm 3 outputs a function with sufficiently small error.

*Proof of Theorem D.5.* Denote by $W^*$ the $K$-dimensional subspace defining $f$. We show that Algorithm 3, with high probability, returns a hypothesis $h$ with $L_2^2$ error at most $(\sqrt{\tau + \epsilon} + \sqrt{\text{OPT}})^2 + \epsilon$. Denote by $\mathbf{w}^{*(1)}, \ldots, \mathbf{w}^{*(K)} \in \mathbb{R}^d$ an orthonormal basis of $W^*$. Let $L_t$ be the list of vectors updated by the algorithm (Line 3c of Algorithm 3) and $V_t = \text{span}(L_t)$, $\dim(V_t) = k_t$. Also, let $\mathcal{S}_t$ for $t \in [T]$ be arbitrary $\epsilon_1$-approximating partitions of $V_t$ where $\epsilon_1$ the value set at Line 1. Let $h_t : \mathbb{R}^d \to [K]$ be a piecewise constant functions, defined as $h_t = h_{\mathcal{S}_t}$ according to Definition D.4 for the distribution $D$.

To prove the correctness of Algorithm 3, we need to show that if $h_t$ has significant error, then the algorithm improves its approximation $V_t$ to $W^*$. For quantifying the improvement at every step, we consider the following potential function $\Phi_t = \sum_{i=1}^K \left\| \mathbf{w}^{*(i)}_{V_t^\perp} \right\|^2$.

We will use the following fact that quantifies how much adding a correlating direction decreases $\Phi_t$.

**Fact D.12** (Potential Decrease, e.g., see Claim 2.16 [DIKZ25] )**.** *Let $\beta \geq 0$. If there exists a unit vectors $\mathbf{v}^{(t)} \in V_{t+1}$ and $\mathbf{w} \in W^*$ such that $\mathbf{w} \cdot \mathbf{v}^{(t)}_{V_t^\perp} \geq \beta$ , then $\Phi_{t+1} \leq \Phi_t - \beta^2$.*

We next prove the following claim which shows that the error of the functions $h_t$ decreases as we add more vectors.

**Claim D.13** (Error Decrease)**.** *For each $t \in [T]$, it holds $\mathbf{E}[(h_{t+1}(\mathbf{x}) - y)^2] \leq \mathbf{E}[(h_t(\mathbf{x}) - y)^2]$.*

*Proof.* Since the statement of Proposition D.9 holds for any $\epsilon_1$-partitions $\mathcal{S}_t$, $t \in [T]$, independent of the choice of basis and threshold points, we can assume that all the approximations $h_t$ are computed with respect to extensions of a common orthonormal basis and that all threshold points are aligned. Hence, $\mathcal{S}_{t+1}$ is a subdivision of $\mathcal{S}_t$. Thus, each set $S \in \mathcal{S}_t$ can be written as a union of sets $S_1, \ldots, S_l \in \mathcal{S}_{t+1}$, i.e., $S = \cup_{i=1}^l S_i$.

By definition, for any set $S \in \mathcal{S}_t$ we have $h_t(\mathbf{x}) = \mathbf{E}_{(\mathbf{x},y)\sim D}[y \mid \mathbf{x} \in S]$ for $\mathbf{x} \in S$, and similarly, for any $S_i \in \mathcal{S}_{t+1}$, $h_{t+1}(\mathbf{x}) = \mathbf{E}_{(\mathbf{x},y)\sim D}[y \mid \mathbf{x} \in S_i]$ for $\mathbf{x} \in S_i$. Thus, for any $S \in \mathcal{S}_t$,

$$
\begin{aligned}
\mathbf{E}_{(\mathbf{x},y)\sim D}[(h_{t+1}(\mathbf{x}) - y)^2, \mathbf{x} \in S] &= \sum_{i=1}^l \mathbf{Pr}[S_i]\,\mathbf{E}_{(\mathbf{x},y)\sim D}[(h_{t+1}(\mathbf{x}) - y)^2 \mid \mathbf{x} \in S_i] \\
&= \sum_{i=1}^l \mathbf{Pr}[S_i]\,\mathbf{E}_{(\mathbf{x},y)\sim D}\left[\left(\mathbf{E}_{(\mathbf{x},y)\sim D}[y \mid \mathbf{x} \in S_i] - y\right)^2 \Big| \mathbf{x} \in S_i\right] \\
&\leq \sum_{i=1}^l \mathbf{Pr}[S_i]\,\mathbf{E}_{(\mathbf{x},y)\sim D}\left[\left(\mathbf{E}_{(\mathbf{x},y)\sim D}[y \mid \mathbf{x} \in S] - y\right)^2 \Big| \mathbf{x} \in S_i\right] \\
&= \mathbf{E}_{(\mathbf{x},y)\sim D}[(h_t(\mathbf{x}) - y)^2, \mathbf{x} \in S] ,
\end{aligned}
$$

where we used the fact that $\mathbf{E}_{x\sim P}[(x - \mathbf{E}[x])^2] \leq \mathbf{E}_{x\sim P}[(x - z)^2]$ for any $z \in \mathbb{R}$ and distribution $P$ supported on $\mathbb{R}$. Summing over all $S \in \mathcal{S}_t$, completes the proof of Claim D.13. $\quad\square$

Finally, we prove the following claim which shows that the population piecewise constant functions are close to the corresponding empirical ones.

**Claim D.14** (Concentration of Piecewise Constant Approximation). *Let $\epsilon, \epsilon', \delta \in (0,1)$ and $k, K \in \mathbb{Z}_+$ with $\epsilon' \leq \epsilon/2$. Let $V \subseteq \mathbb{R}^d$ be a $k$-dimensional subspace, and consider a piecewise constant approximation $h : \mathbb{R}^d \to [K]$ of $D$, with respect to an $\epsilon'$-approximating partition of $V$. Let $\widehat{D}$ be the empirical distribution obtained from $N$ i.i.d. samples drawn from $D$, and let $\widehat{h}$ be a piecewise constant approximation of $\widehat{D}$ defined with respect to the same partition. If $\widehat{h}$ is computed using the median of means estimator (Fact E.13) for each $S \in \mathcal{S}$, then there exists $N = (k/\epsilon')^{O(k)}(M/\epsilon)^{O(1)}\log(1/\delta)$ such that, with probability at least $1 - \delta$, we have $\mathbf{E}[(\widehat{h}(\mathbf{x}) - y)^2] \leq \mathbf{E}[(h(\mathbf{x}) - y)^2] + \epsilon$.*

*Proof.* Denote by $\mathcal{S}$ the approximating partition for the functions $h, \widehat{h}$. First, note that by the definition of an approximating partition (Definition D.2) and by Fact E.2, we have that $\mathbf{Pr}_D[S] = (\epsilon'/k)^{\Omega(k)}$ and hence $|\mathcal{S}| = (k/\epsilon')^{O(k)}$.

Fix $S \in \mathcal{S}$. Note that since $\mathbf{Pr}_D[S]$ is lower bounded we have that the second moment of $y$ conditioned on $S$ can not be arbitrarily large. Specifically, $\mathbf{E}[y^2 \mid \mathbf{x} \in S] = \mathbf{E}[y^2\mathbb{1}(\mathbf{x} \in S)]/\mathbf{Pr}_D[S] = M(k/\epsilon')^{O(k)}$. Therefore, for any $S \in \mathcal{S}$ by applying Fact E.13 we have that, if the number of samples that fall in $S$ is $N_S = C(k/\epsilon')^{Ck}M^2/\epsilon^2\log(1/\delta)$ for a sufficiently large constant $C > 0$, then with probability $1 - \delta$ it holds $|\widehat{h}(\mathbf{x}) - \mathbf{E}[y \mid \mathbf{x} \in S]| \leq \epsilon$ for all $\mathbf{x} \in S$. Noting that by definition $h(\mathbf{x}) = \mathbf{E}[y \mid \mathbf{x} \in S]$ for all $\mathbf{x} \in S$, we have that $|\widehat{h}(\mathbf{x}) - h(\mathbf{x})| \leq \epsilon$ for all $\mathbf{x} \in S$.

Hence, by union bound and Hoeffding's inequality, if $N \geq (k/\epsilon')^{Ck}\log(|\mathcal{S}|/\delta) = (k/\epsilon')^{O(k)}\log(1/\delta)$ for a sufficiently large constant $C$, then with probability $1 - \delta$ we have that $|\mathbf{Pr}_{\widehat{D}}[S] - \mathbf{Pr}_D[S]| \leq \mathbf{Pr}_D[S]/2$ for all $S \in \mathcal{S}$. Hence, it is true that $\mathbf{Pr}_{\widehat{D}}[S] \geq \mathbf{Pr}_D[S]/2$, i.e., the number of samples that fall in each set $S \in \mathcal{S}$ is at least $N\mathbf{Pr}_D[S]/2 = N(\epsilon'/k)^{\Omega(k)}$. Therefore, if $N \geq C(k/\epsilon')^{Ck}M^2/\epsilon^2\log(1/\delta)$ for a sufficiently large constant $C$, then with probability at least $1 - \delta$ for all $S \in \mathcal{S}$ it holds that $|\widehat{h}(\mathbf{x}) - h(\mathbf{x})| \leq \epsilon$ for all $\mathbf{x} \in \mathbb{R}^d$.

As a result, by Cauchy-Schwarz $\mathbf{E}[(\widehat{h}(\mathbf{x}) - y)^2] \leq \mathbf{E}[(h(\mathbf{x}) - y)^2] + \epsilon^2 + \epsilon\sqrt{\mathbf{E}[(h(\mathbf{x}) - y)^2]}$. Moreover, by Jensen's inequality $\mathbf{E}[y^2], \mathbf{E}[h^2(\mathbf{x})] \leq M$, hence $\sqrt{\mathbf{E}[(h(\mathbf{x}) - y)^2]} \leq \sqrt{2M}$. Therefore, if $N \geq C(k/\epsilon')^{Ck}M^3/\epsilon^2\log(1/\delta)$, with probability at least $1 - \delta$ it holds that $\mathbf{E}[(\widehat{h}(\mathbf{x}) - y)^2] \leq \mathbf{E}[(h(\mathbf{x}) - y)^2] + \epsilon$. Which completes the proof of Claim D.14. $\qquad\square$

Note that from Lines 1 and 3 of Algorithm 3, we perform at most $\mathrm{poly}(BKLM2^m/(\epsilon\alpha\sigma))$ iterations. Furthermore, in each iteration, we update the vector set with at most $\mathrm{poly}(BKM2^m/(\epsilon\alpha\sigma))$ vectors. Hence, it follows that $k_t \leq \mathrm{poly}(BKLM2^m/(\epsilon\alpha\sigma))$, for all $t = 1, \ldots, T$.

Assume that $\mathbf{E}_{(\mathbf{x},y)\sim D}[(h_t(\mathbf{x}) - y)^2] > (\sqrt{\tau + \epsilon} + \sqrt{\mathrm{OPT}})^2$ for all $t = 1, \ldots, T$. Using the fact that $N = d^{Cm}2^{(BKLM2^m/(\epsilon\alpha\sigma))^C}\log(1/\delta)$ for a sufficiently large universal constant $C > 0$ (Line 2 of Algorithm 3), we can apply Proposition D.9 and conclude that, with probability $1 - \delta$, there exists unit vectors $\mathbf{v}^{(t)} \in V_{t+1}$ and unit vectors $\mathbf{w}^{(t)} \in W^*$ for $t = [T]$ such that $\mathbf{w}^{(t)} \cdot \mathbf{v}^{(t)}_{V_t^\perp} \geq \mathrm{poly}(\epsilon\sigma\alpha/(BMKL2^m))$. Thus, by Fact D.12, we have that with probability $1 - \delta$, for all $t \in [T]$, $\Phi_t \leq \Phi_{t-1} - \mathrm{poly}(\epsilon\sigma\alpha/(BMKL2^m))$. After $T$ iterations, it follows that $\Phi_T \leq \Phi_0 - T\mathrm{poly}(\epsilon\sigma\alpha/(BMKL2^m)) = K - T\mathrm{poly}(\epsilon\sigma\alpha/(BMKL2^m))$. However, since $T$ is set to be a sufficiently large polynomial of $2^m, B, M, L, K, 1/\epsilon, 1/\alpha$, and $1/\sigma$ we would arrive at a contradiction, since $\Phi_T \geq 0$. Hence, we have that $\mathbf{E}_{(\mathbf{x},y)\sim D}[(h_t(\mathbf{x}) - y)^2] \leq (\sqrt{\tau + \epsilon} + \sqrt{\mathrm{OPT}})^2$, for some $t \in \{1, \ldots, T\}$. Since the error of $h_t$ can only be decreasing by Claim D.13 and $h_t$ is close to its sample variant by Claim D.14, we have that $\mathbf{E}_{(\mathbf{x},y)\sim D}[(h(\mathbf{x}) - y)^2] \leq (\sqrt{\tau + \epsilon} + \sqrt{\mathrm{OPT}})^2 + \epsilon$.

**Sample and Computational Complexity:** From the analysis above we have that the algorithm terminates in $\mathrm{poly}(BKLM2^m/(\epsilon\alpha\sigma))$ iterations and at each iteration we draw of the order of $d^{O(m)}2^{\mathrm{poly}(BKLM2^m/(\epsilon\alpha\sigma))}\log(1/\delta)$ samples. Hence, we have that the total number of samples is $d^{O(m)}2^{\mathrm{poly}(BKLM2^m/(\epsilon\alpha\sigma))}\log(1/\delta)$. Moreover, we use at most $\mathrm{poly}(N)$ time as all operations can be implemented in polynomial time. $\qquad\square$

**Remark D.15.** While our algorithm is sufficient to obtain a polynomial dependence on $d$ for any constant values of $m$ and the other parameters, it is worth looking closer at the exponent of this polynomial. The algorithm that we have presented requires $d^m$ samples in order to accurately estimate the degree-$m$ parts of the relevant indicator functions in Line 2 of Algorithm 4. Observe that there is a quadratic gap between this bound and our SQ lower bound. Recall that our SQ lower bound requires either exponentially many queries or a query of accuracy $d^{-m/4}$, which in turn requires roughly $d^{m/2}$ samples to simulate.

We believe that this gap can be closed with a slightly different algorithm. In particular, instead of finding the polynomial approximation $p_{S,I}$ for each $S$ and $I$ in our discretizing approximation, and combining them into a matrix $\mathbf{U}$ to find directions that are often influential, we instead merely sample $(S, I)$ pairs and find the influential directions for each sampled pair (and note that with high probability we should find one which correlates with $W$). Then instead of estimating $p_{S,I}$ in $L_2$-norm (which requires roughly $d^m$ samples), we can treat it as an $m$-tensor $\mathbf{T}$, which in turn we flatten into a $d^{\lfloor m/2 \rfloor)} \times d^{\lceil m/2 \rceil}$ matrix $\mathbf{M}$. Since by assumption $\mathbf{T}$ has a large component in the $W$-directions, $\mathbf{M}$ must have some singular vectors with relatively large singular values that themselves correspond to polynomials in which the $W$-directions are influential. We can again find these if we estimate a suitable approximation to $\mathbf{M}$, but importantly for our purposes, we only need to estimate $\mathbf{M}$ to small error in operator norm rather than small error in Frobenius norm. This allows the algorithm to succeed with only around $d^{\lceil m/2 \rceil}$ samples.

## D.2 Learning Real-Valued MIMs: Realizable and Random Label Noise

In this section, we demonstrate that our algorithmic approach becomes more efficient when the label $y$ depends only on the projection of $\mathbf{x}$ to a low-dimensional subspace.

Specifically, we assume that there exists a $K$-dimensional subspace $W$ such that $y$ depends on $\mathbf{x}$ only through its projection onto $W$; that is, $\mathbf{Pr}[y = z \mid \mathbf{x} = \mathbf{u}] = \mathbf{Pr}[y = z \mid \mathbf{x}_W = \mathbf{u}_W]$ for all $\mathbf{u} \in \mathbb{R}^d, z \in \mathbb{R}$. This is a setting that captures both the realizable and the independent noise settings. We refer to this as the *MIM distribution* setting, indicating that the random variable $y$ is a MIM, i.e., it depends only on a low-dimensional subspace.

This structural assumption implies that all non-zero moments of the joint distribution are entirely within $W$. Consequently, our algorithm achieves a constant correlation gain in each iteration (unlike the agnostic setting analyzed in Proposition D.9), resulting in only $O(K)$ iterations overall. This leads to significant improvements in sample and computational complexities.

We begin by formally defining the class of distributions for which our algorithm guarantees a satisfactory solution. In Section D.3, we will demonstrate that for several MIM function classes, their distribution of examples belongs to this class.

**Definition D.16** (Well-Behaved MIM Distributions). Fix $d, K, m \in \mathbb{Z}_+$, $\alpha \in (0, 1)$ and $M, \tau, \sigma, \epsilon_1, \epsilon_2 > 0$. We say that a distribution $D$ over $\mathbb{R}^d \times \mathbb{R}$ whose marginal is $\mathcal{N}_d$ is a $(m, \alpha, K, M, \tau, \sigma, \epsilon_1, \epsilon_2)$-well-behaved MIM distribution, if the following conditions hold:

1. There exists a subspace $W \subseteq \mathbb{R}^d$ of dimension at most $K$ such that $y$ depends on $\mathbf{x}$ only through the projection onto $W$, i.e., $\mathbf{Pr}_{(\mathbf{x},y)\sim D}[y = z \mid \mathbf{x} = \mathbf{u}] = \mathbf{Pr}_{(\mathbf{x},y)\sim D}[y = z \mid \mathbf{x}_W = \mathbf{u}_W]$, for all $\mathbf{u} \in \mathbb{R}^d, z \in \mathbb{R}$.

2. The label has bounded variance, i.e., $\mathbf{E}_{(\mathbf{x},y)\sim D}[y^2] \leq M$.

3. For any subspace $V \subseteq \mathbb{R}^d$ with $\dim(V) \leq K$ and for any $(\eta_1, \eta_2)$-approximating discretization $(\mathcal{S}, \mathcal{I})$ with $\eta_1 \leq \epsilon_1, \eta_2 \leq \epsilon_2$

   (a) either $\mathbf{E}_{(\mathbf{x},y)\sim D}[(h_\mathcal{S}(\mathbf{x}_V) - y)^2] \leq \tau$, where $h_\mathcal{S}$ denotes the piecewise constant approximation of $D$ according to Definition D.4.

   (b) or there is a subset $\mathcal{T} \subseteq \mathcal{S}$ such that $\sum_{S \in \mathcal{T}} \mathbf{Pr}[S] \geq \alpha$ and for $U = W_{V^\perp}$, there exists a polynomial $p : \bar{U} \to \mathbb{R}$ of degree at most $m$ and an interval $I \in \mathcal{I}$ such that $\mathbf{E}_{(\mathbf{x},y)\sim D}[p(\mathbf{x}_U)\mathbb{1}(y \in I) \mid \mathbf{x} \in S] > \sigma\|p(\mathbf{x}_U)\|_2$ and $\mathbf{E}_\mathbf{x}[p(\mathbf{x}_U)] = 0$.

We now present the main theorem of this section, which shows that Algorithm 3 achieves improved complexity in this setting.

---

**Learn-MIM-Distributions**: Learning Well-Behaved MIM distributions
**Input:** Accuracy $\epsilon \in (0,1)$, failure probability $\delta \in (0,1)$, sample access to a distribution $D$ over $\mathbb{R}^d \times \mathbb{R}$, and parameters $\theta = (m, \alpha, K, M, \tau, \sigma, \epsilon_1, \epsilon_2)$ for which $D$ is a $\theta$-well-behaved MIM distribution.
**Output:** A hypothesis $h$ such that, with probability at least $1 - \delta$, $\mathbf{E}_{(\mathbf{x},y)\sim D}[(h(\mathbf{x}) - y)^2] \leq \tau + \epsilon$.

1. Let $C$ be a sufficiently large universal constant.

2. Let $L_1 = \emptyset$, $N \leftarrow (dm)^{Cm}(mK/(\epsilon_1\epsilon_2\alpha))^{CK}(M/(\epsilon\sigma))^C \log(1/\delta)$.

3. For $t = 1, \ldots, T$

    (a) Draw a set $S_t$ of $N$ i.i.d. samples from $D$.
    (b) $\mathcal{E}_t \leftarrow$ Algorithm 4$((K|\mathcal{I}|/(\sigma\epsilon\alpha))^C, \epsilon_1/K^4, \epsilon_2, 1/\epsilon_2^2, (\sigma\alpha/K)^C, \mathrm{span}(L_t), S_t, \theta)$.
    (c) Construct $L_{t+1}$ by adding one vector of $\mathcal{E}_t$ to $L_t$.

4. Construct $\mathcal{S}$, an $\epsilon_1$-approximating partition with respect to $\mathrm{span}(L_t)$ (see Definition D.2).

5. Draw $N$ i.i.d. samples from $D$ and construct the piecewise constant function $h_\mathcal{S}$ as follows: For each $S \in \mathcal{S}$, assign the median of $O(\log(1/\delta))$ means of the labels from the samples falling in $S$.

6. Return $h_\mathcal{S}$.

---

**Algorithm 5:** Learning Well-Behaved MIM distributions.

**Theorem D.17** (Learning Well-Behaved MIM Distributions). *Let $D$ be a $(m, \alpha, K, M, \tau, \sigma, \epsilon_1, \epsilon_2)$-well-behaved MIM distribution supported on $\mathbb{R}^d \times \mathbb{R}$. Then, Algorithm 3 draws $N = (dm)^{O(m)}2^{\mathrm{poly}(K)}(m/(\epsilon_1\epsilon_2\alpha))^{O(K)}(M/(\epsilon\sigma))^{O(1)}\log(1/\delta)$ i.i.d. samples from $D$, runs in time $\mathrm{poly}(N)$, and returns a hypothesis $h$ such that, with probability at least $1 - \delta$,*

$$\mathbf{E}_{(\mathbf{x},y)\sim D}[(h(\mathbf{x}) - y)^2] \leq \tau + \epsilon .$$

We now prove the following proposition, which demonstrates that—compared to the agnostic setting (see Proposition D.9)—improved correlation can be achieved in the MIM distribution setting.

**Proposition D.18** (Correctness of Learning Well-Behaved MIM Distributions). *There exists a sufficiently large universal constant $C > 0$ such that the following holds. Let $D$ be a $(m, \alpha, K, M, \tau, \sigma, \epsilon_1, \epsilon_2)$-well-behaved MIM distribution supported on $\mathbb{R}^d \times \mathbb{R}$ and let $W \subseteq \mathbb{R}^d$ with $\dim(W) = K$ be the hidden subspace of $D$. Let $V \subseteq \mathbb{R}^d$ be a $k$-dimensional subspace with $k \leq K$ such that $\|\mathbf{v}_{W^\perp}\| \leq \epsilon' \leq \epsilon(\epsilon_1\epsilon_2\alpha\sigma/K)^C$, for all unit vectors $\mathbf{v} \in V$. Also, let $(\mathcal{S}, \mathcal{I})$ be an $(\epsilon_1/K^4, \epsilon_2)$-approximating dicretization of $V \times \mathbb{R}$. Additionally, let $h_\mathcal{S}$ be a piecewise constant approximation of $D$, with respect to $\mathcal{S}$. There exists $N = (dm)^{O(m)}(K/\epsilon_1)^{O(k)}\log(|\mathcal{I}|/\delta)/\eta^{O(1)}$ such that if $\mathbf{E}_{(\mathbf{x},y)\sim D}[(h_\mathcal{S}(\mathbf{x}) - y)^2] > \tau + \epsilon$, then Algorithm 4, when given $N$ i.i.d. samples from $D$ and parameters $\eta \leq (\epsilon\sigma\epsilon_2\alpha/(mK))^C, \epsilon_1/K^4, \epsilon_2, \lambda = (\sigma\alpha/K)^C$, runs in time $\mathrm{poly}(N)$ and, with probability at least $1 - \delta$, returns a list of unit vectors $\mathcal{E}$ of size $|\mathcal{E}| = \mathrm{poly}(mK/(\epsilon_2\alpha\sigma))$, such that for every vector $\mathbf{v} \in \mathcal{E}$, there exists a unit vector $\mathbf{w} \in W$ with $\mathbf{w} \cdot \mathbf{v} \geq 1 - \epsilon$.*

*Proof.* Let $\widehat{\mathbf{U}}$ be the matrix computed in Line 3 of Algorithm 4 and let $\eta^2$ be the $L_2^2$ error chosen in the polynomial-regression step, i.e., Line 2 of Algorithm 4.

In the next claim, we show that if the sample size for a cubic region $S \in \mathcal{S}$ is sufficiently large, then for every $I \in \mathcal{I}$ the influence matrix, $\mathbf{M}_{S,I} \stackrel{\mathrm{def}}{=} \mathbf{E}_{\mathbf{x}\sim D_\mathbf{x}}[\nabla p_{S,I}(\mathbf{x}_{V^\perp})\nabla p_{S,I}(\mathbf{x}_{V^\perp})^\top \mid \mathbf{x} \in S]$, (see Line 3 of Algorithm 4) has small quadratic form when evaluated for any unit vector in $W^\perp + V$.

**Lemma D.19** (No Bad Vector). *Fix $S \in \mathcal{S}$. Suppose that the number of samples falling in $S$ is $N_S \geq (dm)^{Cm}\log(1/(\delta\epsilon_2))/\eta^C$ for a sufficiently large universal constant $C > 0$. Then, for any unit vector $\mathbf{v} \in (V + W^\perp) \cap V^\perp$ we have that $\mathbf{v}^\top \mathbf{M}_{S,I}\mathbf{v} \lesssim (K^5\epsilon'/\epsilon_1)^2 + m\eta^2$.*

*Proof of Lemma D.19.* Let $N_S, S \in \mathcal{S}$, be the number of samples that land in the set $S$. First, observe that the guarantee obtained at the regression step

$$\mathbf{E}_{(\mathbf{x},y)\sim D}[(\mathbb{1}(y \in I) - p_{S,I}(\mathbf{x}_{V^\perp}))^2 \mid \mathbf{x} \in S] \leq \min_{p' \in \mathcal{P}_m} \mathbf{E}_{(\mathbf{x},y)\sim D}[(\mathbb{1}(y \in I) - p'(\mathbf{x}_{V^\perp}))^2 \mid \mathbf{x} \in S] + \eta^2$$

is equivalent to

$$\mathbf{E}_{(\mathbf{x},y)\sim D^S_{V^\perp}}[(\mathbb{1}(y \in I) - p_{S,I}(\mathbf{x}))^2] \leq \min_{p' \in \mathcal{P}_m} \mathbf{E}_{(\mathbf{x},y)\sim D^S_{V^\perp}}[(\mathbb{1}(y \in I) - p'(\mathbf{x}))^2] + \eta^2 \,,$$

where $D^S_{V^\perp}$ is defined to be the marginal obtained by averaging $D$ over $V$ conditioned on $S$, i.e., $D^S_{V^\perp}(\mathbf{x}_{V^\perp}, y) = \mathbf{E}_{\mathbf{x}_V}[D(\mathbf{x}, y) \mid \mathbf{x} \in S]$. Moreover, by the properties of the Gaussian distribution, we have that the $\mathbf{x}$-marginal of $D^S_{V^\perp}$ is a standard Gaussian and denote by $k' \overset{\text{def}}{=} \dim(V^\perp) = d - k$. Hence, by applying Fact E.9 and the union bound, we have that with $N_S = (dm)^{O(m)} \log(1/(\delta\epsilon_2))/\eta^{O(1)}$ i.i.d. samples and runtime $\text{poly}(N_S, d)$, we can compute polynomials $p_{S,I}$ that satisfy the aforementioned condition with probability at least $1 - \delta$.

For $\beta \in \mathbb{N}^{k'}$, the Hermite coefficients of $p_{S,I}$ and $\mathbb{1}(y \in I)$ are defined by $\widehat{p}_{S,I}(\beta) \overset{\text{def}}{=} \mathbf{E}_{(\mathbf{x},y)\sim D^S_{V^\perp}}[H_\beta(\mathbf{x})p_{S,I}(\mathbf{x})]$ and $\widehat{g}_I(\beta) \overset{\text{def}}{=} \mathbf{E}_{(\mathbf{x},y)\sim D^S_{V^\perp}}[H_\beta(\mathbf{x})\mathbb{1}(y \in I)]$. By the orthogonality of Hermite polynomials, we have

$$\mathbf{E}_{(\mathbf{x},y)\sim D^S_{V^\perp}}[(\mathbb{1}(y \in I) - p_{S,I}(\mathbf{x}))^2] = \sum_{\beta \in \mathbb{N}^{k'}} (\widehat{p}_{S,I}(\beta) - \widehat{g}_I(\beta))^2 \,.$$

Thus, restricting the sum to multi-indices $\beta$ with $1 \leq \|\beta\|_1 \leq m$, it follows that $\sum_{\beta \in \mathbb{N}^{k'}, 1 \leq \|\beta\|_1 \leq m} (\widehat{p}_{S,I}(\beta) - \widehat{g}_I(\beta))^2 \leq \eta^2$.

Note that $\widehat{g}_I(\beta) = \mathbf{E}_{\mathbf{z}\sim\mathcal{N}_d}[H_\beta(\mathbf{z}_{V^\perp})\mathbf{Pr}_{(\mathbf{x},y)\sim D}[y \in I \mid \mathbf{x}_V \in S, \mathbf{z}_{V^\perp} = \mathbf{x}_{V^\perp}]]$. For $z \in V^\perp$ define the function $\widetilde{g}(\mathbf{z}) \overset{\text{def}}{=} \mathbf{Pr}_{(\mathbf{x},y)\sim D}[y \in I \mid \mathbf{x}_V \in S, \mathbf{z} = \mathbf{x}_{V^\perp}]$.

In the following claim, we show that the directional derivative of the function $\widetilde{g}$ in directions within $V + W^\perp$ is small. This implies that the quadratic form of $\mathbf{M}_{S,I}$ in these directions is also small, since $\widetilde{g}$ and $p_{S,I}$ match Hermite coefficients.

**Claim D.20** (Directional-derivative bound on the averaged indicator). *Fix an interval $I \in \mathcal{I}$ and cube $S \in \mathcal{S}$. Let $\mathbf{u} \in (V + W^\perp) \cap V^\perp$ be a unit vector. Then for all $\mathbf{x} \in V^\perp$ it holds that*

$$\left| \frac{d}{dt}\widetilde{g}(\mathbf{x} + t\mathbf{u}) \right| \leq \frac{K^5 \epsilon'}{\epsilon_1}.$$

*Proof of Claim D.20.* Define the function $g(\mathbf{z}) = \mathbf{Pr}_{(\mathbf{x},y)\sim D}[y \in I \mid \mathbf{x} = \mathbf{z}]$ and note that for all $\mathbf{z} \in \mathbb{R}^d$ it holds that $\widetilde{g}(\mathbf{z}_{V^\perp}) = \mathbf{E}_{\mathbf{x}\sim\mathcal{N}_d}[g(\mathbf{x}_V + \mathbf{z}_{V^\perp}) \mid \mathbf{x}_V \in S, \mathbf{x}_{V^\perp} = \mathbf{z}_{V^\perp}]$.

Let $\mathbf{u} = \mathbf{a} + \mathbf{b}$, where $\mathbf{a} \in V$ and $\mathbf{b} \in W^\perp$. Since $\mathbf{u} \in V^\perp$, it holds that $\|\mathbf{a}\|^2 + \mathbf{a} \cdot \mathbf{b} = \mathbf{u} \cdot \mathbf{a} = 0$. Note that by assumption $\|\mathbf{a} - \mathbf{a}_W\| \leq \epsilon' \|\mathbf{a}\|$, thus $\mathbf{a} = \mathbf{a}_W + \epsilon' \|\mathbf{a}\| \mathbf{v}$, for some unit vector $\mathbf{v} \in \mathbb{R}^d$. Hence, we have that $\|\mathbf{a}\|^2 + \mathbf{a}\cdot\mathbf{b} = \|\mathbf{a}\|^2 + \epsilon' \|\mathbf{a}\| (\mathbf{v}\cdot\mathbf{b})$ which implies that $\|\mathbf{a}\| \leq \epsilon' \|\mathbf{b}\|$. Therefore, by triangle inequality $\|\mathbf{b}\| \leq \|\mathbf{a}\| + 1$ as a result $\|\mathbf{a}\| = O(\epsilon')$.

Since $y$ depends on $\mathbf{x}$ only through the projection onto $W$, we have that $g(\mathbf{x} + t\mathbf{u}) = g(\mathbf{x} + t\mathbf{a})$. Thus, shifting $\mathbf{x}$ by $t\mathbf{u}$ is equivalent to shifting by $t\mathbf{a}$.

Denote by $\mathbf{y}$ a standard normal random variable over $V$ and by $\mathbf{x}$ a standard normal random variable over $V^\perp$. From the fact that $g$ is invariant in changes in $W$ we have that

$$\widetilde{g}(\mathbf{x} + t\mathbf{u}) = \frac{\mathbf{E}_{\mathbf{y}}[g(\mathbf{x} + \mathbf{y} + t\mathbf{u})\mathbb{1}(\mathbf{y} \in S)]}{\mathbf{Pr}[\mathbf{y} \in S]} = \frac{\mathbf{E}_{\mathbf{y}}[g(\mathbf{x} + \mathbf{y} + t\mathbf{a})\mathbb{1}(\mathbf{y} \in S)]}{\mathbf{Pr}[\mathbf{y} \in S]} \,.$$

Now define the new random variable $\mathbf{z} \overset{\text{def}}{=} \mathbf{y} + t\mathbf{a}$. By a change of variables we have that

$$\widetilde{g}(\mathbf{x} + t\mathbf{u}) = \frac{\mathbf{E}_{\mathbf{z}}[g(\mathbf{x} + \mathbf{z})\mathbb{1}(\mathbf{z} - t\mathbf{a} \in S)]}{\mathbf{Pr}[\mathbf{y} \in S]}$$

Therefore, shifting the argument by $t\mathbf{u}$ results to shifting the box by $-t\mathbf{a}$. Thus, in order to bound the derivative, it suffices to bound $\mathbf{Pr}[S\Delta(S - t\mathbf{a})]$, where $\Delta$ denotes the symmetric difference of the two sets.

Recall that the edge width of each cube is $\epsilon_1/K^4$, as defined in the statement of the proposition. Denote by $\phi : V \to \mathbb{R}$ the density function of a standard Gaussian random variable in $V$ and by

$\mathbf{v}^{(1)}, \ldots, \mathbf{v}^{(k)}$ the orthonormal basis of $V$ used to define $S$. Note that from the anti-concentration of the Gaussian distribution, it suffices to bound the volume of the symmetric difference. To this end define the following sequence of sets $S = S_0, S_1, \ldots, S_k = S - t\mathbf{a}$, where $S_i$ is equal to $S - t(\sum_{l=1}^{i} (\mathbf{a} \cdot \mathbf{v}^{(l)}) \mathbf{v}^{(l)})$. By the triangle inequality and a simple volume calculation we have that

$$\mathbf{Pr}[S\Delta(S - t\mathbf{a})] = \mathbf{Pr}[\mathbb{1}(\mathbf{x} \in S) \neq \mathbb{1}(\mathbf{x} \in (S - t\mathbf{a}))]$$

$$\leq \sum_{i=1}^{k} \mathbf{Pr}[\mathbb{1}(\mathbf{x} \in S_{i-1}) \neq \mathbb{1}(\mathbf{x} \in S_i)]$$

$$\leq (2/K^{4(k-1)}) \sum_{i \in [k]} \epsilon_1^{k-1} |t(\mathbf{a} \cdot \mathbf{v}^{(i)})| \sup_{\mathbf{x} \in S \cup (S - t\mathbf{a})} \phi(\mathbf{x})$$

$$\leq (2/K^{4(k-1)}) \sqrt{k} \, \|t\mathbf{a}\| \, \epsilon_1^{k-1} \sup_{\mathbf{x} \in S \cup (S - t\mathbf{a})} \phi(\mathbf{x}) \,,$$

where in the last inequality we used the Cauchy-Schwarz inequality. Define the following ratio

$$\rho_S(t) \stackrel{\text{def}}{=} \frac{\sup_{\mathbf{x} \in S \cup (S - t\mathbf{a})} \phi(\mathbf{x})}{\inf_{\mathbf{x} \in S} \phi(\mathbf{x})} \,.$$

Therefore, we have that

$$|\widetilde{g}(\mathbf{x} + t\mathbf{u}) - \widetilde{g}(\mathbf{x})| \leq \frac{\mathbf{Pr}[S\Delta(S - t\mathbf{a})]}{\mathbf{Pr}[\mathbf{x} \in S]} \leq \frac{K^{4k} \inf_{\mathbf{x} \in S} \phi(\mathbf{x})}{\epsilon_1^k} \mathbf{Pr}[S\Delta(S - t\mathbf{a})]$$

$$\lesssim K^4 \frac{\epsilon_1^{k-1} \sqrt{k} \, \|t\mathbf{a}\|}{\epsilon_1^k} \rho_S(t)$$

$$\lesssim K^4 \frac{\sqrt{k}|t|\epsilon'}{\epsilon_1} \rho_S(t) \,,$$

where in the second equation we used the fact that $\mathbf{Pr}[\mathbf{x} \in S] \geq \inf_{\mathbf{x} \in S} \phi(\mathbf{x}) \epsilon_1 / K^{4k}$ and in the third one we substituted our prederived upper bound for $\mathbf{Pr}[S\Delta(S - t\mathbf{a})]$. Finally, note that $\rho_S(t) = \exp((\mathbf{x} - \mathbf{y}) \cdot (\mathbf{x} + \mathbf{y})/2)$ for some $\mathbf{x} \in S$ and $\mathbf{y} \in S \cup (S - t\mathbf{a})$. Hence, $\lim_{t \to 0} \rho_S(t) \leq \exp(\epsilon_1 K^{-4} k^{3/2} \sqrt{\log(k^5/\epsilon_1)})$ which is bounded by a constant since $k \leq K$. This concludes the proof of Claim D.20. $\qquad\square$

Let a unit vector $\mathbf{u} \in (V + W^\perp) \cap V^\perp$, from Claim D.20 we have that $|\mathbf{u} \cdot \nabla \widetilde{g}(\mathbf{x}_{V^\perp})| \lesssim K^5 \epsilon'/\epsilon_1$ for all $\mathbf{x} \in \mathbb{R}^d$. From the rotational invariance of the standard gaussian, let us for simplicity denote $\mathbf{u}$ by $\mathbf{e}_1$. By applying Fact E.6, we have that

$$\sum_{\beta \in \mathbb{N}^{k'}} \beta_1 \widehat{g}_I(\beta)^2 \lesssim \left( \frac{K^5 \epsilon'}{\epsilon_1} \right)^2 \,.$$

Consequently,

$$\mathbf{e}_1^\top \mathbf{E}_{(\mathbf{x},y) \sim D_{V^\perp}^S} [\nabla p_{S,I}(\mathbf{x}) \nabla p_{S,I}(\mathbf{x})^\top] \mathbf{e}_1 = \sum_{\beta \in \mathbb{N}^{k'}} \beta_1 \widehat{p}_{S,I}(\beta)^2 \lesssim \left( \frac{K^5 \epsilon'}{\epsilon_1} \right)^2 + m\eta^2 \,.$$

This completes the proof of Lemma D.19. $\qquad\square$

First note that by the definition of an approximating partition (Definition D.2) and Fact E.2, for all $S \in \mathcal{S}$ we have that $\mathbf{Pr}_D[S] = (\epsilon_1/(Kk))^{\Omega(k)}$ and $|\mathcal{S}| = (kK/\epsilon_1)^{O(k)}$. Therefore, by the union bound and Hoeffding's inequality, it holds that if $N \geq (kK/\epsilon_1)^{Ck} \log(|\mathcal{S}|/\delta) = (kK/\epsilon_1)^{O(k)} \log(1/\delta)$ for a sufficiently large constant $C > 0$, then with probability at least $1 - \delta$ we have that $|\mathbf{Pr}_{\widehat{D}}[S] - \mathbf{Pr}_D[S]| \leq \mathbf{Pr}_D[S]/2$ for all $S \in \mathcal{S}$. Hence, $\mathbf{Pr}_{\widehat{D}}[S] \geq \mathbf{Pr}_D[S]/2$, i.e., the number of samples that fall in each set $S \in \mathcal{S}$ is at least $N\mathbf{Pr}_D[S]/2 = N(\epsilon_1/(Kk))^{\Omega(k)}$.

Therefore, if $N \geq (dm)^{Cm}(kK/\epsilon_1)^{Ck} \log(1/(\delta\epsilon_2))/\eta^C$ for a sufficiently large universal constant $C > 0$, then by applying Lemma D.19 we have the following: with probability $1 - \delta$, for any unit

vector $\mathbf{v} \in (W^\perp + V) \cap V^\perp$

$$\mathbf{v}^\top \widehat{\mathbf{U}} \mathbf{v} \leq \sum_{S \in \mathcal{S}, I \in \mathcal{I}} \left( \left( \frac{K^5 \epsilon'}{\epsilon_1} \right)^2 + m\eta^2 \right) \mathbf{Pr}_{\widehat{D}}[S] \leq |\mathcal{I}| \left( \left( \frac{K^5 \epsilon'}{\epsilon_1} \right)^2 + m\eta^2 \right).$$

Hence, since $\widehat{\mathbf{U}}$ is a symmetric PSD matrix, applying Fact E.8 we have that for all $\mathbf{u} \in \mathcal{E}$ and unit vectors $\mathbf{v} \in (W^\perp + V) \cap V^\perp$ it holds that $|\mathbf{u} \cdot \mathbf{v}| \leq \sqrt{m\eta^2 + (K^5 \epsilon'/\epsilon_1)^2}/\text{poly}(\sigma \epsilon_2 \alpha/K) \leq (m\eta^2 + K^5 \epsilon'/\epsilon_1)/\sqrt{\text{poly}(\sigma \epsilon_2 \alpha/K)}$. As a result, substituting $\epsilon'$ and $\eta$, since $C$ is a sufficiently large constant, we have that for all $\mathbf{u} \in \mathcal{E}$ and any unit vector $\mathbf{v} \in (W^\perp + V) \cap V^\perp$ it holds $|\mathbf{u} \cdot \mathbf{v}| \leq \epsilon$. Note that the subspace space $(W^\perp + V) \cap V^\perp$ is the subspace constructed by projecting every vector of $W^\perp$ to $V^\perp$. Consequently, for any unit vector $\mathbf{v} \in W^\perp$ and any $\mathbf{u} \in \mathcal{E}$, we have that $|\mathbf{v}_{V^\perp} \cdot \mathbf{u}/\|\mathbf{v}_{V^\perp}\|| \leq \epsilon$, which implies that $|\mathbf{v} \cdot \mathbf{u}| \leq \epsilon$, since $\|\mathbf{v}_{V^\perp}\| \leq 1$.

Consequently, by applying Lemma D.10, we have that if $N \geq (dm)^{Cm}(kK/\epsilon_1)^{Ck} \log(1/(\delta \epsilon_2))/\eta^C$ for a sufficiently large universal constant $C > 0$, the set $\mathcal{E}$ in non-empty. Thus, as any $\mathbf{u} \in \mathcal{E}$ is a unit vector and $\|\mathbf{u}_{W^\perp}\| \leq \epsilon$, we have $\|\mathbf{u}_W\| \geq \sqrt{1 - \epsilon^2} \geq 1 - \epsilon$. This completes the proof of Proposition D.18. □

Now we are ready to prove Theorem D.17. Our proof is similar to the proof of Theorem D.5. The difference is that we are going to use Proposition D.18 which provides improved correlation when compared to Proposition D.9.

*Proof of Theorem D.17.* We show that Algorithm 5, with high probability, returns a hypothesis $h$ with $L_2^2$ error at most $\tau + \epsilon$. Let $W$ be the $K$-dimensional subspace that $y$ depends on. Let $L_t$ be the set of vectors maintained by the algorithm (Line 3c) and $V_t = \text{span}(L_t)$, $\dim(V_t) = k_t$. Also let $\epsilon_1$ be the partition width parameter (see Definition D.16), and for $t \in [T]$ let $\mathcal{S}_t$ be arbitrary $\epsilon_1/K^4$-approximating partitions with respect to $V_t$ (see Definition D.2). Let $h_t : \mathbb{R}^d \to [K]$ be piecewise constant functions, defined as $h_t = h_{\mathcal{S}_t}$ according to Definition D.4 for the distribution $D$.

Note that from Lines 1, 3 of Algorithm 5, we perform $T \stackrel{\text{def}}{=} K$ iterations. Furthermore, in each iteration, we update the vector set by adding one vector. Hence, $k_t \leq K$ for all $t \in [T]$.

Assume that $\mathbf{E}_{(\mathbf{x},y) \sim D}[(h_t(\mathbf{x}) - y)^2] > \tau + \epsilon/2$ for all $t = [T]$. Denote by $\mathbf{v}^{(t)} \in V_{t+1} \cap V_t^\perp, t \in [T]$ the unit vectors added at each iteration and let $C$ be a sufficiently large universal constant. Note that in order to add a new vector $\mathbf{v}^{(t)} \in V_{t+t} \cap V_t^\perp$ with $\|(\mathbf{v}^{(t)})^W\| \geq 1 - \rho$ by applying Proposition D.18, we need to already have that every unit vector $\mathbf{v} \in V_{t-1}$ satisfies $\|\mathbf{v}^W\| \geq 1 - \rho(\epsilon_1 \epsilon_2 \alpha/(mK))^C$. Moreover, since $\mathbf{v}^{(t)}$ are orthonormal in this case, for all unit vectors $\mathbf{v} \in V_t$ it holds that $\|\mathbf{v}^W\| \geq 1 - \rho(\epsilon_1 \epsilon_2 \alpha/(mK))^C$. Thus, if the number of samples is sufficiently large, for all iterations $t \in [T]$ applying the proposition for $\rho = (1/(2K))(\epsilon_1 \epsilon_2 \alpha/(mK))^{CK}$ (in place of $\epsilon$) would result to orthonormal vectors $\mathbf{v}^{(t)}$ with $\|(\mathbf{v}^{(t)})^W\| \geq 1 - 1/2K$ for all $t \in [K]$.

Therefore, using the fact that $N = (dm)^{Cm}(mK/(\epsilon_1 \epsilon_2 \alpha))^{CK}(M/(\epsilon \sigma))^C \log(1/\delta)$ (Line 1 of Algorithm 5), we can iteratively apply Proposition D.18 and conclude that, with probability $1 - \delta$, there exist unit vectors $\mathbf{v}^{(t)} \in V_{t+1}$ and unit vectors $\mathbf{w}^{(t)} \in W$ for $t \in [T]$ such that $\mathbf{w}^{(t)} \cdot \mathbf{v}_{V_t^\perp}^{(t)} \geq 1 - 1/(2K)$. Thus, from Fact D.12, we have that with probability $1 - \delta$, for all $t \in [T]$, $\Phi_t \leq \Phi_{t-1} - 1 + 1/(2K)$. After $T$ iterations, it follows that $\Phi_T \leq \Phi_0 - T + T/(2K)$. However, if $T$ is set to be $K + 1$ we would arrive at a contradiction, since $\Phi_T \geq 0$. Hence, we have that $\mathbf{E}_{(\mathbf{x},y) \sim D}[(h_t(\mathbf{x}) - y)^2] \leq \tau + \epsilon/2$, for some $t \in \{1, \dots, T\}$. Since the error of $h_t$ can only be decreasing (see Claim D.13), and $h_t$ is close to its sample variant by Claim D.14, we have that $\mathbf{E}_{(\mathbf{x},y) \sim D}[(h(\mathbf{x}) - y)^2] \leq \tau + \epsilon$.

**Sample and Computational Complexity:** Note that the algorithm terminates in $O(K)$ iterations and at each iteration we draw $N = (dm)^{O(m)} 2^{\text{poly}(K)}(m/(\epsilon_1 \epsilon_2 \alpha))^{O(K)}(M/(\epsilon \sigma))^{O(1)} \log(1/\delta)$ samples. Hence, the total sample size is $(dm)^{O(m)} 2^{\text{poly}(K)}(m/(\epsilon_1 \epsilon_2 \alpha))^{O(K)}(M/(\epsilon \sigma))^{O(1)} \log(1/\delta)$. Moreover we use at most $\text{poly}(N)$ time, as all operations can be implemented in polynomial time. □

### D.3 Algorithmic Applications to Structured Multi-Index Model Classes

In this section, we show that our general algorithm can be leveraged to obtain state-of-the-art guarantees for learning positive-homogeneous Lipschitz MIMs and polynomials in a few relevant directions. The former result is new and subsumes prior work on homogeneous ReLU networks.

#### D.3.1 Learning Positive-Homogeneous Lipschitz MIMs

For each application, we show that the resulting distribution $D$ over examples $(\mathbf{x}, y)$ is a well-behaved MIM distribution with favorable parameters, and we consequently apply Theorem D.17.

First, we recall the target class definition.

**Definition D.21** (Positive-Homogeneous Lipschitz MIMs). For $K \in \mathbb{Z}_+$ and $L > 0$, we define $\mathcal{H}_{K,L}$ to be the class of all $L$-Lipschitz $K$-MIMs $f : \mathbb{R}^d \to \mathbb{R}$ such that $f$ is positive-homogeneous, meaning $f(t\mathbf{x}) = t f(\mathbf{x})$ for all $t > 0$ and $\mathbf{x} \in \mathbb{R}^d$, and $f$ has unit $L_2^2$ norm under the Gaussian distribution, that is, $\mathbf{E}_{\mathbf{x} \sim \mathcal{N}_d}[f^2(\mathbf{x})] = 1$.

This class generalizes the class of Lipschitz and homogeneous ReLU networks of arbitrary depth, since the ReLU activation is itself positive-homogeneous. We prove that by applying our algorithm we can learn the aforementioned class efficiently. Specifically, note the following theorem.

**Theorem D.22** (PAC Learning $\mathcal{H}_{K,L}$). *Let $f : \mathbb{R}^d \to \mathbb{R}$ be a function in $\mathcal{H}_{K,L}$ and let $D$ be the joint distribution of $(\mathbf{x}, f(\mathbf{x}))$, where $\mathbf{x} \sim \mathcal{N}_d$. Then, Algorithm 5 draws $N = d^2 2^{O(K^3 L^2/\epsilon^2)} \log(1/\delta)$ i.i.d. samples from $D$, runs in time $\mathrm{poly}(N)$, and returns a hypothesis $h$ such that, with probability at least $1 - \delta$, it holds $\mathbf{E}_{(\mathbf{x},y) \sim D}[(h(\mathbf{x}) - y)^2] \leq \epsilon$.*

Moreover, consider the following class of bounded depth ReLU Networks.

**Definition D.23** (Lipschitz and Homogeneous ReLU Networks). Let $\mathcal{F}_{S,K,L}$ denote the concept class of $L$-Lipschitz, homogeneous (feedforward) ReLU networks over $\mathbb{R}^d$ of size $S$ that depend only on the projection onto a subspace of dimension at most $K$. Specifically, $f \in \mathcal{F}_{S,K,L}$ if $f$ is $L$-Lipschitz, $\mathbf{E}_{\mathbf{x} \sim \mathcal{N}_d}[f^2(\mathbf{x})] = 1$ and there exist weight matrices $\mathbf{W}_i \in \mathbb{R}^{k_{i+1} \times k_i}$, $i \in [D-1]$ with $k_1 = d$ and $k_D = 1$, $\mathrm{rank}(\mathbf{W}_1) \leq K$, for which $f(\mathbf{x}) = \mathbf{W}_D \phi(\mathbf{W}_{D-1}(\cdots \phi(\mathbf{W}_1 \mathbf{x}) \cdots))$, where $\phi(z) = \max\{z, 0\}$ is the ReLU activation applied entrywise, and $k_1 + \cdots + k_{D-1} = S$.

Since the class of ReLU Networks we defined is positive homogeneous we can apply Theorem D.22 and obtain the following implication.

**Corollary D.24** (Learning Homogeneous ReLU Networks). *Let $f : \mathbb{R}^d \to \mathbb{R}$ be a ReLU network in the class $\mathcal{F}_{S,L,K}$ and let $D$ be the joint distribution of $(\mathbf{x}, f(\mathbf{x}))$, where $\mathbf{x} \sim \mathcal{N}_d$. Then, Algorithm 5 draws $N = d^2 2^{O(K^3 L^2/\epsilon^2)} \log(1/\delta)$ i.i.d. samples from $D$, runs in time $\mathrm{poly}(N)$, and returns a hypothesis $h$ such that, with probability at least $1 - \delta$, it holds $\mathbf{E}_{(\mathbf{x},y) \sim D}[(h(\mathbf{x}) - y)^2] \leq \epsilon$.*

We remark that our primary result about learning general positive-homogeneous Lipschitz functions is not achievable by the algorithm of [CKM22], as it is a proper algorithm that always outputs a homogeneous ReLU network. The fact that we use a general approximation of Lipschitz functions by piecewise constant ones (Definition D.4) makes this result possible. Furthermore, the complexity of [CKM22] depends exponentially on the size of the network $S$, which can be significantly larger than the rank $K$ of the first layer.

Before we prove Theorem D.22, we first present a key structural result for the class $\mathcal{H}_{K,L}$. For a distribution $D$ over $\mathbb{R}^d \times \mathbb{R}$, a function $f : \mathbb{R}^d \to \mathbb{R}$, a scalar $\tau > 0$, and a subspace $V \subseteq \mathbb{R}^d$, define the following matrix:

$$\mathbf{M}_\tau^V \overset{\mathrm{def}}{=} \mathbb{E}_{(\mathbf{x},y) \sim D}\Big[T(\mathbf{x}_V, y)\big(\mathbf{x}_{V^\perp}(\mathbf{x}_{V^\perp})^\top - \Pi_{V^\perp}\big)\Big], \quad T(\mathbf{x}_V, y) = \mathbb{1}(|y - f(\mathbf{x}_V)| > \tau). \quad (3)$$

The following lemma states that this filtered second moment matrix has large correlation with some direction in $W^{\perp V}$.

**Lemma D.25** (Generalization of Lemma 5.5 in [CKM22]). *Let $V, W$ be subspaces of $\mathbb{R}^d$ with $\dim(W) = K, \dim(V) = k$ and $V \subseteq W$. Let $f : \mathbb{R}^d \to \mathbb{R}$ be a function in $\mathcal{H}_{K,L}$ such that $f(\mathbf{x}_W) = f(\mathbf{x})$. Suppose that $\mathbb{E}_{\mathbf{x} \sim \mathcal{N}_d}\big[(f(\mathbf{x}) - f(\mathbf{x}_V))^2\big] \geq \epsilon^2$ for some $\epsilon > 0$. For*

$\tau > 2\sqrt{K-k}\,L$, and $\mathbf{M}_\tau^V \in \mathbb{R}^{d\times d}$ the matrix defined in Equation (3), we have that there exists a unit vector $\mathbf{w} \in W$ such that $\mathbf{w}^\top \mathbf{M}_\tau^V \mathbf{w} \gtrsim e^{-3K\tau^2/\epsilon^2}\,\frac{\tau\epsilon}{\sqrt{K}L^2}$.

*Proof of Lemma D.25.* Let $U$ denote the projection of the subspace $W$ onto $V^\perp$. Note that since $f$ is $L$-Lipschitz the condition $|f(\mathbf{x}) - f(\mathbf{x}_V)| \geq 2\sqrt{K-k}L$ implies that $\|\mathbf{x}_U\|^2 \geq 2(K-k)$. As a result, by taking the trace inner product with the projection matrix onto $W$, we have that there exists a unit vector $\mathbf{w} \in W$ such that

$$\mathbf{w}^\top \mathbf{M}_\tau^V \mathbf{w} \geq (K-k)\mathbf{Pr}[|f(\mathbf{x}) - f(\mathbf{x}_V)| \geq 2\sqrt{K-k}L]\;.$$

Note that since $f$ is a positive-homogeneous function we have that $f(\mathbf{x}) - f(\mathbf{x}_V)$ is positive-homogeneous. Moreover, because $f$ depends only on the projection of $\mathbf{x}$ onto $W$ and $V$ is a subspace of $W$, the function $f(\mathbf{x}) - f(\mathbf{x}_V)$ depends also only on the projection of $\mathbf{x}$ onto $W$. Thus in order to complete the proof it suffices to prove an anticoncentration result about positive-homogeneous functions:

**Claim D.26** (Anticoncentration of Positive-Homogeneous Functions). *If $G : \mathbb{R}^K \to \mathbb{R}$ is positive-homogeneous and $L$-Lipschitz and $\mathbf{E}[G^2] \geq \sigma^2$, then for any $s \geq 0$,*

$$\mathbf{Pr}_{\mathbf{x}\sim\mathcal{N}_K}[|G(\mathbf{x})| > s] \gtrsim \exp(-3Ks^2/\sigma^2)\frac{s\sigma}{\sqrt{K}L^2}\;.$$

*Proof of Claim D.26.* Note that if $\mathbf{x} \sim \mathcal{N}_K$ then it can be decomposed as $\mathbf{x} = \sqrt{r}\mathbf{v}$ where $r \sim \chi_m^2$ and $\mathbf{v}$ is drawn uniformly from $\mathbb{S}^{K-1}$ independent of $r$. First note that by independence of $r$ and $\mathbf{v}$ we have that

$$\sigma^2 = \mathbf{E}_{\mathbf{x}\sim\mathcal{N}_K}[G^2(\mathbf{x})] = \mathbf{E}[r]\mathbf{E}[G^2(\mathbf{v})] = K\mathbf{E}[G^2(\mathbf{v})]\;.$$

Thus, $\mathbf{E}[G^2(\mathbf{v})] = \sigma^2/K$. Hence, by elementary anticoncentration Fact E.14 we have that

$$\begin{aligned}
\mathbf{Pr}_{\mathbf{x}\sim\mathcal{N}_K}[|G(\mathbf{x})| \geq s] \geq \mathbf{Pr}[rG^2(\mathbf{v}) \geq s^2] &\geq \mathbf{Pr}[r \geq 2Ks^2/\sigma^2]\mathbf{Pr}[|G(\mathbf{v})| \geq \sigma/\sqrt{2K}\,] \\
&\geq \mathbf{Pr}[r \geq 2Ks^2/\sigma^2]\frac{\sigma^2}{2KL^2} \\
&\gtrsim \exp(-3Ks^2/\sigma^2)\frac{s\sigma}{\sqrt{K}L^2}\;,
\end{aligned}$$

where we used the well-known fact that $\mathbf{Pr}_{r\sim\chi_m^2}[r \geq x] \geq \mathrm{erfc}(1/\sqrt{x})$ and $\mathrm{erfc}(x) \geq \sqrt{2/\pi}\frac{xe^{-x^2/2}}{x^2+1}$, for all $x \geq 0$. Which concludes the proof of Claim D.26. $\qquad\square$

Therefore, we can apply Claim D.26 for the function $f(\mathbf{x}) - f(\mathbf{x}_V)$, which concludes the proof of Lemma D.25. $\qquad\square$

Now following this structural result, in order to show that the class $\mathcal{H}_{K,L}$ leads to well-behaved MIM distributions and allows the application of Theorem D.17, it suffices to establish the existence of non-trivial moments for a cube-interval pair.

We prove this result in two stages. First, we show that if it is possible to obtain distinguishing moments by conditioning on a region of $V \times \mathbb{R}$ that is well-approximated by cubes and intervals, then there exists a specific cube-interval pair exhibiting distinguishing moments. Consequently, we prove that the region $T$ defined in Equation (3) can indeed be well-approximated by such cube-interval pairs.

**Lemma D.27** (Label Transformation Approximation). *Let $D$ be a distribution supported on $\mathbb{R}^d \times \mathbb{R}$, whose $\mathbf{x}$-marginal is $\mathcal{N}_d$, and let $V$ be a subspace of $\mathbb{R}^d$. Suppose that $T(\mathbf{x}_V, y) : V \times \mathbb{R} \to \{0,1\}$ is a label transformation function and that $p : \mathbb{R}^d \to \mathbb{R}$ is a zero mean, variance one polynomial such that $\mathbf{E}_{(\mathbf{x},y)\sim D}[p(\mathbf{x})T(\mathbf{x}_V, y)] \geq \sigma$. Let $P$ be a partition of $V \times \mathbb{R}$, and let $P' \subseteq P$. Define the approximation $\widehat{T}(\mathbf{x}_V, y) \stackrel{\text{def}}{=} \sum_{R\in P'} \mathbb{1}((\mathbf{x}_V, y) \in R)$. If $\mathbf{Pr}_{(\mathbf{x},y)\sim D}[T(\mathbf{x}_V, y) \neq \widehat{T}(\mathbf{x}_V, y)] \leq \frac{\sigma^2}{4}$, then there exists some $R \in P'$ such that $\mathbf{E}[p(\mathbf{x})\mathbb{1}((\mathbf{x}_V, y) \in R)] \geq \frac{\sigma}{2|P'|}$.*

*Proof.* First we can write

$$\mathbf{E}_{(\mathbf{x},y)\sim D}\big[p(\mathbf{x})T(\mathbf{x}_V,y)\big] = \mathbf{E}_{(\mathbf{x},y)\sim D}\big[p(\mathbf{x})\widehat{T}(\mathbf{x}_V,y)\big] + \mathbf{E}_{(\mathbf{x},y)\sim D}\big[p(\mathbf{x})(T(\mathbf{x}_V,y)-\widehat{T}(\mathbf{x}_V,y))\big].$$

Therefore, we have that

$$\mathbf{E}_{(\mathbf{x},y)\sim D}\big[p(\mathbf{x})\widehat{T}(\mathbf{x}_V,y)\big] \geq \sigma - \left|\mathbf{E}_{(\mathbf{x},y)\sim D}\big[p(\mathbf{x})(T(\mathbf{x}_V,y)-\widehat{T}(\mathbf{x}_V,y))\big]\right|.$$

Since $p(\mathbf{x})$ has variance one, by Cauchy–Schwarz,

$$\left|\mathbf{E}_{(\mathbf{x},y)\sim D}\big[p(\mathbf{x})(T(\mathbf{x}_V,y)-\widehat{T}(\mathbf{x}_V,y))\big]\right| \leq \sqrt{\mathbf{E}[p(\mathbf{x})^2]\mathbf{E}[(T(\mathbf{x}_V,y)-\widehat{T}(\mathbf{x}_V,y))^2]} \leq \sqrt{\sigma/2}\,.$$

Thus,

$$\mathbf{E}_{(\mathbf{x},y)\sim D}\big[p(\mathbf{x})\widehat{T}(\mathbf{x}_V,y)\big] \geq \sigma/2.$$

We can write

$$\mathbf{E}_{(\mathbf{x},y)\sim D}\big[p(\mathbf{x})\widehat{T}(\mathbf{x}_V,y)\big] = \sum_{R\in P'} \mathbf{E}_{(\mathbf{x},y)\sim D}\big[p(\mathbf{x})\mathbb{1}((\mathbf{x}_V,y)\in R)\big].$$

Hence, by the pigeonhole principle, there exists some $R \in P'$ such that

$$\mathbf{E}_{(\mathbf{x},y)\sim D}\big[p(\mathbf{x})\mathbb{1}((\mathbf{x}_V,y)\in R)\big] \geq \frac{1}{|P'|}\mathbf{E}_{(\mathbf{x},y)\sim D}\big[p(\mathbf{x})\widehat{T}(\mathbf{x}_V,y)\big] \geq \frac{\sigma}{2|P'|}.$$

This completes the proof of Lemma D.27. $\qquad\square$

In the following lemma we show that the label transformation defined in Equation (3) can be approximated arbitrary well by a piecewise constant function over a discretization of $V \times \mathbb{R}$ in cubes and intervals.

**Lemma D.28** (Cube-Interval Approximation of $T$). *Let $d,k \in \mathbb{Z}_+, \epsilon, \eta \in (0,1), \tau, L \in \mathbb{R}_+$ with $\tau \geq L \geq \epsilon$. Let $V, W$ be subspaces of $\mathbb{R}^d$ with $V \subseteq W$, $\dim(V) = k$ and $\dim(W) = K$. Let $f : \mathbb{R}^d \to \mathbb{R}$ be function in $\mathcal{H}_{K,L}$ such that $f(\mathbf{x}) = f(\mathbf{x}_W)$, and let $D$ be the joint distribution of $(\mathbf{x}, f(\mathbf{x}))$ where $\mathbf{x} \sim \mathcal{N}_d$. Denote by $T : V \times \mathbb{R} \to \{0,1\}$ the function defined in Equation (3). Let $(\mathcal{S}, \mathcal{I})$ be an $\eta$-approximating discretization of $V \times \mathbb{R}$, with $\eta \leq \epsilon^4/(L\sqrt{k}K)$.*

*There exists a subset of the discretization $P \subseteq \mathcal{S} \times \mathcal{I}$ such that for the function $\widehat{T}(\mathbf{x}_V,y) = \sum_{(S,I)\in P} \mathbb{1}(\mathbf{x}_V \in S, y \in I)$ it holds that $\mathbf{Pr}_{(\mathbf{x},y)\sim D}[T(\mathbf{x}_V,y) \neq \widehat{T}(\mathbf{x}_V,y)] \lesssim \epsilon$.*

*Proof.* Without loss of generality, we can assume that the discretizations $\mathcal{S}$ and $\mathcal{I}$ of the spaces $V$ and $\mathbb{R}$, extend to their entire respective domains (see Definition D.3). This is justified because the partition $\mathcal{S}$ is defined over the subset of $\mathbb{R}^d$ whose coordinates, in an orthonormal basis of $V$, are at most $\sqrt{\log(k/\eta)}$. By the union bound, the probability mass outside this region is at most $\eta$. Similarly, the same holds for $\mathcal{I}$, since $\mathbf{E}_{\mathbf{x}\sim\mathcal{N}_d}[f^2(\mathbf{x})] = 1$, there is at most an $\epsilon$ fraction of the probability mass outside the relevant interval when $|y| \geq 1/\epsilon^2$. As a result, by the union bound, we have

$$\mathbf{Pr}\left[T(\mathbf{x}_V,y) \neq \widehat{T}(\mathbf{x}_V,y), \ \mathbf{x} \notin \bigcup_{S\in\mathcal{S}} S \ \vee \ y \notin \bigcup_{I\in\mathcal{I}} I\right] \leq 2\epsilon.$$

Define the set $A = \{(\mathbf{x}_V, y) : |y - f(\mathbf{x}_V)| = \tau\}$ to be the boundary set of boolean function $T$. We set $P$ to be the subset of the discretization regions $(S,I)$ with $S \in \mathcal{S}$ and $I \in \mathcal{I}$ such that for all points in $(\mathbf{x}_V, y) \in (S,I)$ it holds that $|y - f(\mathbf{x}_V)| > \tau$. Note the diameters of the sets $S$ and $I$ are $\sqrt{k}\eta$ and $\eta$ for all regions $(S,I)$ with $S \in \mathcal{S}$ and $I \in \mathcal{I}$ respectively. Hence, we have that $\mathbf{Pr}_{(\mathbf{x},y)\sim D}[T(\mathbf{x}_V,y) \neq \widehat{T}(\mathbf{x}_V,y)] \leq \mathbf{Pr}_{(\mathbf{x},y)\sim D}[(\mathbf{x}_V,y) \in E]$, where we define $E \stackrel{\text{def}}{=} \{(\mathbf{x}_V, y) \mid \exists(\mathbf{x}'_V, y') \in A : \|\mathbf{x}_V - x'_V\| \leq \sqrt{k}\eta, |y - y'| \leq \eta\}$.

Note that for every $(\mathbf{x}_V, y) \in E$ since $f$ is $L$-Lipschitz we have that $|y - f(\mathbf{x}_V)| = \tau \pm 2L\sqrt{k}\eta$. Therefore, it suffices to upper bound $\mathbf{Pr}_{(\mathbf{x},y)\sim D}[|y - f(\mathbf{x}_V)| = \tau \pm 2L\sqrt{k}\eta]$ or in other words to show anticoncetration of the random variable $y - f(\mathbf{x}_V) = f(\mathbf{x}) - f(\mathbf{x}_V)$.

Define the parameter $\delta = 2L\sqrt{k}\eta$. Since $f$ depends only on the projection onto the $K$-dimensional subspace $W$, the function $g(\mathbf{x}_V) = f(\mathbf{x}) - f(\mathbf{x}_V)$ also depends only on $\mathbf{x}_W$. Moreover, because $f$ is positive-homogeneous, so is $g$. Hence we may define an induced function $\widetilde{g} : \mathbb{R}^K \to \mathbb{R}$ by $\widetilde{g}(\mathbf{z}) = f(\mathbf{z}) - f(\mathbf{z}_V)$, which is likewise positive-homogeneous, and observe that $g(\mathbf{x}) = \widetilde{g}(\mathbf{x}_W)$. Under $\mathbf{x} \sim \mathcal{N}_d$, the projected vector $\mathbf{x}_W$ is distributed as $\mathcal{N}_K$. Therefore $\mathbf{Pr}_{\mathbf{x} \sim \mathcal{N}_d}[\,g(\mathbf{x}) = \tau \pm \delta\,] = \mathbf{Pr}_{\mathbf{z} \sim \mathcal{N}_K}[\,\widetilde{g}(\mathbf{z}) = \tau \pm \delta\,]$, and one can carry out the anticoncentration analysis on the positive-homogeneous function $\widetilde{g} : \mathbb{R}^K \to \mathbb{R}$.

We show anticoncetration of the function $\widetilde{g}$. We do this in two stages. First we show anticoncetration over fibers, i.e. fixed lines that go through the origin, and then we take the expectation over fibers.

We can rewrite a gaussian vector $\mathbf{x}$ as $r\mathbf{v}$, where $\mathbf{v}$ as a uniform unit random vector and $r$ a scalar random variable independent of $\mathbf{v}$ such that $r^2 \sim \chi_K$. Note that for any fixed direction $\mathbf{v}$ it holds that

$$\mathbf{Pr}[\widetilde{g}(\mathbf{x}) = \tau \pm \delta \mid \mathbf{v} = \mathbf{z}] = \mathbf{Pr}[r\widetilde{g}(\mathbf{z}) = \tau \pm \delta]\,.$$

Let $\alpha > 0$ be a parameter to be quantified later. First, consider the case where $|\widetilde{g}(\mathbf{z})| \geq \alpha$ in this case by the Carbery-Wright inequality (see Fact E.5) we have that

$$\mathbf{Pr}[r\widetilde{g}(\mathbf{z}) = \tau \pm \delta \mid \mathbf{v} = \mathbf{z}] \leq \mathbf{Pr}\left[r = \frac{1}{\alpha}(\tau \pm \delta)\right] \lesssim \frac{1}{\alpha}\frac{\sqrt{\tau\delta}}{K^{1/4}}\,.$$

Second, we consider the case where $|\widetilde{g}(\mathbf{z})| < \alpha$. We have that

$$\mathbf{Pr}[r\widetilde{g}(\mathbf{z}) = \tau \pm \delta \mid \mathbf{v} = \mathbf{z}] \leq \mathbf{Pr}[r \geq \tau/(2\alpha)]\,,$$

since $\tau - \delta \geq \tau/2$. Note that setting $\alpha = \delta^{1/4}\tau/(2\sqrt{K})$ by the Gaussian Annulus theorem (Fact E.3) we have that $\mathbf{Pr}[r\widetilde{g}(\mathbf{z}) = \tau \pm \delta \mid \mathbf{v} = \mathbf{z}] \leq e^{-\sqrt{K/\delta}} \leq \sqrt{\delta/K}$.

Thus in both cases we have that $\mathbf{Pr}[r\widetilde{g}(\mathbf{z}) = \tau \pm \delta] \leq (K\delta)^{1/4}$. Taking the expectation over $\mathbf{v}$ completes the proof of Lemma D.28. $\qquad\square$

Moreover, we can also show that $f$ is very close in squared error to a bounded function. This is needed in order for us to be able to approximate $f$ using a finite collection of cubes.

**Lemma D.29** (Functions in $\mathcal{H}_{K,L}$ are almost bounded). *Let $f : \mathbb{R}^d \to \mathbb{R}$ be a function in $\mathcal{H}_{K,L}$ then for any $B \geq C\sqrt{K}L\ln(LK/\epsilon)$, for a sufficiently large constant $C > 0$, there exists a function $f_B : \mathbb{R}^d \to [-B, B]$ such that $\mathbf{E}_{\mathbf{x} \sim \mathcal{N}_d}[(f(\mathbf{x}) - f_B(\mathbf{x}))^2] \leq \epsilon$.*

*Proof.* Define the function $f_B(\mathbf{x}) = \text{sign}(f(\mathbf{x}))\min(|f(\mathbf{x})|, B)$. Note that since $f \in \mathcal{H}_{K,L}$ from Fact E.3 we have that for $t \geq L\sqrt{K}$ and some universal constant $C' > 0$

$$\mathbf{Pr}[|f(\mathbf{x})| \geq t] \leq \mathbf{Pr}[\|\mathbf{x}_W\| \geq t/L] \leq e^{-C't/(L\sqrt{K})}\,.$$

Therefore, by applying this tail bound, we have

$$\mathbf{E}\big[(f(\mathbf{x}) - f_B(\mathbf{x}))^2\big] \leq \mathbf{E}[f^2(\mathbf{x})\mathbb{1}(|f(\mathbf{x})| > B] = B^2\mathbf{Pr}[|f(\mathbf{x})| \geq B] + \int_{B^2}^{\infty}\mathbf{Pr}[|f(\mathbf{x})|^2 \geq t]dt$$

$$= B^2 e^{-C'B/(L\sqrt{K})} + 2\int_{B}^{\infty} t\mathbf{Pr}[|f(\mathbf{x})| \geq t]dt$$

$$= e^{-C'B/(L\sqrt{K})}(B^2 + BL\sqrt{K}/C' + L\sqrt{K}/C')\,.$$

Choosing $B$ to be $C\sqrt{K}L\ln(LK/\epsilon)$, for a sufficiently large constant $C > 0$, completes the proof. $\qquad\square$

Finally, before proceeding to the proof of our theorem, we make the following remark concerning the precise dependence of our algorithm's sample complexity on the dimension.

**Remark D.30.** We remark that the sample complexity bound in Theorem D.17 is $O(d^{O(m)})$, since at each step we perform polynomial regression of degree $m$ (see Fact E.9). In the special case $m = 2$, we can reduce this to $O(d^2)$ by noticing that the polynomial regression task is directly reducible

to covariance estimation in the Frobenius norm. Furthermore, achieving $O(d)$ sample complexity is possible by replacing the regression in Line 2 with a simple covariance-estimation step since covariance estimation in the operator norm requires $O(d)$ samples. Concretely, for each interval $I \in \mathcal{I}$ and region $S \in \mathcal{S}$, define $\mathbf{M}_{S,I} = \mathbf{E}_{(\mathbf{x},y) \sim D}[\mathbb{1}(y \in I)(\mathbf{x}\mathbf{x}^\top - I) \mid \mathbf{x} \in S]$, and let $\mathbf{u}_{I,S}$ be its top eigenvector. Then, following the filtering in Line 3, set $\widehat{\mathbf{U}} = \sum_{I \in \mathcal{I}, S \in \mathcal{S}} \Pi_{V^\perp} \mathbf{u}_{I,S}\, \mathbf{u}_{I,S}^\top \Pi_{V^\perp} \mathbf{Pr}[S]$. By essentially the same argument as in the proof of Proposition D.18, $\widehat{\mathbf{U}}$ provides an arbitrarily accurate projection onto $W$, while it requires only $O(d)$ samples.

Given Lemma D.28, we now prove the main theorem of this section, which shows that the class $\mathcal{H}_{K,L}$ can be learned efficiently using our algorithmic approach.

*Proof of Theorem D.22.* Let $V \subseteq \mathbb{R}^d$ be a subspace of $W$, and let $(\mathcal{S}, \mathcal{I})$ denote an $\eta$-approximating discretization of $V \times \mathbb{R}$, where $\eta \overset{\text{def}}{=} e^{-CK^2L^2/\epsilon^2}$ (see Definition D.3).

The proof consists of showing that $D$ is a $(2, e^{-CK^3L^2/\epsilon^2}, K, 1, \epsilon, e^{-CK^3L^2/\epsilon^2}, e^{-CK^2L^2/\epsilon^2}, e^{-CK^2L^2/\epsilon^2})$-well-behaved MIM distribution, as in Definition D.16, for a sufficiently large constant $C > 0$, and then simply applying Theorem D.17.

First, observe that for the distribution $D$ Conditions (1) and (2) are satisfied with parameters $\mathbf{E}_{(\mathbf{x},y) \sim \mathcal{N}_d}[y^2] = 1$ and dimension of the low dimensional subspace equal to $K$.

Let $W$ be a $K$ dimensional subspace of $\mathbb{R}^d$ such that $f(\mathbf{x}) = f(\mathbf{x}^W)$ (existence of $W$ is guaranteed since $f \in \mathcal{H}_{K,L}$) and let $V$ a subspace of $W$. Notice that since $\eta$ has been chosen appropriately small Fact D.32 and lemmas D.27 and D.28 together imply that if $\mathbf{E}_{\mathbf{x} \sim \mathcal{N}_d}[(f(\mathbf{x}) - f(\mathbf{x}_V))^2] \geq \epsilon$, then there exists $(S, I) \in (\mathcal{S}, \mathcal{I})$ and zero mean, unit variance polynomial $p : W_{V^\perp} \to \mathbb{R}$ of degree at most 2 such that $\mathbf{E}_{(\mathbf{x},y) \sim D}[p(\mathbf{x}_{W_{V^\perp}})\mathbb{1}(y \in I) \mid \mathbf{x}_V \in S] \geq e^{-CK^3L^2/\epsilon^2}$.

We can generalize the above statement for a general subspace $V$ of $\mathbb{R}^d$ with $\dim(V) \leq K$ by noticing that $f$ also depends only on $W' = W + V$. Since $V \subseteq W'$ and $\dim(W') \leq 2K$ we have that if $\mathbf{E}_{\mathbf{x} \sim \mathcal{N}_d}[(f(\mathbf{x}) - f(\mathbf{x}_V))^2] \geq \epsilon$, then there exists $(S, I) \in (\mathcal{S}, \mathcal{I})$ and zero mean, unit variance polynomial $p : W'_{V^\perp} \to \mathbb{R}$ of degree at most 2 such that $\mathbf{E}_{(\mathbf{x},y) \sim D}[p(\mathbf{x}_{W'_{V^\perp}})\mathbb{1}(y \in I) \mid \mathbf{x}_V \in S] \geq e^{-8CK^3L^2/\epsilon^2}$. Noticing that $W'_{V^\perp} = W_{V^\perp}$ gives us the statement for a general subspace $V$ of $\mathbb{R}^d$ of dimension at most $K$.

Therefore, by the assumption that $f$ is $L$-Lipschitz, to verify Condition (3), we can apply the aforementioned statement along with Lemmas D.6 and D.29, imply that if for a piecewise constant approximation $h_\mathcal{S}$ (see Definition D.4) it holds that $\mathbf{E}_{(\mathbf{x},y) \sim D}[(h_\mathcal{S}(\mathbf{x}) - y)^2] \leq \epsilon$, then $\mathbf{E}_{(\mathbf{x},y) \sim D}[p(\mathbf{x}_{W'_{V^\perp}})\mathbb{1}(y \in I) \mid \mathbf{x}_V \in S] \geq e^{-32CK^3L^2/\epsilon^2}$. Consequently, conclude that Condition (3) is satisfied with the specified parameters.

Taking into account Remark D.30 on the sample complexity of the algorithm, concludes the proof of Theorem D.22. $\qquad\square$

### D.3.2 Polynomials in a Few Relevant Directions

In this section, we demonstrate an application of our algorithm to the problem of learning polynomials that depend on only a few directions. Specifically, consider the class of $\alpha$-non-degenerate, low-rank polynomials.

**Definition D.31.** A polynomial $q : \mathbb{R}^d \to \mathbb{R}$ is *$\alpha$-non-degenerate* if

$$\mathbf{M} = \mathbf{E}_{\mathbf{x} \sim \mathcal{N}_d}\left[\nabla q(\mathbf{x})\, \nabla q(\mathbf{x})^\top\right] \quad \text{satisfies} \quad \mathbf{M} \succeq \alpha \|\mathbf{M}\|_2 \mathbf{I}.$$

We say a rank-$K$ polynomial $p : \mathbb{R}^d \to \mathbb{R}$ is non-degenerate if $p$ is non-degenerate in the $K$-dimensional subspace corresponding to the relevant directions. That is, there exist orthonormal vectors $\mathbf{w}^{(1)}, \ldots, \mathbf{w}^{(K)}$ such that $p(\mathbf{x}) = q(\mathbf{w}^{(1)} \cdot \mathbf{x}, \ldots, \mathbf{w}^{(K)} \cdot \mathbf{x})$ and $q$ is non-degenerate. We denote by $\mathcal{P}^\alpha_{K,m}$ the class of $\alpha$ non-degenerate polynomials of rank $K$, degree at most $m$ that have zero mean and unit variance under the standard gaussian.

Note the assumption on the mean and the variance is without loss of generality as we can normalize the samples and obtaining a variance dependency however we assume it for simplicity. We first

present a key structural result of [CM20] for the aforementioned class of polynomials. Specifically, for a distribution $D$ of $\mathbb{R}^d \times \mathbb{R}$, a scalar $\tau > 0$ and subspace $V$ with orthonormal basis $\mathbf{v}^{(1)}, \ldots, \mathbf{v}^{(k)}$, $\dim(V) = k$ define the following matrix:

$$\mathbf{M}_\tau^V = \left( \mathbf{E}_{(\mathbf{x},y) \sim D} \left[ T(\mathbf{x}_V, y) \left( \mathbf{x}_{V^\perp} (\mathbf{x}_{V^\perp})^\top - \Pi_{V^\perp} \right) \right] \right), \; T(\mathbf{x}_V, y) = \mathbb{1}(|y| > \tau, |\mathbf{v}^{(i)} \cdot \mathbf{x}| \le 1, i \in [k]) \tag{4}$$

The following fact states that this label transformation $T$ leads to a non-trivial moment.

**Fact D.32** (e.g., Lemma 4.2 [CM20]). *Let $d, m, K \in \mathbb{Z}_+$ and $\alpha > 0$. There exists constants $\tau$ and $\lambda$ that depend only on $K, m$ and $\alpha$ such that the following holds. Let $V$ and $W$ be a subspaces of $\mathbb{R}^d$ with $\dim(V) < \dim(W) = K$ such that $\|\mathbf{v}_W\| \ge 1 - \lambda$ for all unit vectors $\mathbf{v} \in V$. Let $p : \mathbb{R}^d \to \mathbb{R}$ be a polynomial in the class $\mathcal{P}_{K,m}^\alpha$ (see Definition D.31) with $p(\mathbf{x}) = p(\mathbf{x}_W)$. There exists a unit vector $\mathbf{u} \in W_{V^\perp}$ such that $\mathbf{u}^\top \mathbf{M}_\tau^V \mathbf{u} \ge \lambda$.*

In the following lemma we show that the label transformation defined in Equation (4) can be approximated arbitrary well by a piecewise constant function over a discretization of $V \times \mathbb{R}$ in cubes and intervals.

**Lemma D.33.** *Let $d, k \in \mathbb{Z}_+$ and let $\epsilon > 0$ such that $\epsilon < c$, for a sufficiently small constant $c > 0$. Let $p : \mathbb{R}^d \to \mathbb{R}$ be a polynomial of degree $m$ that has mean zero and variance one under $\mathcal{N}_d$, and let $D$ be the joint distribution of $(\mathbf{x}, p(\mathbf{x}))$ where $\mathbf{x} \sim \mathcal{N}_d$. Let $V$ be a $k$-dimensional subspace of $\mathbb{R}^d$, $k \ge 1$. Denote by $T : V \times \mathbb{R} \to \{0,1\}$ the function defined in (4), and let $(\mathcal{S}, \mathcal{I})$ be an $\epsilon$-approximating discretization of $V \times \mathbb{R}$ (see Definition D.3). Assume that $T$ and $\mathcal{S}$ are defined with respect to the same orthonormal basis of $V$. There exists a subset of the discretization $P \subseteq \mathcal{S} \times \mathcal{I}$ such that for the function $\widehat{T}(\mathbf{x}_V, y) = \sum_{(S,I) \in P} \mathbb{1}(\mathbf{x}_V \in S, y \in I)$ it holds that $\mathbf{Pr}_{(\mathbf{x},y) \sim D}[T(\mathbf{x}_V, y) \ne \widehat{T}(\mathbf{x}_V, y)] = O(k\epsilon + m\epsilon^{1/m})$.*

*Proof.* Let $\mathbf{v}^{(1)}, \ldots, \mathbf{v}^{(k)}$ be a basis of $V$ used to define $T$; this same basis is also used to construct the $\epsilon$-approximating partition $\mathcal{S}$. We construct $P$ as a cartesian product of subsets $\mathcal{S}' \subseteq \mathcal{S}$ and $\mathcal{I}' \subseteq \mathcal{I}$.

Define $R \stackrel{\text{def}}{=} \{\mathbf{x} : |\mathbf{v}^{(i)} \cdot \mathbf{x}| \le 1, i \in [k]\}$. Since $1 \le \sqrt{\log(k/\epsilon)}$ for $\epsilon \le c$, for a sufficiently small constant $c > 0$, it follows from the definition of an $\epsilon$-approximating partition (see Definition D.2) that for all $\mathbf{x} \in R$, there exists some $S \in \mathcal{S}$ such that $\mathbf{x} \in S$.

We define $\mathcal{S}'$ to be the union of the sets $S \in \mathcal{S}$ such that $S \subseteq R$. Note that in order for $\sum_{S \in \mathcal{S}'} \mathbb{1}(\mathbf{x} \in S)$ and $\mathbb{1}(\mathbf{x} \in R)$ to disagree on some point $\mathbf{x} \in \mathbb{R}^d$, it must be that $|\mathbf{v}^{(i)} \cdot \mathbf{x}| \in [1, 1 - \epsilon]$ for some $i \in [k]$. Indeed, if $\mathbf{x}$ satisfies $|\mathbf{v}^{(i)} \cdot \mathbf{x}| \le 1 - 2\epsilon$ for all $i \in [k]$, then the corresponding $S \in \mathcal{S}$ that contains $\mathbf{x}S$ must lie lie entirely within $R$

Using the union bound and the anti-concentration of the Gaussian distribution, we obtain the following bound on the disagreement probability

$$\mathbf{Pr}_{\mathbf{x} \sim \mathcal{N}_d}[\exists i \in [k] : |\mathbf{v}^{(i)} \cdot \mathbf{x}| \in [1, 1 - 2\epsilon]] \le 2k\epsilon.$$

Next, since $\mathbf{E}_{\mathbf{x} \sim \mathcal{N}_d}[p^2(\mathbf{x})] = 1$, by Markov's inequality we have that $\mathbf{Pr}_{\mathbf{x} \sim \mathcal{N}_d}[|y| \ge 1/\epsilon^2] \le \epsilon$. Hence, all but an $\epsilon$ fraction of the probability mass of $y$ that satisfying the condition $|y| \ge \tau$ lies within the discretization $\bigcup_{I \in \mathcal{I}} I$.

We define $\mathcal{I}'$ to be the union of all $I \in \mathcal{I}$ such that for all $y \in I$, we have that $|y| > \tau$. Then, similar to the argument above have that

$$\mathbf{Pr}[\mathbb{1}(y \in \mathcal{I}') \ne \mathbb{1}(|y| > \tau)] \le \mathbf{Pr}[|y| \in (\tau - \epsilon, \tau)] \lesssim m\epsilon^{1/m},$$

where the final inequality follows from the Carbery-Wright inequality (see Fact E.5). Applying the union bound concludes the proof of Lemma D.33. $\qquad\square$

Now given Lemma D.33 we can prove the following theorem which states that polynomials in a few relevant directions can learned by using our algorithmic approach.

**Theorem D.34** (Learning Polynomials in a Few Relevant Directions). *Let $K, m, d \in \mathbb{Z}_+, \alpha > 0$ and $\delta, \epsilon \in (0,1)$. There exists a constant $C(K, m, \alpha)$ that depends only on $K, m$ and $\alpha$ such that the following holds. Let $p : \mathbb{R}^d \to \mathbb{R}$ be a polynomial in $\mathcal{P}^\alpha_{K,m}$ (see Definition D.31) and let $D$ be the joint distribution of $(\mathbf{x}, p(\mathbf{x}))$ where $\mathbf{x} \sim \mathcal{N}_d$. Then, Algorithm 5 draws $N = d^2 \log(1/\delta) C(K, m, \alpha)/\epsilon^{O(mK)}$ i.i.d. samples from $D$, runs in time $\mathrm{poly}(N)$, and returns a hypothesis $h$ such that, with probability at least $1 - \delta$, it holds $\mathbf{E}_{(\mathbf{x},y)\sim D}[(h(\mathbf{x}) - y)^2] \le \epsilon$.*

*Proof.* The proof is very similar to the proof of Theorem D.17. Let $W$ be a $K$-dimensional subspace of $\mathbb{R}^d$ such that $p(\mathbf{x}) = p(\mathbf{x}_W)$ (note that the existence of $W$ is guaranteed by since $p \in \mathcal{P}^\alpha_{K,m}$).

First, observe that for the distribution $D$, Conditions (1) and (2) of Definition D.16 are satisfied for $\mathbf{E}_{(\mathbf{x},y)\sim D}[y^2] = 1$ and dimension of the low dimensional subspace equal to $K$.

We can apply Proposition D.18 together with Lemma D.27 and fact D.32 iteratively $K + 1$ times to obtain subspaces $V_t$, each of dimension $t - 1$ for $t \in [K + 1]$, starting from $V_1 = \{\mathbf{0}\}$ (exactly as in the proof of Theorem D.17). Using $N = d^2 \log(1/\delta) C(K, m, \alpha)/\epsilon^{O(mK)}$ samples and $\mathrm{poly}(N)$ time for all $K$ iterations, we have that every unit vector in $V_{K+1}$ is arbitrarily $\epsilon/(K^2 m)$-close to some unit vector in $W$

Using this we prove that $\|\Pi_W - \Pi_V\| \le \epsilon/(mK)$. Denote by $V \stackrel{\text{def}}{=} V_{K+1}$, by $\{\mathbf{v}^{(i)}\}_1^K$ and $\{\mathbf{w}^{(i)}\}_1^K$ orthonormal basis of $V$ and $W$ respectively. Also denote by $\mathbf{M}_V$ and $\mathbf{M}_W$ matrices that have $\{\mathbf{v}^{(i)}\}_1^K$ and $\{\mathbf{w}^{(i)}\}_1^K$ as column vectors. We have that

$$\|\Pi_W - \Pi_V\| \le K - \|\mathbf{M}_W^\top \mathbf{M}_V\|_F^2 = K - \sum_{i=1}^K \|\mathbf{v}_W^{(i)}\| \le \epsilon/(mK) .$$

Hence, it also holds that $\|\Pi_W \Pi_{V^\perp}\| = \|\Pi_W - \Pi_V\| \le \epsilon/(mK)$.

Therefore, by applying Fact E.6 we have that $\mathbf{E}_{\mathbf{x}\sim\mathcal{N}_d}[\|\nabla p(\mathbf{x})\|^2] \le m$. Hence, the difference $\mathbf{E}[(p(\mathbf{x}_W) - \mathbf{E}[p(\mathbf{z} + \mathbf{x}_{W_{V^\perp}}) \mid \mathbf{z} = \mathbf{x}_{W_V}])]$ can be bounded above by $O(\epsilon)$ using Claim D.8. Thus, we have that after $K$ iterations that there exists a function $g(\mathbf{x}_V)$ that achieves error $\epsilon$.

Finally, note that from the well-known fact that for $t > 2^{O(m)}$ it holds that $\mathbf{Pr}[|p(\mathbf{x})| \ge t] \le \exp(-O(mt^{2/m}))$ by simply integrating we can show that $p$ is $\epsilon$-close in squared error to a function bounded on $[-B, B]$ with $B = m/\epsilon^{O(m)}$. Hence, we can apply Lemma D.6 and claim D.14 for the aforementioned number of samples we conclude that the difference $\mathbf{E}_{(\mathbf{x},y)\sim D}[(h(\mathbf{x}) - y)^2] = O(\epsilon)$, for the output hypothesis $h$. Taking into account Remark D.30 on the sample complexity of the algorithm, concludes the proof of Theorem D.34. □

# E    Omitted Technical Facts

## E.1    Basic Mathematical Facts

**Fact E.1** (see e.g. Claim 2.3 in [DKRS23]). *Let $1 \le m < n$. Let $\mathbf{B} \in \mathbb{R}^{m \times n}$ with $\mathbf{B}\mathbf{B}^\intercal = \mathbf{I}_m$. It holds that $\mathbf{H}_k(\mathbf{B}\mathbf{x}) = \mathbf{B}^{\otimes k} \mathbf{H}_k(\mathbf{x}), \mathbf{x} \in \mathbb{R}^n$.*

**Fact E.2** (Gaussian Density Properties). *Let $\mathcal{N}$ be the standard one-dimensional normal distribution. Then, the following properties hold:*

*1. For any $t > 0$, it holds $e^{-t^2/2}/4 \le \mathbf{Pr}_{z\sim\mathcal{N}}[z > t] \le e^{-t^2/2}/2$.*

*2. For any $a, b \in \mathbb{R}$ with $a \le b$, it holds $\mathbf{Pr}_{z\sim\mathcal{N}}[a \le z \le b] \le (b - a)/\sqrt{2\pi}$.*

**Fact E.3** (Gaussian Annulus Theorem see e.g., [Ver18]). *If $\mathbf{x} \sim \mathcal{N}_d$, with probability at least $1 - \tau$ we have that $\left| \|\mathbf{x}\|^2 - d \right| \lesssim \log\frac{1}{\tau} + \sqrt{d \log\frac{1}{\tau}}$.*

**Fact E.4** (Gaussian Hypercontractivity; see e.g., [O'D14]). *Let $p : \mathbb{R}^d \to \mathbb{R}$ be a polynomial of degree at most $m$ which has zero mean and variance one under the gaussian distribution. For every real number $q \ge 2$, we have $\|p\|_{L^q} = (q - 1)^{\frac{d}{2}} \|p\|_{L^2}$.*

**Fact E.5** (Carbery-Wright Inequality see e.g., [CW01]). *Let $p : \mathbb{R}^d \to \mathbb{R}$ be a polynomial of degree $m$. If $\mathbf{Var}_{\mathbf{x} \sim \mathcal{N}_d}[p(\mathbf{x})] = 1$, then it holds that for any $t \in \mathbb{R}$ and $\epsilon > 0$ $\mathbf{Pr}_{\mathbf{x} \sim \mathcal{N}_d}[|p(\mathbf{x}) - t| \le \epsilon] \lesssim m\epsilon^{1/m}$ .*

**Fact E.6** (see, e.g., Lemma 6 in [KTZ19]). *Let $f \in L^2(\mathbb{R}^d, \mathcal{N}_d)$ with its $k$-degree Hermite expansion $f(\mathbf{x}) = \sum_{\alpha \in \mathbb{N}^d, \|\alpha\|_1 \le k} \widehat{f}(\alpha) H_\alpha(\mathbf{x})$. It holds that $\mathbb{E}_{\mathbf{x} \sim \mathcal{N}_d}\left[(\nabla f(\mathbf{x}) \cdot \mathbf{e}_i)^2\right] = \sum_{\alpha \in \mathbb{N}^d \|\alpha\|_1 \le k} \alpha_i (\widehat{f}(\alpha))^2$.*

## E.2 Omitted Content from Section D.1 to Section D.3

**Fact E.7** (see e.g. Claim B.1 [DIKZ25]). *Let $\mathbf{M} \in \mathbb{R}^{d \times d}$ a symmetric positive semi-definite (PSD) matrix and let $\mathbf{v} \in \mathbb{R}^d$ with $\|\mathbf{v}\| \le 1$ such that $\mathbf{v}^\top \mathbf{M} \mathbf{v} \ge \alpha$. Then there exists a unit eigenvector $\mathbf{u}$ of $\mathbf{M}$ with eigenvalue at least $\alpha/2$ such that $|\mathbf{u} \cdot \mathbf{v}| \gtrsim (\alpha/\|\mathbf{M}\|_F)^{3/2}$ . Moreover, the number of eigenvectors of $\mathbf{M}$ with eigenvalue greater than $\alpha/2$ is at most $4\|\mathbf{M}\|_F/\alpha^2$ .*

**Fact E.8** (see e.g. Claim 4.12 [DIKZ25]). *Let $\mathbf{M} \in \mathbb{R}^{d \times d}$ be a symmetric, PSD matrix and let $\mathbf{v} \in \mathbb{R}^d$ be a unit vector such that $\mathbf{v}^\top \mathbf{M} \mathbf{v} \le \epsilon$. Let $U$ denote the set of unit eigenvectors of $\mathbf{M}$ with eigenvalue at least $\lambda$. Then, for every $\mathbf{u} \in U$, it holds that $|\mathbf{u} \cdot \mathbf{v}| \le \sqrt{\epsilon/\lambda}$.*

**Fact E.9** (see, e.g., Lemma 3.3 in [DKK+21]). *Let $\mathcal{D}$ be a distribution on $\mathbb{R}^d \times \{\pm 1\}$ whose $\mathbf{x}$-marginal is $\mathcal{N}_d$. Let $k \in \mathbb{Z}_+$ and $\epsilon, \delta > 0$. There is an algorithm that draws $N = (dk)^{O(k)} \log(1/\delta)/\epsilon^2$ samples from $\mathcal{D}$, runs in time $poly(N, d)$, and outputs a polynomial $P(\mathbf{x})$ of degree at most $k$ such that $\mathbb{E}_{(\mathbf{x}, y) \sim \mathcal{D}}[(y - P(\mathbf{x}))^2] \le \min_{P' \in \mathcal{P}_k} \mathbb{E}_{(\mathbf{x}, y) \sim \mathcal{D}}[(y - P'(\mathbf{x}))^2] + \epsilon$, with probability $1 - \delta$.*

**Fact E.10** (see, e.g., Lemma 3.3 in [DKK+21]). *Fix $\epsilon \in (0, 1)$ and let $P(\mathbf{x})$ be a degree-$k$ polynomial, such that $\mathbb{E}_{(\mathbf{x}, y) \sim \mathcal{D}}[(y - P(\mathbf{x}))^2] \le \min_{P' \in \mathcal{P}_k} \mathbb{E}_{(\mathbf{x}, y) \sim \mathcal{D}}[(y - P'(\mathbf{x}))^2] + O(\epsilon)$. Let $\mathbf{M} = \mathbb{E}_{\mathbf{x} \sim \mathcal{D}_{\mathbf{x}}}[\nabla P(\mathbf{x}) \nabla P(\mathbf{x})^\top]$ and $V$ be the subspace spanned by the eigenvectors of $\mathbf{M}$ with eigenvalues larger than $\eta$. Then the dimension of the subspace $V$ is $\dim(V) = O(k/\eta)$ and moreover $\operatorname{tr}(\mathbf{M}) = O(k)$.*

**Fact E.11** (Approximation of a Bounded Variation Function using Cubes). *There exists a sufficiently small constant $c > 0$ such that the following holds. Let $f : \mathbb{R}^d \to \mathbb{R}$ continuous and continuous differentiable almost everywhere such that $\mathbf{E}_{\mathbf{x} \sim \mathcal{N}_d}[\|\nabla f(\mathbf{x})\|^2] \le L$. Moreover, assume that there exists a $B > 0$ and $f_B : \mathbb{R}^d \to [-B, B]$ such that $\mathbf{E}_{\mathbf{x} \sim \mathcal{N}_d}[(f(\mathbf{x}) - f_B(\mathbf{x}))^2] \le \rho$. Denote by $h : \mathbb{R}^d \to \mathbb{R}$ the piecewise constant approximation $h(\mathbf{x}) = \mathbf{E}_{\mathbf{x} \sim \mathcal{N}_d}[f(\mathbf{x}) \mid \mathbf{x} \in S]$, for all $\mathbf{x} \in S$ and $S \in \mathcal{S}$, where $\mathcal{S}$ is a collection of consecutive cubes over $\mathbb{R}^d$ of width $\eta \le c\epsilon/(Ld \log(B))$, i.e. $\mathcal{S}$ denotes all subsets of $\mathbb{R}^d$ of the form $\{\mathbf{x} : \mathbf{j}_i \epsilon + t \le \mathbf{x}_i \le (\mathbf{j}_i + 1)\epsilon + t\}, \mathbf{j}_i \in \mathbb{Z}^d, t \in [0, \epsilon/2]$.*

*Then $\mathbf{E}_{\mathbf{x} \sim \mathcal{N}_d}[(f(\mathbf{x}) - h(\mathbf{x}))^2] \le \epsilon + 2\rho$.*

*Proof.* Denote by $\phi(\mathbf{x}) = (2\pi)^{-d/2} \exp(-\|\mathbf{x}\|^2/2)$ and set $\mu_S(\mathbf{x}) = \phi(\mathbf{x})/\mathbf{Pr}_{\mathbf{x} \sim \mathcal{N}_d}[\mathbf{x} \in S]$. For each cube $S \in \mathcal{S}$ define $f_S = \mathbf{E}_{\mathbf{z} \sim \mu_S}[f(\mathbf{z})]$ and $m_S = 1/|S| \int_S f(\mathbf{z}) d\mathbf{z}$, where we denote by $|S|$ the geometric volume of the set $S$. Write

$$\phi_{\min}^S = \inf_{\mathbf{x} \in S} \phi(\mathbf{x}), \quad \phi_{\max}^S = \sup_{\mathbf{x} \in S} \phi(\mathbf{x}), \quad \kappa_S = \frac{\phi_{\max}^S}{\phi_{\min}^S}$$

Fix $R = \sqrt{d \log(B/\epsilon)/c}$ and let $T = \bigcup \{S : \min_{\mathbf{x} \in S} \|\mathbf{x}\| > R\}$. From the Gaussian Annulus Theorem (Fact E.3) we have that $\mathbf{Pr}[\mathbf{x} \in T] \le \epsilon/(8B^2)$.

Moreover, note that since $f$ is close to a bounded function, by Jensen's inequality so is $h$. Indeed $\mathbf{E}[(h(\mathbf{x}) - \mathbf{E}_{\mathbf{x} \sim \mathcal{N}_d}[f_B(\mathbf{x}) \mid \mathbf{x} \in S])^2] \le \rho$. Hence the tail error approximation error of $h$ is bounded the tail error $\mathbf{E}[(f(\mathbf{x}) - h(\mathbf{x}))^2 \mathbb{1}(\mathbf{x} \in T)] \le 2\rho + \epsilon/2$.

Hence without loss of generality we can consider cubes with $\|\mathbf{x}\| \le R$ for all $\mathbf{x} \in S$. Moreover, for those cubes $S \nsubseteq T$ it holds that

$$\kappa_S \le \kappa \stackrel{\text{def}}{=} \exp(R\eta\sqrt{d}) .$$

From the fundamental theorem of calculus, for every $\mathbf{x}, \mathbf{y} \in S$, it holds that

$$f(\mathbf{x}) - f(\mathbf{y}) = \int_0^1 \nabla f(\mathbf{y} + t(\mathbf{x} - \mathbf{y})) \cdot (\mathbf{x} - \mathbf{y}) dt .$$

Hence, using Jensen's inequality we have

$$(f(\mathbf{x}) - f(\mathbf{y}))^2 \leq \|\mathbf{x} - \mathbf{y}\|^2 \int_0^1 \|\nabla f(\mathbf{y} + t(\mathbf{x} - \mathbf{y}))\|^2 \, dt \leq d\,\eta^2 \int_0^1 \|\nabla f(\mathbf{y} + t(\mathbf{x} - \mathbf{y}))\|^2 \, dt$$

Averaging in $\mathbf{y} \in S$ and then in $\mathbf{x} \in S$ yields

$$\frac{1}{|S|} \int_S (f(\mathbf{x}) - m_S)^2 \, d\mathbf{x} \leq d\,\eta^2 \frac{1}{|S|} \int_S \|\nabla f(\mathbf{x})\|^2 \, d\mathbf{x} \,.$$

We can transfer the above bound to the gaussian case for all cubes $S \not\subseteq T$. Specifically,

$$\mathbf{E}_{\mu_S}[(f - m_S)^2] \leq \frac{\phi_{\max}^S |S|}{\mathbf{Pr}[S]} \mathbf{E}_{\mathrm{Unif}(S)}[(f - m_S)^2]$$

$$\leq \frac{\phi_{\max}^S |S|}{\mathbf{Pr}[S]} d\,\eta^2 \, \mathbf{E}_{\mathrm{Unif}(S)}[\|\nabla f\|^2]$$

$$\leq \kappa_S d\eta^2 \mathbf{E}_{\mu_S}[\|\nabla f\|^2].$$

Moreover, since $f_S$ minimizes $\mathbf{E}_{\mu_S}[(f(\mathbf{x}) - m)^2]$, we have

$$\mathbf{E}_{\mu_S}[(f(\mathbf{x}) - f_S)^2] \leq \mathbf{E}_{\mu_S}[(f(\mathbf{x}) - m_S)^2] \,.$$

Averaging over cubes we have

$$\mathbf{E}[(f(\mathbf{x}) - h(\mathbf{x}))^2] \leq \kappa d\eta^2 \mathbf{E}[\|\nabla f(\mathbf{x})\|^2] + 2\rho + \epsilon/2 \leq L d\kappa \eta^2 + 2\rho + \epsilon/2 \,.$$

Setting $\eta = c^2 \epsilon/(Ld \log(B))$ for a sufficiently small constant $c > 0$ concludes the proof of Fact E.11. $\qquad\square$

**Fact E.12** (Discretization over a Subspace). *Let $f : \mathbb{R}^d \to \mathbb{R}$ is continuous, continuous differentiable almost everywhere and satisfies $\mathbf{E}_{\mathbf{x} \sim \mathcal{N}_d}[\|\nabla f(\mathbf{x})\|^2] \leq L$. Moreover, assume that there exists a $B > 0$ and $f_B : \mathbb{R}^d \to [-B, B]$ such that $\mathbf{E}_{\mathbf{x} \sim \mathcal{N}_d}[(f(\mathbf{x}) - f_B(\mathbf{x}))^2] \leq \rho$. Let $V$ be a $k$-dimensional subspace of $\mathbb{R}^d$ and let $\mathbf{v}^{(1)}, \ldots, \mathbf{v}^{(k)}$ be an orthonormal basis of $V$ and let $S$ be the partition of $V$ into axis-aligned cubes of width $\epsilon$. Define $h(\mathbf{x}) = \mathbf{E}_{\mathbf{z}_V \sim \mathcal{N}_k}\left[f(\mathbf{z}_V + \mathbf{x}_{V^\perp}) \mid \mathbf{z}_V \in S\right]$. Then $\mathbf{E}_{\mathbf{x} \sim \mathcal{N}_d}[(f(\mathbf{x}) - h(\mathbf{x}))^2] \leq L\,k\,\epsilon^2$.*

*Proof.* Let $f_B$ be the truncation of $f$ to the interval $[-B, B]$, that is $f_B(\mathbf{x}) = \mathrm{sign}(f(\mathbf{x})) \min(|f(\mathbf{x})|, B)$. Note that $\mathbf{E}[(f_B(\mathbf{x}) - f(\mathbf{x}))^2] \leq \rho$ since $f_B$ is closer to $f$ than any other truncated function.

The function $f_B$ is continuous since we truncate at the level set $B$ continuously. Moreover, $f_B$ is non-differentiable at the points $\{ f(\mathbf{x}) = B\} \cap \{\nabla f(\mathbf{x}) \neq \mathbf{0}\}$. But by the implicit-function theorem, whenever $f(\mathbf{x}_0) = B$ and $\nabla f(\mathbf{x}_0) \neq \mathbf{0}$ there is a neighborhood in which $\{x : f(x) = B\}$ is a $C^1$ submanifold of codimension 1 in $\mathbb{R}^d$, hence of Lebesgue (and thus Gaussian) measure 0. Therefore these extra non-differentiable points lie in a countable union, since each neighborhood contains at least one point in $\mathbb{Q}^d$, of such submanifolds and so form a Gaussian-measure 0 set.

Moreover, almost everywhere

$$\nabla f_B(\mathbf{x}) = \begin{cases} \nabla f(\mathbf{x}), & |f(\mathbf{x})| < B, \\ \mathbf{0}, & |f(\mathbf{x})| > B \text{ or } \big(f(\mathbf{x}) = \pm B \text{ with } \nabla f(\mathbf{x}) = \mathbf{0}\big), \end{cases}$$

and on the remaining set—namely $\{ f(\mathbf{x}) = \pm B\} \cap \{\nabla f(\mathbf{x}) \neq \mathbf{0}\}$ together with the set of original non-differentiable points of $f$—the function fails to be differentiable but that set has Gaussian measure 0. Hence

$$\mathbf{E}_{\mathbf{x} \sim \mathcal{N}_d}\big[\|\nabla f_B(\mathbf{x})\|^2\big] \leq \mathbf{E}\big[\|\nabla f(\mathbf{x})\|^2\big] \leq L.$$

Denote by $h_B(\mathbf{x}) = \mathbf{E}_{\mathbf{z}_V \sim \mathcal{N}_k}\big[f_B(\mathbf{z}_V + \mathbf{x}_{V^\perp}) \mid \mathbf{z}_V \in S\big]$. First, by the law of total expectation and the independence of orthogonal components of the standard gaussian we have that

$$\mathbf{E}[(f_B(\mathbf{x}) - h_B(\mathbf{x}))^2] = \mathbf{E}_{\mathbf{x}_{V^\perp}}[\mathbf{E}_{\mathbf{x}_V}[(f_B(\mathbf{x}) - h_B(\mathbf{x}))^2]].$$

For each fixed $\mathbf{x}_{V^\perp}$, we can apply Fact E.11 in the $k$–dimensional subspace $V$ to the function $\phi_{\mathbf{x}_{V^\perp}}(\mathbf{x}_V) \overset{\text{def}}{=} f_B(\mathbf{x}^V + \mathbf{x}_{V^\perp})$, which is always bounded by $B$. Noting that inequality $\|\nabla_{\mathbf{x}_V} \phi_{\mathbf{x}_{V^\perp}}(\mathbf{x}_V)\| \leq \|\nabla f_B(\mathbf{x})\|$ gives us that that for any fixed $\mathbf{x}_{V^\perp}$

$$\mathbf{E}_{\mathbf{x}_V}[(f_B(\mathbf{x}) - h_B(\mathbf{x}))^2] \leq 2\rho + \epsilon.$$

Taking the outer expectation over $\mathbf{x}_{V^\perp}$ yields

$$\mathbf{E}_{\mathbf{x}}[(f_B(\mathbf{x}) - h_B(\mathbf{x}))^2] \leq 2\rho + \epsilon.$$

Finally, we have that

$$\mathbf{E}[(f(\mathbf{x}) - h(\mathbf{x}))^2]$$
$$\leq 2\mathbf{E}[(f(\mathbf{x}) - f_B(\mathbf{x}))^2] + 2\mathbf{E}[(f_B(\mathbf{x}) - h_B(\mathbf{x}))^2] + 2\mathbf{E}[(h_B(\mathbf{x}) - h(\mathbf{x}))^2] \lesssim \rho + \epsilon \ ,$$

where we used that $\mathbf{E}[(h_B(\mathbf{x}) - h(\mathbf{x}))^2] \leq 2\rho + \epsilon$ because of Jensen's inequality. $\qquad\square$

**Fact E.13** (Median of Means Estimator see, e.g., [BLM13] )**.** *Let $x_1, \ldots, x_n$ be i.i.d. random variables with mean $\mu$ and variance $\sigma^2$. Suppose that $n = mk$, where $m$ and $k$ are positive integers. Define the median-of-means estimator $\widehat{\mu}_n$ as the median of $k = \lceil 8\log(1/\delta) \rceil$ independent sample means. Then, with probability at least $1 - \delta$, we have $|\widehat{\mu}_n - \mu| \leq \sigma\sqrt{\frac{32\log(1/\delta)}{n}}$.*

**Fact E.14** (see e.g. Fact 3.3 [CKM22])**.** *If $Z$ is a random variable for which $|Z| \leq M$ almost surely, and $\mathbf{E}[Z^2] \geq \sigma^2$, then $\mathbf{Pr}|Z| \geq t \geq \frac{1}{M^2}(\sigma^2 - t^2)$.*

