# OpenReview forum: "Algorithms and SQ Lower Bounds for Robustly Learning Real-valued Multi-Index Models"
_NeurIPS.cc/2025/Conference — NeurIPS 2025 spotlight_

### Official Review · Reviewer_AGEu · 2025-06-14

**Clarity:** 3
**Significance:** 3
**Originality:** 3
**Rating:** 4
**Confidence:** 4

**Summary:**

The paper studies learning real-valued Multi-Index Models (MIMs) under Gaussian inputs, where the target depends on an unknown $K$-dimensional subspace. It gives a robust PAC regression algorithm (agnostic setting) that tolerates arbitrary label noise. Under mild regularity (“bounded variation” and “distinguishing low-degree moments”) it runs in time polynomial in $d$ (ambient dimension). A matching SQ lower bound showing any Statistical Query learner must incur roughly $d^{\Omega(m)}$ complexity when the low-degree moment condition fails. There is also a concrete application to positive-homogeneous Lipschitz MIMs (including certain ReLU networks).

**Questions:**

See above.

**Ethical Concerns:**

["NO or VERY MINOR ethics concerns only"]

**Limitations:**

Yes.

**Paper Formatting Concerns:**

No.

**Quality:**

3

**Strengths And Weaknesses:**

The paper is clearly written, and the mathematical arguments are rigorous and not hard to follow. The results are strong, providing decent results on learning MIM, and I recommend a clear acceptance provided the following concern is addressed.
I would point out one more related work (arXiv:2506.05500) that introduces a “generative leap” measure for multi-index models. It would greatly clarify the paper if the authors clarified how their parameter \$m\$ corresponds to this generative leap (which controls the overall learning complexity).
I would be happy to raise my score further once that connection is made explicit.

---

> ### Author Rebuttal · Authors · 2025-07-31
>
> We thank the reviewer for their time, effort, and positive evaluation of our work. We address the reviewer's question below.
>
> The arxiv paper by [Damian et al., 2025] pointed out by the reviewer  (arXiv:2506.05500) appeared online on June 5, three weeks after the NeurIPS submission deadline. As a result, we were unable to include a comparison in our submission.
>
> We provide a brief comparison below, and we would be happy to incorporate it in the revised version of our paper.
>
> At a high-level, while our work and [Damian et al., 2025] study qualitatively similar tasks (“learning multi-index models”), the two works address fundamentally different objectives:
>
> * **PAC Learning vs. Parameter Recovery:** Our submission focuses on the PAC learning task, i.e., the task of obtaining a hypothesis with low prediction error (without requiring to identify the underlying subspace). In contrast, [Damian et al., 2025] focuses on parameter recovery, aiming to precisely estimate the underlying subspace. Note that PAC learning may be feasible in settings where parameter recovery is information-theoretically impossible. Specifically, it is possible to design very simple functions that admit accurate predictors even when the subspace is not identifiable. We mentioned this distinction in our submission, when we compared against the prior work [DBLP24] (which focused on single-index models); see lines 1406-1413.
>
> * **Agnostic Noise:** Our submission provides conditions and provable guarantees for both the realizable/independent label noise and the agnostic settings. In contrast, the complexity measure introduced in [Damian et al., 2025] assumes a realizable, noiseless model.
>
> Despite the above differences, when both works are restricted to a realizable setting and when parameter-recovery is possible, one can show that our Definition 1.3 and the “generative-leap exponent” of [Damian et al., 2025] are essentially equivalent. This can be shown by roughly the same analysis as presented in Appendix C.3 of our submission for the generative exponent (the single-index special case).

---

### Official Review · Reviewer_qTJB · 2025-06-24

**Clarity:** 3
**Significance:** 3
**Originality:** 3
**Rating:** 5
**Confidence:** 3

**Summary:**

The paper studies the problem of learning real-valued multi-index models (MIMs) under Gaussian covariates in the presence of adversarial label noise. The paper proposes an iterative, subspace recovery based algorithm that PAC learns well-behaved MIMs. As a corollary, the authors derive the complexity of learning low rank Relu networks, improving existing results.

**Questions:**

1. The moment condition in part (2) of Definition 1.3 feels quite abstract. Can the authors give a more intuitive explanation or interpretation of what this condition is actually capturing?

2. Corollary 1.7 is said to improve over [CKM22] by removing the dependence on the network size $S$. But isn’t the Lipschitz constant $L$ effectively taking its place? If no prior bound on $L$ is assumed, and ReLU networks naturally have Lipschitz constants that scale with depth and width, doesn’t that reintroduce a kind of hidden dependence on $S$? Can the authors clarify whether $L$ can be controlled independently of $S$?

3. Also, maybe I’m misunderstanding something, but the discussion around optimality of Theorem 1.4 focuses on the $d^m$ term, due to the SQ lower bound. But in practice, isn’t the $2^{\mathrm{poly}(K/\varepsilon\sigma)}$ term a much bigger deal? For small enough $\varepsilon$, this term should easily dominate $d^m$, no?

**Ethical Concerns:**

["NO or VERY MINOR ethics concerns only"]

**Final Justification:**

I am happy with the response provided by the authors regarding my original comments. However, my knowledge in the field of computational learning theory is minimal to non-existent. So, please take my assessment with a grain of salt.

**Limitations:**

yes

**Quality:**

3

**Strengths And Weaknesses:**

This is a technically solid paper. It provides a PAC learner with sample complexity $N$ that runs in $\mathrm{poly}(N)$ time. A SQ lower bound is also given, which qualitatively matches some terms in $N$. So this is, to my knowledge, the first paper to give a partial characterization of efficient PAC learning for MIMs.

My main concern is that the paper doesnot do a great job of contextualizing the results. It says that “our understanding of the efficient learnability of MIMs is somewhat more limited,” but it is unclear what exactly this means. Are there known (possibly inefficient) PAC learners for MIMs? If so, how do the statistical and computational complexities of this paper’s algorithms compare to those?



Based on Definition 1.3, it looks like the authors are working with a subclass of $C^1$. For that class, are there standard covering number-based bounds for agnostic PAC learning. How do those sample complexity bounds compare to the ones proved in this paper? A common issue with traditional sample complexity bounds is that they often rely on ERM over the class (or a discretization), which results in exponential computational complexity in $1/\varepsilon$. But the algorithm from Theorem 1 already has exponential dependence on $1/\varepsilon$, so it’s not clear whether this approach improves on ERM from a computational perspective. Can the authors comment on this?

---

> ### Author Rebuttal · Authors · 2025-07-31
>
> We thank the reviewer for their time, effort, and insightful questions. We address the reviewer's points below.
>
> By the term “efficient learnability of MIMs”, we refer to the computational complexity of learning. Ignoring computational constraints, essentially tight bounds on the sample complexity of PAC learning follow easily from known techniques. On the other hand, prior to our work, no algorithm with runtime subexponential in the dimension $d$ was known (even for constant values of the parameters $L, K, \epsilon$.) We elaborate on these points in the following paragraphs.
>
> **Previously Known (Inefficient) PAC Learners for MIMs:** Via standard uniform convergence arguments, it can be shown that for well-behaved MIMs (in fact, for any bounded‐variation $C^1$ class on $\mathbb{R}^d$) the Empirical Risk Minimizer (ERM) is an agnostic PAC learner with sample complexity linear in the dimension $d$ and polynomial in the parameters $L, K, 1/\epsilon$.  Unfortunately, known implementations of the ERM require searching over an $\epsilon$–cover of the $K$-MIM Lipschitz functions, which takes time $2^{\Omega(d/\epsilon)}$. This runtime is exponential in the dimension $d$, even for constant values of the accuracy parameter. In contrast, the complexity of our algorithm is a fixed-degree polynomial in $d$ (with exponent roughly $m$, the degree of distinguishing moments in Definition 1.3).
>
> **Complexity of ERM vs. Our Algorithm:** As explained above, while the sample complexity of ERM is near-optimal, its computational complexity is exponential in $d$ (even for constant $\epsilon$). Our SQ lower bound strongly suggests an information-computation gap for our learning problem: while the information-theoretically optimal sample complexity is $O(d)$ (times some function of the other parameters), to achieve runtime sub-exponential in the dimension, we need to draw more samples—namely $d^{\Omega(m)}$. Our algorithm qualitatively matches this lower bound (as a function of the dimension), by drawing $d^{O(m)}$ samples (times some function of the other parameters) and running in sample-polynomial time.
>
> **Intuition for the Moment Condition (Def 1.3):** Our submission includes an intuitive explanation regarding this part of Definition 1.3: please see lines 54-56, and 165-178. As mentioned in lines 54–56, the “distinguishing low‑degree moment” condition measures how much improvement one can guarantee when the current subspace guess is off. Concretely, if the guess subspace yields large residual error, then a degree at most $m$ moment will detect that misfit and guide the next update—quantifying exactly how much progress one can make per iteration. If the reviewer has additional suggestions, we are happy to expand on this explanation in the revised version.
>
> **Dependence on Network Size vs. Lipschitz Constant:**
> Regarding the dependence of Corollary 1.7 on the Lipschitz constant $L$, there are a few comments we would like to make. We first want to emphasize that the depth of the network and the Lipschitz constant are independent quantities. Specifically, it is possible to have a depth-one network with arbitrarily large Lipschitz constant; or a very deep network with arbitrarily small Lipschitz constant. While the dependence of our algorithm (establishing Corollary 1.7) on the parameter $L$ may not be optimal, some dependence on $L$ is information-theoretically necessary (since the target function can be supported on a set of probability mass $O(1/L)$). The main contribution of Corollary 1.7 over the prior work [CKM22] is that it completely removes the dependence on the network size (parameter $S$, see Line 97, which is bounded below by the depth and may be much larger) while having qualitatively similar dependence on the Lipschitz constant $L$.
>
> **Relative Impact of the $2^{\mathrm{poly}(1/\epsilon)}$ vs. $d^m$ Terms:**
> The focus and main contribution of our paper is to qualitatively characterize the complexity of PAC learning MIMs as a function of the dimension $d$. Specifically, our work establishes that the $d^m$ term in the complexity of our algorithm is qualitatively optimal in the SQ model (see lines 80-81). We did not make any claims about the optimality of our methods in terms of the other parameters.
>
> Notably, prior to our work, no algorithm with sub-exponential dependence on $d$ was known in our setting, even for constant values of the parameters $\epsilon, L, K$.  As the first algorithmic result in this setting, our primary goal was to establish the correct dimension‐dependence.
>
> As pointed out by the reviewer, for $\epsilon$ very small, the $2^{\mathrm{poly}(1/\epsilon)}$ term may dominate the $d^m$ term. (It is worth pointing out here that, as mentioned in lines 75-76, the version of our algorithm for the case of independent noise has a $\mathrm{poly}(1/\epsilon)^K$ complexity.) Developing algorithms with polynomial dependence in all parameters is left as an interesting open problem.

---

> > ### Comment · Reviewer_qTJB · 2025-08-01
> >
> > I thank the authors for the clarification. My concerns have been largely addressed, and I will maintain my original score.

---

### Official Review · Reviewer_rcRM · 2025-06-27

**Clarity:** 3
**Significance:** 3
**Originality:** 3
**Rating:** 5
**Confidence:** 3

**Summary:**

In this paper, the authors study the problem of PAC learning multi-index functions. They show if the multi-index function $f$ has
distinguishing moments of degree at most $m$, then there is an algorithm that uses at most $d^{O(m)}$ i.i.d. samples
that can learn this function (even in the presence of adversarial noises). As an application of this generic result,
they show that homogeneous Lipschitz ReLU networks can be learned using $O(d^2)$ samples. They also complement their upper
bounds with a Statistical Query (SQ) lower bound which says that if degree $m$ distinguishing moments do not exist for
some subspaces, then any efficient algorithm will require a SQ oracle with tolerance smaller than $d^{-\Omega(m)}$.

**Questions:**

* Can you obtain bounds on the Lipschitz constant of ReLU networks (probably under some other conditions)?
  I am asking this because Corollary 1.7 depends on the Lipschitz constant of the network, which is expected to become
  worse as the network becomes deeper.
* Do your results cover [DIKZ25] as special cases if we treat the discrete label space as a subset of the real line?

**Ethical Concerns:**

["NO or VERY MINOR ethics concerns only"]

**Final Justification:**

The authors have resolved my concern (the improvement of Corollary 1.7 over the [CMK22]) during the rebuttal. I maintain my positive score (5).

**Limitations:**

Yes

**Quality:**

4

**Strengths And Weaknesses:**

### Strengths

* Overall, this is a well-written paper. The presentation is clear, and the proof sketches are easy-to-follow.
* The authors provide an algorithm that learns general multi-index functions with polynomial samples (assuming $m = O(1)$)
  under fairly general assumptions, and show the corresponding SQ lower bounds. These can potentially be used as baseline
  results in later analysis of multi-index models with more structures.
* The authors show that their complexity measure (the lowest degree of the distinguishing moments) is approximately
  equivalent to the Generative Exponent in the single-index case. In some sense, this can be viewed as an evidence
  that they are using the right/a reasonable measure.
* The RNGCA framework introduced in the proof of the lower bounds explicitly encodes the label subspace, which not only
  looks more natural for this type of learning tasks, but also makes the lower bounds applicable to the realizable/noiseless settings.

### Weakness

* The authors claim that Corollary 1.7 (Learning ReLU networks) is depth-independent. However, Corollary 1.7 requires the
  target class to be $L$-Lipschitz and has a super-exponential dependence on $L$, and it is not clear how $L$ grows when
  the network becomes deeper.
* Some parts of this paper, such as Definition 1.3 and Algorithm 1 and 2, are extended/refined/continuous version of the
  counterparts in [DIKZ25]. The authors should mention this more explicitly in the main text.


### Typos
* Line 45: should be $\mathrm{err}_D(f)$.
* Line 307: $\| v_W \|$. Do you mean something like $v_{W \cap V^\perp}$?

[DIKZ25] Ilias Diakonikolas, Giannis Iakovidis, Daniel M. Kane, Nikos Zarifis. Robust Learning of Multi-index Models via Iterative Subspace Approximation. 2025.

---

> ### Author Rebuttal · Authors · 2025-07-31
>
> We thank the reviewer for their time, effort, and insightful questions. We address the points and questions by the reviewer below.
>
> **Dependence on Network Size vs. Lipschitz Constant:** Regarding the dependence of Corollary 1.7 on the Lipschitz constant $L$, there are a few comments we would like to make. We first want to emphasize that the depth of the network and the Lipschitz constant are independent quantities. Specifically, it is possible to have a depth-one network with arbitrarily large Lipschitz constant; or a very deep network with arbitrarily small Lipschitz constant.
>
> While the dependence of our algorithm (establishing Corollary 1.7) on the parameter $L$ may not be optimal, some dependence on $L$ is information-theoretically necessary (since the target function can be supported on a set of probability mass $O(1/L)$). The main contribution of Corollary 1.7 over the prior work [CKM22] is that it completely removes the dependence on the network size (parameter $S$, see Line 97, which is bounded below by the depth and may be much larger) while having qualitatively similar dependence on the Lipschitz constant $L$.
>
> **Discrete Label Case:** While the main results of [DIKZ25] could be morally viewed as a special case of ours, certain non-trivial technical differences between the settings make it difficult to conclude this immediately. Specifically, our paper uses the $L_2$ loss, rather than the $L_0$ loss (misclassification error) in [DIKZ25]. Consequently, our setting allows agnostic noise that produces output values not in the given discrete set.

---

> > ### Comment · Reviewer_rcRM · 2025-08-02
> >
> > I thank the authors for the clarification. I'll maintain my positive score (5), though I still don't think the interpretation of Corollary 1.7 is entirely valid, as it is not the main contribution of the paper.
> >
> > I know it's easy to construct shallow networks with a large Lipschitz constant, but when the network is shallow, Corollary 1.7 doesn't yields any improvement over [CKM22] anyway, so what I'm actually asking is, can you find a non-trivial example where the Lispchitz constant does not grow exponentially with the depth? Even in the deep linear case with all weights bounded, the Lipschitz constant can still blow up. If we can't find such an example, then removing the dependence on the depth while keeping the (super-)exponential dependence on the Lipchitz constant doesn't seem to be a true improvement, unless you could argue that the bound in Cor. 1.7 is sharper than [CKM22] even when the Lipschitz constant is exponential in the depth.
> >
> > As a side note, I don't think (global) Lipschitzness is a good quantity to characterize the regularity of a deep network, as it is too strong/pessimistic and we know, because of the EoS mechanism, the local Lipschitzness of the model along the training trajectory is somewhat constrained by the learning rate.

---

> > > ### Author Response · Authors · 2025-08-04
> > >
> > > We thank the reviewer for engaging in this discussion. As the reviewer notes, Corollary 1.7 is not the main contribution of our work: it is an immediate corollary of Theorem 1.6—which applies to a more general class of functions—and is itself an application of our main algorithmic result, Theorem 1.4 (and its analogue for the independent noise). Still, we believe it is useful to further clarify this point.
> > >
> > > To avoid potential confusion, we start by pointing out that the parameter $S$ (appearing exponentially in the sample complexity and running time of the [CKM22] algorithm) is the *size* of the network, not the depth. This is basically the number of ReLU gates in the network (see lines 94-97 for the setup and definition). Looking closely at Theorem 5.2 in the arxiv version of [CKM22], we note that:
> > >
> > > * The sample complexity of their algorithm (i) contains a $2^S$ term, (ii) additionally contains a $\exp(K^3 L^2/\epsilon^2)$ term. The sample complexity of our algorithm includes term (ii) but not term (i).
> > > * The running time of their algorithm contains a $\exp(K^3 S^2 L^2/\epsilon^2)$. The runtime of our algorithm is $\exp(K^3 L^2/\epsilon^2)$, i.e., the same function of $K, L, \epsilon$ and independent of $S$.
> > >
> > > Moreover, even for small constant-depth ReLU networks, it is possible that $S$ is super-polynomially larger than $K, L$. A simple known example in the literature is the family of CSQ-hard instances in [DKKZ20]. This is a depth-2 ReLU network (linear combination of ReLUs) for which $K=2$ and $L = O(1)$, while $S$ is unbounded.  Furthermore it is not hard to construct ReLU networks that have the width of the second layer (which bounds $K$) small, while having the width of the third layer be much much larger, and still having only a small constant number of layers. In this case, $L$ should usually also be relatively small.

---

> > > > ### Comment · Reviewer_rcRM · 2025-08-05
> > > >
> > > > Thank you for the further clarification! I see the difference between the two bounds now.

---

### Official Review · Reviewer_woY5 · 2025-07-02

**Clarity:** 3
**Significance:** 4
**Originality:** 4
**Rating:** 5
**Confidence:** 4

**Summary:**

The paper presents a robust and efficient algorithm for PAC learning real-valued multi-index models, even under adversarial label noise. It matches its algorithmic sample complexity upper bounds with nearly tight Statistical Query lower bounds, showing the qualitative optimality of the proposed method in terms of dimension and noise tolerance.

**Questions:**

### Suggestions
- In Section 1.4, the authors present their techniques as an generalization of [DPLB24] to MIMs. It would be relevant and helpful to extend this comparison to the recent paper [Damian et al., 2025] on MIMs, which addresses analogous questions. Moreover, the cited work [TDD+24] presents a first extension of the "generative exponent" concept to MIMs (in the proportional regime). This connection is worth mentioning in the same paragraph for completeness.
- Points 1. and 4. in the Weaknesses section.



Damian et al., "The Generative Leap: Sharp Sample Complexity for Efficiently Learning Gaussian Multi-Index Models", 2025.

**Ethical Concerns:**

["NO or VERY MINOR ethics concerns only"]

**Final Justification:**

The authors have addressed the comments, I mantain my positive score.

**Limitations:**

Yes

**Paper Formatting Concerns:**

No concerns.

**Quality:**

4

**Strengths And Weaknesses:**

### Strenghts
The paper is well-written, theoretically sound, and significant. It provides strong results on the learnability of multi-index models for a broader class of functions than in previous works. The exposition is clear and includes detailed algorithmic descriptions, comparisons with prior literature, and explanations of the results.

### Weaknesses
No major weaknesses were identified. Minor points:
1. The second condition in Definition 1.3 is quite technical and may be difficult to parse for non-experts; an intuitive explanation would improve accessibility.
2. Line 45: is the symbol $c$ intended to be $f$?
3. Typo on line 202: "Specificalaly" → "Specifically".
4. The paper would benefit from a simple illustrative example to help readers build intuition around the main ideas and techniques.

---

> ### Author Rebuttal · Authors · 2025-07-31
>
> We thank the reviewer for their careful reading of our work and for pointing out typos. We will fix these minor issues in the revised manuscript. We address the reviewer’s points and suggestions below.
>
> **Intuition for the Moment Condition (Def 1.3):**
> We start by pointing out that our submission includes an intuitive explanation regarding this part of Definition 1.3: please see lines 54-56, and 165-178. As mentioned in lines 54–56, the “distinguishing low‑degree moment” condition measures how much improvement one can guarantee when the current subspace guess is off. Concretely, if the guess subspace yields large residual error, then a degree at most $m$ moment will detect that misfit and guide the next update—quantifying exactly how much progress one can make per iteration. If the reviewer has additional suggestions, we are happy to expand on this explanation in the revised version.
>
> **Comparison with [DBLP24]:**
> As mentioned in lines 245-246 of our submission and pointed out by the reviewer, our Multi‑Index Model (MIM) results can be viewed as a generalization of analogous results for the case of Single‑Index Model (SIMs) studied in [DBLP24]. However, as we explain in our technical overview, the transition to the MIM setting introduces qualitatively new challenges—both in theoretical analysis and in algorithm design—which necessitate substantially novel techniques. Moreover, as reviewer xGhv also noted, our work handles both the noiseless (realizable) and agnostic settings; whereas [DBLP24] treated only the noiseless case. Finally, we emphasize that our focus has been on the task of PAC learning, i.e., computing a hypothesis with low prediction error, as opposed to parameter recovery (i.e., identifying the hidden subspace); see lines 243-245 and 1406-1413. The prior work [DBLP24] on SIMs had focused on identifying the hidden direction (in a setting where it is identifiable).
>
> **Comparison to [Damian et al. (2025)]:**
> We thank the reviewer for pointing us to that arxiv manuscript (arXiv:2506.05500). That paper appeared online on June 5, 2025, i.e., three weeks after the NeurIPS submission deadline.  As a result, we were unable to include a comparison in our submission. We are happy to provide a comparison in the revised version of our manuscript. Please see our response to reviewer **AGEu** for such a comparison.
>
> **Incorporating [TDD+24]:**
> We appreciate the pointer to [TDD+24] as the first proportional‑regime generalization of the generative exponent to multi‑index models. We note that [TDD+24] is already cited in the supplementary material of our submission. We will include a detailed explanation and comparison in the revised version.

---

> > ### Comment · Reviewer_woY5 · 2025-08-05
> >
> > I thank the authors for their reply and the clarifications.

---

### Official Review · Reviewer_xGhv · 2025-07-12

**Clarity:** 4
**Significance:** 3
**Originality:** 3
**Rating:** 5
**Confidence:** 3

**Summary:**

This paper studies the problem of learning K-Multi-Index-Models (K-MIMs), which are models that depend on a K-dimensional subspace of the input.

The input distribution assumed in this paper is isotropic Gaussian, and the cases of agnostic learning, and independent noise are considered.

The paper provides:

1. a new algorithm for learning a K-MIM $f$ which satisfies the following condition.
Condition: for any subspace V,
- either (i) a good approximation to $f$ can be found that depends only on the input projected to V
- or, roughly speaking, (ii) if we subdivide the data space $\mathbb{R}^d \times \mathbb{R}$ into fine-grained buckets depending on the projection to V and the label $y$, then with non-trivial probability a degree-$m$ polynomial applied to data from a randomly chosen bucket will be able to distinguish between the case that $V$ contains all of the directions on which $f$ mostly depends, and the case that there are more directions on which $f$ depends.

2. The paper also proves matching SQ complexity bounds, showing that the dependence on the number of degree $m$ of the distinguishing polynomial is optimal.

As a corollary of the first result, the paper provides a new algorithm for learning homogeneous Lipschitz functions (which includes ReLU networks without biases), improving over the previous result of [CKM20] by removing an exponential dependence on the size of the network.

**Questions:**

No questions -- am in favor of acceptance

**Ethical Concerns:**

["NO or VERY MINOR ethics concerns only"]

**Final Justification:**

The paper is an interesting contribution to the theoretical literature on feature learning.

**Limitations:**

Yes

**Quality:**

4

**Strengths And Weaknesses:**

Clarity: despite the technical nature of this paper, it is quite clear and easy to read. The exposition is well done, and the results are presented in logical order.

Originality: as the paper itself emphasizes, some of the techniques draw heavily from the very recent paper "Robust Learning of Multi-index Models via Iterative Subspace Approximation" by Diakonikolas et al. 2025. The main difference between these papers appears to be (as the paper indicates), a generalization from discrete labels to continuous labels. This necessitates new techniques, which make the technical contributions of this paper sufficiently substantial in my opinion to count as original.

Significance: understanding the complexity of learning multi-index functions is a central question in understanding feature learning in neural networks, as multi-index functions are a central toy case in that field. This paper helps clarify the picture by providing a new learning algorithm for K-MIMs, along with matching SQ bounds.

Quality: the mathematics that I checked seemed correct

---

> ### Author Rebuttal · Authors · 2025-07-31
>
> We would like to thank the reviewer for the appreciation of our work and for providing helpful feedback.

---

### Decision · Program_Chairs · 2025-09-17

**Decision:**

Accept (spotlight)

**Comment:**

This paper studies the problem of learning real-valued multi-index models where the input distribution is standard Gaussian and the output depends only on a low-dimensional subspace of the input.
The main contributions of the paper are two-fold: first, under some technical conditions on the target function, the authors provide an agnostic PAC learning algorithm that works for a broad class of multi-index models, in the potential presence of adversarial noise; second, they establish a lower bound for the sample complexity using the statistical query framework.
The upper and lower bounds nearly match in terms of the dependence on the dimension of the input space.

Overall the paper is well-written and presents its ideas and results clearly.
The technical results are significant and contribute to our understanding of the fundamental learning problem for multi-index models.
While some of the techniques are based on [DIKZ25], the paper extends these ideas in non-trivial ways.
This adds to the recent literature on characterizing the computational complexity for learning single- and multi-index models.

Finally, the paper has received unanimous support from all reviewers.
The authors are encouraged to address the minor comments and suggestions and incorporate the feedback provided by the reviewers to further improve the paper.